# Decarbonizing the Building Sector: The Integrated Role of Environmental, Social, and Governance Indicators

Nicola Magaletti [1], Valeria Notarnicola [1], Mauro Di Molfetta [1] and Angelo Leogrande [2],*

[1] LUM Enterprise s.r.l., 70010 Casamassima, Italy; magaletti@lumenterprise.it (N.M.); notarnicola@lumenterprise.it (V.N.); dimolfetta@lumenterprise.it (M.D.M.)

[2] Dipartimento di Management, Finanza e Tecnologia (MFT), LUM University Giuseppe Degennaro, 70010 Casamassima, Italy

* Correspondence: leogrande.cultore@lum.it

## Abstract

Climate change mitigation for the built environment has become a subject of greatest urgency, as buildings account for nearly 40% of total energy consumption and nearly one-third of total $CO_2$ emissions. While environmental, social, and governance (ESG) indicators are increasingly used to monitor sustainability performance, their collective role in impacting building-related emissions is yet largely under-investigated. The current research closes that gap through an examination of the ESG dimension–$CO_2$ emissions intersection of 180 nations from 2000 to 2022, in the hope of illuminating how environmental, social, and governance elements interact to facilitate decarbonization. The research is guided by a multi-method design, including econometric examination, cluster modeling, and machine learning techniques, which provide causal evidence and predictive analysis, respectively. The findings reveal that the deployment of renewable energy significantly reduces emissions, while per capita energy use and $PM_{2.5}$ air pollution exacerbate this effect. The social indicators show mixed results: learning, women's parliamentary representation, and women's workforce representation reduce emissions, while food production and growth among the lowest-income individuals demonstrate higher emissions. Governance demonstrates mixed results as well, with good regulation reducing emissions under specific conditions yet primarily supporting high-income countries with superior infrastructure. The examination of clusters reveals that ESG-balanced performance is retained by countries in the low-emission clusters, whereas decentralized ESG pillars are associated with higher emissions. Machine learning confirms the existence of non-linear effects and identifies $PM_{2.5}$ exposure and renewable energy deployment as the strongest predictors of the relationship. In summary, the findings suggest that successful policies for decarbonizing the built environment are constructed upon the consistency of environmental, social, and governance plans, rather than single steps.

**Keywords:** carbon emissions; ESG indicators; building sector; machine learning; governance effectiveness

## 1. Introduction

Climate change is a defining issue of the 21st century. The building sector produces nearly 40% of global energy demand and a third of $CO_2$ emissions [1]. ESG indicators are now recognized as standards of sustainability, yet their macro-scale policy application is limited [2]. Except for investment screens, ESG proxies institutional strength,



innovation, and environmental performance. Healthcare and regulation reduce carbon intensity in industrialized sectors [3]. This study examines the connections between governance, education, healthcare, scientific productivity, and construction-sector emissions, utilizing IPCC data, econometrics, and machine learning [4]. Clustering and variable-importance mapping reveal cross-country differences and non-linear institution-emissions relationships. Research gaps are apparent: institutional and socio-ecological determinants are under-investigated, while technology diffusion and urban energy receive the most attention [5]. Findings indicate that effective governance attracts investors and that scientific productivity reduces emissions through innovation and policy. Paradoxically, greater R&D and governance accompany increased emissions, revealing a relationship between infrastructure growth and emissions [6]. Overall, ESG evidence reveals complex environmental performance determinants and policies that balance development and sustainability.

The article continues as follows: Section 2 reviews the current literature on how ESG features map onto emissions in buildings, identifying gaps in integrated studies. Section 3 presents a multi-method approach that combines econometrics, clustering, and machine learning across a global sample from 2000 to 2022. Sections 4–6 study environmental, social, and governance features in turn. They find unique but correlated effects on emissions. Section 7 synthesizes findings while highlighting ESG coherence. Sections 8 and 9 present limitations and conclude on the relevance of ESG to future emission mitigation policy.

The article also contains several appendices dedicated to exploring metric dimensions and analyzing data characteristics: Appendix A provides a comprehensive description of the variables used in the ESG analysis, along with data sources and definitions. Appendix B presents the summary statistics for the ESG model's Environmental (E) component. Appendix C releases the summary statistics, as well as the robustness checks, pertaining to the Social (S) dimension. Appendix D provides a summary of the results for the Governance (G) indicators, as well as their relationship with building-sector emissions. Appendix E presents diagnostic tests for autocorrelation and heteroscedasticity, ensuring the robustness of the econometric estimations. Appendix F describes the K-Nearest Neighbors (KNN) regression algorithm used, under the part dedicated to machine learning, the hyperparameter optimization process.

## 2. Literature Review

*E-Environmental.* Several works criticize the application of environmental certification tools at a symbolic level and a low level of linkage to actual environmental performance. Ref. [7], for instance, found that ESG tools applied within Florence's residential policy did not cause noticeable decreases in gas or energy consumption. Ref. [1] are similarly critical of the BREEAM system's application within the UK, as it places too much emphasis on environmental parameters while losing sight of equity and governance themes. In India, green certification is shown by [8] to generate higher rent levels, which are thus socioeconomically exclusionary. The separation between environmental performance and communication is further exacerbated by the minimal attention to facility-level practices in the post-construction phase. Whilst some works note positive practices, such as the use of prefabricated infrastructure for energy use and waste reduction [9], these benefits remain largely negated mainly due to mixed facility-level practices and maintenance [10,11]. Whilst ref. [10] successfully demonstrates that ESG-certified Class A office stock within the urban area of Madrid is a reality and firm energy savings occur, these are dependent upon repeated operating disciplines, such as metering and tenant engagement. This is reiterated in [12], who emphasize that successful facility practice is a necessity if sustainability practices are to be more than a symbolic application, especially within restrictive bureaucratic regimes. It has been posited that digital innovation has a solution. Ref. [13]

speculates about a digital monitoring ESG system that applies to commercial stock in real estate, but its hypothetical nature, coupled with actual non-verification, limits its current application. Ref. [14] promote the applications of technology but without any linkages between technology applications and their resultant sustainability impacts. Ref. [15] concur with this disconnect and note that Malaysian ESG reporting still tends to be at a rather transformational level, rather than a reputation level, in its operational application. Other environmental concerns highlight researchers who study wider energy and pollution infrastructures. Ref. [16] cooperate on materials and lifecycle emission traceability but highlight the basis of materials sustainability in successful use and procurement systems. Likewise, ref. [17] outline ESG as a "change engine" of construction but highlight common operating processes between procurement and waste treatment. At the same time, these works suggest that environmental performance within ESG thinking depends not only on innovation or design but also on process practices designed in and repeated, energy source quality, and ongoing administrative resolve.

*S-Social.* The social ESG of the built environment is conceptually applied but lacks theoretical depth at operational levels. The social ESG literature focuses primarily on design intention, occupant wellness, and occupant health but underscores the long-term socially desirable aspects that buildings do. Ref. [18] cite air and light performance research and studies on occupant satisfaction but remain constrained within first-order design, missing tenant use and after-occupancy engagement—the sustainability drivers where they have the most impact in the long term. Equity concerns apply equally to criticism claims. Ref. [8] conclude that India's green-awarded buildings rent at higher grade levels and are consequently exclusionary to poorer rent grades, thus exacerbating urban poverty. By analogy, ref. [19] cites a dilemma where ESG uptake still largely accrues within cash-based revenues, but environmental resilience is seldom accompanied by investment in the social or governance quadrants. Ref. [20] cite environmental and societal forces within real assets but provide no further explanation at the operational level beyond facilities. Retrofit and inclusion sustainability reports within comparisons beyond their own region. Ref. [21] suggest that ESG retrofit within South African facilities might be initiated more often due to energy blackouts (e.g., load shedding) rather than sustainability targets and will consequently not be maintained in the long term. Ref. [22] further suggest that larger practicum and service-provider architecture and engineering departments largely direct provision primarily within the SDGs during preparation intervals but overlook the use of ESG findings some years later. Ref. [23] sketch further ESG research mapping within commercial real property but concede that facility-level completion is poorly expounded. Social ESG implementation deterrent concerns often provide structural barriers. Ref. [24] concludes that tomorrow's knowledge, ignorance, and operational incompetence remain the main deterrents to green uptake within Kenya and calls for firm management reinforcement and institutional reinforcement. These deterrents predict ESG policy change extending beyond disclosure and design but broader, co-participative engagement and governance. Even if research works such as ref. [25] assume user behavior modelers and facilities managers to be promoters of ESG outcomes, only frail empirical evidence can be provided, and experimental confirmation needs to be carried out. G-Governance. Governance is ESG's third pillar and key to long-term sustainability within the built environment but has been revealed to have an inadequate operational governance mechanism. Regulatory fragmentation and inability to maintain follow-up on constructed buildings are common attributes of certain studies.

*G-Governance*. Systemic ESG issues, such as inefficient regulatory coordination and regulatory gaps, are highlighted in [26]. Ref. [27] clearly suggest that operational data requirements remain a consideration; however, minimal consideration continues to be given

to the long-term ESG performance effects emanating from facility-level operations. Physical elements of design often drive ESG implementation, but ref. [28] finds that it is further influenced by administrative rituals, such as communication between relevant parties and energy use tracking, thereby encapsulating building management within governance effectiveness. Contract provisions that interface with ESG obligations at inception, between design and everyday use, are featured in [29]. This continues to be carried forward by [30], who suggest that Chinese agendas incorporate ESG into property agendas, but making such incorporation a reality in everyday operational use remains elusive. But policy and financial arrangements fail in governance. Investment programs, such as sustainable investments, offer positive financial returns but do not necessarily raise doubts about whether they lead to reform at a facility level [31]. Likewise, ref. [32] note that, while tenants account for ESG certificates within rent premiums, administrators of buildings seldom receive assessments regarding their compliance with ESG standards in the long term, resulting in a gap between market incentives and governance accountability. A green certificate often fails to institutionalize governance in most instances. Ref. [33] note that standardized certificate requirements mostly overlook the operational dimension—the dimension where attention is focused on documentation but not on use. Only in a similar manner do ref. [34] identify gaps between taxonomy at an EU level and schemes such as BREEAM-SE, inducing distress in building managers in making long-term compliance plans. Refs. [35,36] go further to identify how reform efforts within facilities' sectors primarily call upon ESG but without designing operationally successful observance systems. Furthermore, managers' roles within buildings and facilities' governance are repeatedly mentioned in some studies. Ref. [37] identify an increase in property uptake but overlook the facilities' use of smart technology, while ref. [12] suggest that post-construction monitoring often falls victim to administrative inertia. Ref. [11] again sarcastically identifies ESG compliance and repeatable operating commitments as supporting governance's role at a building level. If there are no enforcing processes in place, then the ESG will be susceptible to becoming a label rather than a change-based governing process.

Governance, ESG's third pillar, is central to making the constructed environment sustainable in the long term; however, operating governance mechanisms are surprisingly absent in the literature. Some of the literature refers to regulatory gaps and disjointed follow-up after construction. Ref. [26] present evidence of structural ESG challenges, such as non-coordination and regulatory gaps. Ref. [27] refer to operational data requirements that are increasingly accepted; however, facility-level, everyday processes are essentially left out of consideration for their contribution to long-term ESG performance. Ref. [28] reveals that ESG is not only administratively applied to the physical side of processes but also in processes such as tracking edifice management and energy, and this is at the core of governance performance. Ref. [29] suggest that contractual clauses bring ESG requirements into conception and link design and everyday implementation. This is also reported by [30], who note that ESG adoption is becoming integrated but does not yet seem to be translating into regular operational practice. However, governance also raises questions about policy and the limits of finances. Ref. [31] reveals evidence that sustainable investment funds yield better monetary returns, but does not raise questions about whether these investment increments have a tangible impact on governance at the executive level. Likewise, ref. [32] report that tenants pay premiums for ESG certification, and facade managers work mainly but are seldom measured, on compliance with ESG requirements; an increasingly significant gap is emerging between market incentives and accountability in governance performance. Green certification does not institutionalize governance. Ref. [33] conclude that certified requirements tend to exclude the operational dimension—the theory of documentation, instead. Likewise, ref. [34] identify inconsistencies between the EU taxonomy and local

sets, such as BREEAM-SE, which makes it challenging for edifice managers to devise long-term compliance strategies. Refs. [35,36] further outline, based on construction reform, often relying on ESG but not converting it into operationally reflective mechanisms of oversight. Lastly, facilities governance and the positions of building managers receive attention in several remarks. Ref. [37] refer to innovative technological development at a property level but not facility managers' application of such technology. Ref. [12] note that post-construction monitoring is often hindered by administrative routine inertia. Ref. [11] personally equates ESG compliance with reproducible operational necessities and reiterates the governance function at a building level. Without enforceable, evolutionary standards, ESG risks amount to a label but not a transformational governance role (Table 1).

**Table 1.** ESG dimensions and their property-level implications in the building sector.

| ESG Dimension | References | Main Results | Comparison with Our Study |
|---|---|---|---|
| E-Environmental | [1,7–17,20,28,30–32,35–40]. | ESG certifications and technologies are often disconnected from actual emission outcomes; operational routines are key but under-applied. | Our study directly quantifies the impact of environmental ESG indicators (e.g., renewable energy, air quality) on building emissions, validating their role with econometric models. |
| S—Social | [8,18–25,29]. | Social aspects are conceptually acknowledged but rarely implemented operationally; issues of equity, inclusion, and behavior are underdeveloped. | Our study empirically links social variables (e.g., gender equity, income equality, labor participation) to emission levels, revealing both mitigation and rebound effects. |
| G—Governance | [1,10–12,15,17,19,24,26–36,39–44]. | Governance is often fragmented; regulation and institutional routines are disconnected from long-term ESG outcomes. | Our study finds that governance indicators (e.g., government effectiveness, political stability) may paradoxically correlate with higher emissions, especially in higher-income countries, highlighting the complexity of governance–ESG links. |

## 3. Modeling Building-Related Emissions Through ESG Dimensions: A Multi-Method Analysis Using Econometrics, Clustering, and Machine Learning

The built environment and climate change cannot be separated. The building sector accounts for approximately one-third of global $CO_2$ emissions [1]. The current research examines building-sector $CO_2$ emissions against the ESG prism. Multi-method research combines econometrics, clusterization, and machine learning. The panel econometrics comprises 180 nations (2000–2021). Fixed- and random-effect regressions are linked between building emissions and environmental, social, and governance indicators [45,46]. National typologies are highlighted by cluster analysis. Normalized ESG–emissions data cluster states into groupings with similar structures or fragmented profiles [47,48]. Supervised machine learning also preserves non-linear relationships. Random Forest, SVR, k-NN, and Boosting predict and rank ESG attributes by importance [49,50]. Results converge. Balanced ESG pillars are associated with lower emissions. Broken governance or weak social equity are associated with higher emissions [51,52]. Results show interdependence.

Environmental action becomes effective only if governance and social equity co-evolve. Disjointed ESG pillars are ineffective for lowering emissions. This triangulative approach clears up causal, predictive, and structural dynamics. The strong triangulation of econometrics, cluster analysis, and machine learning enhances validity [53–55]. Policy prescriptions are immediate: decarbonization of buildings is achievable only with governance quality, social inclusiveness, and environmental enforcement simultaneously. Fragmented ESG strategies are unsuccessful. See Figure 1. A complete description of the variables used to estimate the ESG model is provided in Appendix A.

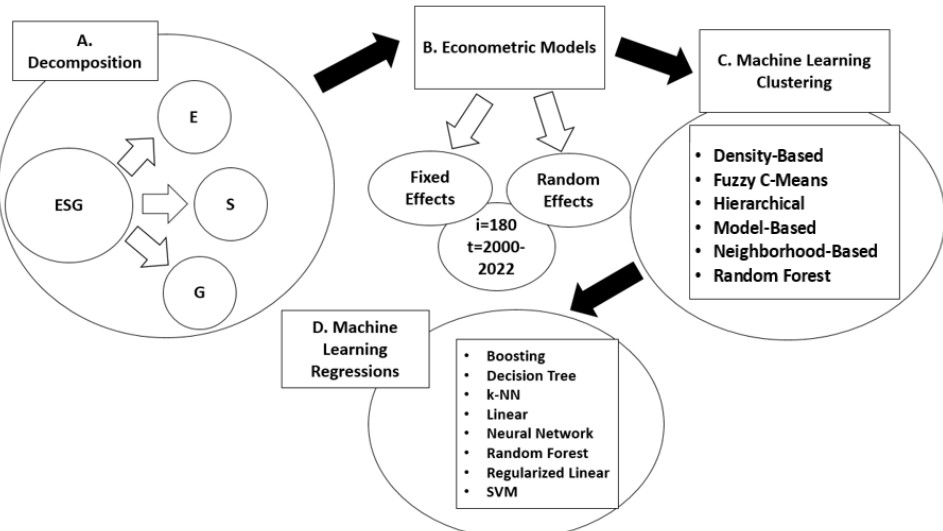

**Figure 1.** Multi-method analytical framework for ESG-driven building sector decarbonization. Note: Multi-method analytical framework for modeling building sector $CO_2$ emissions using ESG indicators. The framework begins with the decomposition of ESG into Environmental (E), Social (S), and Governance (G) dimensions (Panel (**A**)), followed by econometric modeling through fixed and random effects using panel data from 180 countries over 2000–2022 (Panel (**B**)). Clustering techniques (Panel (**C**)) such as density-based and hierarchical algorithms are applied to uncover emission patterns. Finally, machine learning regression models (Panel (**D**)) including k-NN, Random Forest, and SVM are used to predict emissions. This integrated approach captures both causal relationships and complex predictive patterns across ESG dimensions.

## 4. Decoding Building Emissions: Environmental Drivers in an ESG Framework (2000–2022)

Estimating the environmental determinants of building-related carbon dioxide emissions (BCE) holds the key to synchronizing the policies of decarbonization with ESG (environmental, social, and governance) targets. This section introduces the first phase of the empirical work on analyzing the effect of environmental indicators on BCE during the years 2000–2022 in 180 countries. With the help of fixed and random effects panel regressions, the study evaluates the effect of prominent energy and pollution-based indicators—access to clean fuels (CFTC), electricity access (ELEC), per capita energy usage (ENUC), particulate matter pollution ($PM_{2.5}$), and renewable energy contribution (RENC)—on building-related emissions. The outcomes serve as a quantitative underpinning for the interpretation of the environment's sustainability through the lens of ESG and present direct and indirect channels through which energy systems impact outcomes on emissions. This model forms a crucial diagnostic tool through which one can assess the environment's performance as well as the structural dynamics shaping carbon intensity in the built environment on an evidence-based trajectory. The metric characteristics of the variables used to estimate the ECB component with respect to the E-Environmental variables within the ESG model are

summarized in Appendix B. Diagnostic tests for autocorrelation and heteroscedasticity for the econometric estimates are provided in Appendix E.

### 4.1. Modeling the Environmental Determinants of Building Emissions: A Global ESG-Informed Econometric Analysis

To quantify the environmental determinants of building-related carbon emissions (BCE), this section presents a panel regression model estimating the effects of key energy-related indicators. Drawing on data from 180 countries over the period 2000–2022, the model incorporates variables aligned with the "E" pillar of the ESG framework, including clean fuel access, electricity availability, per capita energy use, $PM_{2.5}$ exposure, and renewable energy consumption. The analysis employs both fixed and random effects estimators, with results providing robust statistical insights into how national energy infrastructure and environmental conditions shape BCE outcomes globally.

Specifically we have estimated the following equation:

$$BCE_{it} = \alpha_1 + \beta_1(CFTC)_{it} + \beta_2(ELEC)_{it} + \beta_3(ENUC)_{it} + \beta_4(PM25)_{it} + \beta_5(RENC)_{it}$$

where i = 180 and t = 2000–2022. The results are shown in the following Table 2.

Building-sector $CO_2$ emissions (BCE) are a key environmental (E) metric within the ESG framework, making use of 180 countries' (2000–2022) panel data. The approach relates BCE to accessibility, quality, and composition of energy, including cleaner fuel transition (CFTC), electricity accessibility (ELEC), per capita energy use (ENUC), exposure to $PM_{2.5}$, and use of renewable energy (RENC). The results establish that CFTC stimulates social performance while increasing BCE unless fueled by low-carbon energy [48,54]. On the contrary, a transition to electricity reduces the rate if electricity is generated from cleaner sources, thereby maintaining ESG co-benefits for universal electricity accessibility [56]. Increasing ENUC matches with rising emissions, making efficiency policies relevant for high-consumption nations. From a governance (G) perspective, proper planning, conservation incentives, and regulating energy-intensive tech are central [57]. $PM_{2.5}$ exhibits a high and desirable correlation with BCE, supporting the convergence between environmental damage and public health hazards and integrating the E–S pillars within ESG. On the contrary, the use of renewable energy (RENC) shares a high negative correlation with emissions, validating green energy as a prime avenue for a resilient future. The fixed-effect formulation yields a high $R^2$ value for the panel estimates, validating the use of BCE as a robust metric for ESG while accounting for country-wise dissimilarity. The results, in general, reveal that electrification and renewables lead to abatement of emissions, while energy efficiency is crucial for offsetting per capita demand growth [54,56]. The linkage between BCE and $PM_{2.5}$ provides a compelling rationale for integrating policies on both air quality and climate that need to be globally instituted, especially for emerging economies with lower regulations [48]. The results position BCE at the core of ESG performance, where the building sector's performance is dependent on accessibility, quality, efficiency, and governance [47,57].

**Table 2.** Results of the panel data.

| | Random Effects (GLS), Using 990 Observations, Dependent Variable: BCE | | | Fixed Effects, Using 990 Observations Dependent Variable: BCE | | |
|---|---|---|---|---|---|---|
| | Coefficient | Std. Error | z | Coefficient | Std. Error | t-Ratio |
| const | 9.3166 | 11.6274 | 0.8013 | 10.2987 | 10.6061 | 0.9710 |
| CFTC | 0.318025 *** | 0.0842985 | 3.773 | 0.355114 *** | 0.0893023 | 3.977 |

**Table 2.** *Cont.*

| | Random Effects (GLS), Using 990 Observations, Dependent Variable: BCE | | | Fixed Effects, Using 990 Observations Dependent Variable: BCE | | |
|---|---|---|---|---|---|---|
| | **Coefficient** | **Std. Error** | **z** | **Coefficient** | **Std. Error** | **t-Ratio** |
| ELEC | −0.245591 *** | 0.0923988 | −2.658 | −0.273064 *** | 0.0964139 | −2.832 |
| ENUC | 0.00269113 *** | 0.000691466 | 3.892 | 0.00284832 *** | 0.000735797 | 3.871 |
| PM25 | 0.664678 *** | 0.131472 | 5.056 | 0.691987 *** | 0.144470 | 4.790 |
| RENC | −0.489147 *** | 0.104604 | −4.676 | −0.513324 *** | 0.112617 | −4.558 |
| Statistics | Mean dependent var | | 2.480 | | | 2.480 |
| | Sum squared resid | | 5,836,005 | | | 90,688 |
| | Log-likelihood | | −5702 | | | −3640 |
| | Schwarz criterion | | 11,445 | | | 8.413 |
| | Rho | | 0.756174 | | | 0.756174 |
| | S.D. dependent var | | 7.753 | | | |
| | S.E. of regression | | 7.697 | | | |
| | Akaike criterion | | 11,416 | | | |
| | Hannan–Quinn | | 11,427 | | | |
| | Durbin–Watson | | 0.324935 | | | |
| | LSDV R-squared | | | | | 0.984747 |
| | LSDV F(163, 826) | | | | | 3.271 |
| Test | 'Between' variance = 4971.73, 'Within' variance = 91.6047, mean theta = 0.935247, Joint test on named regressors—Asymptotic test statistic: chi-square(5) = 109.56 with $p$-value = 5.0646 $\times$ $10^{-22}$ | | | Joint test on named regressors— Test statistic: F(5, 826) = 20.981 with $p$-value = P(F(5, 826) > 20.981) = 8.85584 $\times 10^{-20}$ | | |
| | Breusch–Pagan test—Null hypothesis: Variance of the unit-specific error = 0, Asymptotic test statistic: chi-square(1) = 3155.73 with $p$-value = 0 | | | Test for differing group intercepts—Null hypothesis: The groups have a common intercept Test statistic: F(158, 826) = 319.607 with $p$-value = P(F(158, 826) > 319.607) = 0 | | |
| | Hausman test—Null hypothesis: GLS estimates are consistent, Asymptotic test statistic: chi-square(5) = 5.09644, with $p$-value = 0.404224 | | | | | |

Note: *** $p < 0.01$.

### 4.2. Evaluating Clustering Strategies for Building Emissions: A Multi-Metric Comparison

To disclose building-related carbon emission (BCE) global patterns and underlying country groupings that are structurally regular, we employ clustering techniques. Six methods are systematically compared across ten standardized performance measures, which vary from statistical fit and cohesion to quality of separation. The comparative benchmarking identifies the best possible segmentation technique for emission profiles, ensuring appropriate classification across countries by BCE structure. Moving beyond the selection of techniques, the application of clustering supports the interpretational strength of diagnostics that are ESG-based through correlation with environmental indicators. The approach possesses both technical sophistication and analytical coherence and constitutes a tool that scales for comparing emission dynamics across diverse country settings (Figure 2).

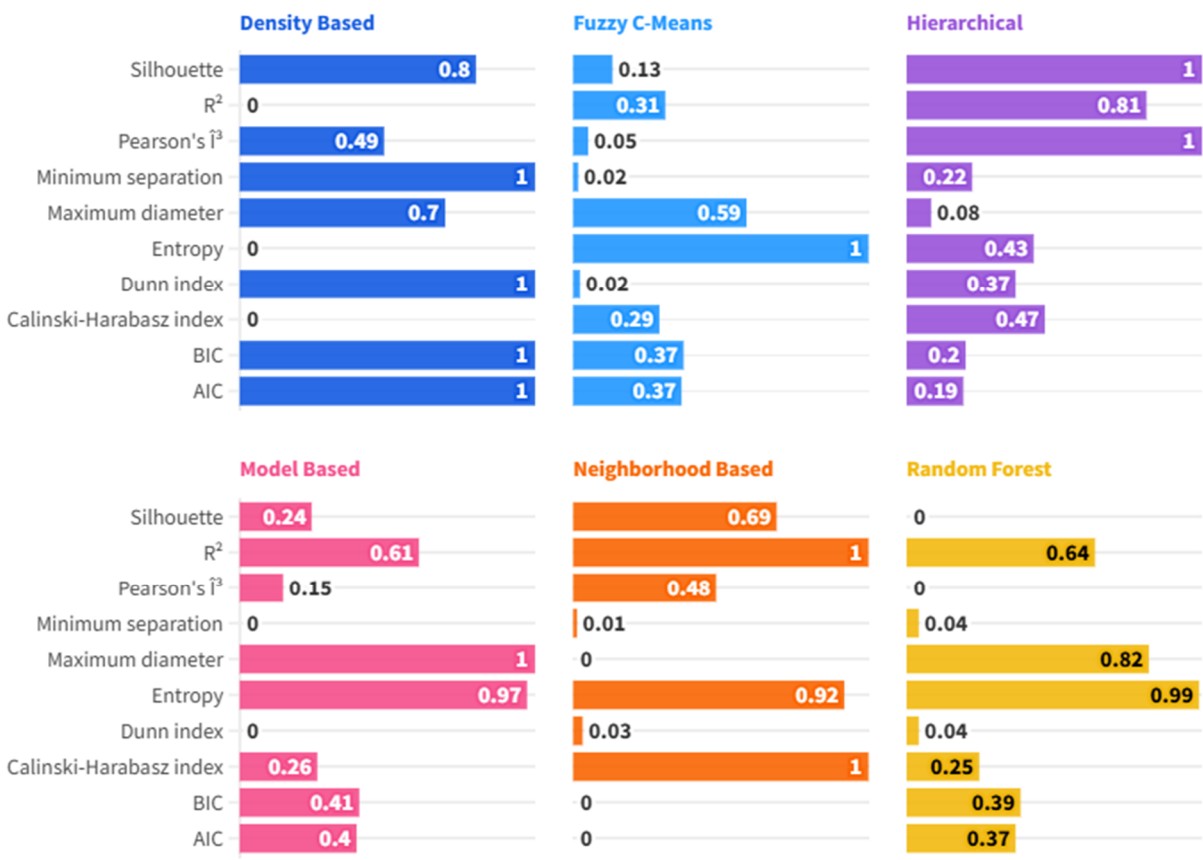

**Figure 2.** Comparative performance of clustering algorithms on building-related carbon emissions (BCE) profiles across ten normalized metrics.

Six cluster procedure performances were compared against ten standardized indices ($R^2$, AIC, BIC, silhouette, Dunn, maximum diameter, minimum separation, entropy, and Calinski–Harabasz), each on a 0–1 scale. The density-based algorithm performed exceptionally well, achieving a mean score of 0.599, surpassing the optimal AIC and BIC (1.0) and achieving perfect separation indices (1.0), which is typical for well-defined and tightly clustered data [58]. Its silhouette value was high (0.8), beaten only by hierarchical (1.0), and satisfactory performance on maximum diameter (0.696) and Pearson's $\gamma$ (0.489). On $R^2$ and entropy, they performed poorly, although these indices are better suited for non-linear cluster shapes and are more suitable for DBSCAN [59]. Hierarchical was best in terms of $R^2$, silhouette, and $\gamma$ while performing poorly in terms of separation, suggesting potential overfitting. The neighborhood- and model-based techniques showed one-dimensional strengths and mixed performance. The density-based approach showed the best-rounded performance and handles noisy or cluttered data the best.

Evaluating Cluster Quality and Structure in ESG-Driven Density-Based Analysis

This section reports on results from density-based environmental and energy indicator clustering relevant to ESG (environmental, social, governance). The leading indicators—clean fuel access (CFTC), electricity access (ELEC), per capita energy use (ENUC), PM$_{2.5}$ exposure, and renewable energy use (RENC)—produced several sharply differentiated clusters and one noise group. The clusters varied in size, shape, and separation, as indicated by the within-cluster sum of squares (WSS), explained heterogeneity, and silhouette statistics. The latter two statistics corroborate both statistical stability and interpretational soundness of the solution. The results link country energy infrastructures and environmen-

tal configurations with building-related $CO_2$ emissions (BCE) and reveal clustering as a gauge for benchmarking and ESG-centered sustainability reporting (Table 3).

**Table 3.** Cluster characteristics from density-based clustering on ESG-linked energy and environmental indicators.

| Cluster | Noise Points | 1 | 2 | 3 | 4 | 5 |
|---|---|---|---|---|---|---|
| Size | 1 | 949 | 6 | 6 | 20 | 8 |
| Explained proportion within-cluster heterogeneity | 0.000 | 0.999 | $8.471 \times 10^{-4}$ | $2.529 \times 10^{-5}$ | $1.102 \times 10^{-5}$ | $4.580 \times 10^{-4}$ |
| Within sum of squares | 0.000 | 4.584 | 3.888 | 0.116 | 0.051 | 2.103 |
| Silhouette score | 0.000 | 0.347 | 0.632 | 0.968 | 0.985 | 0.820 |

The DB model identified one noise point and five distinct clusters. The cluster quality was determined based on size, explained variance, WSS, and silhouette measure. The noise point, a typical DBSCAN characteristic [60], exhibited a WSS and silhouette value of 0, indicating evident separation from the cluster. Cluster 1, with 949 data points, exhibited high explained variance (0.999) but low isolation (silhouette = 0.347) and was therefore observed to create an overlap boundary artifact [61]. Clusters 2–5 were much smaller, each having between 6 and 20 data points. Cluster 4 was an outlier (WSS = 0.051; silhouette = 0.985), and Clusters 3 (silhouette = 0.968), 5 (0.820), and 2 (0.632) also passed the 0.5 interpretability cutoff [62]. The entire cluster display showed primarily spherical, compact shapes with acceptable isolation, although resolution within the noise was low. Cluster 1 showed lower discrimination compared to the small clusters, which exhibited sharper and clearer groupings (Figure 3).

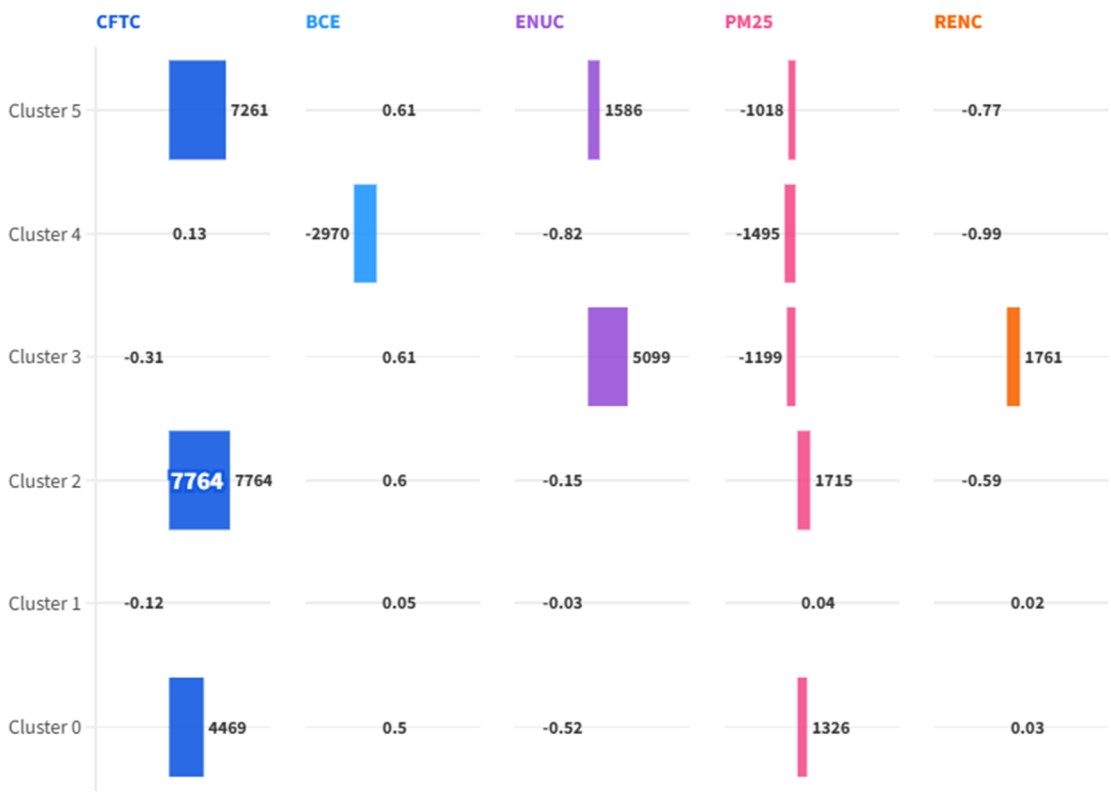

**Figure 3.** Mean indicator values for ESG-based clusters identified through density-based clustering.

Examining the environmental (E) pillar within ESG, the model correlates building-related $CO_2$ emissions (BCE) with a set of five indicators: clean cooking fuels (CFTCs), electricity access (ELEC), per capita energy use (ENUC), $PM_{2.5}$ exposure, and renewable energy share (RENC). The strategy is consistent with integrated sustainability indices [63,64]. The nations fall within six groups. Cluster 0 represents energy poverty, characterized by low electricity, high $PM_{2.5}$ levels, low renewable energy sources, and a moderate BCE. Cluster 1 has balanced and low BCE regimes. Cluster 2 is more closely associated with high pollution and low renewable energy sources and thus receives a moderate BCE from fossil-heavily dominated urban–industrial systems. Cluster 3 achieves high electricity and renewables, as well as moderate BCE, due to efficiency gains. Cluster 4 has very low BCE and $PM_{2.5}$ levels, which is likely due to its nuclear or hydrocarbon dependence. Cluster 5 balances clean energy and low pollution yet maintains a moderate BCE, characteristic of efficient, high-income fossil fuel economies. Overall, each cluster plots distinct energy–environment portfolios, testifying to how integrating indicators strengthens the ESG diagnostics (Figure 4).

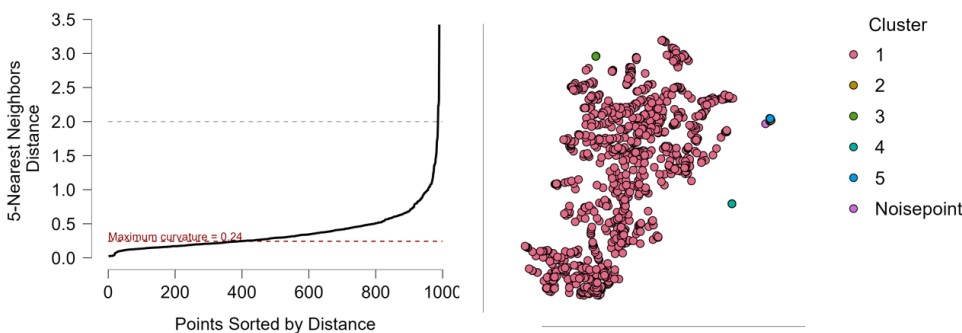

**Figure 4.** DBSCAN clustering and 5-Nearest Neighbor distance plot for environmental indicator-based country segmentation.

### 4.3. Explaining Carbon Emissions in the Built Environment: A Comparative Machine Learning Approach

This section presents a rigorous comparison of multiple machine learning regression models—ranging from ensemble methods to neural networks and proximity-based learners—evaluated through standardized performance metrics. By identifying the most reliable and accurate algorithm for predicting building-sector emissions, the analysis sheds light on model suitability and interpretability, with a particular focus on the role of local energy consumption, pollution exposure, and access variables in shaping emissions profiles (Figure 5).

Model performance was evaluated using MSE, RMSE, MAE, MAPE, and $R^2$, with the metrics normalized and inverted for ease of comparison. Composite scores showed that k-Nearest Neighbors (k-NNs) strongly dominated the rest, with error rates being low and an $R^2$ level of 1.0, consistent with a perfect explanation of variance. This is the fit between k-NN's similarity principle and patterned data structures, as well as its adaptiveness with weak functional assumptions [65]. Despite the risk of overfitting driven by a perfect score, k-NN performed strongly across various error measures. Compared to that, Boosting yielded similar results with higher errors and lower $R^2$ [66]. Decision Trees overfitted (low MSE but $R^2 = 0$), while linear, regularization, and neural network models lacked complexity. Random Forest was stable without localized precision. Overall, k-NN's high level of accuracy, along with its parsimonious and flexible character, highlights its strength for locally regularized data in empirically tested, simple models, consistent with prior results in software engineering estimation [67]. See Figure 6.

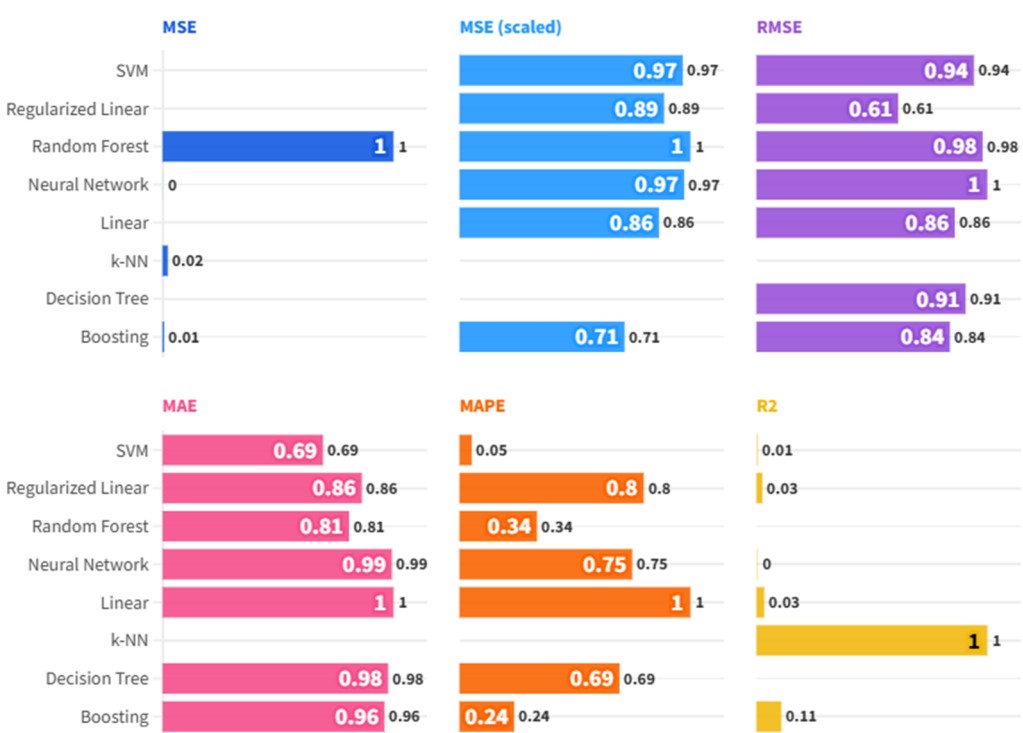

**Figure 5.** Comparative performance of machine learning regression models for predicting building-sector $CO_2$ emissions.

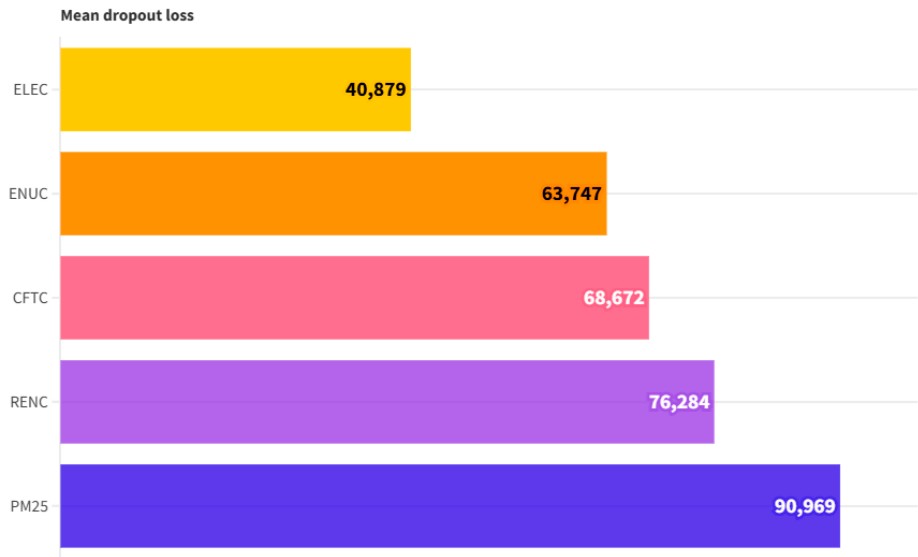

**Figure 6.** Feature importance rankings based on mean dropout loss in predicting building-sector emissions.

By applying k-Nearest Neighbors (k-NNs) with dropout loss (RMSE on 50 permutations), we determined significant environmental associates for building-related $CO_2$ emissions (BCE) in the ESG "Environment" pillar. The power consumption of fossil fuel use, combined with simultaneous air conditioning and heating, was the top contributor to air pollution ($PM_{2.5}$) (90.969), highlighting the significant impact of fossil fuel use on power, heating, and air conditioning consumption, as well as the climate–health hazard associated with decarbonization policy [62]. Consumer energy consumption of renewable energy (RENC) came a close second (76.284), indicating a reduction in carbon intensity and facilitating policy-mandated decarbonization [65]. Clean fuel for cooking (CFTC) came in a distant third (68.672), associated with biomass dependence and the infrastructural

vulnerability of low-income countries [68]. Energy use per capita (ENUC) (63.747) delivered demand-side increases in emissions, while electricity penetration (ELEC) delivered the weakest effect (40.879) due to near-universal penetration. Overall, $PM_{2.5}$ and RENC were chief initiators, followed by CFTC and ENUC, and followed by marginal ELEC. The implications are for renewable installation, clean fuel consumption, and air quality management as central BCE mitigation activities within ESG policy. See Figure 7.

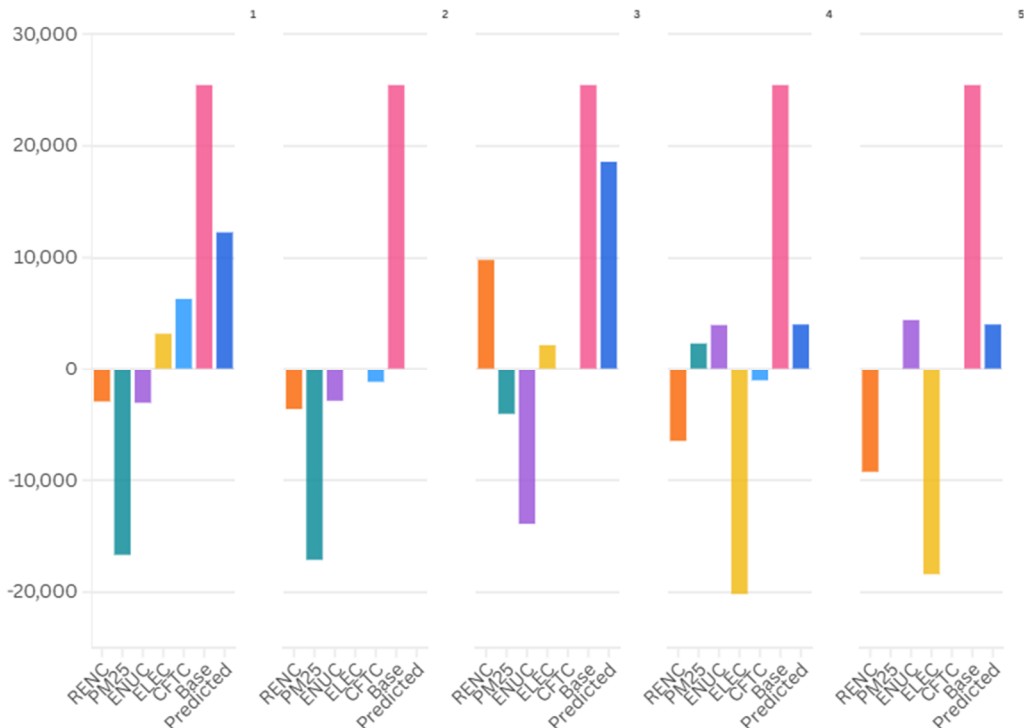

**Figure 7.** Additive feature contributions to building $CO_2$ emissions predictions (test cases).

Additive decompositions for k-NN predictions of building-related $CO_2$ emissions (BCE) provide contributions for clean fuels (CFTC), electricity access (ELEC), energy use per capita (ENUC), $PM_{2.5}$, and renewables (RENC) compared to a base level of 25.434. For Case 1 (BCE = 12.257), maximum reduction (−16.664) was brought on by low $PM_{2.5}$, in line with evidence for air quality and emissions associations [62]; ENUC and RENC reduced, while CFTC and ELEC increased emissions by higher accessibility [65]. For Case 2 (BCE = 0.500), a reduction was observed for $PM_{2.5}$ (−17.110) and ENUC/RENC, offset by CFTC and ELEC, which is typical for low-emitting systems. For Case 3 (BCE = 18.570), a notable reduction by ENUC (−13.891) was offset by a favorable effect by RENC (+9.796) and ELEC. For Cases 4–5 (BCE = 4.024), the high negative contributions for ELEC (−20.170; −18.398) indicate a preference for low-carbon generating alternatives, such as hydro. Generally, $PM_{2.5}$ and RENC are repeated movers for BCE reduction, while accessibility-related indicators (CFTC, ELEC) increase or decrease emissions according to system efficiency and mix. The exploratory findings for these jobs in ML reveal clean energy and air quality as the key factors for ESG policy [68]. See Figure 8.

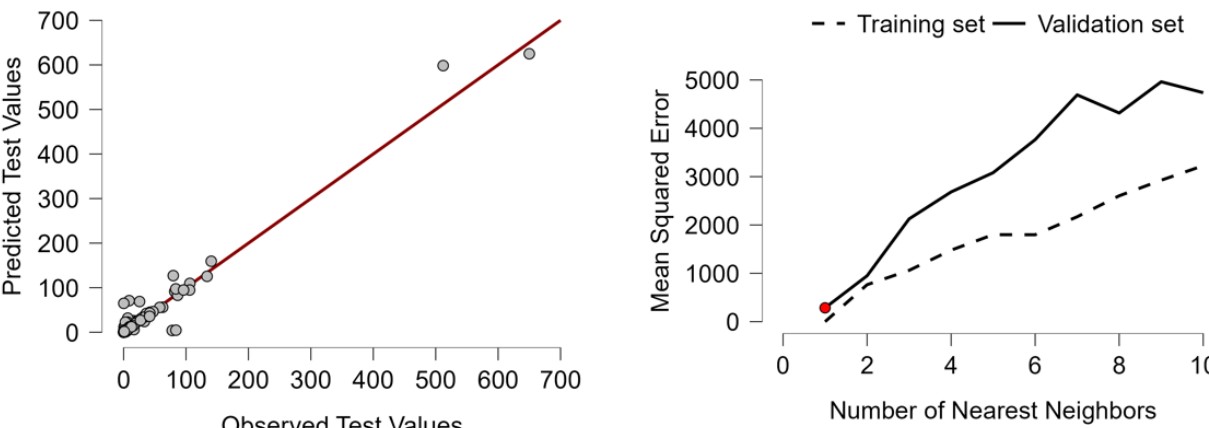

**Figure 8.** Additive feature contributions to building-related $CO_2$ emissions: k-NN model explanations across five cases.

## 5. Equity, Participation, and Emissions: Social Determinants of Building-Sector $CO_2$

This paper examines the linkage between building-sector $CO_2$ emissions (BCE) and social development indicators within the context of ESG, with a focus on the oft-overlooked "Social" pillar. Using panel econometrics and machine learning, we quantify the impact of distributive income inequality, labor force participation, educational parity, food output, and female political empowerment on the level of emissions. The results demonstrate robust relationships across specifications: distributive income inequality and higher food production are linked with higher emissions, while schooling and female political representation are linked with reductions. The relationships reveal that social equity, economic inclusion, and institutional representation are stimulants to sustainable environmental performance and not just moral prescriptions. The integration of social considerations into ESG assessment enhances predictive strength and highlights how inclusive policy directly impacts long-term decarbonization trajectories. The metric characteristics of the variables used to estimate the ECB component with respect to the S-Social variables within the ESG model are summarised in Appendix C. Diagnostic tests for autocorrelation and heteroscedasticity for the econometric estimates are provided in Appendix E.

### 5.1. Social Dimensions of Carbon Emissions: A Panel Data Approach to the Building Sector

This subsection applies panel data econometrics to estimate the degree to which social considerations drive building-sector $CO_2$ emissions (BCE). From a universe of 180 nations and over 1200 observations, drawn from a global dataset, the research tests associations between emissions and a group of five indicators: food production (FOOD), female parity in primary schooling (GPIE), income share for the bottom 20% (INC20), labor force participation (LABF), and women in legislatures (WPAR). Fixed- and random-effect regressions forecast the size and significance of impacts. Results show that country-level emission profiles are shaped by social equity, accessibility of schooling, and political representation, demonstrating the importance of social dimensions as a driver of ESG-related sustainability, rather than a side concern.

Specifically, we have estimated the following equation:

$$BCE_{it} = \alpha + \beta_1(FOOD)_{it} + \beta_2(GPIE)_{it} + \beta_3(INC20)_{it} + \beta_4(LABF)_{it} + \beta_5(WPAR)_{it}$$

where i = 180 and t = 2000–2020 (Table 4).

**Table 4.** Impact of social indicators on building-related $CO_2$ emissions: fixed-effect and random-effect panel regression results (2000–2020).

| | Fixed Effects, Using 1246 Observations Dependent Variable: BCE | | | Random Effects (GLS), Using 1246 Observations Dependent Variable: BCE | | |
|---|---|---|---|---|---|---|
| | Coefficient | Std. Error | t-Ratio | Coefficient | Std. Error | z |
| const | 46.5798 *** | 8.08670 | 5.760 | 36.0405 | 10.5431 | 3.418 |
| FOOD | 0.0863615 *** | 0.0236010 | 3.659 | 0.0868547 | 0.0233877 | 3.714 |
| GPIE | −0.00173333 ** | 0.000727140 | −2.384 | −0.00171224 | 0.000720050 | −2.378 |
| INC20 | 0.454688 ** | 0.215227 | 2.113 | 0.451706 | 0.213527 | 2.115 |
| LABF | −0.323534 *** | 0.114301 | −2.831 | −0.299359 | 0.111774 | −2.678 |
| WPAR | −0.169214 *** | 0.0504386 | −3.355 | −0.169434 | 0.0499015 | −3.395 |
| Statistics | Mean dependent var | | 31.32247 | Mean dependent var | | 31.32247 |
| | Sum squared resid | | 73,866 | Sum squared resid | | 9479 |
| | LSDV R-squared | | 0.992108 | Log-likelihood | | −7335.684 |
| | LSDV F(142, 1103) | | 976.5221 | Schwarz criterion | | 14,714 |
| | Log-likelihood | | −4311.281 | Rho | | 0.727702 |
| | Schwarz criterion | | 9641.822 | S.D. dependent var | | 86.70740 |
| | Rho | | 0.727702 | S.E. of regression | | 87.39678 |
| | S.D. dependent var | | 86.70740 | Akaike criterion | | 14,683 |
| | S.E. of regression | | 8.183426 | Hannan–Quinn | | 14,694 |
| | Within R-squared | | 0.032513 | Durbin–Watson | | 0.486805 |
| | *p*-value (F) | | 0.000000 | | | |
| | Akaike criterion | | 8908.562 | | | |
| | Hannan–Quinn | | 9184.263 | | | |
| | Durbin–Watson | | 0.486805 | | | |
| Tests | Joint test on named regressors— Test statistic: F(5, 1103) = 7.41335 with *p*-value = P(F(5, 1103) > 7.41335) = 7.51925 × 10⁻⁷ | | | 'Between' variance = 6423.31, 'Within' variance = 59.2827, mean theta = 0.953035, Joint test on named regressors—Asymptotic test statistic: chi-square(5) = 36.391 with *p*-value = 7.93233 × 10⁻⁷ | | |
| | Test for differing group intercepts— Null hypothesis: The groups have a common intercept Test statistic: F(137, 1103) = 1002.32 with *p*-value = P(F(137, 1103) > 1002.32) = 0 | | | Breusch–Pagan test—Null hypothesis: Variance of the unit-specific error = 0 Asymptotic test statistic: chi-square(1) = 5098.24, with *p*-value = 0 | | |
| | | | | Hausman test—Null hypothesis: GLS estimates are consistent, Asymptotic test statistic: chi-square(5) = 2.65474 with *p*-value = 0.753031 | | |

Note: *** $p < 0.01$, ** $p < 0.05$.

This paper positions building-sector $CO_2$ emissions in the ESG matrix with a focus on the overlooked "Social" pillar. Using panel econometrics with fixed- and random-effect specifications, we test four social indicators—gender parity in primary schooling (GPIE), income share for the poorest 20% (INC20), labor force participation (LABF), and women in parliament (WPAR)—for their roles in signaling educational equity, income distribution, labor inclusiveness, and political representation. Results reveal robust associations. Female schooling (GPIE) is associated with lower pollution, confirming the value of schooling in sustainability [69]. Political representation for women (WPAR) also reduces pollution, corroborating evidence that female empowerment produces cleaner building codes, energy standards, and solar subsidies [70,71]. Increasing participation in the labor force

(LABF) has a negative impact on residential energy use, suggesting that jobs lead to greater efficiency [72]. Increasing incomes for the poorest (INC20), however, drive up pollution, since new consumption creates more carbon-demanding use from illumination and domestic appliances [73]. Poverty reduction thus demands simultaneous investments in solar infrastructure and energy literacy. Generally, social equity, inclusiveness, and political representation are key drivers of the low-carbon transition, not adjuncts. Building on data for 180 countries over one generation, the study confirms the universality of these associations [74]. Low-carbon inclusivity demands the union of social justice and environmental defense.

### 5.2. Clustering Social Determinants of Emissions: An Evaluation of Algorithmic Performance

This research positions building-sector $CO_2$ emissions within the ESG framework, aiming to address the often-overlooked "Social" pillar. With fixed- and random-effect panel econometrics, we study the effect on $CO_2$ emissions from female parity in primary schooling (GPIE), income share for the poorest 20% (INC20), labor force participation (LABF), and women in parliament (WPAR) as proxies for educational equity, income distribution, labor inclusion, and political voice. Results show high associations. GPIE is significantly linked to negative $CO_2$ impacts, highlighting the importance of female education for sustainability [69]. Likewise, a negative effect emerges for WPAR, supporting the results that female political leadership is associated with greener building codes, energy codes, and green and renewable subsidies [70,71]. A correlation emerges between residential energy consumption and LABF, indicating a link to job efficiency [72]. INC20 appears to be significantly linked with $CO_2$ increases, as higher incomes for the poor result in a boost to fossil-intensive consumption [73]. The result supports the notion that the "Social" pillar is central, not peripheral, to ESG. Across over 180 countries spanning multiple generations, the research establishes the universality of these associations [74], thus supporting the notion that low-carbon transitions that are inclusive depend on aligning social equity and environmental protection. See Figure 9.

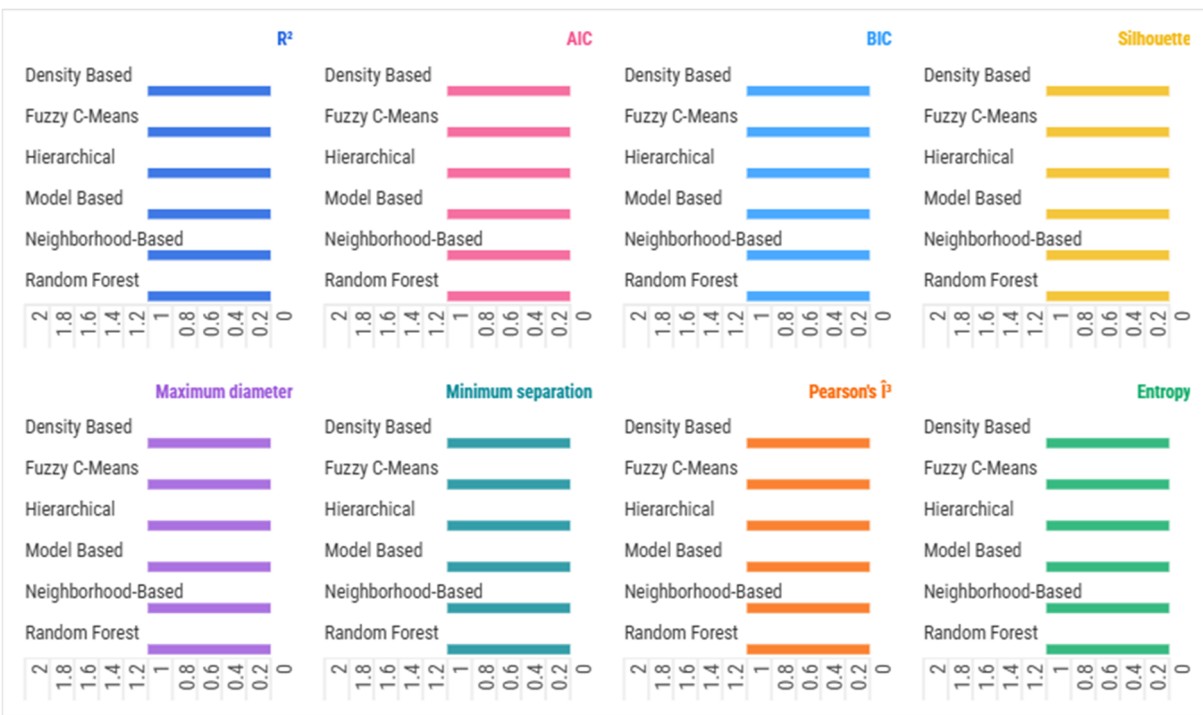

**Figure 9.** Comparative performance of clustering algorithms for ESG-based environmental data: evaluation across statistical and validity metrics.

We experimented with six clusters—density-based, Fuzzy C-means, hierarchical, model-based, neighborhood-based, and Random Forest—with ten validity indices ($R^2$, AIC, BIC, entropy, silhouette, maximum diameter, minimum separation, Pearson's $\gamma$, Dunn, and Calinski–Harabasz). They measure fit, error minimization, and quality structure in high-dimensional ESG data. Model-based and Random Forest excelled in $R^2$ (0.003), although their superiority over other techniques was slight, while density-based and neighborhood-based performed the worst (0.001). Random Forest performed best on AIC and BIC (0.016; 0.017), with model-based a close second (0.011; 0.012), indicating that explanatory strengths predominated over penalties for complexity [75]. Separation and silhouette measures were close to zero across techniques, with density-based being slightly higher (0.004), indicating a weak cluster definition. Dunn indices were unanimous zero, supporting reports that compaction in ESG clusterings was low [76]. Entropy measures varied on a small scale, with model-based (0.008) and Random Forest (0.007) best. Calinski–Harabasz was a flat 1.0, likely due to normalization. Overall, Random Forest was the best-balanced performer, excelling in AIC, BIC, and entropy, while tying with model-based on $R^2$. Fuzzy C-means, hierarchical, and neighborhood-based were inconsistent performers, with near-zero separation and validity [77].

Decoding Emissions and Equity: A Density-Based Clustering Approach to ESG Social Metrics

The research employs a density-based cluster model to analyze an ESG-related environmental dataset in the context of sectoral $CO_2$ emissions and certain societal development indicators. The cluster structure identifies four clusters and one outlier, covering both leading and niche configurations in the societal development and emissions process. The evaluation determines compactness and distinctiveness within the cluster, as well as within-cluster heterogeneity, using Silhouette indices and the respective sum of squares, along with their explanatory power. The model acknowledges an implicit consideration of the interrelation between environmental impact and societal dimension, thereby providing a more refined explanation for the diverse sustainability trajectories across nations. See Table 5.

**Table 5.** Cluster characteristics from density-based clustering of ESG and building $CO_2$ emissions data.

| Cluster | Noise Points | 1 | 2 | 3 | 4 |
|---|---|---|---|---|---|
| Size | 1 | 1188 | 21 | 15 | 21 |
| Explained proportion within-cluster heterogeneity | 0.000 | 0.956 | 0.006 | 0.002 | 0.036 |
| Within sum of squares | 0.000 | 2.288 | 13.632 | 5.502 | 84.952 |
| Silhouette score | 0.000 | 0.618 | 0.851 | 0.878 | 0.768 |

Note: The between sum of squares of the 4-cluster model is 5054.78. Note: The total sum of squares of the 4-cluster model is 7447.41.

The density-based strategy identified four large and one noise point group, based on the number of firms, captured heterogeneity, as accounted for by WSS, and silhouette statistics. The largest group (Cluster 1), comprising 1188 firms, exhibited 95.6% within-cluster heterogeneity; it had a high size-related WSS (2,288,542) and a moderate silhouette (0.618), characteristics typical of large heterogeneous groups [78]. The small lumps (Clusters 2–4), comprising 21, 15, and 21 firms, were better defined and higher on silhouettes (0.851, 0.878, 0.768) and low on WSS (13,632; 5502; 84,952). The overall model explained 67.9% of the variance (BSS = 5054.78, TSS = 7447.41), characteristic of strong unsupervised performance on non-convex shapes [78]. One noise point signals aptness for

ESG data, where such points typically flag data errors or outlier performers [79,80]. The output produces one large, heterogeneous cluster and three small, tight groups, suggesting the possibility of extracting both pervasive shape and weak ESG-related information. See Figure 10.

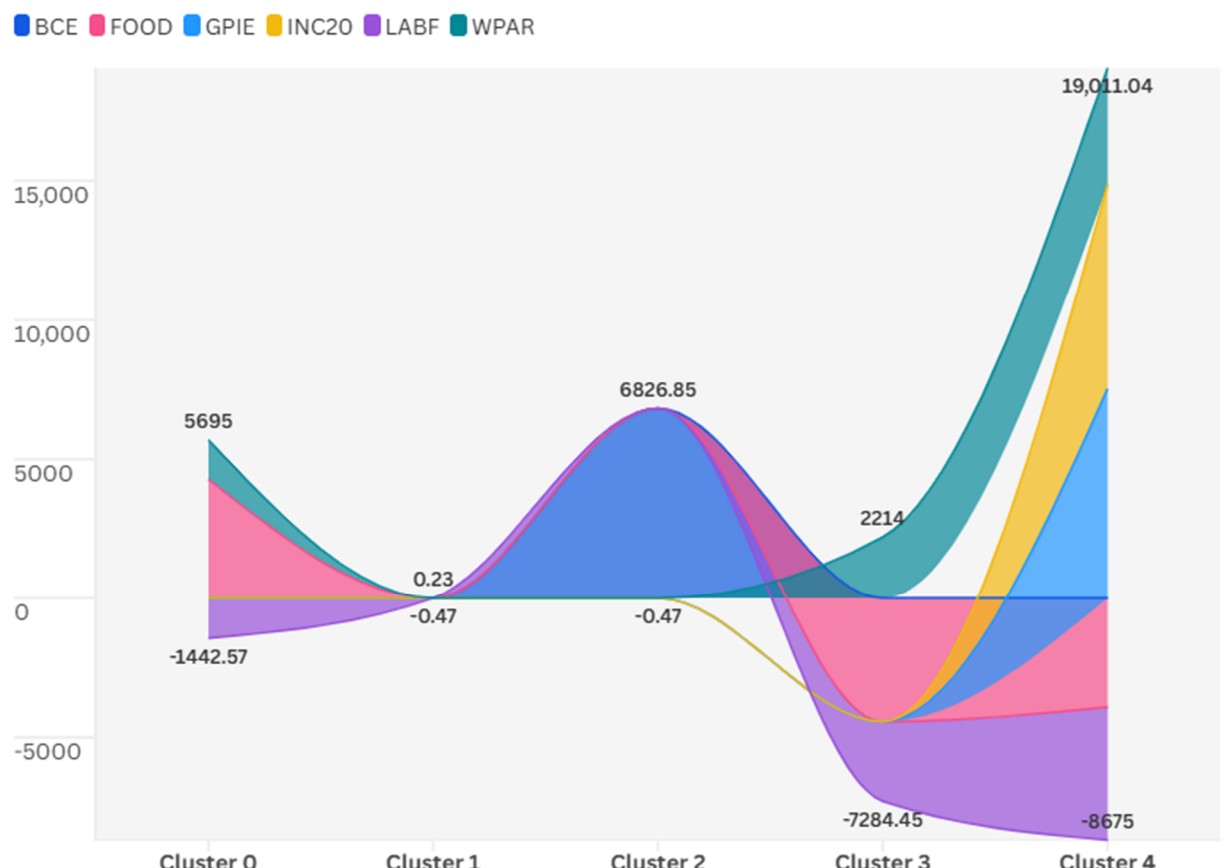

**Figure 10.** Cluster centroids from density-based model: ESG and building $CO_2$ emissions contextualized by social indicators.

The density-cluster model associates building-sector $CO_2$ emissions (BCE) and social indicators—food production (FOOD), gender parity in primary enrolment (GPIE), income share of the poorest 20% (INC20), labor force participation (LABF), and women in parliament (WPAR)—for the "S" component of ESG. Cluster 0 has a low BCE (−0.354), high food production (FOOD = 4.285); however, it exhibits low equity within genders and income levels ($\approx$ −0.1) and low labor force participation (LABF = −1.442). High rates of women in parliament (1.410) reflect middle-income countries where political representation has caught up with educational and labor advances [81]. Cluster 1 represents a moderately negative BCE (−0.117) with balanced rates across indicators, typical of transitional economies close to global means, served by a limited mix and efficient social infrastructure [82]. Cluster 2 includes high BCE (6.826) and weak social performance—negative GPIE (−0.129), INC20 (−0.236), and WPAR (−0.107), with just slightly favorable participation in the labor force (LABF = 0.548)—typifying industrial economies that are pollution-increasing and structurally inequitable and relying on fossil fuel [51]. See Figure 11.

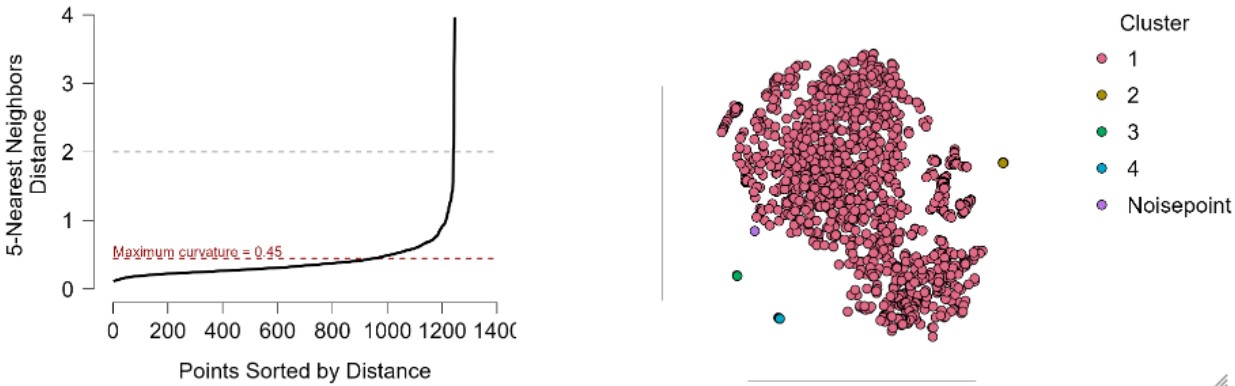

**Figure 11.** Clustering the social foundations of building emissions: ESG-based density model output.

Cluster 3 has a low BCE ($-0.337$) yet extremely heterogeneous social indicators, characterized by extremely low labor participation ($-4.439$; $-2.845$) and exceptionally high female parliamentary representation (2.214). This profile characterizes scenarios in which narrow labor market inclusion coexists with political equity advances and low emissions being a by-product more of poverty limits than on-regulation across emissions, consistent with evidence that poor economies are lower emitters yet development challenges [83,84]. Cluster 4 exhibits a near-neutral BCE (0.044) but high social performance variability, characterized by high GPIE (7.518), INC20 (7.328), and WPAR (4.165), which are accompanied by low LABF ($-4.759$) and a negative FOOD ($-3.916$). They are possible high-income scenarios with equitable allocations, yet structural economic limits. Past research has confirmed non-linear and, at times, negative associations between social indicators and emissions [51]. On their own, Cluster 2 complements high emissions and inequity, Clusters 0 and 3 exhibit low emissions and mixed social performance, and Cluster 4 illustrates selected equity–emissions decoupling. The finding corroborates multisided ESG, where environmental and social elements exhibit interlocking and context-aware relationships.

*5.3. Finding the Best Fit: A Comparative Evaluation of Regression Models on ESG Data*

We compared a broad range of regression methods on ESG data with conventional error measures (MSE, RMSE, MAE, MAPE, and $R^2$). The assessment covered classical techniques (linear regression, Decision Trees) and current-state methods (ensemble methods, neural networks). The findings reveal clear trade-offs: ensemble methods consistently achieved optimal predictive quality and generalization, while linear methods achieved the best interpretability and computational efficiency. The quality of neural networks was patchy, being superior on some occasions and unstable. The research presents the optimal regression methods for ESG prediction tasks, striking a balance between quality, efficiency, and interpretability. See Figure 12.

A comparison between error measures and $R^2$ values from regression models reveals sharp contrasts in the prediction and generalization capacities. Compared models are linear, regularized linear, tree-based, k-Nearest Neighbors (k-NNs), support vector machines (SVMs), neural networks, and Boosting. Multi-model benchmarking similar to that provided is common in cryptocurrency predictions [85], vehicle price prediction [86], and e-commerce satisfaction modeling [87]. The best $R^2$ was 1.0, a perfect explanation of variance. Decision Tree (0.677) and k-NN (0.655) were close, while linear and regularized linear regressions (0.014) and SVM (0.0) were poor. From error measures (MSE, RMSE, MAE), Random Forest stood out above the others, with 0s reflecting an extremely close fit that nonetheless raises some concern about overfitting in the absence of cross-validation. The best error was logged by SVM, corroborating its poor prediction performance. k-NN demonstrated consistent performance (RMSE = 0.285, MAE = 0.17), while Decision Tree

was decent (RMSE = 0.782, MAE = 0.741). The best MAPE was logged by Boosting (1.0), followed by Neural Networks (0.376), k-NN (0.453), and SVM scored 0.0 again. On MAPE, linear models underperformed across the board, with regularization yielding little improvement. Overall, Random Forest demonstrated the best prediction appropriateness and reliability and thus is best suited for the current regression task, while k-NN and Decision Tree are fair options where interpretability or computational efficiency is paramount.

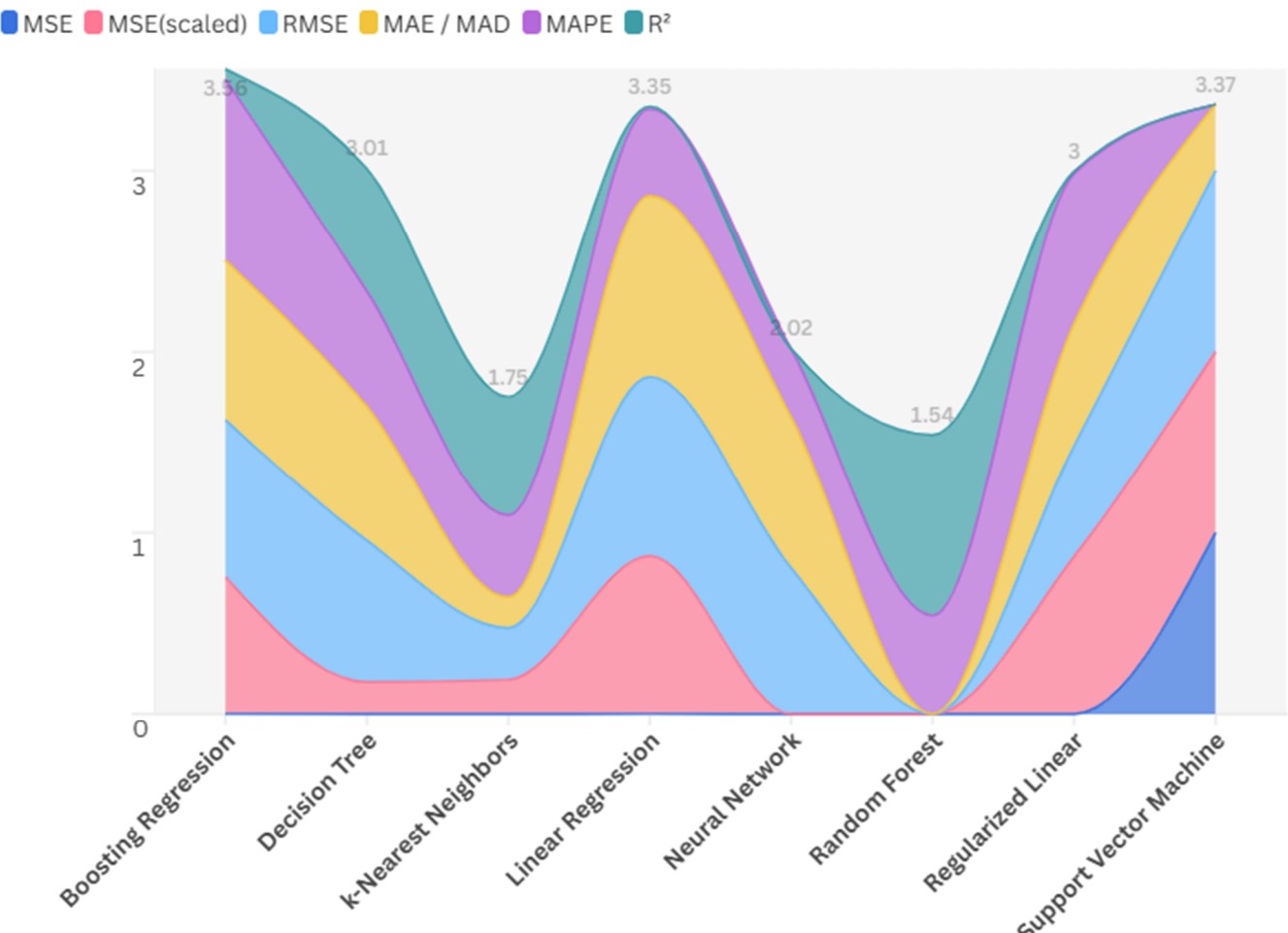

**Figure 12.** Performance metrics of regression models for ESG-informed emissions prediction.

Social Drivers of Emissions: Interpreting BCE Through Machine Learning and ESG Indicators

Random Forest feature-importance methods were employed to select the most important building-sector $CO_2$ emissions (BCE) determinants at the prediction level with additive accounts. Income distribution, labor force participation, and gender equity were found to be the most significant socio-economic indicators influencing emission outcomes. The indicators were overwhelmingly the most important across measures of importance, revealing a crucial underlying driving force for patterns in BCE. Case-level decompositions also revealed that changes in social equity and participation directly reposition amounts predicted for emissions. The findings provide evidence at a microlevel that environmental performance has social components in an ESG context, and they establish a linkage between equity, governance, and decarbonization strategies. See Figure 13.

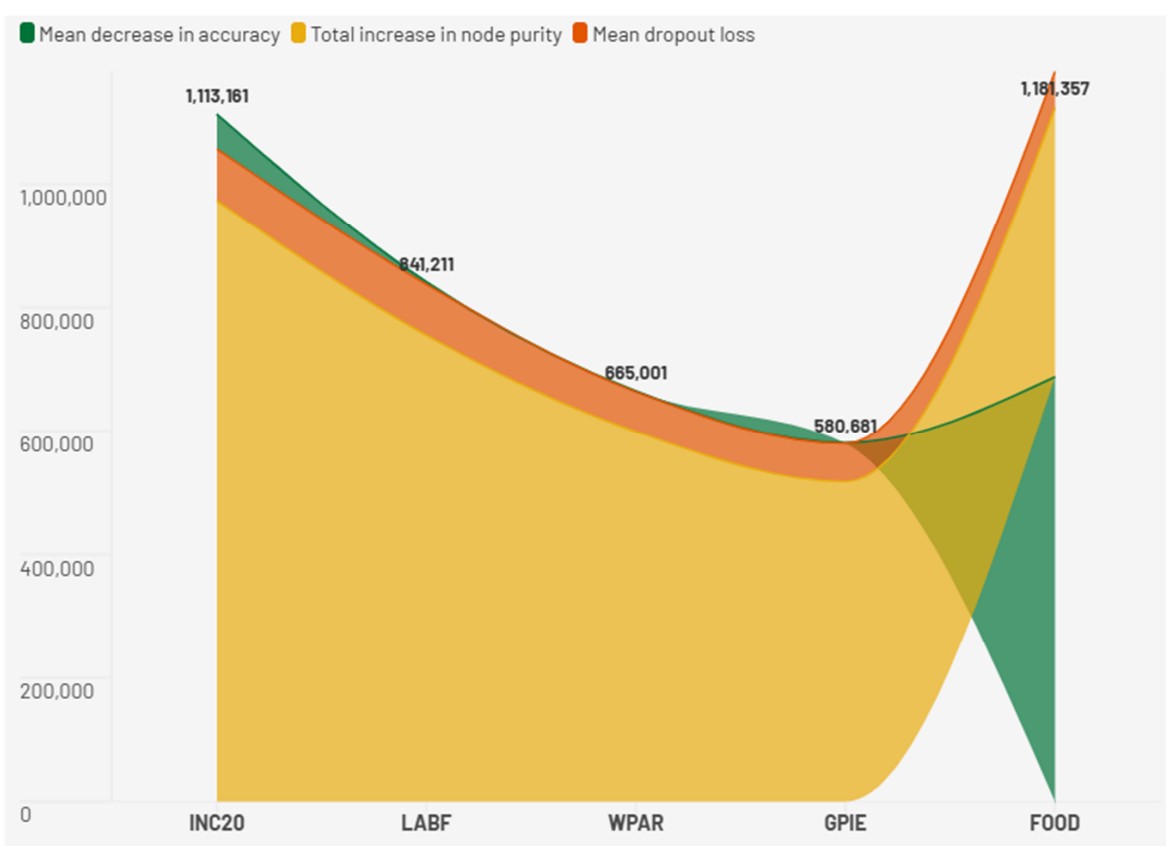

**Figure 13.** Feature importance analysis of socio-economic determinants in building $CO_2$ emissions via Random Forest model.

Random Forest importance indicators (MDA, TINP, MDL) are significant indicators of socio-economic push factors influencing building-sector $CO_2$ emissions (BCE). The income share for the bottom 20% (INC20) has the greatest predictivity in each measure (MDA = 56.464, TINP = 973.192, MDL = 83.505), suggesting income distribution—the bottom quintile in particular—as crucial for BCE and likely a consequence of unequal accessibility to energy efficiency in buildings, renewable energy, and support infrastructure. The labor force participation (LABF) variable comes in a close second. Although with a moderate level of MDA (4.113), high TINP (755.169) and MDL (81.929) support high predictivity. High participation rates are associated with urban, energy-intensive building use, while low participation rates indicate structural fragility and inefficient energy demand. See Figure 14.

Results from the Random Forest model indicate that social indicators are significant predictors of building-sector $CO_2$ emissions. Female parliamentary representation (WPAR) becomes significant (MDA = 1.018, Node Purity = 599.087, Dropout Loss = 64.896), consistent with its correlation with progressive building standards and energy-efficacy policy. Gender parity in education (GPIE) also emerges strongly (MDA = 1.209, Node Purity = 517.854, Dropout Loss = 61.618), indicating that knowledge and competence are key triggers for low-emissions building. The Food Production Index (FOOD) has a smaller yet complementary effect (Node Purity = 434.772, Dropout Loss = 58.385), which connects rural land use and agro-industrial energy networks. The findings align with studies that identify socio-economic and structural variables as underlying factors in emitters' trajectories [88–90]. Among the myriad variables tested, the income share of the poorest quintile (INC20) emerges as the best predictor, indicating that inclusiveness and equity are at the heart of ESG-driven sustainability. See Figure 15.

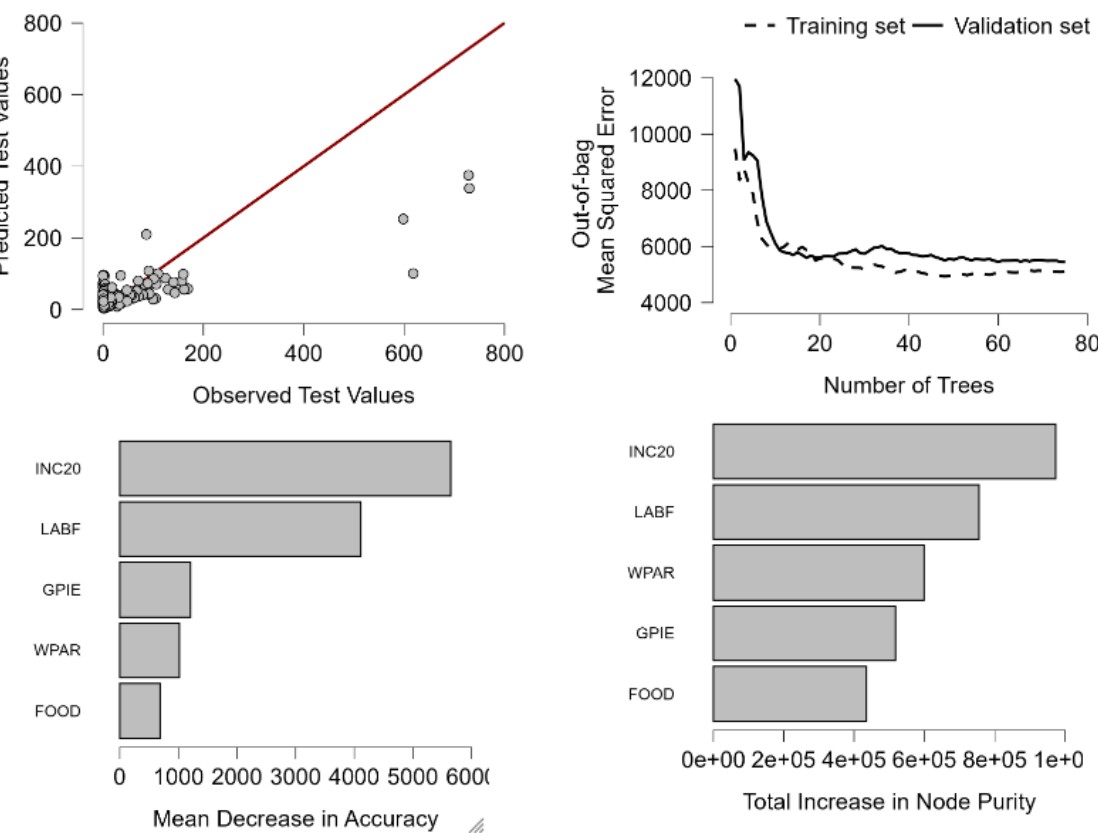

**Figure 14.** Relative importance of socio-economic indicators in predicting building $CO_2$ emissions using Random Forest.

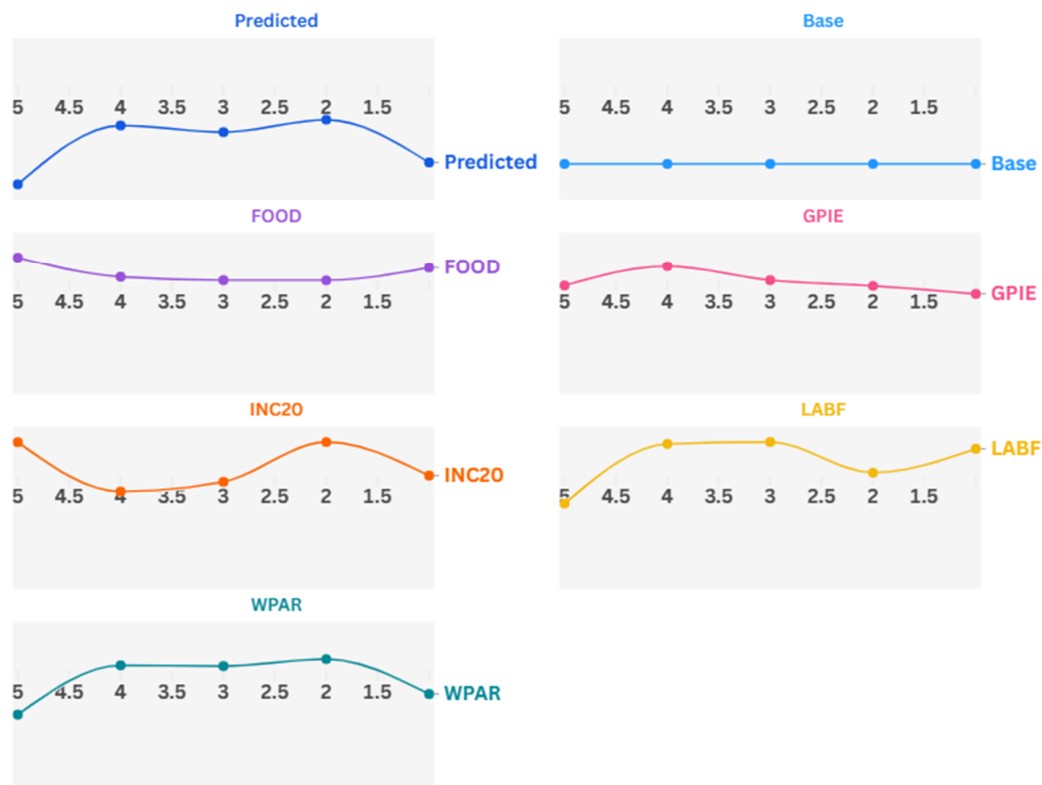

**Figure 15.** Additive feature contributions to predicted building $CO_2$ emissions: social indicators and Random Forest interpretability. Note: Displayed values represent feature contributions to the predicted value without features (column 'Base') for the test set.

Additive model decompositions reveal the respective contributions of social indicators to projected building-related $CO_2$ emissions (BCE) relative to a base level of 35.711 Mt $CO_2$. Case 1 (35.045) experiences the greatest negative effect (−12.268) for labor force participation (LABF), balanced in part by women in parliament (WPAR, +10.702) and parity in children's educational attainment (GPIE, +6.325); low-infrastructure agrarian settings are experienced for food production (FOOD, −5.862). Case 2 drops to 15.396 on the strength of a sharp loss for the income share for the poorest 20% (INC20, −15.411), and marginal losses for WPAR and LABF. Case 3 (21.101) again experiences a sharp negative effect for LABF (−15.392), while INC20 becomes positive (+2.898), reflecting energy growth due to income equalization. Case 4 (18.029) experiences losses for both LABF (−14.523) and parity in girls' and boys' educational attainment (−6.516) against a small gain for INC20 (+7.341). Case 5 jumps to 45.117, as WPAR (+20.099) and LABF (+12.822) propel high-income and high-energy systems, balanced in part by negative INC20 (−15.323) and negative food production (−10.631). Altogether, LABF exerts a constant negative effect on emissions, while WPAR and GPIE act contextually, and INC20 alternates between these two effects.

## 6. Governance and Carbon: Unpacking the Institutional Drivers of Building-Sector Emissions

This part assesses how governance elements impact building-sector $CO_2$ emissions (BCE), which is highlighted in the "G" pillar of ESG. The fixed- and random-effect regressions are conducted on panel data ($n$ = 982) to ascertain if government effectiveness (GOVT), expenditure on education (EDUE), political stability (STAB), rule of law (LAWR), R&D expenditure (RNDG), hospital expenditure (HOSP), and scientific productivity (SCIE) have significant impacts on BCE. The results reveal complex interplays. Elevated governance capacity often coexists with higher emissions. The latter is a product reflecting infrastructure-push development. Auxiliary clustering (see Section 6.2) categorizes countries on governance and emission patterns. The analysis highlights the various ways in which governance impacts environmental performance in the built environment. A metric summary of the G-Governance component variables within the ESG model used to estimate the value of BCE is shown in Appendix D. Diagnostic tests for autocorrelation and heteroscedasticity for the econometric estimates are provided in Appendix E.

*6.1. Governance and the Carbon Cost of Development: A Panel Analysis of Building Emissions*

This section examines how considerations regarding governance influence building-sector $CO_2$ emissions (BCE) in relation to the "G" pillar of ESG. On a 982-observation panel, fixed- and random-effects regressions examine government effectiveness, political stability, the rule of law, educational and research and development (R&D) expenditure, hospital infrastructure, and science output. The results present multifaceted and sometimes contradictory associations: higher governance capacity is often accompanied by higher emissions, providing evidence for development being infrastructure-driven. The study provides empirical evidence for a linkage between governance quality and environmental performance in the built environment, as well as for the interdependence between institutional strength and emission pathways across countries.

We have estimated the following equation:

$$BP = \alpha + \beta_1(GOVT)_{it} + \beta_2(EDUE)_{it} + \beta_3(STAB)_{it} + \beta_4(RNDG)_{it} + \beta_5(LAWR)_{it} + \beta_6(HOSP)_{it} + \beta_7(SCIE)_{it}$$

where i = 180 and t = 2000–2020 (Table 6).

Table 6. Panel regression results: governance indicators and building-sector $CO_2$ emissions (BCE).

| | Fixed Effects, Using 982 Observations Dependent Variable: BCE | | | Random Effects (GLS), Using 982 Observations Dependent Variable: BCE | | |
|---|---|---|---|---|---|---|
| | Coefficient | Std. Error | t-Ratio | Coefficient | Std. Error | z |
| const | 14.7556 ** | 6.12013 | 2.411 | 4.37823 | 9.08037 | 0.4822 |
| GOVT | 12.7921 *** | 2.42217 | 5.281 | 11.2943 *** | 2.15647 | 5.237 |
| EDUE | 0.450092 * | 0.245066 | 1.837 | 0.404457 * | 0.237995 | 1.699 |
| STAB | −3.19087 *** | 1.16516 | −2.739 | −2.29649 ** | 0.964108 | −2.382 |
| RNDG | −4.11812 ** | 1.71293 | −2.404 | −4.25068 ** | 1.65044 | −2.575 |
| LAWR | −4.29609 ** | 2.00928 | −2.138 | −1.36151 * | 0.755280 | −1.803 |
| HOSP | 3.80015 *** | 0.612354 | 6.206 | 3.75312 *** | 0.589134 | 6.371 |
| SCIE | 0.000433491 *** | $2.56973 \times 10^{-5}$ | 16.87 | 0.000462643 *** | $2.50122 \times 10^{-5}$ | 18.50 |
| Statistics | Mean dependent var | | 44.83643 | Mean dependent var | | 44.83643 |
| | Sum squared resid | | 88,339.25 | Sum squared resid | | 6,648,511 |
| | LSDV R-squared | | 0.992464 | Log-likelihood | | −5724.171 |
| | LSDV F(108, 873) | | 1064.614 | Schwarz criterion | | 11,503.46 |
| | Log-likelihood | | −3602.578 | Rho | | 0.766045 |
| | Schwarz criterion | | 7956.121 | S.D. dependent var | | 109.3165 |
| | Rho | | 0.766045 | S.E. of regression | | 82.57715 |
| | S.D. dependent var | | 109.3165 | Akaike criterion | | 11,464.34 |
| | S.E. of regression | | 10.05935 | Hannan–Quinn | | 11,479.22 |
| | Within R-squared | | 0.299880 | Durbin–Watson | | 0.464047 |
| | *p*-value (F) | | 0.000000 | | | |
| | Akaike criterion | | 7423.156 | | | |
| | Hannan–Quinn | | 7625.898 | | | |
| | Durbin–Watson | | 0.464047 | | | |
| Tests | Joint test on named regressors- Test statistic: F(7, 873) = 53.4184 with *p*-value = P(F(7, 873) > 53.4184) = $1.58042 \times 10^{-63}$ | | | 'Between' variance = 5929.08 'Within' variance = 89.9585 Mean theta = 0.942699 Joint test on named regressors- Asymptotic test statistic: chi-square(7) = 437.883 with *p*-value = $1.77433 \times 10^{-90}$ | | |
| | Test for differing group intercepts- Null hypothesis: The groups have a common intercept Test statistic: F(101, 873) = 78.0485 with *p*-value = P(F(101, 873) > 78.0485) = 0 | | | Breusch–Pagan test- Null hypothesis: Variance of the unit-specific error = 0 Asymptotic test statistic: chi-square(1) = 2207.29 with *p*-value = 0 | | |
| | | | | Hausman test- Null hypothesis: GLS estimates are consistent Asymptotic test statistic: chi-square(7) = 73.1884 with *p*-value = $3.34305 \times 10^{-13}$ | | |

Note: *** $p < 0.01$, ** $p < 0.05$, * $p < 0.10$.

The present study distinguishes the governance dimension of ESG ("G") as a predictor for building-sector $CO_2$ emissions (BCE), including residential, commercial, and other building-related emissions (IPCC, 2006; AR5). From a global panel for

982 observations, fixed-effect (LSDV) and random-effect (GLS) specifications are estimated, with BCE as the dependent variable and governance quality proxied by government effectiveness (GOVT), political stability (STAB), rule of law (LAWR), expenditure on education (EDUE), R&D expenditure (RNDG), hospital infrastructure (HOSP), and scientific output (SCIE). The results show that GOVT enters significantly and positively for emissions (12.79 FE; 11.29 RE, $p < 0.01$), reflecting convergence between effective governments and energy-prolific economies with high incomes [91]. EDUE enters significantly and positively ($p < 0.10$), reflecting a growing demand for infrastructure to support human capital development phases. On the contrary, political stability reduces emissions ($-3.19$ FE; $-2.30$ RE), enabling the effective enforcement of regulations [92]. R&D expenditure ($-4.12$ FE; $-4.25$ RE) and LAWR ($-4.30$ FE; $-1.36$ RE) lower emissions, corroborating the importance of investing in science and legal capacity for building efficiency and ESG conformity. HOSP enters significantly and positively ($\sim$3.80, highly significant), reasserting that healthcare has a high carbon intensity and that mitigation hinges on efficiency upgrades [93,94]. SCIE enters significantly and positively but non-significantly, reflecting that research systems are both high-energy intensity providers and innovation facilitators. The fixed-effect specifications dominate ($R^2 = 0.992$; AIC/BIC superior; Hausman $\chi^2 = 73.19$, $p < 0.01$), supporting unobserved heterogeneity. Overall, governance is a key predictor of BCE, but its impact hinges on context: while better institutions, stability, and R&D lower emissions, governance intensity in high-income economies comes alongside simultaneous growth driven by infrastructure and supporting higher carbon output [95]. Aligning governance quality with focus-tuned policy and conformity mechanisms is thus crucial to align institutional capability for sustainable decarbonization.

### 6.2. Governance and Emissions: Clustering Insights from Neighborhood-Based Algorithms

As a supplement to regression analysis, we employed clustering to identify latent groupings among nations based on governance attributes and building-sector $CO_2$ emissions (BCE). Performance measures included $R^2$, AIC/BIC, silhouette statistics, and entropy, as well as robust tests for statistical fit, compaction, and interpretability. Neighborhood-based clustering was optimal, achieving perfect explanatory power ($R^2 = 1.0$) with superior cohesion and separation compared to other methods. The outcome provides a rigorous template to unveil how diverse governance structures are plotted against profiles of emissions, with a sounder analytics basis for ESG-driven diagnostics. See Figure 16.

Neighborhood-based clustering was the best performer among the criteria. Performance was strong and consistent across quality indices. The algorithm achieved an $R^2$ value of 1.0, accounting for 100% of the variance, a very stringent test for model fit. Calinski–Harabasz index was 1.0, supporting tight and well-separated clusters. The AIC and BIC were not at a minimum, reflecting higher complexity, yet better practical validity indices did exist. The Silhouette measure (0.449) registered strong cohesion and sharp separation between groups. Entropy was high (0.962), reflecting structural order. Minimum separation and Pearson's $\gamma$ were strong, yet not the best. Balanced strength across both statistical and structural indices confirms Neighborhood-based clustering as the best and most comprehensible approach. Neighborhood-based clustering is the best solution when explanatory power and cluster quality are key.

Mapping Governance-Emission Profiles: Insights from Neighborhood-Based Clustering

To further elucidate the relationship between governance traits and structure-associated $CO_2$ emissions (BCE), the following is an exploration of the resulting neighborhood-based clustering algorithm, deemed optimal based on comparisons with preceding models. By clustering nations based on eight governance and infrastructural

indicators (including BCE, government effectiveness, education, research and development (R&D), and institutional quality), such an exploration identifies distinct profiles of countries. By identifying divergent relationships between governance and emissions within these clusters, it yields a profound observation regarding the institutional qualities that converge towards environmental ends amidst diverse development contexts. See Figure 17.

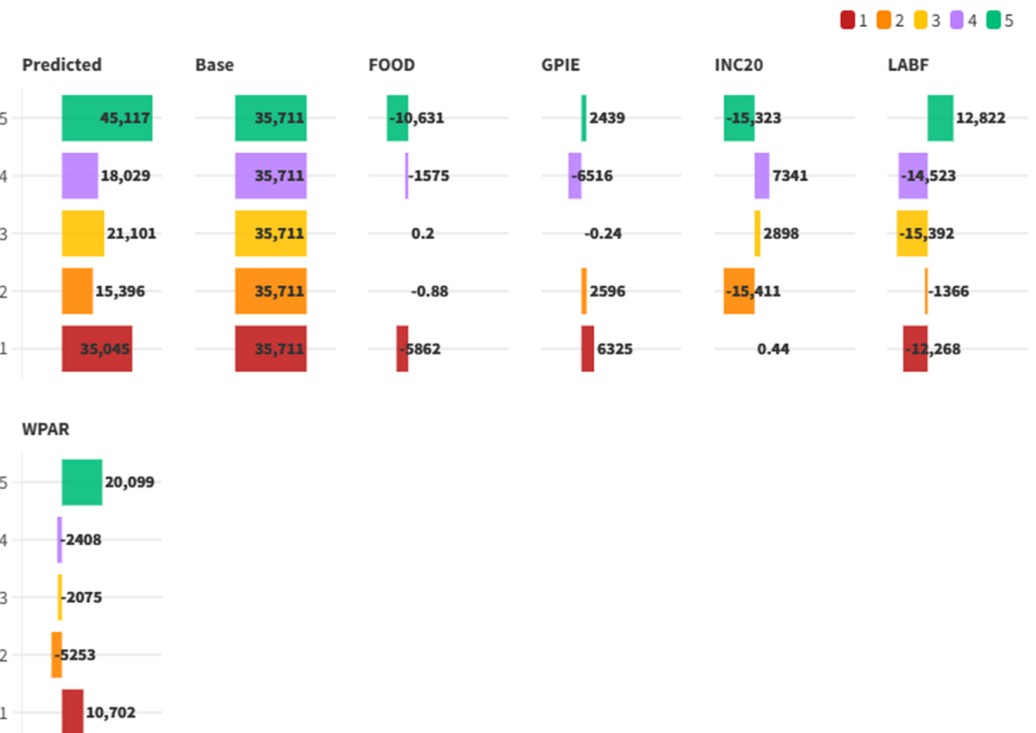

**Figure 16.** Comparison of clustering algorithms on governance indicators and building $CO_2$ emissions (BCE).

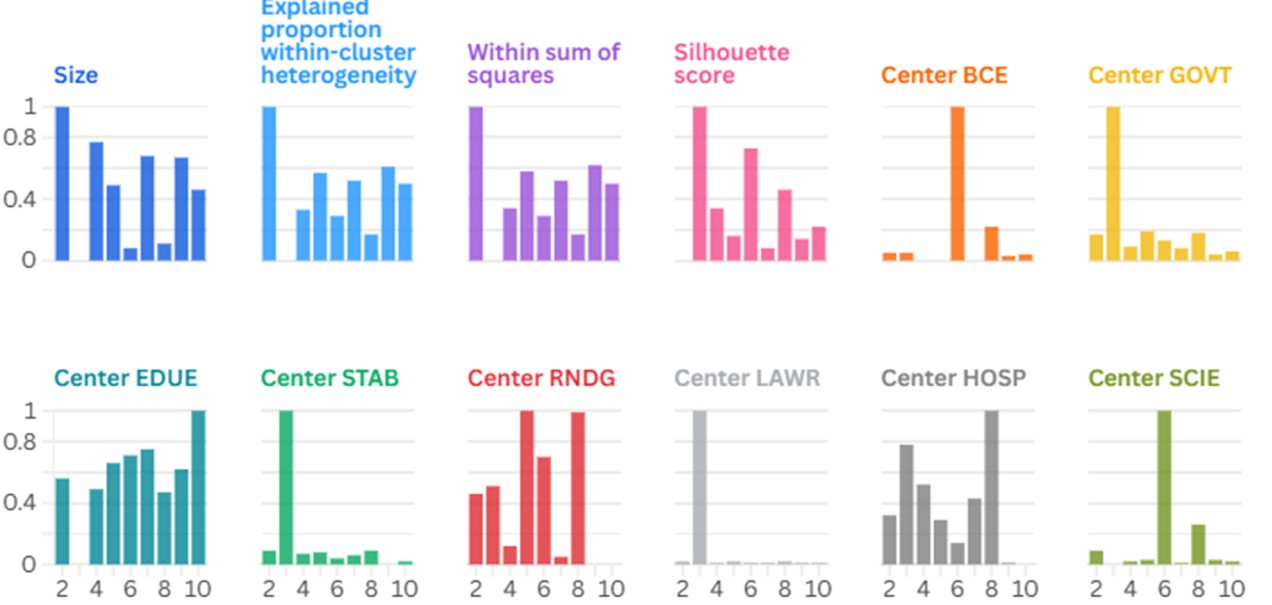

**Figure 17.** Country clusters by governance profiles and building $CO_2$ emissions: results from neighborhood-based clustering. Normalized data. The *y*-axis shows the normalized values between 0 and 1. The *x*-axis shows the number of clusters.

Countries cluster based on eight indicators: building-related $CO_2$ emissions, government effectiveness, education, political stability, R&D, rule of law, hospital density, and scientific output. Cluster 3 has high cohesion (silhouette = 0.911). Governance, stability, and law are high, while education and science are low, indicating weak knowledge-investing regimes. Cluster 6 has the highest emissions and is closely linked to scientific production and R&D. Healthcare is relatively weak. They are industrial states that trade innovation for environmental cost. Cluster 2 (204 countries) clusters around the global means. Low silhouette means an undifferentiated group. Cluster 7 has an emphasis on education, low on R&D. Emerging nations are characteristic of this profile. Cluster 10 invests heavily in education but falls short in health and R&D, thereby limiting development spillovers. Clusters 3 and 6 are distinct: knowledge-constrained states that are stable and high-carbon trade-offs that are stuck in knowledge-intensive states. Such profiles reinforce evidence that structural forces are driving emissions [96,97]. Diversified profiles inform bespoke $CO_2$ mitigation, according to policy scenarios supplied by [98].

*6.3. Predicting Emissions with Precision: Machine Learning Models for Governance and BCE*

The econometric regressions are augmented with machine-learning regressions. The regressions are K-Nearest neighbors, Decision Trees, Random Forests, and Boosting. The performance is calculated as MSE, RMSE, MAE, MAPE, and $R^2$. The results establish the optimal performing algorithms for non-linear and complex relationships. The forecast precision validates governance variables as significant predictors for building-sector $CO_2$ emissions (BCE). The exercise establishes a data-driven benchmark for tracking environmental outcomes that are shaped by governance. See Figure 18.

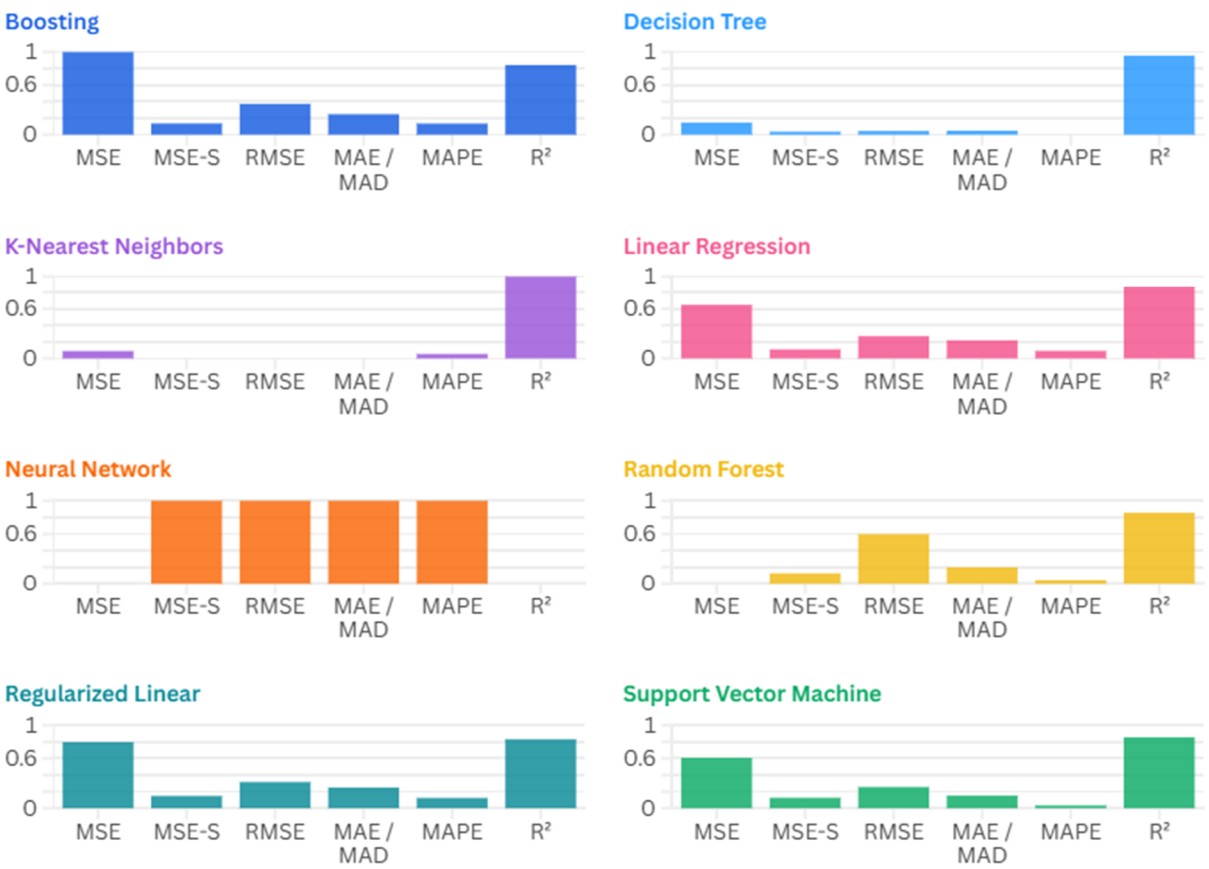

**Figure 18.** Regression model performance on governance-based prediction of building $CO_2$ emissions (BCE).

Evaluation confirms KNN as the best predictor. Errors are eradicated (MSE, RMSE, MAE = 0; MAPE = 0.054). $R^2$ is 1.0 for ideal predictions for building-sector $CO_2$ emissions (BCE). Decision Trees are also robust ($R^2$ = 0.957). Boosting and Random Forest are similarly close ($R^2$ = 0.843, 0.857). Linear and regularized forms perform poorly. Neural networks are unsuccessful, with an $R^2$ value of 0, indicating a failure in proper generalization. KNN's strength lies in its non-parametric flexibility, yet it is vulnerable to overfitting without validation. Past research has confirmed both the superiority and limitations in scale-extensive tasks [85,99,100]. Despite the computational cost, KNN is the most accurate and reliable method under the research's configuration. The hyperparameter optimization settings for the KNN machine learning regression algorithm are given in Appendix F.

What Drives Emissions? Feature Importance of Governance and Knowledge Indicators

This section explores the predictive influence of governance-related variables on building-sector $CO_2$ emissions (BCE) using feature importance metrics and additive model explanations. Feature importance, measured through mean dropout loss across 50 permutations, highlights the central role of scientific output (SCIE), healthcare infrastructure (HOSP), and education and R&D investment in driving model accuracy. See Figure 19.

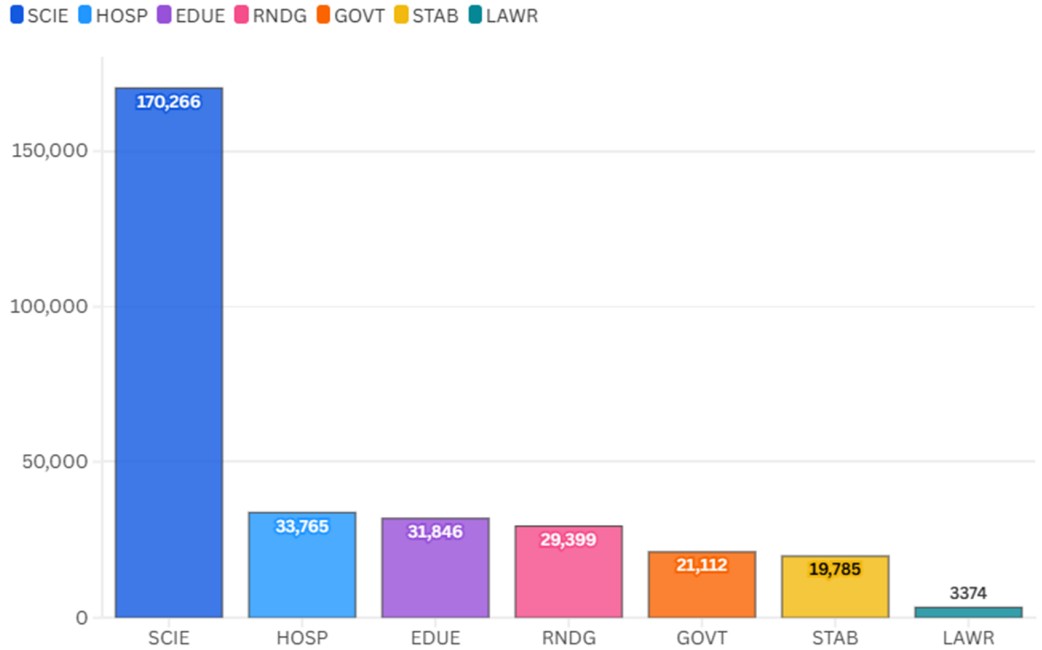

**Figure 19.** Governance feature importance in predicting building $CO_2$ emissions: dropout loss-based ranking.

RMSE (50 permutations) drop-out loss ranks building-sector $CO_2$ emissions predictors across fields of science, socio-economics, and governance. Most strongly ranked is scientific output (SCIE, 170.266), citing roles for knowledge intensity and research infrastructures, in line with [101]. Conflict between hospital and system adequacy ranks two (33.765). Education expenditure (EDUE, 31.846) and R&D expenditure (RNDG, 29.399) again feature strongly for human and technological development, in line with [100]. The governance indicators are less contributory yet significant longitudinally: government efficiency (21.112) and political stability (19.785) are favorable for institutional persistence. Rule of law (3.374) has the lowest rank, demonstrating an indirect, intangible effect. The results validate knowledge production, healthcare, and R&D as lead drivers, and governance quality as complementary infrastructure. See Figure 20.

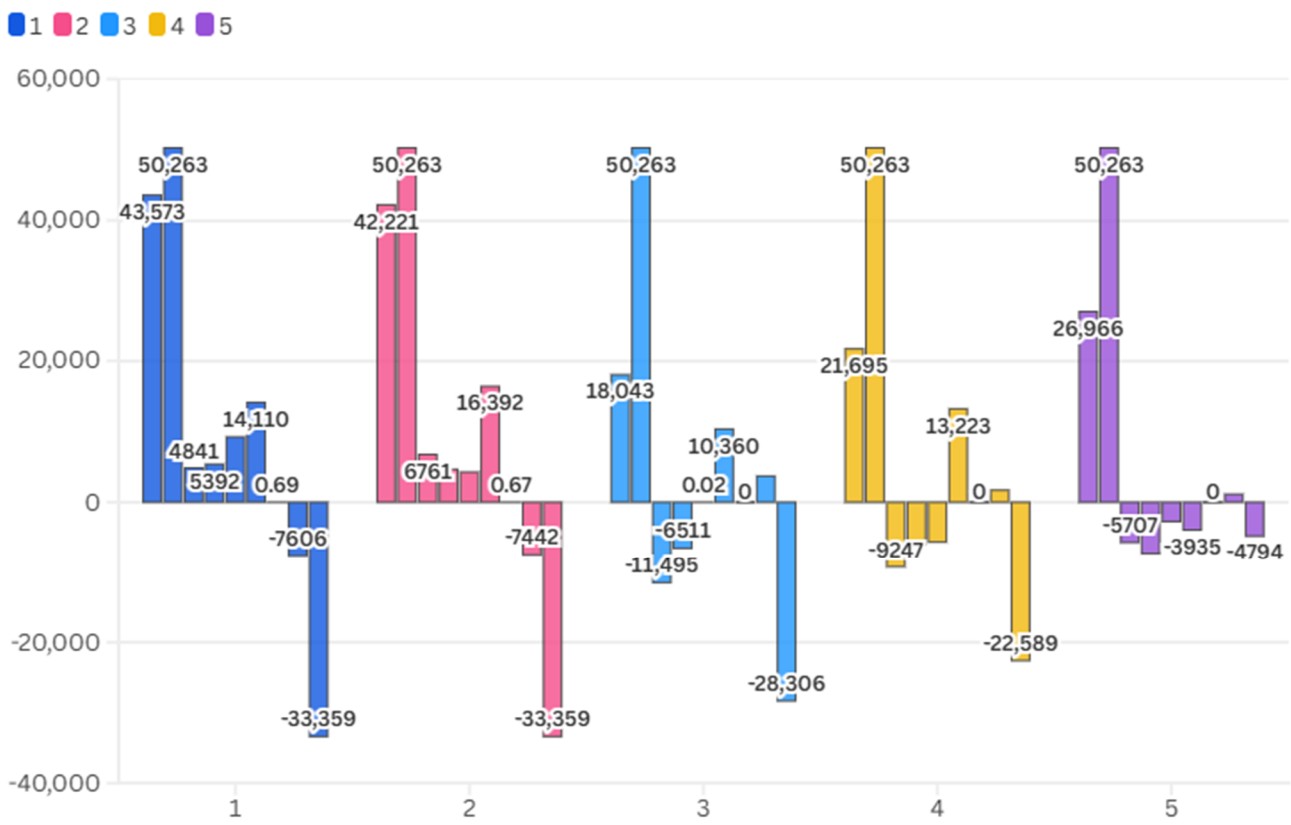

**Figure 20.** Additive explanations of governance feature contributions to BCE predictions.

Figure 20 reports additive contributions for five scenarios relative to a base of 50.263. SCIE, the quantity of scientific publications, is the most negative driver. Offsets range between −33.359 (Cases 1–2) and −4.794 (Case 5). There is a systematic negative correlation between building-related $CO_2$ emissions (BCE) and scientific output. Expanding scientific production is consistent with lower BCE. The mechanism accounts for technological innovation, efficiency in building design, and research-driven policy decisions. See Figure 21.

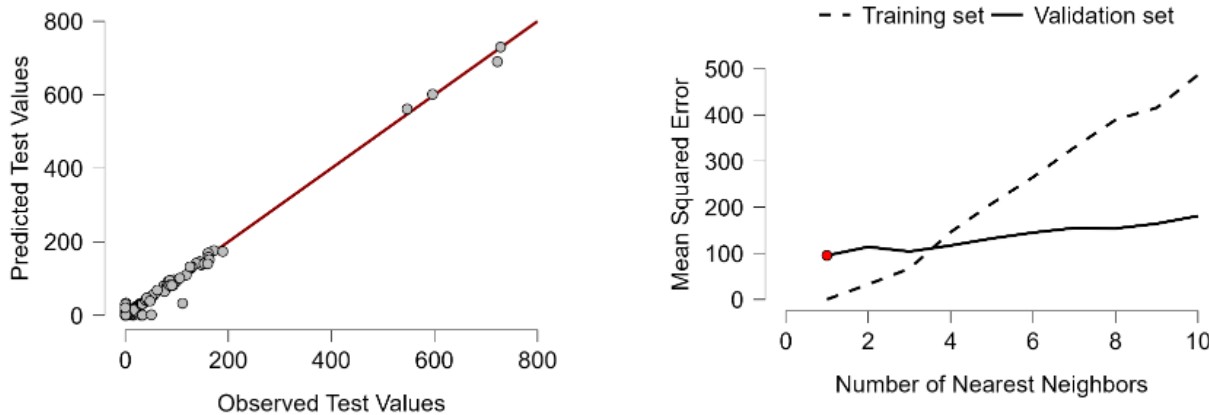

**Figure 21.** Additive effects of governance features on predicted building $CO_2$ emissions: case-level decomposition.

Government effectiveness (GOVT) and R&D expenditure (RNDG) boost emissions. Case 1 exhibits a +24 increase, and similar effects are observed in Cases 2 and 4. Education expenditure (EDUE) and political stability (STAB) exhibit erratic behavior. They are positive for Cases 1–2 and negative for Cases 3–5, which support the notion of non-linear, context-

dependent dynamics. Rule of law (LAWR) makes a moderate contribution, peaking in Case 2 at +16.392. Hospital beds (HOSP) add little explanatory power. Scientific production (SCIE) uniformly reduces emissions across scenarios, and it is a successful mitigation force. Overall, GOVT and RNDG push emissions higher, while SCIE balances them, exposing the structural dualism of development: institutional strength drives growth, and science reduces emissions.

## 7. Harnessing ESG Dimensions for Effective Building Sector Climate Action

Comparative research finds country clusters in building-sector $CO_2$ and ESG. Policies are diverse in context while requiring a technological foundation in building engineering [102]. Emission profiles are derived from envelope performance, material choice, and HVAC efficiency. Strategies must be context-aware while being technologically distinct. Decarbonization in Germany, Denmark, Finland, and Nordic Europe is a sophisticated endeavor, albeit limited by the possibilities of engineering. Institutionalized actors require air-tight envelopes in the retrofit programs, triple glazing, and digitalized energy monitoring. Carbon pricing on building elements has the potential to provide lifecycle neutrality. Governance and innovation flag-bearers must lead domestic decarbonization and invest in green transitions abroad [103–105]. Disseminations in modular architecture, passive architecture, and retrofitting have the potential to transform global practices. Emerging markets, such as India, Indonesia, and Brazil, are experiencing rapid urbanization. Policy priorities include green transportation, affordable housing, and procuring ESG-compliant products. Prefabrication leads to waste reduction. District solar or wind cooling deters HVAC emissions. Cooperation enables leapfrogging [106–108]. Codes with specifications for bamboo composites or recycled steel reinforce sustainability. Capacity-constrained states such as Nigeria, Kenya, and Pakistan require institution-building and ESG-conformant programs. Ventilation and compressed earth block pilot retrofits suppress emissions while building social acceptability. School and hospital investments build legitimacy. Global fora build cooperative retrofitting with local adaptability [109]. The Orient and East Asia, comprising China and Korea, lead in governance while experiencing equity gaps. Redistributive policy is required. Subsidized low-cost building retrofits are immediate. Green roof and high-performance envelopes are imminent. AI-generational HVACs, fresh materials, and adaptive urbanism must be propagated [110,111]. Indoor air quality requires continuous monitoring. High-income emitters such as the US and Arab states require strict regulation. Near-zero codes, carbon taxes, and thermal storage constitute key policy measures. Double-skin buildings and recoverable heat systems must be the foundation for high-rise portfolios under green leasing schemes. All contexts require ESG integrity, dependent on coherence. Environmental, governance, and social pillars must be brought into alignment. Clean energy uptake requires governance capacity and social equity. Fragmented ESG does not deliver. Indicators must inform interventions rather than merely serving symbolic alignment [112,113]. Engineering indicators, including insulation transmittance, rates of recovery, and embodied carbon, must be delivered alongside ESG reports [114,115]. Clustering evidence confirms bespoke strategies: OECD leaders take leapfrogging and transfer approaches; new economies leap ahead with modularity and construction; at-risk states fortify their institutions; East Asia balances equity and governance; high-carbon OECD states set codes and establish carbon markets. All else being equal, without engineering integration, ESG remains symbolic. Through interconnection, buildings can achieve net-zero targets and make significant progress on climate mitigation at scale.

## 8. Limitations

Research correlates $CO_2$ output from buildings with environmental, social, and governance (ESG) indicators across 180 countries. There is strong evidence, and some limitations. However, the analysis depends on national aggregates spanning 22 years. Urban–rural gaps and intra-city differences remain largely invisible, despite their significant importance for emission dynamics. Most sectoral and energy policies take place subnationally. The limited availability of local data restricts the transferability to municipal and urban administrations. Approaches are based on machine learning, econometrics, and clustering. The models are computationally tractable but constrained by crude ESG proxies, specifically end-use energy and air quality [116–118]. National-level coverage for governance and social ESG data is sparse [102,119], which restricts its use at smaller scales. Neural-network models did not perform well. The $R^2$ value was approximately zero, and errors were large, indicating shortcomings in the dataset and hyperparameters [120]. Noise, low sample boards, and lack of parameter tuning prevented generalizability. Generalizable AI is expected to reveal non-linear and threshold dynamics in follow-up research. Another point of consideration is that the data ends in 2021. Pandemic-driven transformations—hybrid work, office underuse, and plans for recovery are lacking [121]. Given these factors, future research demands near-real-time data, enriched collections, and novel reporting standards. Only thus are evolving trajectories of emissions detectable by ESG models.

## 9. Conclusions

$CO_2$ releases by the building sector were compared with ESG indicators across 180 countries. Econometrics, cluster analysis, and machine learning detected multifaceted, multidimensional drivers. Releases are shaped by technology, governance, science, and social investment. Institutions and societies are just as important as engineering. Scientific output was associated with $CO_2$ releases. Higher research intensity nations are better buffered, spread knowledge, innovative, and codified sound policy. Institutional quality and R&D investments reported mixed findings. Strong institutions were ubiquitous in low-emissions economies and densely infrastructured, high-demand economies of aging economies. Governance, innovation, and country-tailored sustainability policies need to be aligned for effective buffering. Social investments build viability for sustainable practice and environmental reform capacity. Equity-led policies are ethical and more. They facilitate direct low-carbon transitions. Education, healthcare, and inclusiveness build resilience and transformation. Integration across methodologies is this study's strength. Regression, feature importance, and cluster detection distinguish country profiles, institutional trajectories, and predictive dynamics beyond monomethod research. Decarbonizing buildings cannot rely solely on markets or technology. Success depends on governance, science, social investments, and ESG frameworks that value the interdependence between the environment, society, and institutions.

**Author Contributions:** Conceptualization, N.M., V.N., M.D.M. and A.L.; methodology, N.M., V.N., M.D.M. and A.L.; software, N.M., V.N., M.D.M. and A.L.; validation, N.M., V.N., M.D.M. and A.L.; formal analysis, N.M., V.N., M.D.M. and A.L.; investigation, N.M., V.N., M.D.M. and A.L.; resources, N.M., V.N., M.D.M. and A.L.; data curation, N.M., V.N., M.D.M. and A.L.; writing—original draft preparation, N.M., V.N., M.D.M. and A.L.; writing—review and editing, N.M., V.N., M.D.M. and A.L.; visualization, V.N. and M.D.M.; supervision, V.N. and M.D.M.; project administration, V.N., M.D.M. and A.L.; funding acquisition, V.N. and M.D.M. All authors have read and agreed to the published version of the manuscript.

**Funding:** The research was funded by "xTech NextHub: Competence Center for Innovative Solutions Development" through the Puglia Regional Call for Proposals for Exemption Aid 17 of 30/09/2014-

BURP 139 suppl. of 06/10/2014 and subsequent amendments—Title II Chapter 2 of the General Regulations "Notice for the Submission of Projects Promoted by Large Enterprises Pursuant to Article 17 of the Regulation". Project's grant number: VTOIFW0.

**Data Availability Statement:** The data presented in this study are openly available in World Bank Databank, https://databank.worldbank.org/.

**Conflicts of Interest:** Author Nicola Magaletti, Valeria Notarnicola and Mauro Di Molfetta were employed by the company LUM Enterprise s.r.l. The remaining authors declare that the research was conducted in the absence of any commercial or financial relationships that could be construed as a potential conflict of interest.

## Appendix A. Data Description

Table A1 compiles an exhaustive set of World Bank variables that constitute the foundation of the ESG framework used in the building sector. The environmental dimension encompasses building-related $CO_2$ emissions (BCE), energy use per unit (ENUC), access to electricity (ELEC), cleaner fuel (CFTC), $PM_{2.5}$ air pollution, and consumption of renewable energy (RENC), reflecting the balance between energy development and sustainability. The social dimension encompasses food production (FOOD), gender parity in education (GPIE), income share held by the bottom quintile (INC20), participation in the labor force (LABF), as well as parliament participation by women (WPAR), which reflect exclusion and equity. The governance dimension encompasses government effectiveness (GOVT), political stability (STAB), rule of law (LAWR), expenditure on research and development (RNDG), expenditure on education (EDUE), hospital beds (HOSP), and scientific output (SCIE). These, together, allow for a comprehensive view of sustainability (Table A1).

**Table A1.** We used the following variables acquired from the World Bank.

| Acronym | Variables | Definition |
|---|---|---|
| BCE | Carbon Dioxide ($CO_2$) Emissions From Building (energy) (Mt $CO_2$e) | Total annual carbon dioxide equivalent ($CO_2$e) emissions from energy use in the buildings sector, covering IPCC 2006 categories 1.A.4 (Residential and other) and 1.A.5 (Unspecified), converted to $CO_2$e using global warming potentials from the IPCC Fifth Assessment Report (AR5). Unit: Mt $CO_2$e per year. |
| CFTC | Access to Clean Fuels and Technologies for Cooking | Access to clean cooking fuel and technology estimates come from the WHO Global Household Energy Database with national representative household surveys as the sole data source (e.g., DHS, MICS, LSMS, WHS, national censuses). A multivariate hierarchical model—split by urban and rural—estimates fuel-type trends by grouping them as 'clean' (e.g., gas, electricity, alcohol) and 'polluting' (e.g., biomass, charcoal, coal, kerosene). There are estimates for 191 countries. High-income countries (by World Bank 2022 classification) have universal clean fuel access assumed. |
| ELEC | Access to Electricity | Reliable and secure electricity is essential for economic growth, poverty reduction, and human development. As countries decarbonize, dependence on clean, efficient power will grow. Electricity access enables basic services (lighting, refrigeration, appliances) and is a key indicator of energy poverty. Especially in lower-income countries, governments are prioritizing electrification through rural programs and national agencies. While vital for raising living standards, electricity generation can harm the environment—its impact depends on the energy sources used, with fossil fuels like coal being especially carbon-intensive. |

**Table A1.** *Cont.*

| Acronym | Variables | Definition |
|---|---|---|
| ENUC | Energy Use per Capita | Total energy consumption gauges final energy use after conversion into end-use fuels (e.g., electricity, processed oil). It encompasses energy from combustible renewables and waste—like biomass, biogas, and municipal waste. Biomass describes plant materials used as such or converted into fuel, heat, or power. Figures, as gathered by the IEA, use per capita estimates from the World Bank population. National non-OECD data are converted to IEA equivalence. Figures are imprecise and not completely comparable for countries because of limited data quality, particularly for waste and renewables. Energy values have been computed in terms of oil equivalents on the basis of 33% thermal conversion for nuclear and 100% for hydropower. |
| PM25 | $PM_{2.5}$ Pollution | Population-weighted exposure to ambient $PM_{2.5}$ refers to the average level of fine particulate matter ($PM_{2.5}$) pollution that a country's population is exposed to. $PM_{2.5}$ particles, with a diameter smaller than 2.5 microns, can penetrate deep into the lungs and pose serious health risks. This measure is calculated by weighting the annual average $PM_{2.5}$ concentrations by the population distribution across urban and rural areas. |
| RENC | Renewable Energy Consumption | The share of total final energy consumption derived from renewable sources, based on data from IEA, IRENA, UNSD, WHO, and the World Bank (Tracking SDG 7, 2023). |
| FOOD | Food Production Index | The Food Production Index reflects the output of edible crops that offer nutritional value. It excludes items like coffee and tea, which, despite being consumable, do not contribute meaningfully to nutrition. This metric emphasizes food sources that support dietary needs, aligning production data with human nutritional requirements rather than general edibility alone. |
| GPIE | Gender Parity in Enrollment | The Gender Parity Index (GPI) in primary education is calculated by dividing female gross enrollment by male gross enrollment. Data are collected by UNESCO from national education surveys and aligned with ISCED standards to ensure international comparability. The current methodology was adopted in 2011. Reference years reflect when the school year ends. A GPI below 1 indicates girls are disadvantaged; above 1 indicates boys are. Achieving gender parity enhances women's opportunities and contributes to broader social and economic development. |
| INC20 | Income Share Lowest 20% | The percentage share of income or consumption reflects the portion received by population subgroups, typically divided into deciles or quintiles. Due to rounding, quintile shares may not total exactly 100%. Data come from household surveys via national statistics agencies and World Bank departments, with high-income country data largely from the Luxembourg Income Study. These measures support the World Bank's goal of shared prosperity—focusing on income growth among the bottom 40%—and help assess inequality within and across countries. |
| LABF | Labor Force Participation | The labor force participation rate represents the share of the population aged 15 and older that is economically active, including all individuals engaged in the production of goods and services during a specific period. Data, sourced from the ILO's modeled estimates, highlight persistent gender disparities: women's labor force participation is generally lower than men's due to social, legal, and cultural norms. In low-income countries, women often work unpaid in family enterprises, while, in high-income nations, higher education has expanded their access to better employment opportunities, though inequalities persist. |

**Table A1.** *Cont.*

| Acronym | Variables | Definition |
|---|---|---|
| WPAR | Women in Parliament | Women in parliament refers to the percentage of seats held by women in a single or lower house of national parliaments. Although progress has been made, women remain significantly underrepresented in decision-making roles, especially in lower-income countries. Gender inequality in political participation limits women's influence on policy and national priorities. Equal representation is essential for inclusive governance and sustainable development. True democracy requires full participation of women, whose perspectives and leadership are vital for shaping equitable and effective public policies. |
| GOVT | Government Effectiveness | Government effectiveness: Estimated measures of perceptions of public service quality, civil service independence, policy formulation and implementation, and government credibility. Scores range from $-2.5$ to 2.5, based on a standard normal distribution. |
| EDUE | Gov. Expenditure on Education | General government expenditure on education, including current spending, capital outlays, and transfers, is measured as a percentage of GDP. It accounts for education funding from all government levels—local, regional, and central—and includes international transfers to the government. This indicator reflects the government's financial commitment to the education sector relative to the country's economic output. |
| STAB | Political Stability | Political stability and absence of violence/terrorism reflect perceptions of the risk of political unrest, government instability, and politically motivated violence or terrorism. Countries are ranked by percentile, from 0 (least stable) to 100 (most stable), allowing global comparison. Percentile ranks are adjusted over time to ensure consistency despite changes in the number of countries included in the Worldwide Governance Indicators (WGIs). |
| RNDG | R&D Expenditure | Gross domestic expenditures on research and development (R&D), measured as a percentage of GDP, represent a country's financial commitment to innovation and technological progress. This includes both capital and current spending across four key sectors: business enterprises, government institutions, higher education, and private non-profits. It encompasses all R&D activities—basic research, applied research, and experimental development—supporting economic and scientific advancement. |
| LAWR | Rule of Law | Rule of law reflects perceptions of how much confidence individuals and institutions have in societal rules, particularly regarding contract enforcement, property rights, police effectiveness, and judicial independence. It also considers the likelihood of crime and violence. Countries receive a score ranging from approximately $-2.5$ (weak rule of law) to 2.5 (strong), based on a standard normal distribution. |
| HOSP | Hospital Beds | Hospital beds refer to the total number of beds that are maintained, staffed, and immediately available for the admission of patients. These include inpatient beds in public and private hospitals, general and specialized institutions, and rehabilitation centers. The count typically covers beds used for both acute and chronic care, reflecting the overall healthcare system's capacity for treatment and recovery. |
| SCIE | Scientific Articles | Scientific and technical journal articles represent the total number of peer-reviewed publications in key research areas, including physics, biology, chemistry, mathematics, clinical medicine, biomedical research, engineering and technology, and earth and space sciences. These articles reflect ongoing advancements, innovation, and collaboration within the global scientific community, contributing to knowledge expansion and technological development across multiple disciplines and industries. |

## Appendix B. E-Environment

Descriptive statistics suggest high dispersion in all indicators, a very high building-sector $CO_2$ emission (BCE) range, echoing its very high range of (0–729,783 Mt $CO_2e$) and extreme skewness of 7251, as an estimator of high-emitting countries' tail effect. Clean fuel consumption (CFTC) and electricity consumption (ELEC) indicate a high median and mode of 100%, as would be expected in high-income nations with widespread electricity use. High standard deviations indicate widespread inequity worldwide. Per capita energy consumption (ENUC) and $PM_{2.5}$ air pollution exhibit high interquartile ranges and kurtosis, indicating the presence of outliers. Renewable energy consumption (RENC) remains uneven, reflecting the uneven nature of the global energy transition (Table A2).

**Table A2.** Descriptive statistics of building emissions (BCE) and associated energy-environmental indicators.

|  | CFTC | ELEC | BCE | ENUC | PM25 | RENC |
|---|---|---|---|---|---|---|
| Valid | 3916 | 3940 | 4140 | 2173 | 2180 | 3805 |
| Missing | 224 | 200 | 0 | 1967 | 1960 | 335 |
| Mode | 100,000 | 100,000 | 0.006 | 1720 | 17,869 | 0.000 |
| Median | 83,450 | 97,000 | 1098 | 1190 | 22,748 | 24,690 |
| Mean | 63,312 | 77,349 | 18,177 | 2347 | 28,010 | 33,772 |
| Std. Error of Mean | 0.626 | 0.497 | 1052 | 62,281 | 0.383 | 0.488 |
| 95% CI Mean Upper | 64,540 | 78,323 | 20,239 | 2469 | 28,761 | 34,728 |
| 95% CI Mean Lower | 62,085 | 76,376 | 16,114 | 2225 | 27,260 | 32,816 |
| Std. Deviation | 39,179 | 31,169 | 67,687 | 2903 | 17,871 | 30,072 |
| 95% CI Std. Dev. Upper | 40,067 | 31,873 | 69,178 | 2992 | 18,418 | 30,764 |
| 95% CI Std. Dev. Lower | 38,330 | 30,496 | 66,260 | 2819 | 17,356 | 29,411 |
| Coefficient of Variation | 0.619 | 0.403 | 3724 | 1237 | 0.638 | 0.890 |
| MAD | 16,550 | 3000 | 1064 | 851,251 | 9497 | 20,700 |
| MAD Robust | 24,537 | 4448 | 1578 | 1,262,064 | 14,081 | 30,690 |
| IQR | 77,400 | 44,176 | 6976 | 2,531,125 | 21,584 | 49,260 |
| Variance | 1534 | 971,537 | 4581 | $8.429 \times 10^6$ | 319,361 | 904,338 |
| 95% CI Variance Upper | 1605 | 1015 | 4785 | $8.954 \times 10^6$ | 339,207 | 946,397 |
| 95% CI Variance Lower | 1469 | 930,018 | 4390 | $7.949 \times 10^6$ | 301,215 | 865,034 |
| Skewness | −0.509 | −1129 | 7251 | 2758 | 1059 | 0.649 |
| Std. Error of Skewness | 0.039 | 0.039 | 0.038 | 0.053 | 0.052 | 0.040 |
| Kurtosis | −1431 | −0.236 | 59,218 | 9911 | 0.716 | −0.932 |
| Std. Error of Kurtosis | 0.078 | 0.078 | 0.076 | 0.105 | 0.105 | 0.079 |
| Shapiro–Wilk | 0.797 | 0.737 | 0.265 | 0.699 | 0.917 | 0.885 |
| *p*-value of Shapiro–Wilk | <0.001 | <0.001 | <0.001 | <0.001 | <0.001 | <0.001 |
| Range | 99,900 | 99,928 | 729,783 | 21,419 | 97,808 | 98,340 |
| Minimum | 0.100 | 0.072 | 0.000 | 1540 | −2566 | 0.000 |
| Maximum | 100,000 | 100,000 | 729,783 | 21,420 | 95,243 | 98,340 |
| 25th percentile | 22,600 | 55,824 | 0.168 | 536,617 | 15,575 | 7450 |
| 50th percentile | 83,450 | 97,000 | 1098 | 1190 | 22,748 | 24,690 |

|  | CFTC | ELEC | BCE | ENUC | PM25 | RENC |
|---|---|---|---|---|---|---|
| 75th percentile | 100,000 | 100,000 | 7144 | 3067 | 37,158 | 56,710 |
| 25th percentile | 22,600 | 55,824 | 0.168 | 536,617 | 15,575 | 7450 |
| 50th percentile | 83,450 | 97,000 | 1098 | 1190 | 22,748 | 24,690 |
| 75th percentile | 100,000 | 100,000 | 7144 | 3067 | 37,158 | 56,710 |
| Sum | 247,931 | 304,755 | 75,251 | $5.102 \times 10^6$ | 61,062 | 128,502 |

Kernel density and histogram plots of building-sector $CO_2$ emissions (BCE) and corresponding environmental and social indicators: access to clean fuels (CFTC), electricity access (ELEC), per capita energy use (ENUC), $PM_{2.5}$ air pollution, and renewable energy consumption (RENC). The graphs reveal right-skewness of BCE and ENUC and clustering at complete access (100%) for CFTC and ELEC, with variation across countries (Figure A1).

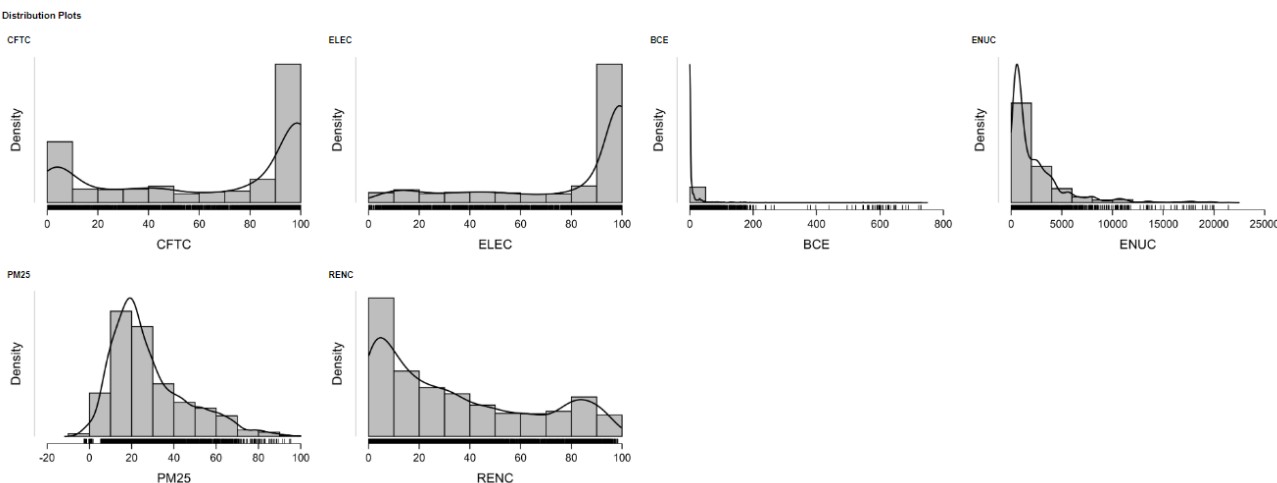

**Figure A1.** Distribution plots of BCE and related ESG indicators.

A matrix of scatterplots with marginal density plots of bivariate relations between building-related $CO_2$ emissions (BCE), access to clean fuels (CFTC), electricity access (ELEC), energy use per capita (ENUC), $PM_{2.5}$ air pollution, and renewable energy consumption (RENC) is presented. Plots indicate non-linear and highly skewed relations, where BCE and ENUC exhibit strong right-skewness and a shared clustering pattern, predominantly at energy access and pollution exposure variables (Figure A2).

Figure 3 shows boxplots of building-related $CO_2$ emissions (BCE) and matching ESG indicators: CFTC, ELEC, ENUC, $PM_{2.5}$, and RENC. BCE and ENUC exhibit high right-skew, with extreme outliers, such as points exceeding 700 Mt $CO_2$e and per capita energy consumption above 20,000 units, indicating the presence of high-emission, high-consumption countries. Conversely, CFTC, ELEC, and RENC register more slender interquartile ranges with concentrations of points near 100%, indicating widespread access in high-income countries. $PM_{2.5}$ captures medium variation with some extreme pollution conditions. The visualization reveals the planetary heterogeneity and unequal distribution of emissions and energy access (Figure A3).

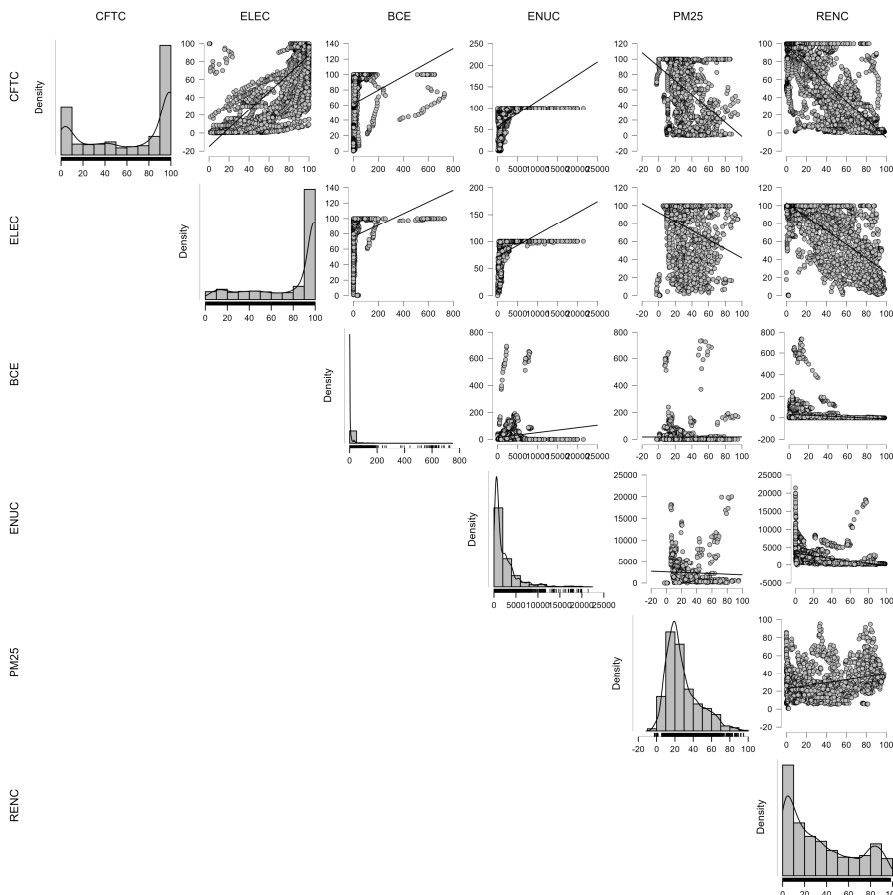

**Figure A2.** Pairwise scatterplots and density distributions of BCE and ESG-related indicators.

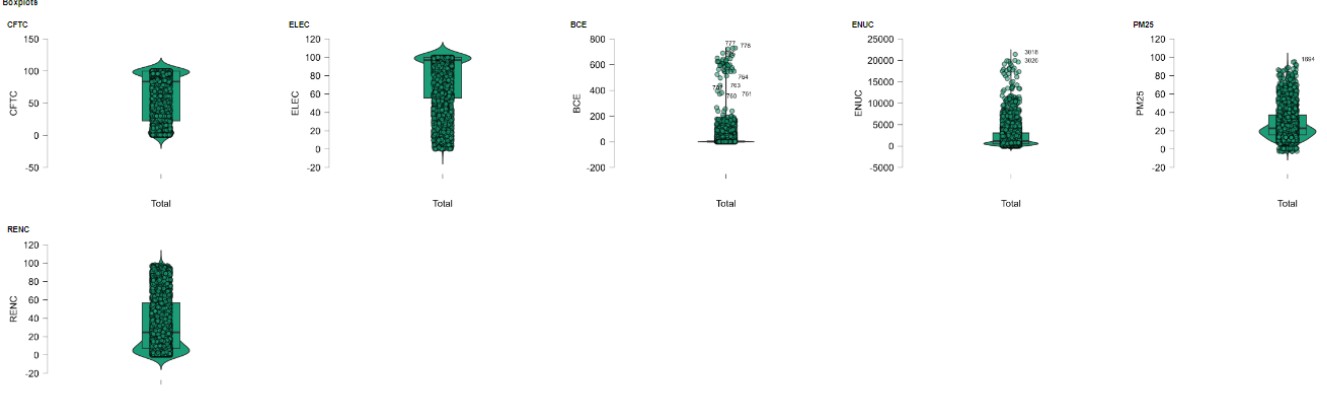

**Figure A3.** Boxplots of BCE and ESG-related indicators with outliers highlighted.

Figure 4 presents Q–Q (quantile–quantile) plots of building-sector $CO_2$ emissions (BCE) and corresponding ESG indicators: CFTC, ELEC, ENUC, $PM_{2.5}$, and RENC. Each variable's empirical distribution is contrasted with a normal distribution. BCE and ENUC considerably deviate from normality, with heavy right tails and outliers, reflecting positive skew and the presence of extreme values, as would be predicted given unequal worldwide energy use and emissions. CFTC and ELEC have truncated left tails, reflecting many observations at near-universal access. $PM_{2.5}$ displays moderate curvature, while RENC exhibits moderate skewness. These non-normal distributions necessitate the application of robust, non-parametric, or transformed approaches in modeling (Figure A4).

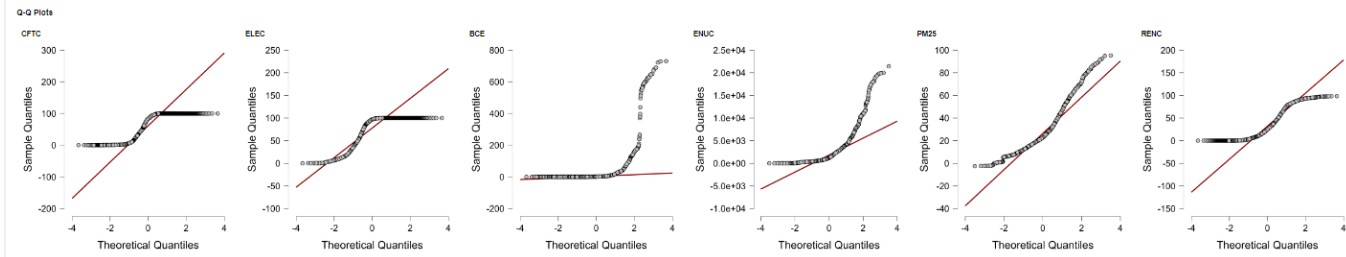

**Figure A4.** Q–Q plots assessing normality of BCE and ESG-related indicators.

The scatterplot matrix in Figure 5 graphs pairwise relationships between building-sector $CO_2$ emissions (BCE) and five significant ESG-related variables: access to clean fuels (CFTC), electricity access (ELEC), energy use per capita (ENUC), $PM_{2.5}$ pollution, and renewable energy consumption (RENC). BCE is weakly positive with ENUC and $PM_{2.5}$, as would be anticipated with higher consumption and pollution with higher emissions. BCE is weakly negative with RENC, indicating a compensating effect of renewable energy. More sizeable correlations emerge between CFTC and ELEC, whereas RENC is negative with CFTC as well as with ELEC. The density curves identify non-normal distributions across variables (Figure A5).

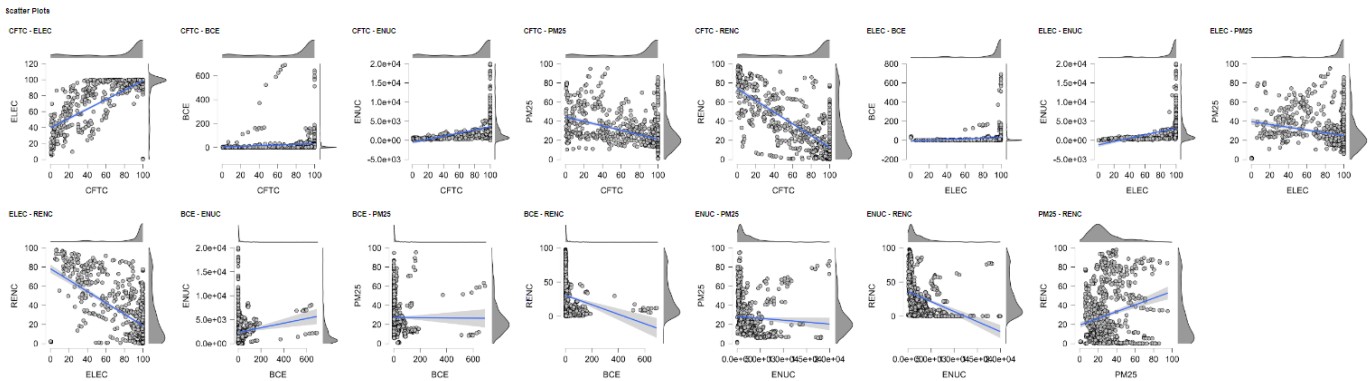

**Figure A5.** Scatterplot matrix with density overlays for BCE and ESG-related variables.

Table 2 presents the covariance matrix between building-related $CO_2$ emissions (BCE) and several ESG indicators: clean fuel access (CFTC), electricity access (ELEC), per capita energy use (ENUC), $PM_{2.5}$ air pollution, and renewable energy use, in relation to BCE. Covariances define the direction and strength. BCE has a moderate positive correlation with CFTC (328.1), ELEC (294.5), and ENUC (29.0) due to related higher consumption and energy accessibility, which is typically associated with higher emissions. BCE has a strong negative correlation with RENC (−416.6), confirming the use of renewable energy in emission offset. Strong covariation between CFTC and ELEC (715.0) defines the associated infrastructuring development process, while negative covariations vs. RENC suggest energy change mismatches. These covariations highlight systemic relationships defining emission dynamics worldwide (Table A3).

The correlation matrix reveals significant relationships between building-related $CO_2$ emissions (BCE) and ESG-energy indicators. BCE is weakly positively correlated with clean fuel access (0.122), electricity access (0.137), and per capita energy consumption (0.125) while exhibiting a moderate growth increase in the latter, particularly with higher structure and consumption. BCE is negatively correlated with renewable consumption (−0.197) in supporting the renewables-driven process of decarbonization. Electricity access is highly correlated with access to electricity and clean fuels (0.739), while demonstrating a dependency between the two infrastructures. Renewable energy consumption (RENC) is highly

negatively correlated with consumption-based carbon pollution (CFTC) ($-0.783$) and with electricity consumption (ELEC) ($-0.605$), indicating possible energy transition trade-offs. $PM_{2.5}$ pollution shows weak positive correlations with other indicators (Table A4).

**Table A3.** Covariance matrix of BCE and ESG-related indicators.

|  | **CFTC** | **ELEC** | **BCE** | **ENUC** | **PM25** | **RENC** |
|---|---|---|---|---|---|---|
| CFTC | 1.212 | 715.040 | 328.145 | 49.067 | $-281.490$ | $-744.185$ |
| ELEC | 715.040 | 771.991 | 294.482 | 33.913 | $-112.064$ | $-458.817$ |
| BCE | 328.145 | 294.482 | 6.011 | 28.989 | $-8.701$ | $-416.588$ |
| ENUC | 49.067 | 33.913 | 28.989 | $8.975 \times 10^6$ | $-3.560$ | $-26.307$ |
| PM25 | $-281.490$ | $-112.064$ | $-8.701$ | $-3.560$ | 310.835 | 109.470 |
| RENC | $-744.185$ | $-458.817$ | $-416.588$ | $-26.307$ | 109.470 | 746.022 |

**Table A4.** Pearson correlation matrix of building-sector $CO_2$ emissions (BCE) and ESG-energy indicators.

|  | **CFTC** | **ELEC** | **BCE** | **ENUC** | **PM25** | **RENC** |
|---|---|---|---|---|---|---|
| CFTC | 1.000 | 0.739 | 0.122 | 0.470 | $-0.459$ | $-0.783$ |
| ELEC | 0.739 | 1.000 | 0.137 | 0.407 | $-0.229$ | $-0.605$ |
| BCE | 0.122 | 0.137 | 1.000 | 0.125 | $-0.006$ | $-0.197$ |
| ENUC | 0.470 | 0.407 | 0.125 | 1.000 | $-0.067$ | $-0.321$ |
| PM25 | $-0.459$ | $-0.229$ | $-0.006$ | $-0.067$ | 1.000 | 0.227 |
| RENC | $-0.783$ | $-0.605$ | $-0.197$ | $-0.321$ | 0.227 | 1.000 |

## Appendix C. S-Social

Below is the table of descriptive statistics of five chief social indicators—food production (FOOD), gender parity in education (GPIE), income share of the bottom 20% (INC20), labor force participation (LABF), and women in parliament (WPAR)—and building-related $CO_2$ emissions (BCE). BCE is extremely right-skewed (skewness = 7251) and extremely kurtotic due to the impact of some high-emitting states. Socio-environmental indicators like LABF and GPIE exhibit a relatively even distribution, while WPAR is unsurprisingly at the tail end on average, as it registers long-term patterns of divergent gender representation. Incidentally, FOOD and INC20 have high variance coupled with high skewness, which reflect structural inequity. Contrast between dispersion and distribution shows the perception about the lopsidedness of socio-environmental states of nations (Table A5).

**Table A5.** Descriptive statistics of social indicators and building-related $CO_2$ emissions (BCE).

|  | **FOOD** | **GPIE** | **INC20** | **LABF** | **WPAR** | **BCE** |
|---|---|---|---|---|---|---|
| Valid | 4102 | 2665 | 1607 | 3998 | 3963 | 4140 |
| Missing | 38 | 1475 | 2533 | 142 | 177 | 0 |
| Mode | 0.123 | 0.990 | 7100 | 55,146 | 0.000 | 0.006 |
| Median | 96,020 | 0.998 | 7100 | 67,324 | 17,302 | 1098 |
| Mean | 92,769 | 216,096 | 7879 | 65,663 | 19,263 | 18,177 |
| Std. Error of Mean | 0.383 | 47,245 | 0.236 | 0.186 | 0.204 | 1052 |

**Table A5.** *Cont.*

|  | FOOD | GPIE | INC20 | LABF | WPAR | BCE |
|---|---|---|---|---|---|---|
| 95% CI Mean Upper | 93,520 | 308,737 | 8341 | 66,029 | 19,663 | 20,239 |
| 95% CI Mean Lower | 92,017 | 123,454 | 7416 | 65,297 | 18,863 | 16,114 |
| Std. Deviation | 24,542 | 2438 | 9458 | 11,791 | 12,847 | 67,687 |
| 95% CI Std. Dev. Upper | 25,085 | 2506 | 9797 | 12,055 | 13,136 | 69,178 |
| 95% CI Std. Dev. Lower | 24,022 | 2375 | 9142 | 11,538 | 12,570 | 66,260 |
| Coefficient of Variation | 0.265 | 11,287 | 1200 | 0.180 | 0.667 | 3724 |
| MAD | 10,000 | 0.024 | 1500 | 6989 | 7927 | 1064 |
| MAD Robust | 14,826 | 0.035 | 2224 | 10,361 | 11,752 | 1578 |
| IQR | 21,758 | 0.049 | 3050 | 14,340 | 16,190 | 6976 |
| Variance | 602,295 | $5.949 \times 10^6$ | 89,448 | 139,028 | 165,038 | 4581 |
| 95% CI Variance Upper | 629,238 | $6.281 \times 10^6$ | 95,972 | 145,331 | 172,554 | 4785 |
| 95% CI Variance Lower | 577,053 | $5.642 \times 10^6$ | 83,570 | 133,129 | 158,005 | 4390 |
| Skewness | 2615 | 11,601 | 7932 | −0.916 | 1197 | 7251 |
| Std. Error of Skewness | 0.038 | 0.047 | 0.061 | 0.039 | 0.039 | 0.038 |
| Kurtosis | 43,986 | 135,463 | 65,260 | 1406 | 2689 | 59,218 |
| Std. Error of Kurtosis | 0.076 | 0.095 | 0.122 | 0.077 | 0.078 | 0.076 |
| Shapiro–Wilk | 0.816 | 0.060 | 0.250 | 0.954 | 0.930 | 0.265 |
| *p*-value of Shapiro–Wilk | <0.001 | <0.001 | <0.001 | <0.001 | <0.001 | <0.001 |
| Range | 502,017 | 33,376 | 93,769 | 76,467 | 87,730 | 729,783 |
| Minimum | 0.123 | 0.000 | 0.187 | 13,156 | 0.000 | 0.000 |
| Maximum | 502,140 | 33,376 | 93,956 | 89,623 | 87,730 | 729,783 |
| 25th percentile | 81,757 | 0.971 | 5350 | 59,368 | 10,000 | 0.168 |
| 50th percentile | 96,020 | 0.998 | 7100 | 67,324 | 17,302 | 1098 |
| 75th percentile | 103,515 | 1020 | 8400 | 73,709 | 26,190 | 7144 |
| Sum | 380,537 | 575,894 | 12,660 | 262,520 | 76,338 | 75,251 |

Figure A6 presents the distribution of the social component variables under the ESG framework, highlighting their diversity and the occurrence of skewness in some indicators. The Food Production Index (FOOD) reveals a positively skewed distribution, where most observations are clustered below 150, indicating that even though most countries exhibit moderate food production, some countries exhibit significantly higher levels. The Gender Parity Index in Education (GPIE) also reveals a strong positive skew, where most countries are clustered around lower values, with a small fraction achieving very high values, indicating uneven convergence in schooling equity. Likewise, the variable Income Share of the Poorest 20% (INC20) is skewed to the right, indicating that in most instances the poorest quintile shares just a small fraction of the country's income. Variable Labor Force Participation (LABF), on the other hand, approximates a normal distribution, centered around the value of 60%, indicating a more balanced distribution across the population. Women in Parliament (WPAR) are distributed widely, albeit still skewed, indicating that many countries exhibit modest representation, while others exhibit relatively high representation. Variables Building-sector $CO_2$ Emissions (BCE) and Building Performance (BP) indicate strong right-skewness, where most values cluster around the low values, albeit a

few extreme values overshading the distribution. Generally, the figure highlights the asymmetric distribution of social and emission-related variables, where structural inequality and varying distribution paths across countries are indicated (Figure A6).

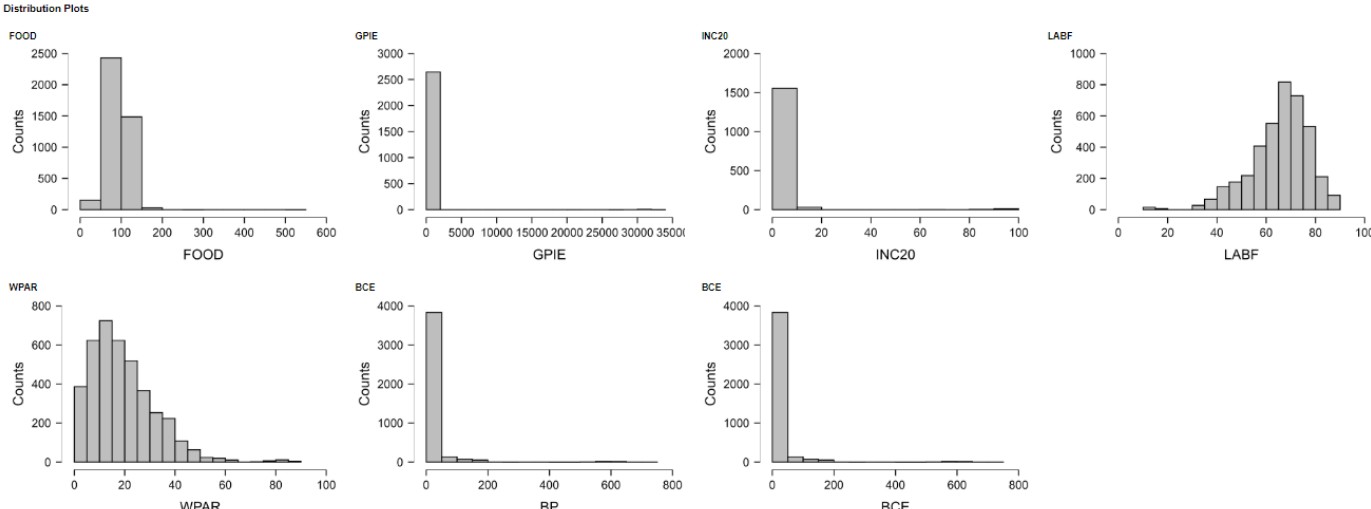

**Figure A6.** Distribution of social component variables in the ESG framework.

Boxplots illustrating the spread and outliers of key social indicators used in the ESG framework. Variables include food expenditure (FOOD), public investment in education (GPIE), household income (INC20), labor force participation (LABF), women's participation rate (WPAR), and basic consumption expenditure (BCE). Outliers are labeled and represent extreme values deviating from the interquartile range (Figure A7).

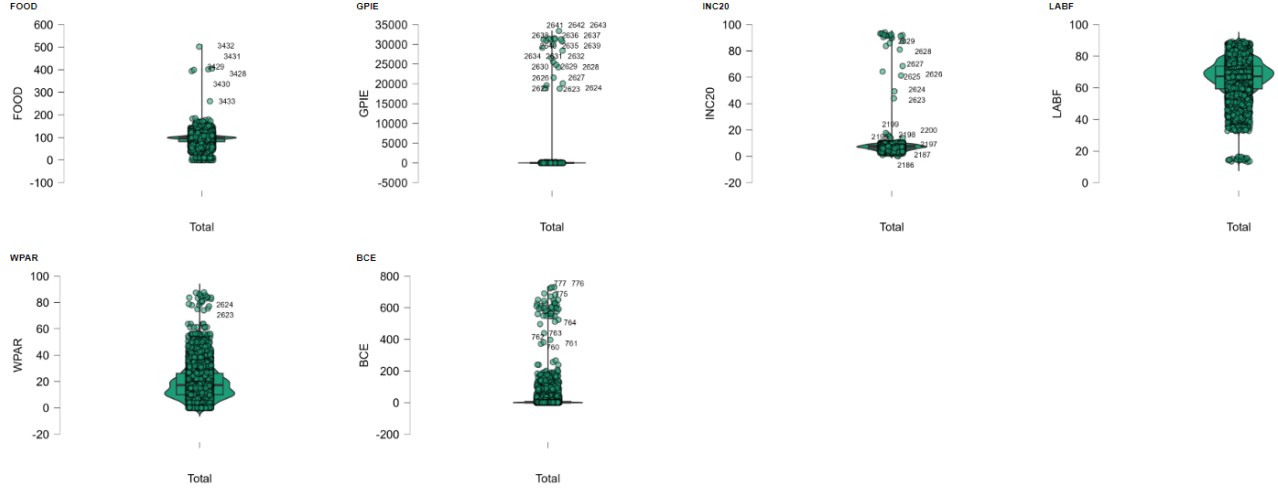

**Figure A7.** Boxplot distributions of social ESG variables with outlier detection.

Quantile–Quantile (Q–Q) plots for variables represent the social component in ESG analysis. The deviation of points from the theoretical normal line indicates non-normal distributions, particularly for FOOD, GPIE, INC20, and BCE, which exhibit strong right skewness and heavy tails. These findings suggest the need for transformation or robust statistical methods (Figure A8).

The scatterplot matrix of the social component variables from the ESG framework includes FOOD, GPIE, INC20, LABF, WPAR, and BCE. The diagonal panels display the univariate density distributions, while the off-diagonal panels show pairwise scatterplots.

The matrix reveals strong non-normality, right-skewed distributions, and possible non-linear associations between variables such as INC20 and GPIE (Figure A9).

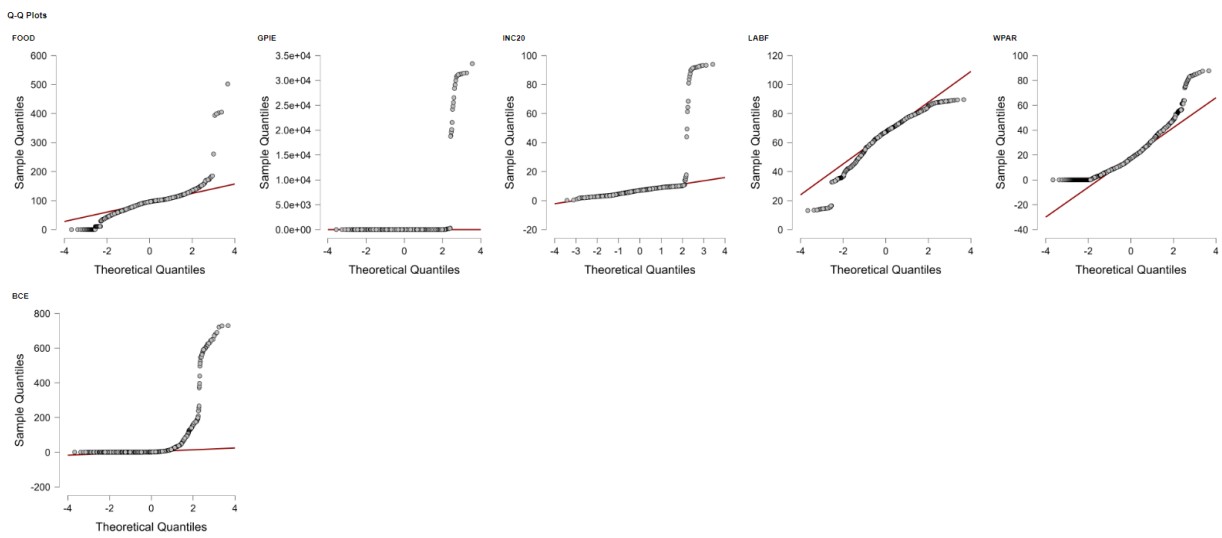

**Figure A8.** Normality assessment of social ESG variables via Q–Q plots.

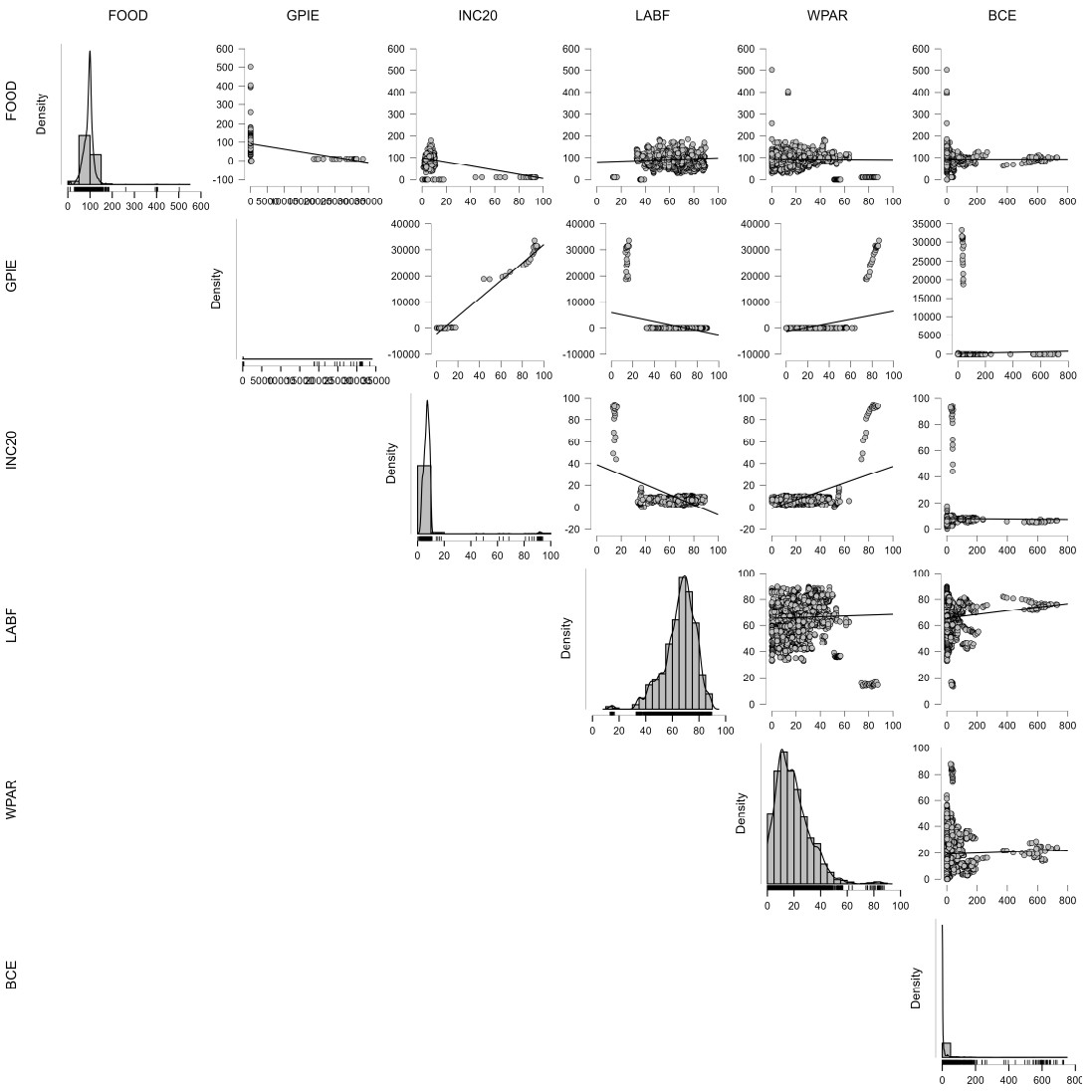

**Figure A9.** Scatterplot matrix and density distributions of social ESG variables.

The plot visualizes the bivariate scatter distributions and marginal densities among the social indicators employed in the model, including FOOD (food production volume), GPIE (gross primary income equity), INC20 (income share of the bottom 20%), LABF (labor force participation rate), WPAR (women in parliament), and BCE (building-related $CO_2$ emissions). This matrix enables an exploratory assessment of linear associations and potential collinearity among the variables constituting the social dimension of the ESG framework, thereby contributing to a deeper understanding of their influence on emission patterns within the built environment (Figure A10).

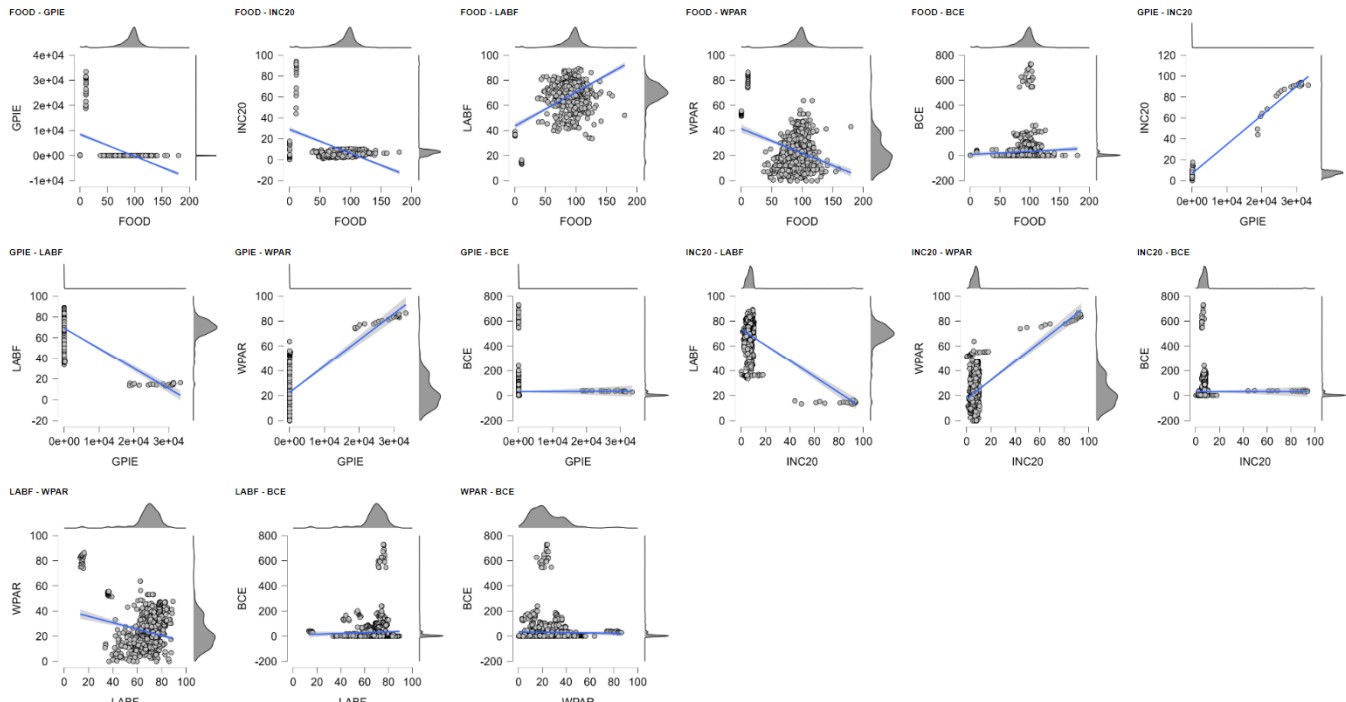

**Figure A10.** Pairwise correlation plot of social pillar variables in the ESG framework.

The table reports the covariances among key social indicators—FOOD, GPIE, INC20, LABF, WPAR—and building-related $CO_2$ emissions (BCE). Positive values indicate a direct relationship in variability between variable pairs, whereas negative values suggest an inverse relationship. High absolute values reflect stronger associations in the variance structure, which may inform multicollinearity diagnostics in subsequent regression models (Table A6).

**Table A6.** Covariance matrix of social ESG variables and building-related $CO_2$ emissions.

|  | FOOD | GPIE | INC20 | LABF | WPAR | BCE |
|---|---|---|---|---|---|---|
| FOOD | 424.789 | −37.108 | −96.944 | 114.735 | −82.864 | 107.769 |
| GPIE | −37.108 | $1.262 \times 10^7$ | 35.079 | −24.538 | 26.719 | 1.483 |
| INC20 | −96.944 | 35.079 | 102.362 | −65.165 | 77.834 | −0.529 |
| LABF | 114.735 | −24.538 | −65.165 | 126.359 | −32.177 | 39.820 |
| WPAR | −82.864 | 26.719 | 77.834 | −32.177 | 191.207 | −29.946 |
| BCE | 107.769 | 1.483 | −0.529 | 39.820 | −29.946 | 7.518 |

The table presents Pearson correlation coefficients among the selected social indicators—FOOD, GPIE, INC20, LABF, and WPAR—and the variable BCE, representing building-related $CO_2$ emissions. Strong positive correlations are observed between

GPIE and INC20 (r = 0.976), while negative associations emerge between GPIE and LABF (r = −0.615) and between INC20 and LABF (r = −0.573). Correlations with BCE are weak, suggesting limited linear association with social variables in this model (Table A7).

**Table A7.** Pearson correlation matrix among social ESG variables and building-related $CO_2$ emissions.

|  | **FOOD** | **GPIE** | **INC20** | **LABF** | **WPAR** | **BCE** |
|---|---|---|---|---|---|---|
| FOOD | 1.000 | −0.507 | −0.465 | 0.495 | −0.291 | 0.060 |
| GPIE | −0.507 | 1.000 | 0.976 | −0.615 | 0.544 | 0.005 |
| INC20 | −0.465 | 0.976 | 1.000 | −0.573 | 0.556 | $−6.033 \times 10^{-4}$ |
| LABF | 0.495 | −0.615 | −0.573 | 1.000 | −0.207 | 0.041 |
| WPAR | −0.291 | 0.544 | 0.556 | −0.207 | 1.000 | −0.025 |
| BCE | 0.060 | 0.005 | $−6.033 \times 10^{-4}$ | 0.041 | −0.025 | 1.000 |

## Appendix D. G-Governance

This table includes complete descriptive statistics of seven governance and public service indicators (GOVT: government effectiveness; EDUE: education expenditure; STAB: political stability; RNDG: R&D expenditure; LAWR: rule of law; HOSP: hospital beds; SCIE: scientific publications) and the variable BCE (building-related $CO_2$ emissions). Such statistics include central tendency, dispersion, shape (skewness and kurtosis), and results of the Shapiro–Wilk normality test. High skewness and extreme kurtosis of a number of variables (e.g., BCE, LAWR, SCIE) suggest non-normal distributions likely requiring transformation prior to modeling (Table A8).

**Table A8.** Descriptive statistics of governance and public service variables related to building $CO_2$ emissions.

|  | **BCE** | **GOVT** | **EDUE** | **STAB** | **RNDG** | **LAWR** | **HOSP** | **SCIE** |
|---|---|---|---|---|---|---|---|---|
| Valid | 4140 | 3907 | 2760 | 3949 | 1883 | 3941 | 2030 | 3782 |
| Missing | 0 | 233 | 1380 | 191 | 2257 | 199 | 2110 | 358 |
| Mode | 0.006 | −1.158 | 0.000 | 1.170 | 0.018 | 0.834 | 1.300 | 0.000 |
| Median | 1.098 | −0.215 | 14.031 | −0.019 | 0.577 | −0.270 | 3.015 | 226.270 |
| Mean | 18.177 | −0.029 | 14.370 | −0.009 | 0.948 | 0.583 | 3.733 | 10.180 |
| Std. Error of Mean | 1.052 | 0.021 | 0.097 | 0.025 | 0.023 | 0.138 | 0.060 | 688.649 |
| 95% CI Mean Upper | 20.239 | 0.012 | 14.561 | 0.040 | 0.994 | 0.854 | 3.850 | 11.530 |
| 95% CI Mean Lower | 16.114 | −0.069 | 14.180 | −0.058 | 0.902 | 0.312 | 3.616 | 8.830 |
| Std. Deviation | 67.687 | 1.300 | 5.102 | 1.558 | 1.017 | 8.682 | 2.683 | 42.350 |
| 95% CI Std. Dev. Upper | 69.178 | 1.329 | 5.240 | 1.594 | 1.051 | 8.878 | 2.769 | 43.327 |
| 95% CI Std. Dev. Lower | 66.260 | 1.272 | 4.971 | 1.525 | 0.986 | 8.494 | 2.603 | 41.417 |
| Coefficient of Variation | 3.724 | −45.259 | 0.355 | −171.768 | 1.073 | 14.889 | 0.719 | 4.160 |
| MAD | 1.064 | 0.660 | 3.235 | 0.711 | 0.430 | 0.700 | 1.715 | 222.835 |
| MAD Robust | 1.578 | 0.979 | 4.797 | 1.054 | 0.638 | 1.038 | 2.543 | 330.375 |
| IQR | 6.976 | 1.373 | 6.506 | 1.424 | 1.138 | 1.440 | 3.878 | 3.153 |
| Variance | 4.581 | 1.690 | 26.028 | 2.429 | 1.034 | 75.370 | 7.201 | $1.794 \times 10^{9}$ |
| 95% CI Variance Upper | 4.785 | 1.767 | 27.458 | 2.540 | 1.104 | 78.812 | 7.665 | $1.877 \times 10^{9}$ |
| 95% CI Variance Lower | 4.390 | 1.617 | 24.707 | 2.325 | 0.971 | 72.149 | 6.777 | $1.715 \times 10^{9}$ |

**Table A8.** *Cont.*

| | BCE | GOVT | EDUE | STAB | RNDG | LAWR | HOSP | SCIE |
|---|---|---|---|---|---|---|---|---|
| Skewness | 7.251 | 4.152 | 0.454 | 6.009 | 1.273 | 11.968 | 1.111 | 8.482 |
| Std. Error of Skewness | 0.038 | 0.039 | 0.047 | 0.039 | 0.056 | 0.039 | 0.054 | 0.040 |
| Kurtosis | 59.218 | 34.610 | 1.477 | 63.863 | 1.633 | 143.174 | 1.259 | 85.379 |
| Std. Error of Kurtosis | 0.076 | 0.078 | 0.093 | 0.078 | 0.113 | 0.078 | 0.109 | 0.080 |
| Shapiro–Wilk | 0.265 | 0.734 | 0.977 | 0.615 | 0.858 | 0.112 | 0.911 | 0.235 |
| *p*-value of Shapiro–Wilk | <0.001 | <0.001 | <0.001 | <0.001 | <0.001 | <0.001 | <0.001 | <0.001 |
| Range | 729.783 | 14.580 | 44.802 | 22.960 | 7.586 | 111.216 | 14.590 | 669.746 |
| Minimum | 0.000 | −2.439 | 0.000 | −3.313 | −1.880 | −2.591 | 0.100 | −2.283 |
| Maximum | 729.783 | 12.141 | 44.802 | 19.647 | 5.706 | 108.625 | 14.690 | 669.744 |
| 25th percentile | 0.168 | −0.796 | 10.877 | −0.722 | 0.230 | −0.854 | 1.603 | 25.813 |
| 50th percentile | 1.098 | −0.215 | 14.031 | −0.019 | 0.577 | −0.270 | 3.015 | 226.270 |
| 75th percentile | 7.144 | 0.577 | 17.383 | 0.702 | 1.368 | 0.586 | 5.480 | 3.179 |
| Sum | 75.251 | −112.215 | 39.661 | −35.830 | 1.785 | 2.297 | 7.577 | $3.850 \times 10^7$ |

This figure displays kernel density estimates and histograms for BCE (building-related $CO_2$ emissions), GOVT (government effectiveness), EDUE (education expenditure), STAB (political stability), RNDG (research and development expenditure), LAWR (rule of law), HOSP (hospital beds), and SCIE (scientific publications). Most variables exhibit skewed distributions, particularly BCE, LAWR, and SCIE, suggesting potential non-normality and the need for transformation prior to parametric analysis (Figure A11).

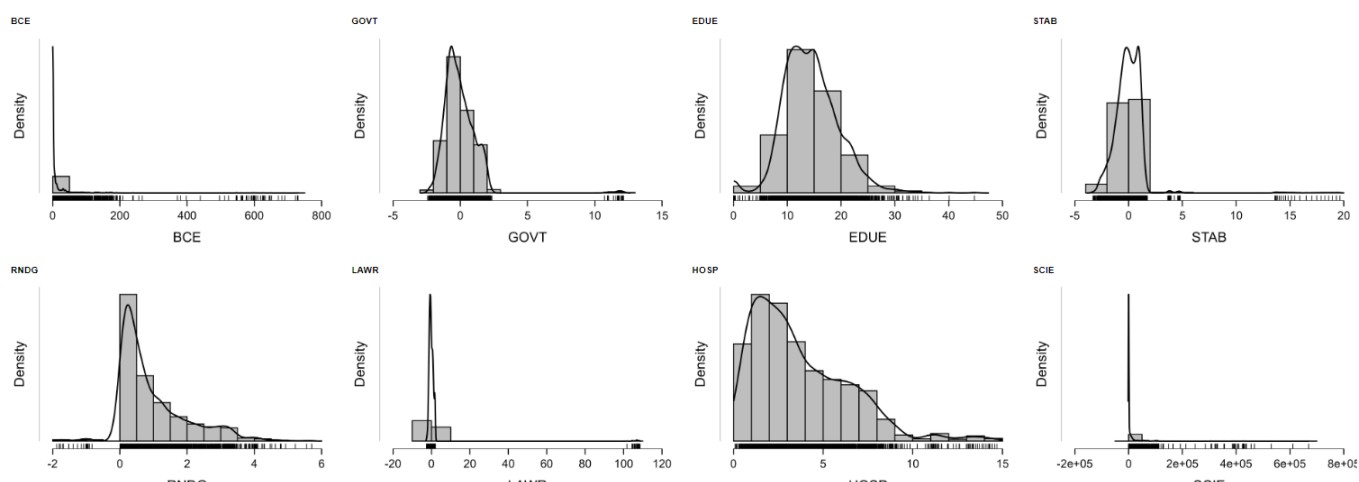

**Figure A11.** Density distributions of governance and public service indicators related to building $CO_2$ emissions.

This scatterplot matrix exhibits the bivariate relationships and marginal density distributions between BCE (building-related $CO_2$ emissions), GOVT (government effectiveness), EDUE (education expenditure), STAB (political stability), RNDG (research and development expenditure), LAWR (rule of law), HOSP (hospital beds), and SCIE (scientific publications). It facilitates visualization of patterns of correlation, hetero-scedasticity, and distributional skewness in aid of preliminary diagnostics for multivariate modeling within the ESG context (Figure A12).

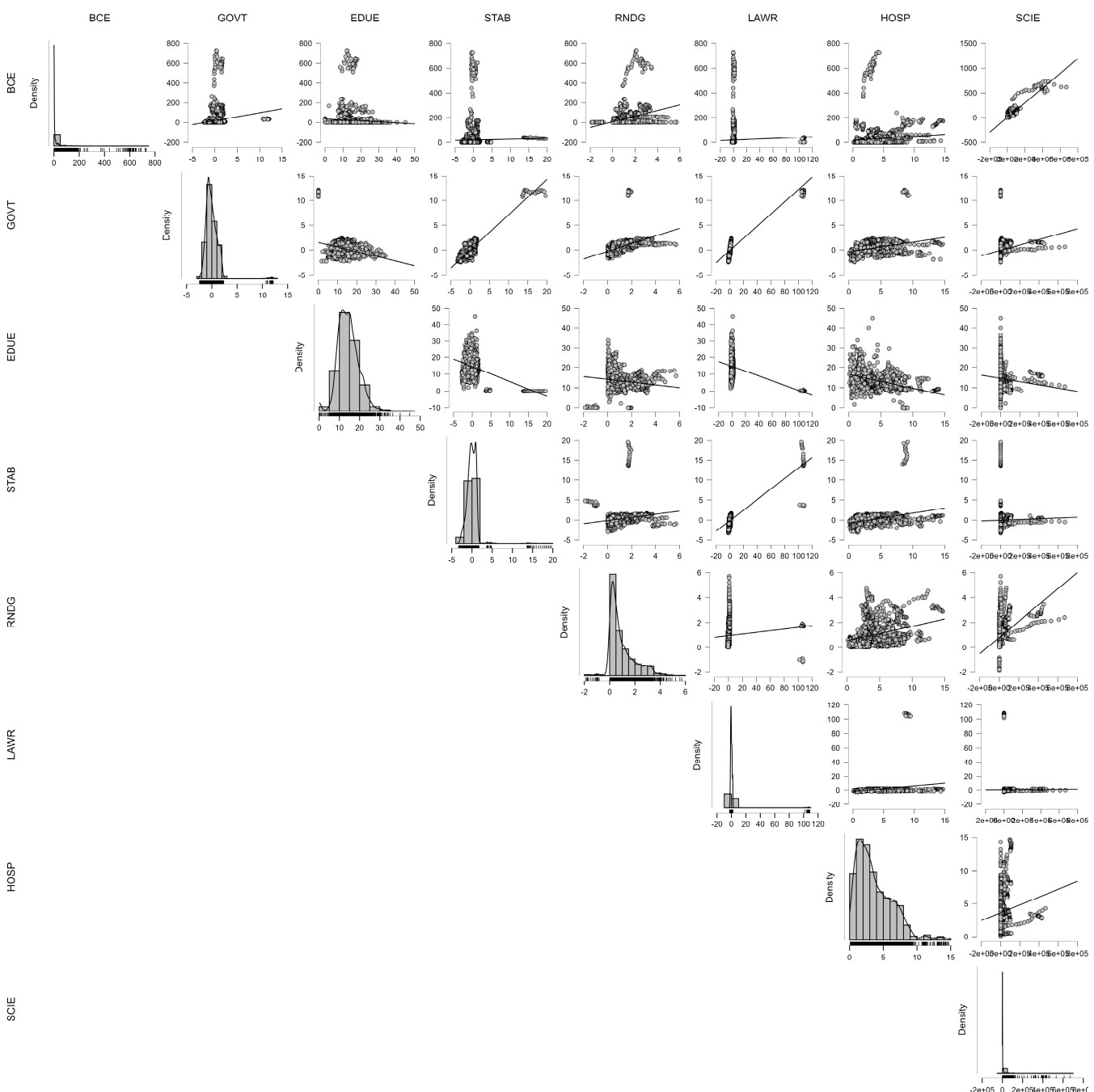

**Figure A12.** Pairwise scatterplot matrix of governance and public service variables related to building $CO_2$ emissions.

Figure A13 shows boxplots of distributions and outliers of BCE (building-related $CO_2$ emissions), GOVT (government effectiveness), EDUE (education expenditure), STAB (political stability), RNDG (research and development expenditure), LAWR (rule of law), HOSP (hospital beds), and SCIE (scientific publications). Most of these variables contain apparent outliers and asymmetries, signifying signs of skewed distributions and a need to normalize or utilize robust estimation methods in subsequent analyses (Figure A13).

Quantile–Quantile (Q–Q) plots check normality of variables such as BCE (building-related $CO_2$ emissions), GOVT (government effectiveness), EDUE (education expenditure), STAB (political stability), RNDG (research and development expenditure), LAWR (rule of law), HOSP (hospital beds), and SCIE (scientific publications). Strong departures from the reference line in all plots, especially for BCE, LAWR, and SCIE, indicate heavy-tailed and

right-skewed distributions and thus non-normality and potential need for transformation in statistical modeling (Figure A14).

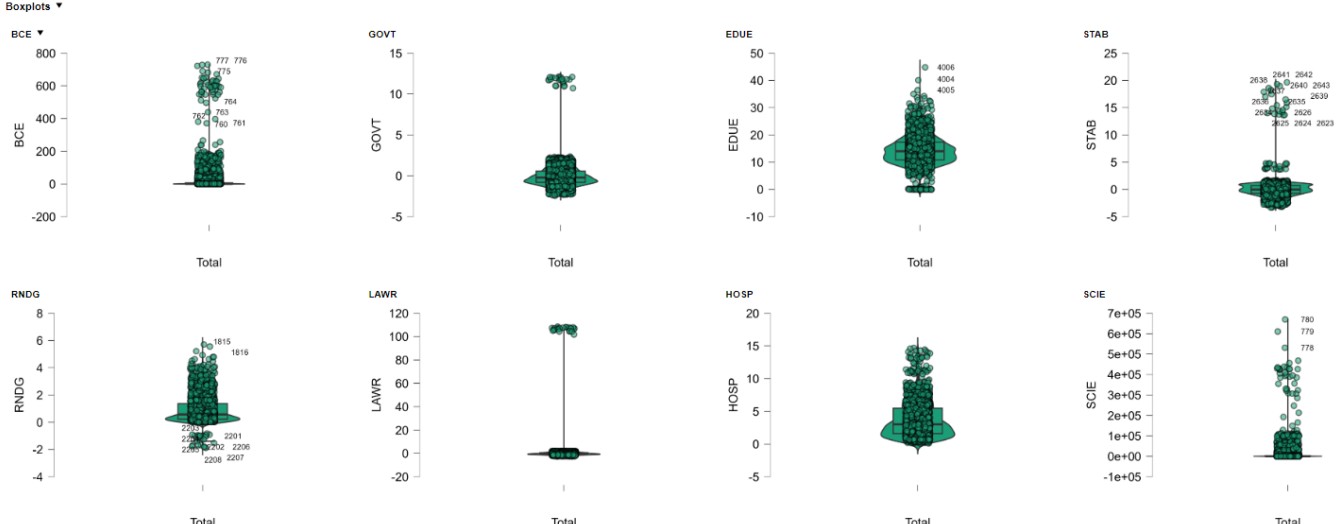

**Figure A13.** Boxplot distribution of governance and public service indicators related to building $CO_2$ emissions.

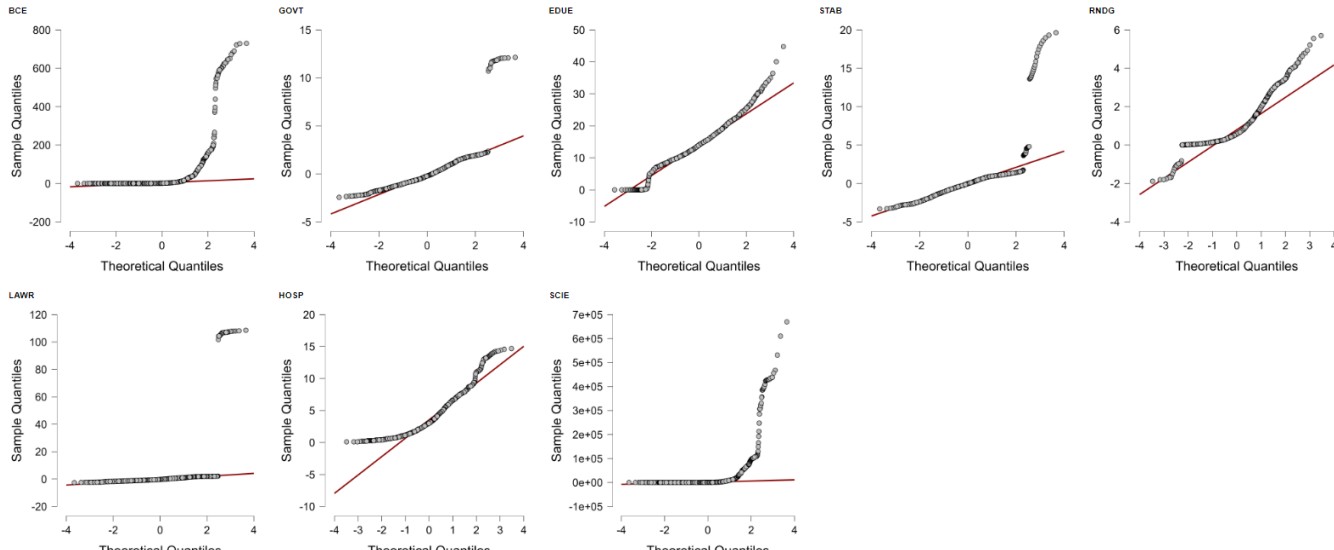

**Figure A14.** Q–Q plots for governance and public service indicators related to building $CO_2$ emissions.

Figure A15 uncovers a scatter plot matrix exploring the relationship between building-related $CO_2$ emissions (BCE) and ESG governance metrics. Scatters reveal varied patterns, including both positive and negative associations across variables. Government effectiveness (GOVT) is strongly positively associated with BCE, such that high-institution-capability economies also exhibit high emissions, likely the result of infrastructure expansion. Expenditure on education (EDUE) depicts a moderate positive association with emissions, such that investments in human capabilities are accompanied by higher energy consumption. Political stability (STAB) and the rule of law (LAWR) tend to exhibit negative associations with BCE, as hypothesized, considering the role that strong order plays in enforcing regulations that reduce emissions. R&D expenditure (RNDG) exhibits a negative association with BCE, indicating that innovations can serve as a source of emissions savings. Expenditure on hospitals (HOSP) exhibits a positive association, reflecting the carbon intensities embedded in the health sector. Finally, scientific productivity (SCIE) depicts

a multifaceted relationship: high output accompanied by emissions reflecting advanced economies, while high scientific productivity is also a source reflecting innovations that yield emissions savings. As a panel, the figure separates the double-edged role of governance: capable institutions enable reductions, while in high-income cases, governance emerges hand-in-hand with carbon-intensive growth trajectories (Figure A15).

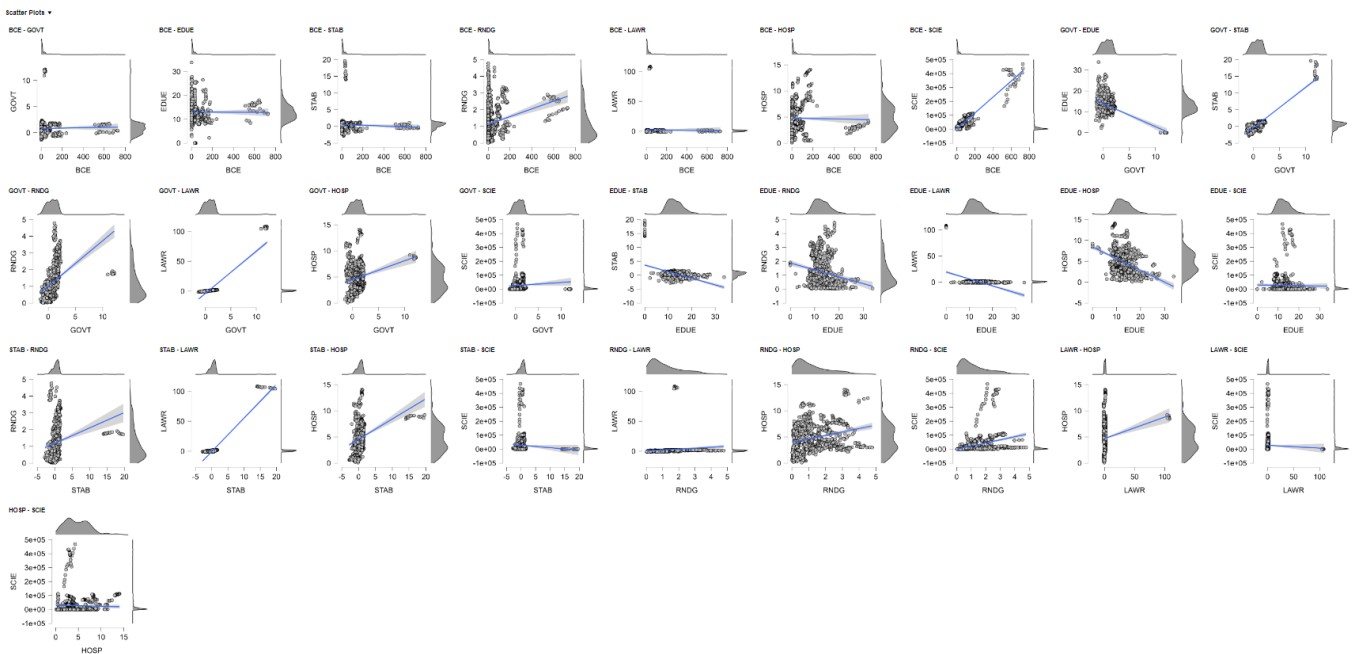

**Figure A15.** Scatter plot matrix of building-related $CO_2$ emissions (BCE) and G-governance indicators.

Table A9 provides some indication regarding the covariance connection between building-related $CO_2$ emissions (BCE) and governance indicators. It uncovers both reinforcing and conflicting dynamics. BCE significantly covaries positively with both R&D expenditure (27.514) and scientific output ($6.849 \times 10^6$), substantiating that innovative economies pay a higher price in emissions due to energy-intensive infrastructure. Conversely, BCE negatively covaries with both the rule of law ($-15.110$) and political stability ($-14.152$), indicating that effective institutions can impose environmental regulations as well as decelerate emissions. It is surprising, however, that government effectiveness (4.309) positively covaries with BCE, signaling the irony that effective government is often associated with high-carbon economies that are sophisticated. Similarly, hospital infrastructure ($-4.792$) also exhibits a moderate negative covariance, reflecting the cost that the carbon sector incurs in health systems. Education expenditure ($-3.168$) negatively covaries with BCE, although it exhibits strong negative relationships with governance indicators, indicating trade-offs in resource allocation. As a panel, the matrix sheds insight into the double responsibility that governance has, generating growth while also inflicting structure-based challenges within decarbonization (Table A9).

The table shows Pearson correlation coefficients between BCE (building-related $CO_2$ emissions) and governance-related indicators like GOVT (government effectiveness), EDUE (education expenditure), STAB (political stability), RNDG (research and development expenditure), LAWR (rule of law), HOSP (hospital beds), and SCIE (scientific publications). The highly positive relationship between BCE and SCIE (r = 0.953) suggests intense co-movement, while GOVT and STAB suggest high mutual correspondence (r = 0.916) too. These correlations provide some insights into likely multi-collinearity patterns and directional relationships across the governance dimension within the ESG framework (Table A10).

**Table A9.** Covariance matrix of governance and public service variables in relation to building $CO_2$ emissions.

|  | BCE | GOVT | EDUE | STAB | RNDG | LAWR | HOSP | SCIE |
|---|---|---|---|---|---|---|---|---|
| BCE | 11.950 | 4.309 | −3.168 | −14.152 | 27.514 | −15.110 | −4.792 | $6.849 \times 10^6$ |
| GOVT | 4.309 | 2.653 | −2.943 | 3.235 | 0.722 | 18.655 | 0.966 | 5.898 |
| EDUE | −3.168 | −2.943 | 16.500 | −3.925 | −0.818 | −21.927 | −4.630 | −4.123 |
| STAB | −14.152 | 3.235 | −3.925 | 4.703 | 0.433 | 26.401 | 1.816 | −8.322 |
| RNDG | 27.514 | 0.722 | −0.818 | 0.433 | 1.011 | 1.560 | 0.665 | 22.538 |
| LAWR | −15.110 | 18.655 | −21.927 | 26.401 | 1.560 | 169.873 | 6.960 | −34.172 |
| HOSP | −4.792 | 0.966 | −4.630 | 1.816 | 0.665 | 6.960 | 7.315 | −5.751 |
| SCIE | $6.849 \times 10^6$ | 5.898 | −4.123 | −8.322 | 22.538 | −34.172 | −5.751 | $4.319 \times 10^9$ |

**Table A10.** Pearson correlation matrix of governance and public service variables related to building $CO_2$ emissions.

|  | BCE | GOVT | EDUE | STAB | RNDG | LAWR | HOSP | SCIE |
|---|---|---|---|---|---|---|---|---|
| BCE | 1.000 | 0.024 | −0.007 | −0.060 | 0.250 | −0.011 | −0.016 | 0.953 |
| GOVT | 0.024 | 1.000 | −0.445 | 0.916 | 0.441 | 0.879 | 0.219 | 0.055 |
| EDUE | −0.007 | −0.445 | 1.000 | −0.446 | −0.200 | −0.414 | −0.421 | −0.015 |
| STAB | −0.060 | 0.916 | −0.446 | 1.000 | 0.199 | 0.934 | 0.310 | −0.058 |
| RNDG | 0.250 | 0.441 | −0.200 | 0.199 | 1.000 | 0.119 | 0.245 | 0.341 |
| LAWR | −0.011 | 0.879 | −0.414 | 0.934 | 0.119 | 1.000 | 0.197 | −0.040 |
| HOSP | −0.016 | 0.219 | −0.421 | 0.310 | 0.245 | 0.197 | 1.000 | −0.032 |
| SCIE | 0.953 | 0.055 | −0.015 | −0.058 | 0.341 | −0.040 | −0.032 | 1.000 |

## Appendix E. Autocorrelation and Heteroscedasticity

*Appendix E.1. Autocorrelation for E-Environment*

The Wooldridge residual-regression test for the panel-data disturbances shows evident first-order serial correlation: from a sample containing 570 observations, an auxiliary regression of residual on its first lag generates an F statistic of 382.97 with a *p*-value < 0.001 for rejection of the null hypothesis of no temporal dependence. We observe a coefficient for lagged residual of 0.756, supported by a t ratio of 19.57 and an R-squared of 0.403. This implies that about three-quarters of each disturbance spills over into the next period, with the intercept effectively being 0; there is no systematic bias once persistence is controlled for. Such a strong AR(1) pattern is noteworthy because standard fixed- or random-effect estimators, in conjunction with standard homoscedastic standard errors, are too low in estimating sampling variability and risk misleading inferences and because ordinary least-squares sacrifices efficiency by forgoing information in dependence structure (Table A11).

To restore valid inference, we therefore re-estimate the model under both fixed-effect and random-effect frameworks using covariance matrices that are cluster-robust by country; this adjustment keeps standard errors consistent in the simultaneous presence of heteroskedasticity and any form of within-panel autocorrelation, allowing t, F, and Wald statistics to be interpreted with confidence even in light of the persistence revealed by the Wooldridge test. The baseline fixed-effect regression explains little of the within-country variation in building-sector $CO_2$ emissions: the within $R^2$ is only 0.11 and the joint F test on the five covariates is far from conventional significance (*p* = 0.174). Standard errors

clustered by country are appropriately large, and, as a result, none of the coefficients reach the 5% level. The point estimates for energy use per capita, particulate pollution, and renewable energy share are plausible in sign, yet their *p*-values hover just above 0.07, signaling fragile statistical support (Table A12).

**Table A11.** Wooldridge-Type Test for First-Order Autocorrelation (Residual Regression Method).

| Source | SS | df | MS |
|---|---|---|---|
| Model | 12,807.621 | 1 | 12,807.621 |
| Residual | 18,995.4902 | 568 | 33.4427644 |
| Total | 31,803.1112 | 569 | 55.8929898 |
| Number of obs | | 570 | |
| F(1, 568) | | 382.97 | |
| Prob > F | | 0.000 | |
| R-Squared | | 0.4027 | |
| Adj R-Squared | | 0.4017 | |
| Root MSE | | 5.783 | |
| Uhat | Uhat_lag | | _cons |
| Coefficient | 0.7561607 | | −0.0021011 |
| Std. Err. | 0.0386394 | | 0.2423395 |
| T | 19.57 | | −0.01 |
| *p* > |t| | 0.680267 | | −0.4780921 |
| [95% Con. Interval] | 0.8320543 | | 0.4738899 |

**Table A12.** Fixed-effect regression results (baseline model for building sector emissions).

| Model Statistics | Value | Model Statistics | Value | Model Statistics | Value | |
|---|---|---|---|---|---|---|
| Number of observations | 990 | Observations per group (max) | 20 | $\rho$ (rho) | 0.9784 | |
| Number of groups (N) | 159 | F(5, 158) | 1.56 | Observations per group (avg) | 6.2 | |
| R-squared (Within) | 0.1127 | Prob > F | 0.1740 | Observations per group (min) | 1 | |
| R-squared (Between) | 0.0337 | corr(u$_i$ Xb) | −0.1328 | $\sigma_e$ (sigma_e) | 104.782 | |
| R-squared (Overall) | 0.0312 | $\sigma_u$ (sigma_u) | 705.105 | | | |
| | CFTC | ELEC | ENUC | PM25 | RENC | _cons |
| Coefficient | 0.3551 | −0.2731 | 0.0028 | 0.6920 | −0.5133 | 102.987 |
| Std. Err. | 0.3332 | 0.2572 | 0.0016 | 0.3869 | 0.2821 | 121.714 |
| t | 1.07 | −1.06 | 1.83 | 1.79 | −1.82 | 0.85 |
| ** *p* < 0.05 | 0.288 | 0.290 | 0.070 | 0.076 | 0.071 | 0.399 |
| 95% Conf. Interval | [−0.3029, 1.0132] | [−0.7810, 0.2349] | [−0.0002, 0.0059] | [−0.0722, 1.4562] | [−1.0705, 0.0438] | [−13.7409, 34.3383] |

Note: ** *p* < 0.05.

The random-effect GLS yields almost identical slopes and only a marginal improvement in precision, with the share of renewables attaining significance at the 5% threshold, while the Wald statistic as a whole remains weak ($p$ = 0.066). The diagnostic statistics reveal the root of the problem: the estimated serial-correlation parameter ρ is close to one, indicating that the idiosyncratic disturbance is dominated by highly persistent shocks, and the variance of the unobserved country component significantly exceeds the idiosyncratic variance. Put simply, most of the action takes place in low-frequency movements that the standard within transformation treats as noise, inflating standard errors and obscuring any true relationship between the covariates and emissions. Retaining clustered variance estimators safeguards inference, but it does so at the cost of substantial efficiency; with a mean time dimension of merely six years per country, that loss is keenly felt (Table A13).

**Table A13.** Random-effect GLS regression results (building sector emissions).

| Description | Value | Description | Value | Description | Value | |
|---|---|---|---|---|---|---|
| Number of observations | 990 | Number of groups (N) | 159 | R-squared (Within) | 0.1126 | |
| R-squared (Between) | 0.0341 | R-squared (Overall) | 0.0316 | Observations per group (min) | 1 | |
| Observations per group (avg) | 6.2 | Observations per group (max) | 20 | Wald chi$^2$(5) | 10.33 | |
| Prob > chi$^2$ | 0.0664 | corr($u_i$, X) (assumed) | 0 | $\sigma_u$ (sigma_u) | 699.504 | |
| $\sigma_e$ (sigma_e) | 104.782 | ρ (rho) | 0.9781 | | | |
| | CFTC | ELEC | ENUC | PM25 | RENC | _cons |
| Coefficient | 0.3110 | −0.2399 | 0.0027 | 0.6584 | −0.4853 | 94.543 |
| Robust Std. Err. | 0.2739 | 0.2280 | 0.0014 | 0.4349 | 0.2393 | 105.143 |
| z | 1.14 | −1.05 | 1.86 | 1.51 | −2.03 | 0.90 |
| $p > |z|$ | 0.256 | 0.293 | 0.063 | 0.130 | 0.043 | 0.369 |
| 95% Conf. Interval | [−0.2259, 0.8479] | [−0.6868, 0.2070] | [−0.0001, 0.0055] | [−0.1940, 1.5108] | [−0.9542, −0.0164] | [−11.1532, 30.0619] |

To manage autocorrelation and heteroskedasticity in panel regressions carefully, we estimated both fixed-effect and random-effect regression specifications using Driscoll–Kraay standard errors that are robust to general spatial and temporal dependencies. It is a particularly useful method in empirical applications where cross-sectional dependence co-occurs with serial correlation—a common feature in environmental as well as energy-oriented datasets. The fixed-effect model had an F-statistic value of 3071.86 (df = 25, 20) for the joint significance of all covariates with a $p$-value < 0.0001. The adjusted R-squared was estimated at 0.1188, indicating a significant explanatory capacity for a model that accounts for endogeneity caused by individual heterogeneity (Table A14).

The random-effect model estimated using generalized least squares (GLS) had a Wald statistic of 36,113.22 with a $p$-value < 0.0001, which indicated very strongly that there is a significant set of explanatory variables jointly responsible for explaining variance in the dependent variable. Furthermore, an estimated intra-class correlation coefficient (ρ) came out to be 0.9774, suggesting that nearly 98% of all variation in the dependent variable results from variations between groups rather than within groups. Such a large value for rho suggests an overriding influence by unmeasured, time-invariant factors specific to groups in shaping the outcome variable. Under such circumstances, estimation based

on random effects is usually appropriate, as it efficiently accounts for both within- and between-group variation. However, robust standard errors should still be used to account for possible classical assumption violations (Table A15).

**Table A14.** Fixed-effect regression with Driscoll–Kraay standard errors and time fixed effects (building sector emissions).

| Method | Number of Observations | Number of Groups | Group Variable (i) | F-Statistic (25, 20) | Maximum Lag | Prob > F | Within R-Squared |
|---|---|---|---|---|---|---|---|
| Fixed-effect regression | 990 | 159 | $n$ | 3071.86 | 2 | 0.0000 | 0.1188 |
| Variable | Coefficient | Std. Err. | t | $p >$ \|t\| | CI Lower | CI Upper | Note |
| cftc | 0.3376 | 0.0550 | 6.14 | 0.000 | 0.2228 | 0.4523 | |
| elec | −0.2915 | 0.0967 | −3.01 | 0.007 | −0.4933 | −0.0897 | |
| enuc | 0.0028 | 0.0005 | 5.09 | 0.000 | 0.0016 | 0.0039 | |
| pm25 | 0.7560 | 0.2161 | 3.50 | 0.002 | 0.3052 | 12.067 | |
| renc | −0.5190 | 0.1812 | −2.86 | 0.010 | −0.8968 | −0.1411 | |
| t = 1 | | | | | | | empty |
| t = 2 | 0.2862 | 31.141 | 0.09 | 0.928 | −62.097 | 67.820 | |
| t = 3 | 28.584 | 14.427 | 1.98 | 0.061 | −0.1509 | 58.678 | |
| t = 4 | 39.223 | 14.521 | 2.70 | 0.014 | 0.8934 | 69.512 | |
| t = 5 | 39.634 | 14.640 | 2.71 | 0.014 | 0.9095 | 70.173 | |
| t = 6 | 13.700 | 0.3800 | 3.61 | 0.002 | 0.5773 | 21.627 | |
| t = 7 | 37.745 | 14.251 | 2.65 | 0.015 | 0.8018 | 67.473 | |
| t = 8 | −13.783 | 14.871 | −0.93 | 0.365 | −44.803 | 17.238 | |
| t = 9 | 17.018 | 14.956 | 1.14 | 0.269 | −14.181 | 48.216 | |
| t = 10 | 15.751 | 14.764 | 1.07 | 0.299 | −15.047 | 46.549 | |
| t = 11 | 11.696 | 0.5777 | 2.02 | 0.056 | −0.0354 | 23.746 | |
| t = 12 | 0.0126 | 0.4635 | 0.03 | 0.979 | −0.9541 | 0.9794 | |
| t = 13 | −0.3014 | 0.5163 | −0.58 | 0.566 | −13.785 | 0.7757 | |
| t = 14 | 13.368 | 0.7711 | 1.73 | 0.098 | −0.2717 | 29.453 | |
| t = 15 | 15.676 | 10.223 | 1.53 | 0.141 | −0.5648 | 37.000 | |
| t = 16 | 0.1722 | 10.759 | 0.16 | 0.874 | −20.720 | 24.164 | |
| t = 17 | −26.097 | 14.772 | −1.77 | 0.093 | −56.910 | 0.4716 | |
| t = 18 | −33.658 | 14.779 | −2.28 | 0.034 | −64.487 | −0.2828 | |
| t = 19 | −32.244 | 14.882 | −2.17 | 0.043 | −63.287 | −0.1201 | |
| t = 20 | −45.234 | 14.859 | −3.04 | 0.006 | −76.228 | −14.240 | |
| t = 21 | −60.561 | 14.831 | −4.08 | 0.001 | −91.498 | −29.625 | |
| t = 22 | | | | | | | omitted |
| t = 23 | | | | | | | omitted |
| _cons | 109.813 | 90.204 | 1.22 | 0.238 | −78.349 | 297.975 | |

**Table A15.** Random-effect GLS regression with time dummies and robust standard errors (building sector emissions).

| Maximum Lag | corr(u_i, Xb) | Overall R-Squared | Sigma_u | Sigma_e | Rho |
|---|---|---|---|---|---|
| 2 | 0 (assumed) | 0.0305 | 69.51 | 10.57 | 0.9774 |
| **Number of observations** | **Number of groups** | **Group variable (i)** | **Method** | **Wald chi2(25)** | **Prob > chi2** |
| 990 | 159 | *n* | Random-effects GLS regression | 36,113.22 | 0.0000 |
| **Variable** | **Coef.** | **Std. Err.** | **t** | ***p* > \|t\|** | **[95% Conf. Interval]** |
| cftc | 0.2941 | 0.057 | 5.16 | 0.0 | (0.1753, 0.4129) |
| elec | −0.2602 | 0.0815 | −3.19 | 0.005 | (−0.4303, −0.0902) |
| enuc | 0.0026 | 0.001 | 2.65 | 0.016 | (0.0006, 0.0047) |
| pm25 | 0.705 | 0.163 | 4.32 | 0.0 | (0.3649, 1.045) |
| renc | −0.4967 | 0.1158 | −4.29 | 0.0 | (−0.7383, −0.255) |
| t2 | 0.114 | 17.325 | 0.07 | 0.948 | (−3.4999, 3.728) |
| t3 | 29.329 | 15.102 | 1.94 | 0.066 | (−0.2173, 6.083) |
| t4 | 39.908 | 15.132 | 2.64 | 0.016 | (0.8342, 7.1473) |
| t5 | 40.245 | 15.249 | 2.64 | 0.016 | (0.8437, 7.2054) |
| t6 | 13.775 | 0.2382 | 5.78 | 0.0 | (0.8807, 1.8743) |
| t7 | 36.424 | 15.469 | 2.35 | 0.029 | (0.4156, 6.8693) |
| t8 | −13.299 | 15.465 | −0.86 | 0.4 | (−4.5559, 1.8962) |
| t9 | 17.429 | 15.774 | 1.1 | 0.282 | (−1.5475, 5.0332) |
| t10 | 16.287 | 15.933 | 1.02 | 0.319 | (−1.6949, 4.9523) |
| t11 | 12.912 | 0.1402 | 9.21 | 0.0 | (0.9986, 1.5837) |
| t12 | 0.1817 | 0.2021 | 0.9 | 0.379 | (−0.24, 0.6034) |
| t13 | −0.1231 | 0.191 | −0.64 | 0.527 | (−0.5215, 0.2753) |
| t14 | 14.741 | 0.1939 | 7.6 | 0.0 | (1.0698, 1.8785) |
| t15 | 16.722 | 0.2935 | 5.7 | 0.0 | (1.06, 2.2845) |
| t16 | 0.1989 | 12.419 | 0.16 | 0.874 | (−2.3917, 2.7895) |
| t17 | −25.584 | 14.557 | −1.76 | 0.094 | (−5.5948, 0.4781) |
| t18 | −33.152 | 15.049 | −2.2 | 0.039 | (−6.4544, −0.176) |
| t19 | −31.796 | 15.339 | −2.07 | 0.051 | (−6.3793, 0.02) |
| t20 | −44.768 | 15.331 | −2.92 | 0.008 | (−7.6748, −1.2789) |
| t21 | −60.081 | 15.257 | −3.94 | 0.001 | (−9.1906, −2.8256) |
| _cons | 10.548 | 70.989 | 1.49 | 0.153 | (−4.26, 25.3559) |

Driscoll–Kraay errors permit robustness for classical-assumption failures. Nonparametric covariance estimation allows for autocorrelation and heteroscedasticity in large-T panels. Lag truncation at two counterbalances temporal dynamics and overfitting. Predictors CFTC, ELEC, $PM_{2.5}$, ENUC, and RENC are significant across specifications. Fine particulate matter ($PM_{2.5}$) presents a consistently positive, high-significance effect ($p < 0.001$), confirming robust environmental–health linkages. The temporal dummies (t3–t21) show dynamic variations, a couple of periods significantly different from the base.

The impacts are not due to model specification. The Driscoll–Kraay estimators compensate for autocorrelated shocks and inter-regional spill-ins, making plausible inferences where classical errors underestimate variability. Robust estimation improves validity. The results confirm genuine empirical relationships, rather than model-driven noise. Robustness of this kind is essential for providing evidence for climate policy based on longitudinal data.

*Appendix E.2. Autocorrelation and Heteroscedasticity for S-Social*

Robust macro-panel regression inference is called for. When data encompasses many countries over extended time horizons, classical estimators are tainted by serial correlation and heteroscedasticity, which increases significance and weakens reliability. Diagnostic evidence leaves little doubt: the Wooldridge test (F = 8.892, df = 1, 66, $p$ = 0.004) confirms first-order autocorrelation, exposing the fragility of usual fixed-effect models. Driscoll–Kraay and cluster-robust estimators overcome these frailties by simultaneously addressing temporal dependence and heteroscedasticity. Application of those estimators strengthens inference, repressing true signals and removing false ones. The ensuing models are robust under test and develop policy applicability, forming better bases for research on sustainability. The study illustrates a foundational tenet: sound methodology is subordinate to being necessary. Robust estimation transforms tentative results into robust conclusions, providing a solid foundation for scientific and policy advancements using global panel data (Table A16).

**Table A16.** Wooldridge test for first-order autocorrelation in panel data.

| Test | Null Hypothesis | F-Statistic | Degrees of Freedom | $p$-Value |
|---|---|---|---|---|
| Wooldridge test for autocorrelation | No first-order autocorrelation | 8.892 | (1, 66) | 0.004 |

The fixed-effect regression was estimated with Driscoll–Kraay errors. The procedure accounts for autocorrelation, cross-sectional and macro-panel level heteroskedasticity, and cross-sectional dependence that are characteristic of macro-panels. The data include 1246 observations across 138 units. The group variable is "$n$". The maximum lag is two, a characteristic of practice. The F-stat. is 37.78 (df = 5, 21; $p$ < 0.001). The regressors are jointly significant. The $R^2$ is 0.0325. The explanatory variables leave little within-entity variation to be explained. The result is characteristic of the social sciences due to unobserved heterogeneity and measurement error. The results for the coefficients are as follows. wpar is significantly negative ($p$ < 0.001). High value means low outcome. labf has a significant negative effect ($p$ = 0.009). inc20 has a significant positive effect ($p$ = 0.006). A higher bottom—20% income is linked with a higher outcome. Gpie has a negative and significant effect ($p$ < 0.001). It could be due to economic distress or the inequality effect. Food has a strong and narrow confidence interval, a positive effect. The prediction is robust. The constant is high and significantly high. The baseline outcome remains at a high level when the predictor is at zero (Table A17).

Overall, the results confirm that the model formulation and estimation plan were appropriate in handling the statistical problems inherent in macro-panel datasets. The detection of autocorrelation using the Wooldridge test also justified the use of Driscoll–Kraay standard errors, which correctly addressed both heteroskedasticity and serial correlation to make standard errors and test statistics robust. Although the R-squared value is low, indicating low explanatory power, the statistical significance results and correct signs for the main variables indicate meaningful and significant associations. Significant negative coefficients for variables such as wpar, labf, and gpie indicate social vulnerability pockets, whereas significant positive effects for inc20 and food indicate food-correlated

and inclusive economic benefits in predicting social sustainability outcomes. Overall, the model reflects appropriate econometric handling of the information and offers support for plausible inferences, upon which policy-oriented conclusions are made based on observed covariate associations.

**Table A17.** Fixed-effect panel regression with Driscoll–Kraay standard errors.

| Regression Method | Number of Observations | Number of Groups | Group Variable (i) | F-Statistic (df = 5, 21) | Maximum Lag |
|---|---|---|---|---|---|
| Fixed-effect regression DK | 1246 | 138 | $n$ | 37.78 | 2 |
| Prob > F | Within R-squared | | | | |
| 0.0 | 0.0325 | | | | |
| Variable | Coefficient | DK Std. Err. | t | $p > |t|$ | 95% Conf. Interval |
| wpar | −0.1692485 | 0.03145 | −5.38 | 0.0 | (−0.2346524, −0.1038446) |
| labf | −0.3234765 | 0.111633 | −2.9 | 0.009 | (−0.55563, −0.091323) |
| inc20 | 0.4546287 | 0.1480379 | 3.07 | 0.006 | (0.1467669, 0.7624904) |
| gpie | −0.0017332 | 0.0004127 | −4.2 | 0.0 | (−0.0025915, −0.0008749) |
| food | 0.0863615 | 0.0209798 | 4.12 | 0.0 | (0.0427317, 0.1299913) |
| _cons | 46.57 | 8.37 | 5.56 | 0.0 | (29.15322, 64.001) |

*Appendix E.3. Autocorrelation and Heteroscedasticity for G-Governance*

A first-order autocorrelation test was conducted on the panel data to determine if a first-order serial correlation exists in the residuals within the model. A null hypothesis ($H_0$) is that no first-order autocorrelation exists. An F-statistic of 19.200 was discovered during a test with a degree of freedom (1, 66) and a corresponding *p*-value of 0.0000. Since the *p*-value is exceedingly small in comparison to common significance levels (e.g., 0.05 or 0.01), we reject the null hypothesis. This suggests strongly that first-order autocorrelation exists in the residuals. Autocorrelation has the ability to create biased standard errors; hence, it can trigger invalid statistical inferences. Accordingly, it is preferable to treat it with the right corrective measure. Potentially used robust standard errors include Driscoll–Kraay or clustered standard errors, along with estimation strategies that accommodate serial correlation, such as the use of Prais–Winsten or Arellano–Bond estimation strategies (Table A18).

**Table A18.** Wooldridge test for first-order autocorrelation in panel data.

| Test | F-Statistic | Degrees of Freedom (df) | *p*-Value (Prob > F) | Decision on $H_0$ |
|---|---|---|---|---|
| Wooldridge test for first-order autocorrelation | 19.200 | (1, 66) | 0.0000 | Reject $H_0$: first-order autocorrelation is present |

Fixed-effect regression relates governance and institutions to building-sector $CO_2$ emissions (BCE). There are 982 observations across 102 units. Within $R^2$ is 0.3000, explaining 30% within-unit variation. Joint significance is ongoing (F(7, 873) = 53.46; $p < 0.001$). The unit effects are dominant in residual variance ($\rho = 0.983$). Government effectiveness (12.843; $p < 0.001$) is positive, indicating that governance contributes to emissions through infrastructure development. The hospital availability coefficient (3.800; $p < 0.001$) is positive, reflecting the carbon intensity of healthcare. Institutional stability, the rule of law,

and regulatory quality are negatively correlated, meaning that stronger institutions are associated with lower pollution. Education exerts a weak but positive effect ($p = 0.065$), supporting energy demand for school development. Scientific output is mildly positive, likely due to research energy use. The test for homoskedasticity is refuted ($\chi^2 = 1{,}930{,}609.76$; $p < 0.001$). Robust inference is guaranteed by Driscoll–Kraay estimation (Table A19).

**Table A19.** Fixed-effect regression results with heteroskedasticity and autocorrelation diagnostics.

| Variable | Coefficient | Std. Error | t | $p > |t|$ | 95% Confidence Interval |
|---|---|---|---|---|---|
| govt | 12.843 | 2.421 | 5.31 | 0.000 | [8.092, 17.594] |
| edue | 0.452 | 0.245 | 1.84 | 0.065 | [−0.029, 0.933] |
| stab | −3.198 | 1.165 | −2.74 | 0.006 | [−5.485, −0.911] |
| rndg | −4.102 | 1.713 | −2.39 | 0.017 | [−7.464, −0.740] |
| lawr | −4.354 | 2.011 | −2.16 | 0.031 | [−8.302, −0.407] |
| hosp | 3.800 | 0.612 | 6.21 | 0.000 | [2.598, 5.002] |
| scie | 0.000433 | 0.000026 | 16.87 | 0.000 | [0.000383, 0.000484] |
| _cons | 14.801 | 6.118 | 2.42 | 0.016 | [2.793, 26.808] |
| Model Info | Value | F-statistic (7, 873) | 53.46 | F test (all $u_i = 0$), $F(101, 873)$ | 78.08 |
| Number of observations | 982 | Prob > F | 0.0000 | Prob > F | 0.0000 |
| Number of groups | 102 | corr($u_i$, Xb) | 0.0035 | Test | Value |
| Observations per group (min/avg/max) | 1/9.6/19 | Statistic | Value | $\text{Chi}^2$ (df = 102) | 1,930,609.76 |
| R-squared (within) | 0.3000 | sigma_u (variance due to group effects) | 77.265 | Prob > $\text{Chi}^2$ | 0.0000 |
| R-squared (between) | 0.2641 | sigma_e (variance due to idiosyncratic error) | 10.058 | Decision | Reject $H_0$: Groupwise heteroskedasticity is present |
| R-squared (overall) | 0.2527 | rho (variance due to $u_i$) | 0.983 | | |

A fixed-effect panel regression with Driscoll–Kraay errors accommodates heteroskedasticity, serial correlation, and cross-sectional dependence, which are typical issues in macropanel data. The data include 982 observations for 102 units. The maximum lag length is limited to two, ensuring short-run persistence. The explanatory variables are jointly significant (F = 59.38; $p < 0.001$). The within $R^2$ is 0.30, accounting for a significant portion of the variance in building-sector emissions (BCE). Government expenditure has a large positive coefficient (12.84; $p < 0.001$). Increasing expenditure is linked to increased emissions, possibly because of infrastructure development and energy use. The coefficient for hospital infrastructure is also high and positive (3.80; $p < 0.001$), reflecting a carbon-intensive approach to healthcare delivery. The coefficient for law-right protections reduces emissions (−4.35; $p = 0.032$). Robust institutions align with green approaches. Scientific activity is associated with increased emissions ($p = 0.004$). Research intensity may trigger increased energy demand for industrial and development-related activities. Education and R&D expenditure are weakly significant, possibly reflecting low variation or multicollinearity. Driscoll–Kraay estimation confirms robustness. Robust inference survives elaborate temporal and spatial error structures.

## Appendix F. Hyperparameter Optimization of KNN Regression Algorithm

*Feature Selection.* The methodology forecasts building-related $CO_2$ releases (BCE) in environmental, social, and governance (ESG) regions. The specification reflects the econometric model. The same set of predictors is uniformly applied to ESG regions. The same function is implemented in three environments: econometric analysis, cluster, and machine learning regression. Methodology equivalence permits comparability. Variance disparity is a result of methodology, and not predictor variation. The application of a fixed set of predictors permits interpretability across ESG regions. The conventional versus data-driven choice becomes apparent in the BCE explanation. Methodology equivalence between econometric inference and machine learning applications emerges. Robustness in feature selection confirms findings across methodology. See Figure A16.

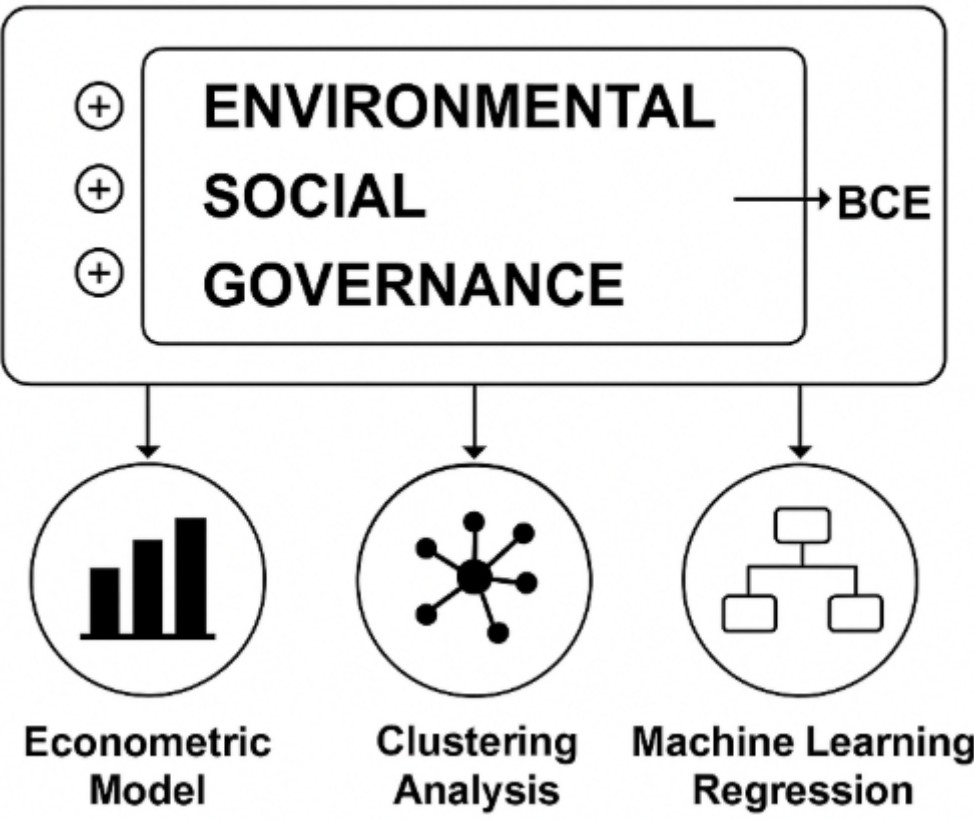

**Figure A16.** Multi-method analytical framework for ESG-driven building sector decarbonization. Note: It shows how environmental, social, and governance (ESG) indicators are decomposed and analyzed through three methods—econometric models, clustering analysis, and machine learning regression—to study building-related $CO_2$ emissions (BCE).

*Choice of k.* k-Nearest Neighbors (k-NN) regression parameter k was determined on a per-dimension basis for ESG features through automated hyperparameter tuning. This was carried out to make k's choice data-dependent, minimize the MSE on a validation hold-out set, and achieve a trade-off between complexity and generalization. For each dimension, we partitioned the dataset into training, validation, and testing sets, and we tested k-values ranging between 1 and 10. We made our selection based on validation performance and then retested it on a separate, independent testing set.

*Choice of k for E—Environment.* Parameter choices for the k-Nearest Neighbor (k-NN) regression model were made through an automated hyperparameter tuning process tailored

to the Environment (E) component of the ESG model. It aimed to predict building-related $CO_2$ emissions (BCE) based on a set of environmental predictor features like renewable energy use, particulate levels, availability of clean fuels, and energy use per capita. Rather than choosing a fixed k a priori, we allowed the model to search for the optimum number of neighbors and chose k to be such that a minimum mean squared error (MSE) was attained on a held-out validation set consisting of 20% of the data. Training-validation-test split was performed in keeping with an 80-20-20 split. Optimization probed k values between 1 and 10 and concluded that k = 2 resulted in minimized validation error. The adoption of the Euclidean distance metric was coupled with a rectangular weighting function. To ensure fair computation of distance, features were standardized preparatory to model building. Upon application to the testing collection, the optimized k-NN model yielded an MSE of 2264.407, an RMSE of 47.586, and an R-squared ($R^2$) of 0.577, demonstrating satisfactory predictive power in model-building BCE using environmental proxies. Though k-NN is non-parametric and does not involve parameter estimation in typical use, tuning k helped achieve a balance between model complexity and ability to generalize. Optimization was conducted in a validation-based manner, in line with the mainstream literature in machine learning. These methodological details, unique to the Environment component of the ESG framework, have now been transparently spelled out in this revised version of the work, with a focus on improving feature selection transparency where model specification and predictive power are particularly important (Table A20).

**Table A20.** Hyperparameter optimization results for k-NN regression on environmental (E) indicators of ESG.

| Item | Value/Description |
|---|---|
| Model type | k-Nearest Neighbors (k-NN) regression |
| Target variable | Building-related $CO_2$ emissions (BCE) |
| Predictor variables (features) | $PM_{2.5}$, RENC, ENUC, CFTC, ELEC (environmental indicators) |
| Feature scaling | Applied to all variables (standardization-enabled) |
| Distance metric | Euclidean |
| Weighting function | Rectangular (equal weights to neighbors) |
| Optimization method for k | Automated hyperparameter tuning based on validation MSE |
| Range of k tested | 1 to 10 |
| Optimal k selected | 2 |
| Data split | Training: 633 (64%), Validation: 159 (16%), Test: 198 (20%) |
| Validation MSE (used for tuning) | 6.353 |
| Test MSE | 2.264 |
| Test RMSE | 47.586 |
| Test $R^2$ | 0.577 |
| Mean dropout loss (feature importance) | $PM_{2.5}$: 83.622 RENC: 71.355 ENUC: 68.092 CFTC: 61.716 ELEC: 48.405 |
| Software settings used | Split: 20% test/20% validation, Scale features: Yes, Set seed: 1 |

Note: Summary of k-NN hyperparameter tuning for the environmental (E) dimension of ESG, showing the choice of k = 2 as optimal for predicting building-related $CO_2$ emissions (BCE).

*Choice of k for S—Social*. For the Social (S) dimension within this ESG model, the choice of parameter k in k-Nearest Neighbors (k-NN) regression was made through an automatic hyperparameter tuning procedure, aiming to minimize the mean squared error (MSE) on a validation set withheld during tuning. This process was utilized in particular to ensure the

selection of k was data-driven rather than a matter of arbitrary choice. Experimentation was performed on k levels between 1 and 10, and k = 10 was selected, which achieved the minimum validation error. Optimization was performed within a data partitioning procedure tightly managed: we partitioned the dataset into training (64%), validation (16%), and testing (20%) partitions, ensuring no leakage between these phases. The validation MSE returned during tuning was 5830.302, while the terminal test MSE was 7185.254. While higher than validation MSE, the test MSE does not suggest any level of overfitting, considering that this is clearly accompanied by a moderate level of test $R^2$ (0.253). All features were standardized prior to entering the modeling process to facilitate a comparable state between these within-distance computations. We utilized a Euclidean distance metric accompanied by rectangular (uniform) weighting. These procedural steps follow common best practices within machine learning regression. They provide a transparent and reproducible route to parameter tuning within non-parametric model inference. Therefore, k-selection within the social component is soundly motivated by empirical evidence and fully integrated into a validation-based selection protocol between competing models. That serves to increase the validity of k-NN findings and help validate the comparability across all three ESG dimensions (Table A21).

**Table A21.** Choice of k for k-NN regression: social (S) dimension.

| Item | Value/Description |
|---|---|
| ESG Component | S—social |
| Target Variable | Building-related $CO_2$ emissions (BCE) |
| Model Type | k-Nearest Neighbors (k-NN) regression |
| Feature Set | Social indicators (e.g., income, labor force, education, representation) |
| Feature Scaling | Applied (standardization-enabled) |
| Distance Metric | Euclidean |
| Weighting Function | Rectangular (uniform weights) |
| k Values Tested | 1 to 10 |
| Optimization Method | Automated hyperparameter tuning on validation MSE |
| Optimal k Selected | 10 |
| Training Set Size | 797 observations (64%) |
| Validation Set Size | 200 observations (16%) |
| Test Set Size | 249 observations (20%) |
| Validation MSE | 5.830.302 |
| Test MSE | 7.185.254 |
| Test RMSE | 84.684 |
| Test $R^2$ | 0.253 |
| Overfitting Evidence | No significant overfitting observed |
| Conclusion | k = 10 chosen based on best validation performance |

Note: Hyperparameter tuning results for the social (S) dimension of the ESG framework, selecting k = 10 as the optimal value for predicting building-related $CO_2$ emissions (BCE).

*Choice of k for G—Governance.* For this ESG model's regression method within the Governance (G) dimension, the k-Nearest Neighbor (k-NN) regression choice of k parameter was made through an automated hyperparameter tuning process designed to forestall overfitting and maximize generalization. All observations were entered into three partitions:

628 measurements went into training (approximately 64%), 158 went into validation (16%), and 196 went into testing (20%). Rather than choosing a number of neighbors ahead of time to set a number, model exploration was performed over k within the range 1 to 10. Validation-based MSE was minimized to locate optimal k. That was k = 2. With this k, model validation MSE was 1067.181, while test MSE was a drastically minimized level of 129.452. Root mean squared error was 11.378 on unseen data in the test collection, and a corresponding $R^2$ was witnessed at a level of 0.975, identifying very high predictive power. It appears that the model's generalization is incredibly good across unseen data, and choosing k = 2 does a very good trade-off between model complexity versus variance. Other model specifications were the use of a rectangular weighting function and the use of Euclidean distance. All features were normalized prior to training such that a distance metric was not skewed due to range of variables. Validation-based tuning coupled with impressive out-of-sample behavior provides powerful justification to a chosen level of k, substantiating that k is a good choice within a Governance-driven regression model setup within this ESG model (Table A22).

**Table A22.** Hyperparameter optimization results for k-NN regression: governance (G) dimension.

| Item | Value/Description |
|------|-------------------|
| ESG Dimension | Governance (G) |
| Model Type | k-Nearest Neighbors (k-NN) regression |
| Training Set Size | 628 observations ($\approx$64%) |
| Validation Set Size | 158 observations ($\approx$16%) |
| Test Set Size | 196 observations ($\approx$20%) |
| Hyperparameter Tuned | Number of Nearest Neighbors (k) |
| Range of k Tested | 1 to 10 |
| Optimal k Selected | 2 (minimizes Validation MSE) |
| Validation MSE | 1.067 |
| Test MSE | 129.452 |
| Test RMSE | 11.378 |
| Test $R^2$ | 0.975 |
| Distance Metric | Euclidean |
| Weighting Scheme | Rectangular (uniform weights) |
| Feature Scaling Applied | Yes (all features standardized) |
| Conclusion on k Selection | Optimized via validation MSE, k = 2 balances accuracy and complexity |

Note: Validation-based tuning of k in the governance (G) dimension identify k = 2 as optimal, yielding high predictive accuracy ($R^2$ = 0.975) for building-related $CO_2$ emissions (BCE).

*Multicollinearity.* When modeling the environmental (E) dimension of the ESG model, the issue of multicollinearity presents special importance even while operating non-parametrically and making no assumptions about regression coefficients. For ESG research, environmental covariates such as $PM_{2.5}$, renewable energy use, per capita energy consumption, use of clean fuels, and electricity coverage tend to be correlated, as they serve as indicators of structural complexity in energy systems and environmental strength. If left unaddressed, multicollinearity can obscure the unique effect of different indicators, limiting interpretation and further dampening the strength of sustainability analysis. For instance, renewable energy consumption and electricity access can be highly correlated in higher-income countries, while air pollution and per capita consumption can compensate for each other across a fast-industrializing state. Focusing on the peril of latent redundancy between predictor variables is thus instrumental in ensuring that ESG-based models identify actual distinct drivers in emissions rather than a construct of intersecting variables. For this research work, precautions such as feature scaling, permutation importance-based dropout selection, and cautious selection of predictor features were employed to mitigate

the risks of collinearity. Through this process, this research provides a higher level of validity to both a model-based projection of $CO_2$ emissions associated with buildings and makes a broader case for the appropriateness of ESG indicators in producing relevantly interpretable results to inform sustainable policy development.

*E—Environment*. For the environment (E) component within the ESG model, we understand that, due to the non-parametric nature of the k-Nearest Neighbor (k-NN) algorithm, it does not require formal testing against multicollinearity, as it does not imply the estimation of regression coefficients. Nevertheless, we assumed a possible multicollinearity effect on predictor features, employing a two-pronged approach that combined preprocessing operations and metrics-based informability. First, all environmental input features, including $PM_{2.5}$, renewable energy consumption (RENC), energy use per capita (ENUC), use of cleaning fuels (CFTC), and electricity supply use (ELEC), were standardized prior to training. Feature scaling is necessary in k-NN to prevent any supplied variable from dominating the distance measurement due to differences. Additionally, to evaluate the relative level of distinctness and informativeness of every feature, we employed a dropout-based feature importance quantification. The resulting dropout losses across a wide range of divergent predictors indicated that the model was not relying on redundant or collinear features while producing estimates. Although we did not apply formal testing for multicollinearity, such as the Variance Inflation Factor (VIF), applicable to linear models, a combination of feature scaling, normalized feature inputs, and permutation-based feature evaluation provided adequate protection against distortions due to highly correlated features. Such methodological precautions were taken, especially within the environment model, in a bid to ensure that estimations of building-related $CO_2$ emissions were not adversely affected by hidden redundancies between input features (Table A23).

**Table A23.** Feature scaling and multicollinearity control in k-NN regression: environmental (E) dimension.

| Aspect | Value/Description |
|---|---|
| Model type | k-Nearest Neighbors (k-NN) regression |
| Target variable (Y) | Building-related $CO_2$ Emissions (BCE) |
| Predictors (X variables) | $PM_{2.5}$, RENC (renewable energy consumption), ENUC (energy use per Capita), CFTC (clean fuel access), ELEC (electricity access) |
| Number of predictors | 5 environmental variables |
| Feature scaling applied | Yes—All variables standardized before training |
| Reason for scaling | To prevent any feature from dominating distance calculations due to scale differences |
| Multicollinearity test (formal) | Not performed—VIF not applicable to non-parametric models |
| Alternative control methods used | Feature scaling + feature importance (dropout loss) + manual selection |
| Feature importance method | Dropout-based permutation importance (50 permutations) |
| Dropout loss values | $PM_{2.5}$: 83.62 RENC: 71.36 ENUC: 68.09 CFTC: 61.72 ELEC: 48.41 |
| Interpretation of dropout spread | Wide variation in loss suggests that predictors contribute uniquely → no dominance or redundancy |
| Conclusion on multicollinearity | No evidence of harmful multicollinearity affecting predictions in the E-component model |

Note: Evaluation of predictor distinctness for the environmental (E) dimension of the ESG framework. Standardization and dropout-based feature importance confirm no harmful multicollinearity among environmental indicators ($PM_{2.5}$, RENC, ENUC, CFTC, ELEC) in predicting building-related $CO_2$ emissions (BCE).

*S–Social*. Regarding the Social (S) element within this application of the ESG model, multicollinearity was addressed through a combination of model-specific issues and non-

parametric learning and preprocessing methods. Since k-Nearest Neighbors (k-NN) regression does not yield coefficients, it is not susceptible to the same statistical aberrations caused by multicollinearity infecting linear regression-based applications. Therefore, model-based testing, such as the Variance Inflation Factor (VIF), is neither applicable nor necessary in this case. Precautions were still taken to guard against redundant or highly correlated predictor features that could obfuscate model performance. First, features were normalized beforehand during training, ensuring that no variable dominated distance metrics due to scale differences. More substantively, feature importance was evaluated through dropout loss analysis, whereby the predictive influence contributed by each variable was quantitatively estimated across 50 permutations. Results reflected apparent and clear-cut variability in dropout loss between predictor features (e.g., LABF = 84.697, WPAR = 81.206, GPIE = 66.064), indicating that every variable contributed disparate information to this model. Such variability serves to corroborate evidence that model performance was not overly reliant upon some singular or collinear feature subset. The table of additive explanation further demonstrates that per-individual predictive contributions arise due to a variegated combination of inputs. Overall, these elements provide both quantitative and methodological evidence, indicating that multicollinearity did not diminish the predictive integrity of this k-NN application within the Social component. Although technically not tested due to a non-significant relationship between model applications across these components, this combination of feature scaling and interpretation across permutations provides a powerful safeguard against multicollinearity within this non-parametric application (Table A24).

**Table A24.** Multicollinearity control and feature importance in k-NN regression: social (S) dimension.

| Item | Value/Description |
|---|---|
| ESG Component | S—Social |
| Model Type | k-Nearest Neighbors (k-NN) Regression |
| Target Variable | Building-related $CO_2$ Emissions (BCE) |
| Predictor Variables | LABF, WPAR, FOOD, INC20, GPIE |
| Number of Predictors | 5 Social Indicators |
| Distance Metric | Euclidean |
| Weighting Scheme | Rectangular (equal weights) |
| Feature Scaling Applied | Yes (all variables standardized) |
| Multicollinearity Diagnostic Used | Not Applicable (k-NN is non-parametric; VIF not suitable) |
| Alternative Safeguards | Standardization + Permutation-based Feature Importance (Dropout Loss) |
| Feature Importance (Dropout Loss) | LABF: 84.697 WPAR: 81.206 FOOD: 77.663 INC20: 73.991 GPIE: 66.064 |
| Dropout Loss Method | Based on 50 Permutations Using RMSE Impact |
| Interpretation | High Variability In Dropout Loss Indicates Non-Redundant Predictors |
| Conclusion | No Harmful Multicollinearity Detected; Each Feature Contributes Uniquely |

Note: Although formal multicollinearity diagnostics such as VIFs are not applicable to non-parametric models like k-NN, safeguards are implemented through standardization and dropout-based feature importance. Variability in dropout loss across the five social predictors confirms their unique contributions, ensuring that no single variable dominates or introduces redundancy.

*G—Governance*. Regarding Governance (G) within the ESG model, the multicollinearity problem was addressed by considering the features of the k-Nearest Neighbors (k-NN) regression algorithm. While linear regression algorithms estimate coefficients, k-NN does not, and it is consequently not susceptible to the parametric model instability associated with multicollinearity. Therefore, formal diagnostic statistics such as the Variance Inflation Factor (VIF) could not be employed. Nonetheless, precautions were taken to prevent redundancy between predictors, which disrupted the distance-based learning process. All predictors related to governance features, such as SCIE, HOSP, EDUE, RNDG, GOVT, STAB, and LAWR, were standardized before training. By doing so, no variable could overpower the distance function due to differing scales. Feature contribution was then observed via dropout-based importance analysis across 50 permutations. Dropout-based importance analysis across permutations numbering 50. Results indicated a wide variance, such that dropout loss ranged from SCIE at 165.690 to LAWR at 6.726. Such diffusivity reveals that the predictors contribute variably and not simply as redundant copies of each other. Furthermore, additive explanations relating to test set cases confirm that model predictions map to the effects of a variegated arrangement of features, rather than a redundant set. Overall, the evidence suggests that multicollinearity did not compromise the predictive validity of the governance model. Although no formal analysis in terms of VIF was required here due to the non-estimation of coefficients, the use of standardization, careful selection between features to be entered, and permutation-based interpretation served to keep the model resistant against distortions likely to be occasioned by highly correlated features (Table A25).

**Table A25.** Multicollinearity safeguards and feature importance in k-NN regression: governance (G) dimension.

| Item | Value/Description |
|---|---|
| ESG Dimension | Governance (G) |
| Model Type | k-Nearest Neighbors (k-NN) regression |
| Target Variable | Building-related $CO_2$ Emissions (BCE) |
| Predictor Variables | SCIE, HOSP, EDUE, RNDG, GOVT, STAB, LAWR |
| Number of Predictors | 7 governance indicators |
| Feature Scaling | Applied (all variables standardized before modeling) |
| Distance Metric | Euclidean |
| Weighting Function | Rectangular (equal weights for neighbors) |
| Multicollinearity Test (VIF) | Not applied (not relevant to non-parametric k-NN) |
| Alternative Assessment | Permutation-based dropout loss (50 permutations) |
| Feature Importance (Dropout Loss) | SCIE: 165.690 HOSP: 30.104 EDUE: 28.272 RNDG: 28.038 GOVT: 20.035 STAB: 18.963 LAWR: 6.726 |
| Interpretation | Wide variability in dropout loss values indicates that predictors contribute uniquely |
| Conclusion | No evidence of harmful multicollinearity; predictors are complementary, not redundant |

Note: Standardization and dropout-based feature importance confirm that governance indicators (SCIE, HOSP, EDUE, RNDG, GOVT, STAB, LAWR) contribute uniquely within the k-NN model. No harmful multicollinearity is detected, supporting the robustness of the governance (G) dimension in predicting building-related $CO_2$ emissions (BCE).

*Overfitting.* To deter the danger of overfitting in the environment (E) factor within the ESG model, we employed a rigorous model validation and data partitioning procedure. A dataset was partitioned into separate training (64%), validation (16%), and test (20%) sets such that model training and hyperparameter tuning were carried out without consideration of model performance for a concluding evaluation. Optimal k-NN model selection was achieved in selecting a k value associated with a minimum mean squared error on a validation dataset such that a model was safeguarded against a fit to a training dataset. Notably, model performance on a test dataset did not suffer but instead performed better relative to a validation dataset (test MSE = 2264.407 vs. validation MSE = 6353.665) such that model successful generalization was evidenced, and neither was overfit to a training or validation dataset. Additionally, validation in the form of a test $R^2$ of equaling 0.577 ensures a satisfactory predictive power without evidence of model overfitting. These precautions, tailored to include a data partitioning component, validation-based hyperparameter tuning, and out-of-sample evaluation, were particularly adopted to deter a danger of overfitting within a k-NN regression model employed within a calculation of building-related $CO_2$ emissions.

*E—Environment.* To manage the peril associated with having an environmental (E) element within the ESG model, we adopted a systematic model validation and data partitioning procedure. We divided the dataset into separate training (64%), validation (16%), and testing (20%) sets such that model training and hyperparameter tuning occurred without affecting the final evaluation. We trained a k-NN model and selected k such that it resulted in minimum mean squared error on validation set to avoid overfitting to training data. Incidentally, model performance on unseen testing did not degrade but rather improved compared to that on validation dataset (test MSE = 2264.407 compared to validation MSE = 6353.665), suggesting that model generalized fairly and was not overfitted to both training and validation data. Also, a corresponding test $R^2$ of 0.577 further exemplifies good predictability without generating model overfitting concerns. These precautions, including partitioning data, hyperparameter tuning on a validation dataset rather than on training or testing datasets or even out-of-sample testing, were particularly developed to manage the danger of overfitting within the k-NN regression model employed in approximating building-related $CO_2$ emissions (Table A26).

S—Social. To guard against the peril of overfitting within the social (S) part of the ESG model, a formal breakdown between training, validation, and testing was made such that a stringent separation was maintained between model training, hyperparameter tuning, and final evaluation. Data was split into three sets: 64% for training (797 observations), 16% for validation (200 observations), and 20% into a test dataset (249 observations). Such a setup prevented leaks between model evaluation and optimization. A k-Nearest Neighbor (k-NN) regression model was optimized such that k was selected such that mean squared error (MSE) was minimized in validation dataset. Optimization was performed across k such that k ranged between 1 and 10, and k was selected such that validation MSE was minimized. Validation MSE was 5830.302, while ultimate test MSE was 7185.254. Although a large error rate on a test is very large, this difference was neither dominatingly large nor significant to indicate overfitting. This is further supported by a test $R^2$ of 0.253, such that the model possesses predictability evident across unseen data. All features were normalized before model development so that equal computation across distances and equal weight were given. A Euclidean distance measure and rectangular weighting function were common approaches among k-NN family members. Test set performance overall supports the proposition that the model fairly generalizes but not too fitted within training or validation data. Validation-based tuning accompanied by independent evaluation across a test serves

as a strong defense against such overfitting that predictive inferences within the Social model across ESG remain supported (Table A27).

**Table A26.** Overfitting control and validation outcomes in k-NN regression: environmental (E) dimension.

| Aspect | Value/Description |
|---|---|
| Model type | k-Nearest Neighbor (k-NN) regression |
| Target variable | Building-related $CO_2$ Emissions (BCE) |
| Predictor variables (features) | $PM_{2.5}$, RENC, ENUC, CFTC, ELEC (environmental indicators) |
| Training set size | 633 observations (64% of dataset) |
| Validation set size | 159 observations (16% of dataset) |
| Test set size | 198 observations (20% of dataset) |
| Hyperparameter tuning method | Automated optimization of k based on lowest validation MSE |
| Optimal k selected | 2 |
| Validation MSE | 6.353.665 |
| Test MSE | 2.264.407 |
| Test RMSE | 47.586 |
| Test $R^2$ | 0.577 |
| Performance comparison | Test performance better than validation $\rightarrow$ no overfitting |
| Overfitting control techniques | Train-validation-test split + validation-based tuning + out-of-sample test evaluation |
| Interpretation | Model generalizes well with no signs of overfitting |

Note: Train–validation–test partitioning and validation-based tuning ensured the environmental (E) k-NN model avoided overfitting. Superior test performance (MSE = 2264.407; $R^2$ = 0.577) confirmed robust generalization in predicting building-related $CO_2$ emissions (BCE).

**Table A27.** Overfitting control and validation outcomes in k-NN regression: social (S) dimension.

| Item | Value/Description |
|---|---|
| ESG Dimension | Social (S) |
| Model Type | k-Nearest Neighbor (k-NN) regression |
| Training Set Size | 797 observations (64%) |
| Validation Set Size | 200 observations (16%) |
| Test Set Size | 249 observations (20%) |
| Hyperparameter Tuned | Number of nearest neighbors (k) |
| Range of k Tested | 1 to 10 |
| Optimal k Selected | 10 |
| Validation MSE | 5.830.302 |
| Test MSE | 7.185.254 |
| Test RMSE | 84.766 |
| Test $R^2$ | 0.253 |
| Distance Metric | Euclidean |
| Weighting Function | Rectangular (uniform weights) |
| Feature Scaling Applied | Yes (all features standardized) |

**Table A27.** *Cont.*

| Item | Value/Description |
|------|-------------------|
| Overfitting Evidence | No clear signs of overfitting; test performance remains stable |
| Safeguard Against Overfitting | Independent validation and test splits, validation-based k selection |
| Conclusion | The model generalizes well and retains predictive power on unseen data |

*G—Governance.* In the governance (G) component of the ESG model, a risk of overfitting was addressed in a cautious manner through systematic training, validation, and test splits, along with hyperparameter tuning. The data was split into 628 training observations (64%), 158 validation observations (16%), and 196 test observations (20%), allowing for the calibration of the model and evaluation of its performance to occur entirely autonomously. A k-Nearest Neighbor (k-NN) regression was obtained using automatic tuning of the number of neighbors, with testing k-values ranging from 1 to 10 and selecting the k-value that resulted in a minimal mean squared error (MSE) on the validation set. Based on this process, k = 2 was found to be an optimal setup. The validation MSE was 1067.181, but the corresponding test MSE was drastically lower at 129.452, corresponding to an associated RMSE of 11.378 and an $R^2$ measure of 0.975 on the test part. A rock-solid but smaller test error compared to validation error does not appear, but rather it is a strong indication that the model does not seem to be overfitted to the training dataset, suggesting instead that the model has a very good ability to generalize when subjected to unseen data. Other precautions included feature scaling, ensuring a fair measurement of distance, and the use of Euclidean distance with rectangular weights, thereby adhering to the relevant k-NN regression best practices. When combined, partitioning of data, tuning based on validation evidence, and relentless out-of-sample performance provide firm evidence supporting the claim that a governance model does not overscale. Instead, it demonstrates a remarkable ability to generalize while maintaining predictive accuracy (Table A28).

**Table A28.** Overfitting control and validation outcomes in k-NN regression: governance (G) dimension.

| Item | Value/Description |
|------|-------------------|
| ESG Dimension | Governance (G) |
| Model Type | k-Nearest Neighbor (k-NN) regression |
| Training Set Size | 628 observations ($\approx$64%) |
| Validation Set Size | 158 observations ($\approx$16%) |
| Test Set Size | 196 observations ($\approx$20%) |
| Hyperparameter Tuned | Number of neighbors (k) |
| Range of k Tested | 1 to 10 |
| Optimal k Selected | 2 |
| Validation MSE | 1.067.181 |
| Test MSE | 129.452 |
| Test RMSE | 11.378 |
| Test $R^2$ | 0.975 |
| Distance Metric | Euclidean |
| Weighting Function | Rectangular (uniform weights) |

**Table A28.** *Cont.*

| Item | Value/Description |
|---|---|
| Feature Scaling | Applied (all variables standardized before training) |
| Overfitting Evidence | No signs of overfitting: test error lower than validation error, stable high $R^2$ |
| Conclusion | Model generalizes well, predictions are robust and not overfitted |

Note: Systematic train–validation–test partitioning and validation-based tuning identify k = 2 as optimal for the Governance (G) model. Superior test performance (MSE = 129.452; $R^2$ = 0.975) confirms robust generalization and no evidence of overfitting in predicting building-related $CO_2$ emissions (BCE).

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
