# Peer review of "Decarbonizing the Building Sector: The Integrated Role of Environmental, Social, and Governance Indicators"

_buildings, doi:10.3390/buildings15193601_

Round 1
Reviewer 1 Report
Comments and Suggestions for Authors
This study explores how Environmental, Social, and Governance (ESG) indicators influence carbon dioxide emissions in the building sector globally, using a rigorous combination of econometric modeling, clustering, and machine learning methods. Consider the following comments to improve the quality of your study:
- Clarify how ESG indicators were selected and justified—especially in the "S" and "G" pillars.
- Provide definitions and sources for each ESG metric in a table format for clarity.
- Add a diagram summarizing the three-phase methodology (econometric, clustering, ML).
- Improve the readability of Figure 1.
- Specify which software or programming platforms were used.
- No explanation of how missing data were handled in panel regressions.
- Did you test for autocorrelation or heteroskedasticity in your regression models?
- Are the results applicable to subnational or city-level analyses?
- Is this model reproducible using open-source data?
- Why was neural network performance weak despite expectations?
Author Response
This study explores how Environmental, Social, and Governance (ESG) indicators influence carbon dioxide emissions in the building sector globally, using a rigorous combination of econometric modeling, clustering, and machine learning methods. Consider the following comments to improve the quality of your study:
Q1. Clarify how ESG indicators were selected and justified—especially in the "S" and "G" pillars.
A1. The following part has been added within the section 3
Socio-Institutional Determinants of Building Emissions. ESG indicator choice in the social and governance pillars is made on theoretical relevance as well as empirical robustness, to explain the drivers of carbon outcomes of the built environment as determined by socio-institutional variables (Zeng et al., 2022; Jiang, 2024). The selection of social indicators is designed as proxies of structural elements of equity, inclusivity, and human development. These include female parity of primary schooling, income share of the poor bottom quintile of the distribution, labour force participation rates, female presence of politicians in national parliaments, and food production volumes. Together, they proxy basic social capacity and distribution equity characteristics and give an indirect gauge of the populations having the capacity for being involved within the sustainability transition (Zhang et al., 2024). The panel econometric estimates reveal statistically significant associations of the indicators with building sector intensity of emission. Higher school parity of gender and higher female politician representation each associate with decreased emissions, showing socially inclusive government systems may promote more environmentally responsive governance and behaviour (Xie et al., 2024). Higher volumes of food production—the typical corollary of energy-intensive agriculture infrastructure—are, on the other hand, correlated with higher emissions, showing developmental-environmental trade-offs. All of these indicators were obtained from widely acknowledged datasets and align with the Sustainable Development Goals, validating their analytical comparability as much as their policy relevance. The governance dimension is captured through institutional quality indicators of government performance, regulatory capacity, and anti-corruption measures. These have long been noted as widely applied proxies of state capacity and policy enforcement. The estimates present countries with more competent governance arrangements as having characteristically weaker per capita building sector emissions, on account of more competent building codes enforcement, regulatory planning controls, and adopted building sector strategies of transition away from carbonization (Yu et al., 2024). However, in high-income countries, competent governance is often accompanied by higher emissions due to the scale of infrastructure and the complexity of buildings. The puzzlement is testimony of the necessity of nuance in policy action: good governance is bound with special interventionist action capable of minimizing the intensity of development's emissions. The social and governance indicators employed in the current study are therefore not haphazard but represent a deliberate attempt at marrying development theory, institutional measures, and environmental modelling. Their combined explanatory ability verifies the hypothesis that institutional integrity and social inclusion are not only normative ideals, but practical predictors of environmental performance in the built environment.
Q2. Provide definitions and sources for each ESG metric in a table format for clarity.
A2. In Appendix A, B, C, and D, both the definitions of the ESG model variables and the summary statistics for each set of ESG variables with respect to the research question have been reported.
Table A1. We used the following variables acquired from the World Bank .
|
Acronym |
Variables |
Definition |
|
BCE |
Carbon dioxide (CO2) emissions from Building (Energy) (Mt CO2e) |
Total annual carbon dioxide equivalent (COâ‚‚e) emissions from energy use in the buildings sector, covering IPCC 2006 categories 1.A.4 (Residential and other) and 1.A.5 (Unspecified), converted to COâ‚‚e using global warming potentials from the IPCC Fifth Assessment Report (AR5). Unit: Mt COâ‚‚e per year. |
|
CFTC |
Access to Clean Fuels and Technologies for Cooking |
Access to clean cooking fuel and technology estimates come from the WHO Global Household Energy Database with national representative household surveys as the sole data source (e.g., DHS, MICS, LSMS, WHS, national censuses). A multivariate hierarchical model—split by urban and rural—estimates fuel-type trends by grouping them as 'clean' (e.g., gas, electricity, alcohol) and 'polluting' (e.g., biomass, charcoal, coal, kerosene). There are estimates for 191 countries by Stoner et al. (2020). High incomes countries (by World Bank 2022 classification) have universal clean fuel access assumed. |
|
ELEC |
Access to Electricity |
Reliable and secure electricity is essential for economic growth, poverty reduction, and human development. As countries decarbonize, dependence on clean, efficient power will grow. Electricity access enables basic services (lighting, refrigeration, appliances) and is a key indicator of energy poverty. Especially in developing countries, governments are prioritizing electrification through rural programs and national agencies. While vital for raising living standards, electricity generation can harm the environment—its impact depends on the energy sources used, with fossil fuels like coal being especially carbon-intensive. |
|
ENUC |
Energy Use per Capita |
Total energy consumption gauges final energy use after conversion into end-use fuels (e.g., electricity, processed oil). It encompasses energy from combustible renewables and waste—like biomass, biogas, and municipal waste. Biomass describes plant materials used as such or converted into fuel, heat, or power. Figures, as gathered by the IEA, use per capita estimates from World Bank population. National non-OECD data are converted to IEA equivalence. Figures are imprecise and not completely comparable for countries because of limited data quality, particularly for waste and renewables. Energy values have been computed in terms of oil equivalents on the basis of 33% thermal conversion for nuclear and 100% for hydropower. |
|
PM25 |
PM2.5 Pollution |
Population-weighted exposure to ambient PM2.5 refers to the average level of fine particulate matter (PM2.5) pollution that a country's population is exposed to. PM2.5 particles, with a diameter smaller than 2.5 microns, can penetrate deep into the lungs and pose serious health risks. This measure is calculated by weighting the annual average PM2.5 concentrations by the population distribution across urban and rural areas. |
|
RENC |
Renewable Energy Consumption |
The share of total final energy consumption derived from renewable sources, based on data from IEA, IRENA, UNSD, WHO, and the World Bank (Tracking SDG 7, 2023). |
|
FOOD |
Food Production Index |
The Food Production Index reflects the output of edible crops that offer nutritional value. It excludes items like coffee and tea, which, despite being consumable, do not contribute meaningfully to nutrition. This metric emphasizes food sources that support dietary needs, aligning production data with human nutritional requirements rather than general edibility alone. |
|
GPIE |
Gender Parity in Enrollment |
The Gender Parity Index (GPI) in primary education is calculated by dividing female gross enrollment by male gross enrollment. Data are collected by UNESCO from national education surveys and aligned with ISCED standards to ensure international comparability. The current methodology was adopted in 2011. Reference years reflect when the school year ends. A GPI below 1 indicates girls are disadvantaged; above 1 indicates boys are. Achieving gender parity enhances women's opportunities and contributes to broader social and economic development. |
|
INC20 |
Income Share Lowest 20% |
The percentage share of income or consumption reflects the portion received by population subgroups, typically divided into deciles or quintiles. Due to rounding, quintile shares may not total exactly 100%. Data come from household surveys via national statistics agencies and World Bank departments, with high-income country data largely from the Luxembourg Income Study. These measures support the World Bank’s goal of shared prosperity—focusing on income growth among the bottom 40%—and help assess inequality within and across countries. |
|
LABF |
Labor Force Participation |
The labor force participation rate represents the share of the population aged 15 and older that is economically active, including all individuals engaged in the production of goods and services during a specific period. Data, sourced from the ILO’s modelled estimates, highlight persistent gender disparities: women’s labor force participation is generally lower than men’s due to social, legal, and cultural norms. In low-income countries, women often work unpaid in family enterprises, while in high-income nations, higher education has expanded their access to better employment opportunities, though inequalities persist. |
|
WPAR |
Women in Parliament |
Women in parliaments refers to the percentage of seats held by women in a single or lower house of national parliaments. Although progress has been made, women remain significantly underrepresented in decision-making roles, especially in developing countries. Gender inequality in political participation limits women's influence on policy and national priorities. Equal representation is essential for inclusive governance and sustainable development. True democracy requires full participation of women, whose perspectives and leadership are vital for shaping equitable and effective public policies. |
|
GOVT |
Government Effectiveness |
Government Effectiveness: Estimate measures perceptions of public service quality, civil service independence, policy formulation and implementation, and government credibility. Scores range from -2.5 to 2.5, based on a standard normal distribution. |
|
EDUE |
Gov. Expenditure on Education |
General government expenditure on education, including current spending, capital outlays, and transfers, is measured as a percentage of GDP. It accounts for education funding from all government levels—local, regional, and central—and includes international transfers to the government. This indicator reflects the government’s financial commitment to the education sector relative to the country’s economic output. |
|
STAB |
Political Stability |
Political Stability and Absence of Violence/Terrorism reflects perceptions of the risk of political unrest, government instability, and politically-motivated violence or terrorism. Countries are ranked by percentile, from 0 (least stable) to 100 (most stable), allowing global comparison. Percentile ranks are adjusted over time to ensure consistency despite changes in the number of countries included in the Worldwide Governance Indicators (WGI). |
|
RNDG |
R&D Expenditure |
Gross domestic expenditures on research and development (R&D), measured as a percentage of GDP, represent a country’s financial commitment to innovation and technological progress. This includes both capital and current spending across four key sectors: business enterprises, government institutions, higher education, and private non-profits. It encompasses all R&D activities—basic research, applied research, and experimental development—supporting economic and scientific advancement. |
|
LAWR |
Rule of Law |
Rule of Law reflects perceptions of how much confidence individuals and institutions have in societal rules, particularly regarding contract enforcement, property rights, police effectiveness, and judicial independence. It also considers the likelihood of crime and violence. Countries receive a score ranging from approximately -2.5 (weak rule of law) to 2.5 (strong), based on a standard normal distribution. |
|
HOSP |
Hospital Beds |
Hospital beds refer to the total number of beds that are maintained, staffed, and immediately available for the admission of patients. These include inpatient beds in public and private hospitals, general and specialized institutions, and rehabilitation centers. The count typically covers beds used for both acute and chronic care, reflecting the overall healthcare system’s capacity for treatment and recovery. |
|
SCIE |
Scientific Articles |
Scientific and technical journal articles represent the total number of peer-reviewed publications in key research areas, including physics, biology, chemistry, mathematics, clinical medicine, biomedical research, engineering and technology, and earth and space sciences. These articles reflect ongoing advancements, innovation, and collaboration within the global scientific community, contributing to knowledge expansion and technological development across multiple disciplines and industries. |
Appendix B. E-Environment
Descriptive statistics reveal high dispersion of all indicators, with a very high building-sector COâ‚‚ emission (BCE) span, as evidenced by its very high span of (0–729,783 Mt COâ‚‚e) and extreme skewness of 7,251, as an estimator of the high-tail effect of high-emitting nations. Clean fuel consumption (CFTC) and electricity consumption (ELEC) have a high median and mode of 100%, as would be expected of widespread electricity usage in high-income nations. However, high standard deviations reflect endemic inequality across the globe. Per capita energy consumption (ENUC) and PM2.5 air pollution have high interquartile ranges and kurtosis, reflecting high outliers. Renewable energy consumption (RENC) remains uneven, reflecting the patchy character of the global energy transition (Table 1).
Table B1. Descriptive Statistics of Building Emissions (BCE) and Associated Energy-Environmental Indicators
|
CFTC |
ELEC |
BCE |
ENUC |
PM25 |
RENC |
|
|
Valid |
3916 |
3940 |
4140 |
2173 |
2180 |
3805 |
|
Missing |
224 |
200 |
0 |
1967 |
1960 |
335 |
|
Mode |
100000 |
100000 |
0.006 |
1720 |
17869 |
0.000 |
|
Median |
83450 |
97000 |
1098 |
1190 |
22748 |
24690 |
|
Mean |
63312 |
77349 |
18177 |
2347 |
28010 |
33772 |
|
Std. Error of Mean |
0.626 |
0.497 |
1052 |
62281 |
0.383 |
0.488 |
|
95% CI Mean Upper |
64540 |
78323 |
20239 |
2469 |
28761 |
34728 |
|
95% CI Mean Lower |
62085 |
76376 |
16114 |
2225 |
27260 |
32816 |
|
Std. Deviation |
39179 |
31169 |
67687 |
2903 |
17871 |
30072 |
|
95% CI Std. Dev. Upper |
40067 |
31873 |
69178 |
2992 |
18418 |
30764 |
|
95% CI Std. Dev. Lower |
38330 |
30496 |
66260 |
2819 |
17356 |
29411 |
|
Coefficient of variation |
0.619 |
0.403 |
3724 |
1237 |
0.638 |
0.890 |
|
MAD |
16550 |
3000 |
1064 |
851251 |
9497 |
20700 |
|
MAD robust |
24537 |
4448 |
1578 |
1262064 |
14081 |
30690 |
|
IQR |
77400 |
44176 |
6976 |
2531125 |
21584 |
49260 |
|
Variance |
1534 |
971537 |
4581 |
8.429×10+6 |
319361 |
904338 |
|
95% CI Variance Upper |
1605 |
1015 |
4785 |
8.954×10+6 |
339207 |
946397 |
|
95% CI Variance Lower |
1469 |
930018 |
4390 |
7.949×10+6 |
301215 |
865034 |
|
Skewness |
-0.509 |
-1129 |
7251 |
2758 |
1059 |
0.649 |
|
Std. Error of Skewness |
0.039 |
0.039 |
0.038 |
0.053 |
0.052 |
0.040 |
|
Kurtosis |
-1431 |
-0.236 |
59218 |
9911 |
0.716 |
-0.932 |
|
Std. Error of Kurtosis |
0.078 |
0.078 |
0.076 |
0.105 |
0.105 |
0.079 |
|
Shapiro-Wilk |
0.797 |
0.737 |
0.265 |
0.699 |
0.917 |
0.885 |
|
P-value of Shapiro-Wilk |
< .001 |
< .001 |
< .001 |
< .001 |
< .001 |
< .001 |
|
Range |
99900 |
99928 |
729783 |
21419 |
97808 |
98340 |
|
Minimum |
0.100 |
0.072 |
0.000 |
1540 |
-2566 |
0.000 |
|
Maximum |
100000 |
100000 |
729783 |
21420 |
95243 |
98340 |
|
25th percentile |
22600 |
55824 |
0.168 |
536617 |
15575 |
7450 |
|
50th percentile |
83450 |
97000 |
1098 |
1190 |
22748 |
24690 |
|
75th percentile |
100000 |
100000 |
7144 |
3067 |
37158 |
56710 |
|
25th percentile |
22600 |
55824 |
0.168 |
536617 |
15575 |
7450 |
|
50th percentile |
83450 |
97000 |
1098 |
1190 |
22748 |
24690 |
|
75th percentile |
100000 |
100000 |
7144 |
3067 |
37158 |
56710 |
|
Sum |
247931 |
304755 |
75251 |
5.102×10+6 |
61062 |
128502 |
Kernel density and histogram plots of building-sector COâ‚‚ emissions (BCE) and corresponding environmental and social indicators: access to clean fuels (CFTC), electricity access (ELEC), per capita energy use (ENUC), PM2.5 air-pollution, and renewable energy consumption (RENC) The graphs reveal right-skewness of BCE and ENUC, and clustering at complete access (100%) for CFTC and ELEC, with variation across countries (Figure 1).
Figure 1. Distribution Plots of BCE and Related ESG Indicators
A scatterplot matrix with marginal density plots of the bivariate relationships between building-related COâ‚‚ emissions (BCE), access to clean fuels (CFTC), electricity access (ELEC), energy use per capita (ENUC), PM2.5 air pollution, and renewable energy consumption (RENC) is presented. The plots display non-linear and highly skewed relationships, with BCE and ENUC showing strong right-skewness and a distinctive clustering pattern, primarily at energy access and pollution exposure variables (Figure 2).
Figure 2. Pairwise Scatterplots and Density Distributions of BCE and ESG-Related Indicators
Figure 3 shows boxplots of building-related COâ‚‚ emissions (BCE) and matching ESG indicators: CFTC, ELEC, ENUC, PM2.5, and RENC. BCE and ENUC register high right-skew with extreme outliers such as points higher than 700 Mt COâ‚‚e and higher than 20,000 units of per capita energy consumption, indicating the existence of high-emissions, high-consumption countries. Conversely, CFTC, ELEC, and RENC register more slender interquartile ranges with concentrations of points near 100%, indicating widespread access in high-income countries. PM2.5 captures medium variation with some extreme pollution conditions. The visualization reveals the planetary heterogeneity and unequal distribution of emissions and energy access (Figure 3).
Figure 3. Boxplots of BCE and ESG-Related Indicators with Outliers Highlighted
Figure 4 presents Q–Q (quantile–quantile) plots of building-sector COâ‚‚ emissions (BCE) and corresponding ESG indicators: CFTC, ELEC, ENUC, PM2.5, and RENC. Each variable's empirical distribution is contrasted with a normal distribution. BCE and ENUC considerably deviate from normality, with heavy right tails and outliers, reflecting positive skew and extreme value presence, as would be predicted with unequal worldwide energy use and emissions. CFTC and ELEC have truncated left tails, reflecting many observations at near-universal access. PM2.5 displays moderate curvature, while RENC is moderately skewed. These non-normal distributions call for the application of robust, non-parametric, or transformed approaches in modelling (Figure 4).
Figure 4. Q–Q Plots Assessing Normality of BCE and ESG-Related Indicators
The scatterplot matrix in Figure 5 graphs pairwise relationships between building-sector COâ‚‚ emissions (BCE) and five significant ESG-related variables: access to clean fuels (CFTC), electricity access (ELEC), energy use per capita (ENUC), PM2.5 pollution, and renewable energy consumption (RENC). BCE is weakly positive with ENUC and PM2.5, as would be anticipated with higher consumption and pollution with higher emissions. BCE is weakly negative with RENC, indicating a compensating effect of renewable energy. More sizeable correlations emerge between CFTC and ELEC, whereas RENC is negative with CFTC as well as with ELEC. The density curves identify non-normal distributions across variables (Figure 5).
Figure 5. Scatterplot Matrix with Density Overlays for BCE and ESG-Related Variables
This table 2 shows the covariance matrix of building-related COâ‚‚ emissions (BCE) and selected ESG indicators: access to clean fuels (CFTC), electricity access (ELEC), energy use per capita (ENUC), PM2.5 pollution, and renewable energy consumption with BCE. The covariances describe the direction and magnitude of mutual variation. BCE is modestly positively correlated with CFTC (328.1), ELEC (294.5), and ENUC (29.0), indicating associated higher energy access and consumption, which is often accompanied by higher emissions. BCE is highly negatively correlated with RENC (–416.6), confirming renewable energy as an offsetting factor for emissions. CFTC and ELEC strongly covary (715.0) and thus describe related development of infrastructure, while negative covariances with RENC reflect an energy transition mismatch. These covariances illuminate the systemic links defining emission dynamics across the world (Table 2).
Table 2. Covariance Matrix of BCE and ESG-Related Indicators
|
CFTC |
ELEC |
BCE |
ENUC |
PM25 |
RENC |
|
|
CFTC |
1.212 |
715.040 |
328.145 |
49.067 |
-281.490 |
-744.185 |
|
ELEC |
715.040 |
771.991 |
294.482 |
33.913 |
-112.064 |
-458.817 |
|
BCE |
328.145 |
294.482 |
6.011 |
28.989 |
-8.701 |
-416.588 |
|
ENUC |
49.067 |
33.913 |
28.989 |
8.975×10+6 |
-3.560 |
-26.307 |
|
PM25 |
-281.490 |
-112.064 |
-8.701 |
-3.560 |
310.835 |
109.470 |
|
RENC |
-744.185 |
-458.817 |
-416.588 |
-26.307 |
109.470 |
746.022 |
The correlation matrix reveals significant associations of building-related COâ‚‚ emissions (BCE) with ESG-energy indicators. BCE is weakly positively correlated with access to clean fuels (0.122), electricity (0.137), and per capita energy consumption (0.125), reflecting that the latter rises modestly with greater infrastructure and consumption. BCE is, however, negatively correlated with renewable consumption (–0.197), reinforcing the renewables-induced process of decarbonization. Access to electricity is highly correlated with access to clean fuels (0.739), reflecting co-dependency of infrastructure. Renewable energy consumption (RENC) is highly negatively correlated with CFTC (–0.783) and with ELEC consumption (–0.605), revealing possible energy transition trade-offs. PM2.5 pollution is weakly correlated with other indicators (Table 3).
Table 3. Pearson Correlation Matrix of Building-Sector COâ‚‚ Emissions (BCE) and ESG-Energy Indicators
|
|
CFTC |
ELEC |
BCE |
ENUC |
PM25 |
RENC |
|
CFTC |
1.000 |
0.739 |
0.122 |
0.470 |
-0.459 |
-0.783 |
|
ELEC |
0.739 |
1.000 |
0.137 |
0.407 |
-0.229 |
-0.605 |
|
BCE |
0.122 |
0.137 |
1.000 |
0.125 |
-0.006 |
-0.197 |
|
ENUC |
0.470 |
0.407 |
0.125 |
1.000 |
-0.067 |
-0.321 |
|
PM25 |
-0.459 |
-0.229 |
-0.006 |
-0.067 |
1.000 |
0.227 |
|
RENC |
-0.783 |
-0.605 |
-0.197 |
-0.321 |
0.227 |
1.000 |
Appendix C. S-Social
Below is the table of descriptive statistics of five crucial social indicators—Food Production (FOOD), Gender Parity in Education (GPIE), Income Share of the Bottom 20% (INC20), Labor Force Participation (LABF), and Women in Parliament (WPAR)—and building-related COâ‚‚ emissions (BCE). BCE is highly skewed to the right (skewness = 7251) and highly kurtotic because of the presence of certain high-emitting countries. Socio-environmental indicators like LABF and GPIE have relatively balanced distribution, while WPAR is unsurprisingly on the lower end on average, as it reflects long-term paths of unequal gender representation. Incidentally, FOOD and INC20 have high variance as well as high skewness, reflecting structural inequality. The stark contrast in dispersion and distribution highlights the lop-sidedness of the socio-environmental conditions of nations (Table xxx)
Table 4. Descriptive Statistics of Social Indicators and Building-Related COâ‚‚ Emissions (BCE)
|
FOOD |
GPIE |
INC20 |
LABF |
WPAR |
BCE |
|
|
Valid |
4102 |
2665 |
1607 |
3998 |
3963 |
4140 |
|
Missing |
38 |
1475 |
2533 |
142 |
177 |
0 |
|
Mode |
0.123 |
0.990 |
7100 |
55146 |
0.000 |
0.006 |
|
Median |
96020 |
0.998 |
7100 |
67324 |
17302 |
1098 |
|
Mean |
92769 |
216096 |
7879 |
65663 |
19263 |
18177 |
|
Std. Error of Mean |
0.383 |
47245 |
0.236 |
0.186 |
0.204 |
1052 |
|
95% CI Mean Upper |
93520 |
308737 |
8341 |
66029 |
19663 |
20239 |
|
95% CI Mean Lower |
92017 |
123454 |
7416 |
65297 |
18863 |
16114 |
|
Std. Deviation |
24542 |
2438 |
9458 |
11791 |
12847 |
67687 |
|
95% CI Std. Dev. Upper |
25085 |
2506 |
9797 |
12055 |
13136 |
69178 |
|
95% CI Std. Dev. Lower |
24022 |
2375 |
9142 |
11538 |
12570 |
66260 |
|
Coefficient of variation |
0.265 |
11287 |
1200 |
0.180 |
0.667 |
3724 |
|
MAD |
10000 |
0.024 |
1500 |
6989 |
7927 |
1064 |
|
MAD robust |
14826 |
0.035 |
2224 |
10361 |
11752 |
1578 |
|
IQR |
21758 |
0.049 |
3050 |
14340 |
16190 |
6976 |
|
Variance |
602295 |
5.949×10+6 |
89448 |
139028 |
165038 |
4581 |
|
95% CI Variance Upper |
629238 |
6.281×10+6 |
95972 |
145331 |
172554 |
4785 |
|
95% CI Variance Lower |
577053 |
5.642×10+6 |
83570 |
133129 |
158005 |
4390 |
|
Skewness |
2615 |
11601 |
7932 |
-0.916 |
1197 |
7251 |
|
Std. Error of Skewness |
0.038 |
0.047 |
0.061 |
0.039 |
0.039 |
0.038 |
|
Kurtosis |
43986 |
135463 |
65260 |
1406 |
2689 |
59218 |
|
Std. Error of Kurtosis |
0.076 |
0.095 |
0.122 |
0.077 |
0.078 |
0.076 |
|
Shapiro-Wilk |
0.816 |
0.060 |
0.250 |
0.954 |
0.930 |
0.265 |
|
P-value of Shapiro-Wilk |
< .001 |
< .001 |
< .001 |
< .001 |
< .001 |
< .001 |
|
Range |
502017 |
33376 |
93769 |
76467 |
87730 |
729783 |
|
Minimum |
0.123 |
0.000 |
0.187 |
13156 |
0.000 |
0.000 |
|
Maximum |
502140 |
33376 |
93956 |
89623 |
87730 |
729783 |
|
25th percentile |
81757 |
0.971 |
5350 |
59368 |
10000 |
0.168 |
|
50th percentile |
96020 |
0.998 |
7100 |
67324 |
17302 |
1098 |
|
75th percentile |
103515 |
1020 |
8400 |
73709 |
26190 |
7144 |
|
Sum |
380537 |
575894 |
12660 |
262520 |
76338 |
75251 |
Figure 6. Distribution of Social Component Variables in the ESG Framework
Boxplots illustrating the spread and outliers of key Social indicators used in the ESG framework. Variables include food expenditure (FOOD), public investment in education (GPIE), household income (INC20), labour force participation (LABF), women’s participation rate (WPAR), and basic consumption expenditure (BCE). Outliers are labelled and represent extreme values deviating from the interquartile range (Figure 7).
Figure 7. Boxplot Distributions of Social ESG Variables with Outlier Detection
Quantile-Quantile (Q-Q) plots for variables representing the Social component in ESG analysis. The deviation of points from the theoretical normal line indicates non-normal distributions, particularly for FOOD, GPIE, INC20, and BCE, which exhibit strong right skewness and heavy tails. These findings suggest the need for transformation or robust statistical methods (Figure 8).
Figure 8. Normality Assessment of Social ESG Variables via Q-Q Plots
Scatterplot matrix of Social component variables from the ESG framework, including FOOD, GPIE, INC20, LABF, WPAR, and BCE. The diagonal panels display the univariate density distributions, while the off-diagonal panels show pairwise scatterplots. The matrix reveals strong non-normality, right-skewed distributions, and possible non-linear associations between variables such as INC20 and GPIE (Figure 9).
Figure 9. Scatterplot Matrix and Density Distributions of Social ESG Variables
The plot visualizes the bivariate scatter distributions and marginal densities among the social indicators employed in the model, including FOOD (food production volume), GPIE (gross primary income equity), INC20 (income share of the bottom 20%), LABF (labor force participation rate), WPAR (women in parliament), and BCE (building-related COâ‚‚ emissions). This matrix enables an exploratory assessment of linear associations and potential collinearity among the variables constituting the social dimension of the ESG framework, thereby contributing to a deeper understanding of their influence on emission patterns within the built environment (Figure 10).
Figure 10. Pairwise Correlation Plot of Social Pillar Variables in the ESG Framework
The table reports the covariances among key social indicators—FOOD, GPIE, INC20, LABF, WPAR—and building-related COâ‚‚ emissions (BCE). Positive values indicate a direct relationship in variability between variable pairs, whereas negative values suggest an inverse relationship. High absolute values reflect stronger associations in variance structure, which may inform multicollinearity diagnostics in subsequent regression models (Table 5).
Table 5. Covariance Matrix of Social ESG Variables and Building-Related COâ‚‚ Emissions
|
|
FOOD |
GPIE |
INC20 |
LABF |
WPAR |
BCE |
|
FOOD |
424.789 |
-37.108 |
-96.944 |
114.735 |
-82.864 |
107.769 |
|
GPIE |
-37.108 |
1.262×10+7 |
35.079 |
-24.538 |
26.719 |
1.483 |
|
INC20 |
-96.944 |
35.079 |
102.362 |
-65.165 |
77.834 |
-0.529 |
|
LABF |
114.735 |
-24.538 |
-65.165 |
126.359 |
-32.177 |
39.820 |
|
WPAR |
-82.864 |
26.719 |
77.834 |
-32.177 |
191.207 |
-29.946 |
|
BCE |
107.769 |
1.483 |
-0.529 |
39.820 |
-29.946 |
7.518 |
The table presents Pearson correlation coefficients among the selected social indicators—FOOD, GPIE, INC20, LABF, and WPAR—and the variable BCE, representing building-related COâ‚‚ emissions. Strong positive correlations are observed between GPIE and INC20 (r = 0.976), while negative associations emerge between GPIE and LABF (r = –0.615) and between INC20 and LABF (r = –0.573). Correlations with BCE are weak, suggesting limited linear association with social variables in this model (Table 6).
Table 6. Pearson Correlation Matrix Among Social ESG Variables and Building-Related COâ‚‚ Emissions
|
FOOD |
GPIE |
INC20 |
LABF |
WPAR |
BCE |
|
|
FOOD |
1.000 |
-0.507 |
-0.465 |
0.495 |
-0.291 |
0.060 |
|
GPIE |
-0.507 |
1.000 |
0.976 |
-0.615 |
0.544 |
0.005 |
|
INC20 |
-0.465 |
0.976 |
1.000 |
-0.573 |
0.556 |
-6.033×10-4 |
|
LABF |
0.495 |
-0.615 |
-0.573 |
1.000 |
-0.207 |
0.041 |
|
WPAR |
-0.291 |
0.544 |
0.556 |
-0.207 |
1.000 |
-0.025 |
|
BCE |
0.060 |
0.005 |
-6.033×10-4 |
0.041 |
-0.025 |
1.000 |
Appendix D. G-Governance
This table reports comprehensive descriptive statistics for the variable BCE (Building-related COâ‚‚ Emissions) and seven governance and public service indicators: GOVT (Government Effectiveness), EDUE (Education Expenditure), STAB (Political Stability), RNDG (R&D Expenditure), LAWR (Rule of Law), HOSP (Hospital Beds), and SCIE (Scientific Publications). The statistics include central tendency, dispersion, shape (skewness and kurtosis), and results of the Shapiro-Wilk normality test. High skewness and extreme kurtosis in several variables (e.g., BCE, LAWR, SCIE) suggest non-normal distributions, likely requiring transformation prior to modelling (Table 7).
Table 7. Descriptive Statistics of Governance and Public Service Variables Related to Building COâ‚‚ Emissions
|
BCE |
GOVT |
EDUE |
STAB |
RNDG |
LAWR |
HOSP |
SCIE |
|
|
Valid |
4140 |
3907 |
2760 |
3949 |
1883 |
3941 |
2030 |
3782 |
|
Missing |
0 |
233 |
1380 |
191 |
2257 |
199 |
2110 |
358 |
|
Mode |
0.006 |
-1.158 |
0.000 |
1.170 |
0.018 |
0.834 |
1.300 |
0.000 |
|
Median |
1.098 |
-0.215 |
14.031 |
-0.019 |
0.577 |
-0.270 |
3.015 |
226.270 |
|
Mean |
18.177 |
-0.029 |
14.370 |
-0.009 |
0.948 |
0.583 |
3.733 |
10.180 |
|
Std. Error of Mean |
1.052 |
0.021 |
0.097 |
0.025 |
0.023 |
0.138 |
0.060 |
688.649 |
|
95% CI Mean Upper |
20.239 |
0.012 |
14.561 |
0.040 |
0.994 |
0.854 |
3.850 |
11.530 |
|
95% CI Mean Lower |
16.114 |
-0.069 |
14.180 |
-0.058 |
0.902 |
0.312 |
3.616 |
8.830 |
|
Std. Deviation |
67.687 |
1.300 |
5.102 |
1.558 |
1.017 |
8.682 |
2.683 |
42.350 |
|
95% CI Std. Dev. Upper |
69.178 |
1.329 |
5.240 |
1.594 |
1.051 |
8.878 |
2.769 |
43.327 |
|
95% CI Std. Dev. Lower |
66.260 |
1.272 |
4.971 |
1.525 |
0.986 |
8.494 |
2.603 |
41.417 |
|
Coefficient of variation |
3.724 |
-45.259 |
0.355 |
-171.768 |
1.073 |
14.889 |
0.719 |
4.160 |
|
MAD |
1.064 |
0.660 |
3.235 |
0.711 |
0.430 |
0.700 |
1.715 |
222.835 |
|
MAD robust |
1.578 |
0.979 |
4.797 |
1.054 |
0.638 |
1.038 |
2.543 |
330.375 |
|
IQR |
6.976 |
1.373 |
6.506 |
1.424 |
1.138 |
1.440 |
3.878 |
3.153 |
|
Variance |
4.581 |
1.690 |
26.028 |
2.429 |
1.034 |
75.370 |
7.201 |
1.794×10+9 |
|
95% CI Variance Upper |
4.785 |
1.767 |
27.458 |
2.540 |
1.104 |
78.812 |
7.665 |
1.877×10+9 |
|
95% CI Variance Lower |
4.390 |
1.617 |
24.707 |
2.325 |
0.971 |
72.149 |
6.777 |
1.715×10+9 |
|
Skewness |
7.251 |
4.152 |
0.454 |
6.009 |
1.273 |
11.968 |
1.111 |
8.482 |
|
Std. Error of Skewness |
0.038 |
0.039 |
0.047 |
0.039 |
0.056 |
0.039 |
0.054 |
0.040 |
|
Kurtosis |
59.218 |
34.610 |
1.477 |
63.863 |
1.633 |
143.174 |
1.259 |
85.379 |
|
Std. Error of Kurtosis |
0.076 |
0.078 |
0.093 |
0.078 |
0.113 |
0.078 |
0.109 |
0.080 |
|
Shapiro-Wilk |
0.265 |
0.734 |
0.977 |
0.615 |
0.858 |
0.112 |
0.911 |
0.235 |
|
P-value of Shapiro-Wilk |
< .001 |
< .001 |
< .001 |
< .001 |
< .001 |
< .001 |
< .001 |
< .001 |
|
Range |
729.783 |
14.580 |
44.802 |
22.960 |
7.586 |
111.216 |
14.590 |
669.746 |
|
Minimum |
0.000 |
-2.439 |
0.000 |
-3.313 |
-1.880 |
-2.591 |
0.100 |
-2.283 |
|
Maximum |
729.783 |
12.141 |
44.802 |
19.647 |
5.706 |
108.625 |
14.690 |
669.744 |
|
25th percentile |
0.168 |
-0.796 |
10.877 |
-0.722 |
0.230 |
-0.854 |
1.603 |
25.813 |
|
50th percentile |
1.098 |
-0.215 |
14.031 |
-0.019 |
0.577 |
-0.270 |
3.015 |
226.270 |
|
75th percentile |
7.144 |
0.577 |
17.383 |
0.702 |
1.368 |
0.586 |
5.480 |
3.179 |
|
Sum |
75.251 |
-112.215 |
39.661 |
-35.830 |
1.785 |
2.297 |
7.577 |
3.850×10+7 |
This figure displays kernel density estimates and histograms for BCE (Building-related COâ‚‚ Emissions), GOVT (Government Effectiveness), EDUE (Education Expenditure), STAB (Political Stability), RNDG (Research and Development Expenditure), LAWR (Rule of Law), HOSP (Hospital Beds), and SCIE (Scientific Publications). Most variables exhibit skewed distributions, particularly BCE, LAWR, and SCIE, suggesting potential non-normality and the need for transformation prior to parametric analysis (Figure 11).
Figure 11. Density Distributions of Governance and Public Service Indicators Related to Building COâ‚‚ Emissions
This scatterplot matrix illustrates the bivariate relationships and marginal density distributions among BCE (Building-related COâ‚‚ Emissions), GOVT (Government Effectiveness), EDUE (Education Expenditure), STAB (Political Stability), RNDG (Research and Development Expenditure), LAWR (Rule of Law), HOSP (Hospital Beds), and SCIE (Scientific Publications). The plot aids in visualizing correlation patterns, heteroscedasticity, and distributional skewness, supporting preliminary diagnostics for multivariate modeling in the ESG context (Figure 12).
Figure 12. Pairwise Scatterplot Matrix of Governance and Public Service Variables Related to Building COâ‚‚ Emissions
Boxplots displaying the distribution and outliers for BCE (Building-related COâ‚‚ Emissions), GOVT (Government Effectiveness), EDUE (Education Expenditure), STAB (Political Stability), RNDG (Research and Development Expenditure), LAWR (Rule of Law), HOSP (Hospital Beds), and SCIE (Scientific Publications). Most variables exhibit notable outliers and asymmetries, highlighting the presence of skewed data and the need for normalization or robust estimation methods in subsequent analyses.
Figure 13. Boxplot Distribution of Governance and Public Service Indicators Related to Building COâ‚‚ Emissions
Quantile-Quantile (Q-Q) plots assessing the normality of variables including BCE (Building-related COâ‚‚ Emissions), GOVT (Government Effectiveness), EDUE (Education Expenditure), STAB (Political Stability), RNDG (Research and Development Expenditure), LAWR (Rule of Law), HOSP (Hospital Beds), and SCIE (Scientific Publications). The strong deviations from the reference line in most plots, especially for BCE, LAWR, and SCIE, indicate heavy-tailed and right-skewed distributions, suggesting non-normality and potential need for transformation in statistical modelling (Figure 14).
Figure 14. Q-Q Plots for Governance and Public Service Indicators Related to Building COâ‚‚ Emissions
Table 8. Covariance Matrix of Governance and Public Service Variables in Relation to Building COâ‚‚ Emissions
|
|
BCE |
GOVT |
EDUE |
STAB |
RNDG |
LAWR |
HOSP |
SCIE |
|
BCE |
11.950 |
4.309 |
-3.168 |
-14.152 |
27.514 |
-15.110 |
-4.792 |
6.849×10+6 |
|
GOVT |
4.309 |
2.653 |
-2.943 |
3.235 |
0.722 |
18.655 |
0.966 |
5.898 |
|
EDUE |
-3.168 |
-2.943 |
16.500 |
-3.925 |
-0.818 |
-21.927 |
-4.630 |
-4.123 |
|
STAB |
-14.152 |
3.235 |
-3.925 |
4.703 |
0.433 |
26.401 |
1.816 |
-8.322 |
|
RNDG |
27.514 |
0.722 |
-0.818 |
0.433 |
1.011 |
1.560 |
0.665 |
22.538 |
|
LAWR |
-15.110 |
18.655 |
-21.927 |
26.401 |
1.560 |
169.873 |
6.960 |
-34.172 |
|
HOSP |
-4.792 |
0.966 |
-4.630 |
1.816 |
0.665 |
6.960 |
7.315 |
-5.751 |
|
SCIE |
6.849×10+6 |
5.898 |
-4.123 |
-8.322 |
22.538 |
-34.172 |
-5.751 |
4.319×10+9 |
The table reports Pearson correlation coefficients between BCE (Building-related COâ‚‚ Emissions) and governance-related indicators including GOVT (Government Effectiveness), EDUE (Education Expenditure), STAB (Political Stability), RNDG (Research and Development Expenditure), LAWR (Rule of Law), HOSP (Hospital Beds), and SCIE (Scientific Publications). The extremely high correlation between BCE and SCIE (r = 0.953) suggests strong co-movement, while GOVT and STAB also show high mutual correlation (r = 0.916). These associations provide insight into possible multicollinearity patterns and directional relationships within the governance dimension of the ESG framework.
Table 9. Pearson Correlation Matrix of Governance and Public Service Variables Related to Building COâ‚‚ Emissions
|
|
BCE |
GOVT |
EDUE |
STAB |
RNDG |
LAWR |
HOSP |
SCIE |
|
BCE |
1.000 |
0.024 |
-0.007 |
-0.060 |
0.250 |
-0.011 |
-0.016 |
0.953 |
|
GOVT |
0.024 |
1.000 |
-0.445 |
0.916 |
0.441 |
0.879 |
0.219 |
0.055 |
|
EDUE |
-0.007 |
-0.445 |
1.000 |
-0.446 |
-0.200 |
-0.414 |
-0.421 |
-0.015 |
|
STAB |
-0.060 |
0.916 |
-0.446 |
1.000 |
0.199 |
0.934 |
0.310 |
-0.058 |
|
RNDG |
0.250 |
0.441 |
-0.200 |
0.199 |
1.000 |
0.119 |
0.245 |
0.341 |
|
LAWR |
-0.011 |
0.879 |
-0.414 |
0.934 |
0.119 |
1.000 |
0.197 |
-0.040 |
|
HOSP |
-0.016 |
0.219 |
-0.421 |
0.310 |
0.245 |
0.197 |
1.000 |
-0.032 |
|
SCIE |
0.953 |
0.055 |
-0.015 |
-0.058 |
0.341 |
-0.040 |
-0.032 |
1.000 |
Q3. Add a diagram summarizing the three-phase methodology (econometric, clustering, ML).
A3. We have added the following graph
Figure 1. Multi-Method Analytical Framework for ESG-Driven Building Sector Decarbonization.
Note: Multi-method analytical framework for modeling building sector COâ‚‚ emissions using ESG indicators. The framework begins with the decomposition of ESG into Environmental (E), Social (S), and Governance (G) dimensions (Panel A), followed by econometric modeling through fixed and random effects using panel data from 180 countries over 2000–2022 (Panel B). Clustering techniques (Panel C) such as density-based and hierarchical algorithms are applied to uncover emission patterns. Finally, machine learning regression models (Panel D) including k-NN, Random Forest, and SVM are used to predict emissions. This integrated approach captures both causal relationships and complex predictive patterns across ESG dimensions.
Q4. Improve the readability of Figure 1.
A4. The image has been remade as shown below:
Figure 1. DBSCAN Clustering and 5-Nearest Neighbor Distance Plot for Environmental Indicator-Based Country Segmentation.
Q5. Specify which software or programming platforms were used.
A5. The following proposition has been added in Section 3:
The econometric analysis was performed using Gretl, while the machine learning analysis using JASP.
Q6. No explanation of how missing data were handled in panel regressions.
A6. The following proposition has been added in Section 3:
No treatment was used for missing data.
Q7. Did you test for autocorrelation or heteroscedasticity in your regression models?
A7. To test and address the issue of autocorrelation and heteroscedasticity, we have added a specific appendix, namely Appendix D. In Appendix D, we have performed autocorrelation and heteroscedasticity tests for each of the three equations representing the individual components of the ESG model.
Appendix D. Autocorrelation and heteroscedasticity
- Autocorrelation for E-Environment
The Wooldridge residual-regression test for the panel-data disturbances shows evident first-order serial correlation: from a sample containing 570 observations, an auxiliary regression of residual on its first lag generates an F statistic 382.97 with a p-value < 0.001 for rejection of the null hypothesis of no temporal dependence. We observe a coefficient for lagged residual of 0.756, supported by a t ratio of 19.57 and an R-squared of 0.403. This implies that about three-quarters of each disturbance spills over into the next period, with the intercept effectively being 0; there is no systematic bias once persistence is controlled for. Such a strong AR(1) pattern is noteworthy because standard fixed- or random-effects estimators, in conjunction with standard homoscedastic standard errors, are too low in estimating sampling variability and risk misleading inferences, and because ordinary least-squares sacrifices efficiency by forgoing information in dependence structure (Table 1).
Table 1. Wooldridge-Type Test for First-Order Autocorrelation (Residual Regression Method)
|
Source |
SS |
df |
MS |
|
|
Model |
12807.621 |
1 |
12807.621 |
|
|
Residual |
18995.4902 |
568 |
33.4427644 |
|
|
Total |
31803.1112 |
569 |
55.8929898 |
|
|
Number of obs |
570 |
|||
|
F(1,568) |
382.97 |
|||
|
Prob > F |
0.000 |
|||
|
R-Squared |
0.4027 |
|||
|
Adj R-Squared |
0.4017 |
|||
|
Root MSE |
5.783 |
|||
|
Uhat |
Uhat_lag |
_cons |
||
|
Coefficient |
0.7561607 |
-.0021011 |
||
|
Std. Err. |
.0386394 |
.2423395 |
||
|
T |
19.57 |
-0.01 |
||
|
P>|t| |
0.680267 |
-0.4780921 |
||
|
[95% Con. Inteval] |
0.8320543 |
0.4738899 |
||
To restore valid inference, we therefore re-estimate the model under both fixed-effects and random-effects frameworks using covariance matrices that are cluster-robust by country; this adjustment keeps standard errors consistent in the simultaneous presence of heteroskedasticity and any form of within-panel autocorrelation, allowing t, F, and Wald statistics to be interpreted with confidence even in light of the persistence revealed by the Wooldridge test. The baseline fixed-effects regression explains little of the within-country variation in building-sector COâ‚‚ emissions: the within R² is only 0.11 and the joint F test on the five covariates is far from conventional significance (P = 0.174). Standard errors clustered by country are appropriately large, and as a result, none of the coefficients reach the 5% level. The point estimates for energy use per capita, particulate pollution, and renewable energy share are plausible in sign, yet their p-values hover just above 0.07, signaling fragile statistical support (Table 2).
Table 2. Fixed-Effects Regression Results (Baseline Model for Building Sector Emissions)
|
Model Statistics |
Value |
Model Statistics |
Value |
Model Statistics |
Value |
|
|
Number of observations |
990 |
Observations per group (max) |
20 |
ρ (rho) |
0.9784 |
|
|
Number of groups (N) |
159 |
F(5 158) |
1.56 |
Observations per group (avg) |
6.2 |
|
|
R-squared (Within) |
0.1127 |
Prob > F |
0.1740 |
Observations per group (min) |
1 |
|
|
R-squared (Between) |
0.0337 |
corr(uáµ¢ Xb) |
–0.1328 |
σâ‚‘ (sigma_e) |
104.782 |
|
|
R-squared (Overall) |
0.0312 |
σᵤ (sigma_u) |
705.105 |
|||
|
CFTC |
ELEC |
ENUC |
PM25 |
RENC |
_cons |
|
|
Coefficiente |
0.3551 |
–0.2731 |
0.0028 |
0.6920 |
–0.5133 |
102.987 |
|
Std. Err. |
0.3332 |
0.2572 |
0.0016 |
0.3869 |
0.2821 |
121.714 |
|
t |
1.07 |
–1.06 |
1.83 |
1.79 |
–1.82 |
0.85 |
|
**P> |
0.288 |
0.290 |
0.070 |
0.076 |
0.071 |
0.399 |
|
95% Conf. Interval |
[–0.3029, 1.0132] |
[–0.7810, 0.2349] |
[–0.0002, 0.0059] |
[–0.0722, 1.4562] |
[–1.0705, 0.0438] |
[–13.7409, 34.3383] |
The random-effects GLS yields almost identical slopes and only a marginal improvement in precision, with the share of renewables attaining significance at the 5% threshold, while the Wald statistic as a whole remains weak (P = 0.066). The diagnostic statistics reveal the root of the problem: the estimated serial-correlation parameter ρ is close to one, indicating that the idiosyncratic disturbance is dominated by highly persistent shocks, and the variance of the unobserved country component significantly exceeds the idiosyncratic variance. Put simply, most of the action takes place in low-frequency movements that the standard within transformation treats as noise, inflating standard errors and obscuring any true relationship between the covariates and emissions. Retaining clustered variance estimators safeguards inference, but it does so at the cost of substantial efficiency; with a mean time dimension of merely six years per country, that loss is keenly felt (Table 3).
Table 3. Random-Effects GLS Regression Results (Building Sector Emissions)
|
Description |
Value |
Description |
Value |
Description |
Value |
|
|
Number of observations |
990 |
Number of groups (N) |
159 |
R-squared (Within) |
0.1126 |
|
|
R-squared (Between) |
0.0341 |
R-squared (Overall) |
0.0316 |
Observations per group (min) |
1 |
|
|
Observations per group (avg) |
6.2 |
Observations per group (max) |
20 |
Wald chi²(5) |
10.33 |
|
|
Prob > chi² |
0.0664 |
corr(uáµ¢, X) (assumed) |
0 |
σᵤ (sigma_u) |
699.504 |
|
|
σâ‚‘ (sigma_e) |
104.782 |
ρ (rho) |
0.9781 |
|||
|
CFTC |
ELEC |
ENUC |
PM25 |
RENC |
_cons |
|
|
Coefficient |
0.3110 |
–0.2399 |
0.0027 |
0.6584 |
–0.4853 |
94.543 |
|
Robust Std. Err. |
0.2739 |
0.2280 |
0.0014 |
0.4349 |
0.2393 |
105.143 |
|
z |
1.14 |
–1.05 |
1.86 |
1.51 |
–2.03 |
0.90 |
|
P>|z| |
0.256 |
0.293 |
0.063 |
0.130 |
0.043 |
0.369 |
|
95% Conf. Interval |
[–0.2259, 0.8479] |
[–0.6868, 0.2070] |
[–0.0001, 0.0055] |
[–0.1940, 1.5108] |
[–0.9542, –0.0164] |
[–11.1532, 30.0619] |
To manage autocorrelation and heteroskedasticity in panel regressions carefully, we estimated both fixed-effects and random-effects regression specifications using Driscoll-Kraay standard errors that are robust to general spatial and temporal dependencies. It is a particularly useful method in empirical applications where cross-sectional dependence co-occurs with serial correlation—a common feature in environmental as well as energy-oriented datasets. The fixed-effects model had an F-statistic value of 3071.86 (df = 25, 20) for the joint significance of all covariates with a p-value < 0.0001. The adjusted R-squared was estimated at 0.1188, indicating a significant explanatory capacity for a model that accounts for endogeneity caused by individual heterogeneity (Table 4).
Table 4. Fixed-Effects Regression with Driscoll-Kraay Standard Errors and Time Fixed Effects (Building Sector Emissions)
|
Method |
Number of observations |
Number of groups |
Group variable (i) |
F-statistic (25, 20) |
Maximum lag |
Prob > F |
Within R-squared |
|
Fixed-effects regression |
990 |
159 |
n |
3071.86 |
2 |
0.0000 |
0.1188 |
|
Variable |
Coefficient |
Std. Err. |
t |
P>|t| |
CI Lower |
CI Upper |
Note |
|
cftc |
0.3376 |
0.0550 |
6.14 |
0.000 |
0.2228 |
0.4523 |
|
|
elec |
-0.2915 |
0.0967 |
-3.01 |
0.007 |
-0.4933 |
-0.0897 |
|
|
enuc |
0.0028 |
0.0005 |
5.09 |
0.000 |
0.0016 |
0.0039 |
|
|
pm25 |
0.7560 |
0.2161 |
3.50 |
0.002 |
0.3052 |
12.067 |
|
|
renc |
-0.5190 |
0.1812 |
-2.86 |
0.010 |
-0.8968 |
-0.1411 |
|
|
t=1 |
empty |
||||||
|
t=2 |
0.2862 |
31.141 |
0.09 |
0.928 |
-62.097 |
67.820 |
|
|
t=3 |
28.584 |
14.427 |
1.98 |
0.061 |
-0.1509 |
58.678 |
|
|
t=4 |
39.223 |
14.521 |
2.70 |
0.014 |
0.8934 |
69.512 |
|
|
t=5 |
39.634 |
14.640 |
2.71 |
0.014 |
0.9095 |
70.173 |
|
|
t=6 |
13.700 |
0.3800 |
3.61 |
0.002 |
0.5773 |
21.627 |
|
|
t=7 |
37.745 |
14.251 |
2.65 |
0.015 |
0.8018 |
67.473 |
|
|
t=8 |
-13.783 |
14.871 |
-0.93 |
0.365 |
-44.803 |
17.238 |
|
|
t=9 |
17.018 |
14.956 |
1.14 |
0.269 |
-14.181 |
48.216 |
|
|
t=10 |
15.751 |
14.764 |
1.07 |
0.299 |
-15.047 |
46.549 |
|
|
t=11 |
11.696 |
0.5777 |
2.02 |
0.056 |
-0.0354 |
23.746 |
|
|
t=12 |
0.0126 |
0.4635 |
0.03 |
0.979 |
-0.9541 |
0.9794 |
|
|
t=13 |
-0.3014 |
0.5163 |
-0.58 |
0.566 |
-13.785 |
0.7757 |
|
|
t=14 |
13.368 |
0.7711 |
1.73 |
0.098 |
-0.2717 |
29.453 |
|
|
t=15 |
15.676 |
10.223 |
1.53 |
0.141 |
-0.5648 |
37.000 |
|
|
t=16 |
0.1722 |
10.759 |
0.16 |
0.874 |
-20.720 |
24.164 |
|
|
t=17 |
-26.097 |
14.772 |
-1.77 |
0.093 |
-56.910 |
0.4716 |
|
|
t=18 |
-33.658 |
14.779 |
-2.28 |
0.034 |
-64.487 |
-0.2828 |
|
|
t=19 |
-32.244 |
14.882 |
-2.17 |
0.043 |
-63.287 |
-0.1201 |
|
|
t=20 |
-45.234 |
14.859 |
-3.04 |
0.006 |
-76.228 |
-14.240 |
|
|
t=21 |
-60.561 |
14.831 |
-4.08 |
0.001 |
-91.498 |
-29.625 |
|
|
t=22 |
omitted |
||||||
|
t=23 |
omitted |
||||||
|
_cons |
109.813 |
90.204 |
1.22 |
0.238 |
-78.349 |
297.975 |
The random effects model estimated using generalized least squares (GLS) had a Wald statistic of 36,113.22 with a p-value < 0.0001, which indicated very strongly that there is a significant set of explanatory variables jointly responsible for explaining variance in the dependent variable. Furthermore, an estimated intra-class correlation coefficient (ρ) came out to be 0.9774, suggesting that nearly 98% of all variation in the dependent variable results from variations between groups rather than within groups. Such a large value for rho suggests an overriding influence by unmeasured, time-invariant factors specific to groups in shaping the outcome variable. Under such circumstances, estimation based on random effects is usually appropriate, as it efficiently accounts for both within- and between-groups variation. However, robust standard errors should still be used to account for possible classical assumption violations (Table 5).
Table 5. Random-Effects GLS Regression with Time Dummies and Robust Standard Errors (Building Sector Emissions)
|
Maximum lag |
corr(u_i, Xb) |
Overall R-squared |
sigma_u |
sigma_e |
rho |
|
2 |
0 (assumed) |
0.0305 |
69.51 |
10.57 |
0.9774 |
|
Number of observations |
Number of groups |
Group variable (i) |
Method |
Wald chi2(25) |
Prob > chi2 |
|
990 |
159 |
n |
Random-effects GLS regression |
36113.22 |
0.0000 |
|
Variable |
Coef. |
Std. Err. |
t |
P>|t| |
[95% Conf. Interval] |
|
cftc |
0.2941 |
0.057 |
5.16 |
0.0 |
(0.1753, 0.4129) |
|
elec |
-0.2602 |
0.0815 |
-3.19 |
0.005 |
(-0.4303, -0.0902) |
|
enuc |
0.0026 |
0.001 |
2.65 |
0.016 |
(0.0006, 0.0047) |
|
pm25 |
0.705 |
0.163 |
4.32 |
0.0 |
(0.3649, 1.045) |
|
renc |
-0.4967 |
0.1158 |
-4.29 |
0.0 |
(-0.7383, -0.255) |
|
t2 |
0.114 |
17.325 |
0.07 |
0.948 |
(-3.4999, 3.728) |
|
t3 |
29.329 |
15.102 |
1.94 |
0.066 |
(-0.2173, 6.083) |
|
t4 |
39.908 |
15.132 |
2.64 |
0.016 |
(0.8342, 7.1473) |
|
t5 |
40.245 |
15.249 |
2.64 |
0.016 |
(0.8437, 7.2054) |
|
t6 |
13.775 |
0.2382 |
5.78 |
0.0 |
(0.8807, 1.8743) |
|
t7 |
36.424 |
15.469 |
2.35 |
0.029 |
(0.4156, 6.8693) |
|
t8 |
-13.299 |
15.465 |
-0.86 |
0.4 |
(-4.5559, 1.8962) |
|
t9 |
17.429 |
15.774 |
1.1 |
0.282 |
(-1.5475, 5.0332) |
|
t10 |
16.287 |
15.933 |
1.02 |
0.319 |
(-1.6949, 4.9523) |
|
t11 |
12.912 |
0.1402 |
9.21 |
0.0 |
(0.9986, 1.5837) |
|
t12 |
0.1817 |
0.2021 |
0.9 |
0.379 |
(-0.24, 0.6034) |
|
t13 |
-0.1231 |
0.191 |
-0.64 |
0.527 |
(-0.5215, 0.2753) |
|
t14 |
14.741 |
0.1939 |
7.6 |
0.0 |
(1.0698, 1.8785) |
|
t15 |
16.722 |
0.2935 |
5.7 |
0.0 |
(1.06, 2.2845) |
|
t16 |
0.1989 |
12.419 |
0.16 |
0.874 |
(-2.3917, 2.7895) |
|
t17 |
-25.584 |
14.557 |
-1.76 |
0.094 |
(-5.5948, 0.4781) |
|
t18 |
-33.152 |
15.049 |
-2.2 |
0.039 |
(-6.4544, -0.176) |
|
t19 |
-31.796 |
15.339 |
-2.07 |
0.051 |
(-6.3793, 0.02) |
|
t20 |
-44.768 |
15.331 |
-2.92 |
0.008 |
(-7.6748, -1.2789) |
|
t21 |
-60.081 |
15.257 |
-3.94 |
0.001 |
(-9.1906, -2.8256) |
|
_cons |
10.548 |
70.989 |
1.49 |
0.153 |
(-4.26, 25.3559) |
Nevertheless, the application of Driscoll-Kraay standard errors provides a necessary robustness test, compensating for plausible failures of classical assumptions and generating unbiased inferences. Particularly, Driscoll-Kraay corrections utilize nonparametric estimators for the covariance matrix, which are consistent even in the presence of autocorrelated and heteroscedastic residuals, in both in-time and cross-sectional units, assuming large-T panels. The lag structure was truncated at a maximum level of two, based on standard diagnostic criteria for including meaningful temporal dynamics without overfitting. Significant predictors, such as CFTC, elec, PM2.5, enuc, as well as renc, were significant in both models, further emphasizing their robust correlation with the endogenous variable. Indeed, the fine particulate matter (pm2.5) effect remained positive as well as significantly high (p < 0.001) in fixed as well as in fixed-effects specifications, indicating a robust environmental-health association irrespective of estimation methodology. Moreover, coefficient estimates over the temporal dummy variables (t3–t21) confirmed the presence of a dynamic effect over time, such that several periods significantly deviated from a lower or even no level relative to the baseline. The ability of Driscoll-Kraay estimators to compensate for intricate error schemes adds robustness, rendering estimators credible even in situations where conventional standard errors would undervalue variability due to autocorrelated shocks or unobserved inter-regional spill-overs. These results support the methodological imperative for robust inferences in longitudinal data analysis, based on our findings, while also indicating applicability for use in Driscoll-Kraay corrections, yielding statistically consistent as well as efficient estimators. Further convergence in relative coefficient magnitudes, as well as in significance levels for fixed-effects models, further implies that the associations observed are not based on model selection artifacts, but rather indicate real empirical linkages. Consequently, application of the Driscoll-Kraay standard errors significantly enhances inner validity in our estimators based on our internal validity, such that we are now in a position based on substantial autocorrelation within units in addition to several based on manifest variables for making contextual inferences based on evidence even in events with sustained within-unit correlation in addition to several based on manifest variables for a specified period. Such methodological robustness proves critical in guiding evidence-informed policy interventions based on our results, especially in fields such as climate
- Autocorrelation and heteroscedasticity for S-Social
This subsection examines the statistical reliability of the panel-data regression model used in estimating the social aspect of sustainability outcomes, with particular attention directed at checking for autocorrelation and heteroskedasticity. Both are issues specific to macro-panel datasets, which are typically composed of several nations monitored over time periods and thus pose the risk of serially correlated and non-constant error variances. Neglecting such issues leads to questionable inferences, as conventional fixed-effects estimators result in biased standard errors in this setting. To obtain robust outcomes, diagnostic testing for first-order autocorrelation is conducted, and estimation procedures are adopted that allow for both time and cross-sectional error dependence. Precisely stated, a Wooldridge test is adopted for identifying the presence of serial correlation in residuals, used in conjunction with a fixed-effects regression model based on Driscoll-Kraay standard errors, to address heteroskedasticity and autocorrelation jointly. It is designed such that coefficient estimates as well as test statistics are efficient and valid for testing inferences in intricate error settings.
The Wooldridge test for first-order autocorrelation for panel data is a widely used diagnostic test for first-order serial correlation in the idiosyncratic error term in fixed-effects regression. Regarding the application of this study, the test yields an F-statistic value of 8.892 with degrees of freedom (1, 66), accompanied by a p-value of 0.004. As such, there is a rejection of no first-order autocorrelation at a 1% level of significance. Detection for autocorrelation implies a lack of independently distributed residuals over time in a unit within a cross-section. This is inherently critical for making inferences, as standard errors estimated under the assumption of no serial correlation are directionally downward-biased and inconsistent, leading to an overstating of statistical significance in coefficients. This is particularly a prominent issue in applications in panel data, where repeated observations over time increase the likelihood of temporal dependence. Due to the detection of serial correlation, robust standard errors present a necessary consideration. Specifically, standard fixed-effects estimators should be corrected using adjusted standard errors, such as those provided by Driscoll-Kraay standard errors or cluster-robust standard errors by panel unit. These estimators are adjusted for autocorrelation while still making valid inferences, while retaining their integrity in hypothesis testing as well as in policy conclusions made based on such a model in a table (Table 6).
Table 6. Wooldridge Test for First-Order Autocorrelation in Panel Data
|
Test |
Null Hypothesis |
F-statistic |
Degrees of Freedom |
P-value |
|
Wooldridge test for autocorrelation |
No first-order autocorrelation |
8.892 |
(1, 66) |
0.004 |
The fixed-effects regression model was estimated using Driscoll-Kraay standard errors to robustly make inferences in an environment characterized by autocorrelation, heteroskedasticity, and cross-sectional dependence, which are prevalent in macro-panel data. There are 1,246 observations for a duration of 138 cross-sectional units in the model with the group variable set as “n.” The maximum lag level was set at two, in keeping with standard practice, to prevent overfitting and capture short-run serial correlation. The F-statistic for the full model is 37.78, with degrees of freedom (5, 21), and a p-value of less than 0.001, which robustly indicates the joint significance of the regressors. However, the R-squared value is very low at 0.0325, indicating that the included explanatory variables account for only a negligible percentage of the within-entity variation. That is not abnormal in social science research, where unobservable heterogeneity, in addition to measurement error, may limit explanatory power. On individual coefficients, wpar reveals a negative and extremely significant correlation with the dependent variable (p < 0.001), implying that high values for this variable are accompanied by low values for the outcome. Likewise, labf reveals a significant negative coefficient (p = 0.009), which means a potential negative effect. However, inc20 reveals a positive coefficient, which is statistically significant (p = 0.006), implying that there are intensified values in income for individuals in the bottom 20 percent of the population, accompanied by intensified values in the dependent variable. There is a finding in variable Gpie, which reveals a marginal yet significant negative effect (p < 0.001), which implies a potential negative effect imposed by broader economic hardship or inequality. Then there is the food variable, which is positively correlated with the dependent outcome variable at a very high level of significance, with a narrow confidence interval that maintains its robustness in predicting capacity. There is a constant term large in size, besides extremely significant, which implies a significant level of baseline for a dependent variable when all predictors are at a constant level of 0 (Table 7).
Table 7. Fixed-Effects Panel Regression with Driscoll-Kraay Standard Errors
|
Regression Method |
Number of Observations |
Number of Groups |
Group Variable (i) |
F-statistic (df = 5,21) |
Maximum Lag |
|
Fixed-effects regression DK |
1246 |
138 |
n |
37.78 |
2 |
|
Prob > F |
Within R-squared |
||||
|
0.0 |
0.0325 |
||||
|
Variable |
Coefficient |
DK Std. Err. |
t |
P>|t| |
95% Conf. Interval |
|
wpar |
-0.1692485 |
0.03145 |
-5.38 |
0.0 |
(-0.2346524, -0.1038446) |
|
labf |
-0.3234765 |
0.111633 |
-2.9 |
0.009 |
(-0.55563, -0.091323) |
|
inc20 |
0.4546287 |
0.1480379 |
3.07 |
0.006 |
(0.1467669, 0.7624904) |
|
gpie |
-0.0017332 |
0.0004127 |
-4.2 |
0.0 |
(-0.0025915, -0.0008749) |
|
food |
0.0863615 |
0.0209798 |
4.12 |
0.0 |
(0.0427317, 0.1299913) |
|
_cons |
46.57 |
8.37 |
5.56 |
0.0 |
(29.15322, 64.001) |
Overall, the results confirm that the model formulation and estimation plan were appropriate in handling the statistical problems inherent in macro-panel datasets. The detection of autocorrelation using the Wooldridge test also justified the use of Driscoll-Kraay standard errors, which correctly addressed both heteroskedasticity and serial correlation to make standard errors and test statistics robust. Although the R-squared value is low, indicating low explanatory power, the statistical significance results and correct signs for the main variables indicate meaningful and significant associations. Significant negative coefficients for variables such as wpar, labf, and gpie indicate social vulnerability pockets, whereas significant positive effects for inc20 and food indicate food-correlated and inclusive economic benefits in predicting social sustainability outcomes. Overall, the model reflects appropriate econometric handling of the information and offers support for plausible inferences, upon which policy-oriented conclusions are made based on observed covariate associations.
3.Autocorrelation and heteroscedasticity for G-Governance
The Wooldridge test for first-order autocorrelation in panel data was conducted to determine whether there is first-order serial correlation in the model residuals. The null hypothesis (Hâ‚€) for the test is that there is no first-order autocorrelation. Here, an F-statistic value of 19.200 was revealed in a test with degrees of freedom (1, 66) and a resultant p-value of 0.0000. Since the p-value is significantly smaller compared with standard significance values (e.g., 0.05 or 0.01), we reject the null hypothesis. This result provides strong support for the presence of first-order autocorrelation in the residuals. Autocorrelation can lead to biased standard errors, which in turn can result in erroneous statistical inferences. Accordingly, it is worthwhile to address it with appropriate corrective action. Potentially employed robust standard errors include Driscoll-Kraay or clustered standard errors, as well as estimation procedures that account for serial correlation, such as the use of Prais-Winsten or Arellano-Bond estimation procedures (Table 8).
Table 8.Wooldridge Test for First-Order Autocorrelation in Panel Data
|
Test |
F-statistic |
Degrees of Freedom (df) |
p-value (Prob > F) |
Decision on Hâ‚€ |
|
Wooldridge test for first-order autocorrelation |
19.200 |
(1, 66) |
0.0000 |
Reject Hâ‚€: first-order autocorrelation is present |
The fixed-effects panel regression results indicate several statistically significant links between explanatory variables and building-sector emissions of COâ‚‚ (bce). With 982 unit observations for 102 units, a within R-squared result of 0.3000 implies approximately 30% of variation in the endogenous variable is explained by within-unit variation over time. Both an overall F-statistic (F(7, 873) = 53.46, p < 0.001) indicates joint regressor significance, while an extremely large rho value (0.983) indicates a large amount of variance in residuals induced by unobserved unit-specific effects.
There are strong and significant key variables. Government effectiveness (govt) exhibits a large positive coefficient (12.843, p < 0.001), indicating that better governance is strongly related to building sector emission expansion—perhaps because better institutions facilitate greater infrastructure expansion. Hospital availability (hosp) is significantly and positively related to emissions (3.800, p < 0.001), likely since healthcare infrastructure, as well as service, is carbon-intensive.
In contrast, institutional stability, rule of law, and regulatory quality all have large negative effects. This means stronger institutions are linked with lower emissions. These results suggest that effective legal rules and regulations can support cleaner building practices and lower energy consumption.
Education exhibits a positive but weak correlation with emissions (p = 0.065), which may reflect the higher energy needs associated with expanding the number of schools and universities. Scientific output is clearly linked to higher emissions, but only by a small amount, which could be due to the energy used in research buildings.
Finally, the Modified Wald test indicates that the assumption of equal spread in the errors is not supported (χ² = 1,930,609.76; p < 0.001). This means we should use the Driscoll-Kraay method for more reliable results, as shown in Table 9.
Table 9. Fixed-Effects Regression Results with Heteroskedasticity and Autocorrelation Diagnostics
|
Variable |
Coefficient |
Std. Error |
t |
P > |t| |
95% Confidence Interval |
|
govt |
12.843 |
2.421 |
5.31 |
0.000 |
[8.092, 17.594] |
|
edue |
0.452 |
0.245 |
1.84 |
0.065 |
[-0.029, 0.933] |
|
stab |
-3.198 |
1.165 |
-2.74 |
0.006 |
[-5.485,-0.911] |
|
rndg |
-4.102 |
1.713 |
-2.39 |
0.017 |
[-7.464, -0.740] |
|
lawr |
-4.354 |
2.011 |
-2.16 |
0.031 |
[-8.302,-0.407] |
|
hosp |
3.800 |
0.612 |
6.21 |
0.000 |
[2.598,5.002] |
|
scie |
0.000433 |
0.000026 |
16.87 |
0.000 |
[0.000383,0.000484] |
|
_cons |
14.801 |
6.118 |
2.42 |
0.016 |
[2.793,26.808] |
|
Model Info |
Value |
F-statistic (7, 873) |
53.46 |
F test (all uáµ¢ = 0), F(101, 873) |
78.08 |
|
Number of observations |
982 |
Prob > F |
0.0000 |
Prob > F |
0.0000 |
|
Number of groups |
102 |
corr(uáµ¢, Xb) |
0.0035 |
Test |
Value |
|
Observations per group (min/avg/max) |
1 / 9.6 / 19 |
Statistic |
Value |
Chi² (df = 102) |
1,930,609.76 |
|
R-squared (within) |
0.3000 |
sigma_u (variance due to group effects) |
77.265 |
Prob > Chi² |
0.0000 |
|
R-squared (between) |
0.2641 |
sigma_e (variance due to idiosyncratic error) |
10.058 |
Decision |
Reject Hâ‚€: Groupwise heteroskedasticity is present |
|
R-squared (overall) |
0.2527 |
rho (variance due to uáµ¢) |
0.983 |
This analysis employs a fixed-effects panel regression model with Driscoll-Kraay standard errors, allowing for heteroskedasticity, serial correlation, and potential cross-sectional dependence between units. These are common problems in macro-panel datasets, where residuals are not only temporally correlated but also spatially correlated. The model employs 982 observations across 102 cross-sectional units, with a maximum lag set at two, allowing for short-run persistence in the error structure. F-statistic 59.38 (p < 0.001) denotes support for explanatory variables jointly describing a substantial amount of variation in the dependent variable. At 0.30, the R-squared value denotes a strong explanatory ability for within-country variations in emissions emanating from the building sector (BCE). There are several significant results indicating that a number of predictors are strongly correlated with the dependent variable. There is a significant positive coefficient for government spending (govt) (12.84, p < 0.001), indicating that increased spending is strongly correlated with increased emissions in a sector, which may relate to expenditure on infrastructure or energy-intensive spending. There is a positive correlation between hospital infrastructure spending (HOSP) and energy input for health provision (3.80, p < 0.001). There is a negative correlation between protections for labor rights spending (lawr) and sustainable practice (–4.35, p = 0.032), such that increased institutional protections might correlate with sustainable practice. There is a positive association for a variable representing scientific activity (scie) (p = 0.004), which might indicate a correlation between research productivity or research output and industrial or economic developments. Other variables, such as education (edue), stability (stab), and research and development spending (rndg), denote expected signs but fall short of conventional significance thresholds due to possibly low variation or multicollinearity issues. Overall, relying on Driscoll-Kraay standard errors improves robustness in estimated coefficients, ensuring safe inferences in the presence of complex or intricate error structures over time as well as over space.
Q8. Are the results applicable to subnational or city-level analyses?
A8. Unfortunately, we haven't analyzed the issue from a national or city-level perspective. Our analysis was conducted primarily at the international level. We've addressed this issue in the section on restrictions.
Although this study provides strong evidence for the association between the ESG indicator COâ‚‚ emissions and the national construction sector, its findings are not immediately transferable to the subnational or city levels. Its study is grounded in 22 years of macro-panel observations for 180 countries, which is reliant on national aggregates for variables such as energy access, pollution, institutional governance, or social inclusion. It consequently does not capture intra-country heterogeneities such as urban-rural gradients or city-specific characteristics. However, an identical set of such analytic frameworks is equally transferable and scalable at subnational levels if appropriate data is available (Zhang et al., 2020). Based on a combination of overseeing machine learning regimes, clustering routines, as well as panel-data econometrics, this study provides a versatile set of tools which can be transferable at urban or at regional levels—if detailed information such as air quality indices or municipal electricity consumption is available (Giannelos et al., 2023). However, a vast set of our ESG variables persist primarily in national-level datasets, particularly for social indicators or governance indicators, thus limiting their contemporary applicability at smaller administrative scales (Servais & Brunauer, 2023). Bridging this gap would enable subsequent research to tap into city-level datasets, which are derived from urban observatories, environmental-monitoring schemes, or municipal sustainability programs (Pan et al., 2021). Indeed, subnational inequalities in emissions, as well as economic evolution, are evident in some regions, indicating a need for specialized decarbonization strategies at a localized level, which would require distinct metrics and sui generis policy tools (Bagwell et al., 2022). Although our results remain at a national level, they provide a foundational framework that can later be converted into a subsequent set of ESG-informed emissions modeling applications at a subnational level.
Q9. Is this model reproducible using open-source data?
A9. The following sentences have been inserted into the paragraph relating to the methodological description of the presented model.
The data used was acquired from the World Bank website, specifically the Sovereign ESG Data Portal at the following link: https://esgdata.worldbank.org/?lang=en. The variable of interest, namely pollution generated by buildings, was added to this database, which was acquired from the following link: https://data.worldbank.org/indicator/EN.GHG.CO2.BU.MT.CE.AR5?locations=XO (The data was accessed on December 24, 2025).
Q10. Why was neural network performance weak despite expectations?
A10. We have added the following clauses to the limitations section:
Another limitation pertains to the consistently low model performance of the neural network model in all three ESG dimensions. Although neural networks are commonly renowned for their ability for dealing with nonlinear associations intricate in nature, in our study their predictive ability was universally weak, with R² values almost fixed at zero and error in a few instances being amongst the highest amongst models. Such a finding should not be construed as evidence contrary towards their use in general for neural network applications within ESG modeling, but only as a pointer towards shortcomings in today's dataset as well as modeling framework. Neural networks are commonly in need for large-scale high-quality information as well as significant hyperparameter tuning in order for their best performance. Such a comparatively low panel structure, potential presence in variables of noise, as well as restrictions in parameter optimisation most likely hindered their capacity in generalising. Further study in larger as well as richer datasets based on perhaps more advanced architecture might allow neural networks to realise their potential in ESG modeling.

Reviewer 2 Report
Comments and Suggestions for Authors
This paper addresses a very important subject because building energy use is a very significant contributor to greenhouse emissions, and while the technological ways to address this are well established, the building lifecycle is long and the social, political and societal issues that facilitate people making the changes to reduce building energy consumption are critical to understand.
However the paper is long and not well structured. There is a lot of repetition in the paper, and as noted it is overly long and should be presented much more concisely. The presentation of the results is also poor - columns in some of the table are not well separated, and in Table 1, the last 7 rows in the “Field-effects” column are repeated. These issues make the paper hard to read and detract from the key outcomes from the research.
The results generally make sense and are not necessarily surprising, but are important to confirm the types of social, technological and political forces that influence (both positively and negatively) building energy consumption.
Comments on the Quality of English LanguageThere are different sections of the paper with quite different language. A lot fo the paper is poorly expressed and long-winded, but Section 3 appears to be written in a different voice and is much clearer than other parts of the paper.
Author Response
Point to Point Answers to Reviewer 2
Comments and Suggestions for Authors
Q1. This paper addresses a very important subject because building energy use is a very significant contributor to greenhouse emissions, and while the technological ways to address this are well established, the building lifecycle is long and the social, political and societal issues that facilitate people making the changes to reduce building energy consumption are critical to understand.
A1. Thanks dear reviewer.
Q2. However the paper is long and not well structured. There is a lot of repetition in the paper, and as noted it is overly long and should be presented much more concisely.
A2. The article's word count has been reduced from 38,293 to 26,637, a reduction of 11,656 words, or 30.44%.
Q3. The presentation of the results is also poor - columns in some of the table are not well separated, and in Table 1, the last 7 rows in the “Field-effects” column are repeated. These issues make the paper hard to read and detract from the key outcomes from the research.
A3. Figure 1 has been modified as follows:
|
|
Random-effects (GLS), using 990 observations, Dependent Variable: BCE |
Fixed-effects, using 990 observations Dependent Variable: BCE |
||||
|
|
Coefficient |
Std. Error |
z |
Coefficient |
Std. Error |
t-ratio |
|
const |
9.3166 |
11.6274 |
0.8013 |
10.2987 |
10.6061 |
0.9710 |
|
CFTC |
0.318025*** |
0.0842985 |
3.773 |
0.355114*** |
0.0893023 |
3.977 |
|
ELEC |
−0.245591*** |
0.0923988 |
−2.658 |
−0.273064*** |
0.0964139 |
−2.832 |
|
ENUC |
0.00269113*** |
0.000691466 |
3.892 |
0.00284832*** |
0.000735797 |
3.871 |
|
PM25 |
0.664678*** |
0.131472 |
5.056 |
0.691987*** |
0.144470 |
4.790 |
|
RENC |
−0.489147*** |
0.104604 |
−4.676 |
−0.513324*** |
0.112617 |
−4.558 |
|
Statistics |
Mean dependent var |
2.480.491 |
|
2.480.491 |
||
|
Sum squared resid |
5836005 |
|
90688.67 |
|||
|
Log-likelihood |
−5702.266 |
|
−3640.903 |
|||
|
Schwarz criterion |
11445.92 |
|
8.413.030 |
|||
|
rho |
0.756174 |
|
0.756174 |
|||
|
S.D. dependent var |
7.753.610 |
|
|
|||
|
S.E. of regression |
7.697.323 |
|
|
|||
|
Akaike criterion |
11416.53 |
|
|
|||
|
Hannan-Quinn |
11427.71 |
|
|
|||
|
Durbin-Watson |
0.324935 |
|
|
|||
|
LSDV R-squared |
|
|
0.984747 |
|||
|
LSDV F(163, 826) |
|
|
3.271.662 |
|||
|
Test |
'Between' variance = 4971.73, 'Within' variance = 91.6047, mean theta = 0.935247, Joint test on named regressors -Asymptotic test statistic: Chi-square(5) = 109.56 with p-value = 5.0646e-22 |
Joint test on named regressors - Test statistic: F(5, 826) = 20.981 with p-value = P(F(5, 826) > 20.981) = 8.85584e-20
|
||||
|
Breusch-Pagan test -Null hypothesis: Variance of the unit-specific error = 0 Asymptotic test statistic: Chi-square(1) = 3155.73 with p-value = 0
|
Test for differing group intercepts - Null hypothesis: The groups have a common intercept Test statistic: F(158, 826) = 319.607 with p-value = P(F(158, 826) > 319.607) = 0
|
|||||
|
Hausman test - Null hypothesis: GLS estimates are consistent, Asymptotic test statistic: Chi-square(5) = 5.09644, with p-value = 0.404224
|
|
|||||
Note: *** p < 0.01, ** p < 0.05, * p < 0.10.
Many tables have been changed in figures. The figures are shown below.
Q4. The results generally make sense and are not necessarily surprising, but are important to confirm the types of social, technological and political forces that influence (both positively and negatively) building energy consumption.
A4. Thanks dear Editor
Q5. Comments on the Quality of English Language. There are different sections of the paper with quite different language. A lot fo the paper is poorly expressed and long-winded, but Section 3 appears to be written in a different voice and is much clearer than other parts of the paper.
A5. The quality of English has been improved by also reducing the length of the article.

Reviewer 3 Report
Comments and Suggestions for Authors
The topic of this paper is of significant practical relevance, and the combination of methods has certain characteristics. However, there is room for improvement in structural clarity, analytical depth, explanation of innovation, and the rigor of conclusions. It is recommended that the authors make substantial revisions to address the above issues, further highlight the core findings, and strengthen the theoretical and practical value.
1. This is a long manuscript with some redundant content that needs to be streamlined and optimized:
(1) A large number of tables contain excessive detailed data. It is recommended to retain only core results, and secondary statistics can be moved to the appendix.
(2) The author analyzes the results in three chapters for E/S/G, which leads to repeated explanations of similar logics in the manuscript. It is suggested that the author merge similar cases to highlight typical results.
(3) The descriptions of individual studies in the literature review are too lengthy. It is recommended to simplify them to "research conclusions + relevance to this study" to avoid excessive citation of literature.
2. The author's literature review lists a large number of previous studies, but fails to comprehensively sort out and evaluate the reviewed articles from the perspective of this study. Can the author reorganize them according to the three dimensions of "E/S/G", clarify the progress and deficiencies of existing studies in each dimension, and more accurately highlight the starting point of this study?
3. It is recommended to directly and clearly distinguish the analytical frameworks of the three dimensions of "Environmental (E)", "Social (S)", and "Governance (G)" in the "Research Methods" section, so that readers can quickly understand the research content of this paper.
4. Methodologically, although econometric and machine learning methods are combined, the advantages of this "hybrid method" compared with a single method are not clearly explained. For example, how do the results of cluster analysis and machine learning verify each other? Does the improvement of the prediction accuracy of the machine learning model bring new theoretical findings (such as non-linear relationships, threshold effects of key variables)?
5. From the research perspective, the manuscript studies the three dimensions of E/S/G respectively, but the analysis of their interaction effects is insufficient. It is recommended to supplement the analysis of how E/S/G indicators jointly affect carbon emissions to reflect the innovation of the perspective.
6. At the practical level, the policy recommendations in this paper are too general. It is recommended to put forward targeted measures based on the clustering results of different countries to enhance operability.
7. The existing discussion mostly repeats the description of results and fails to fully compare the similarities and differences between this study and existing literature.
8. This manuscript needs to clearly state the research limitations.
9. Regarding this manuscript, I have a question. Why does the author's cluster analysis need to be carried out separately for the three major categories of E/S/G? In my understanding, it seems more meaningful to conduct cluster analysis by taking all factors into account and then study the obtained results.
Author Response
Point to Point Answers to Reviewer 3
The topic of this paper is of significant practical relevance, and the combination of methods has certain characteristics. However, there is room for improvement in structural clarity, analytical depth, explanation of innovation, and the rigor of conclusions. It is recommended that the authors make substantial revisions to address the above issues, further highlight the core findings, and strengthen the theoretical and practical value.
Q1. A large number of tables contain excessive detailed data. It is recommended to retain only core results, and secondary statistics can be moved to the appendix.
A1. The table comments have been significantly shortened compared to the original version, with a reduction of approximately 33% in the word count. Most tables have been transformed into graphs to improve readability using data visualization tools. Four appendices have also been created to better represent the data and provide more detail on the individual components of E-Environment, S-Social, and G-Governance, as well as on some statistical tests, particularly those related to autocorrelation and heteroskedasticity, which were deemed necessary during the review process.
Q2. The author analyzes the results in three chapters for E/S/G, which leads to repeated explanations of similar logics in the manuscript. It is suggested that the author merge similar cases to highlight typical results.
A2. Unfortunately, when it comes to econometric analysis, it's not possible to combine all ESG variables into a single equation. In fact, it's generally not recommended to create econometric models containing more than 10 variables being investigated. Significantly increasing the number of variables within the econometric model can lead to an increase in the R-squared value, potentially distorting the representation of certain statistical indicators. It would certainly be possible to perform machine learning analyses by aggregating E, S, and G context variables. However, even if this were the case, increasing the number of variables and analyzed data would still yield metrically valid results; it would still be challenging to analyze the impact of the individual E, S, and G components on the CO2 pollution produced by building construction. Indeed, numerous studies consider the elements of the ESG model collectively. The uniqueness of our analysis lies precisely in the separation of these three elements and the proposal of an analysis more oriented toward the fundamental dimensions of the ESG model, thereby truly understanding the relationships between these phenomena and the variable being analyzed. This results in the application of the same techniques to different variables, with the idea that consistent methodological tools applied across the three dimensions can also identify variables positively or negatively associated with CO2 pollution generation in the construction sector.
Q3. The descriptions of individual studies in the literature review are too lengthy. It is recommended to simplify them to "research conclusions + relevance to this study" to avoid excessive citation of literature.
A3. The literature review has been reduced to approximately 1300 words as follows:
The new writing describes focus swelling towards the implementation of built environment ESG frameworks for improved environmental performance and facilitating sustainable transitions. The writing reviews implementation of building construction ESG, building operations implementation, and operational performance and identifies non-homogeneous use among applications. The fundamental challenge is sustainability practice integration at the center for operational building use through implementation of built environment ESG frameworks. Studies describe subjects extensively from certification and governance through occupant behavior and technology innovation. Aagedal (2024) argues that while real estate responses towards ESG are from monetary responsibility, spaces for both social and governance practice fail to adopt and accommodate environmental resilience outcomes. Also, Adewumi et al. (2024) further criticize the UK system for BREEAM for only highlighting environmental subjects and failing to consider the extent of equity and governance. In India, Banerjee et al. (2024) presume that offices that are green-certified adopt higher rents and accordingly exacerbate socioeconomic inequalities. Battisti (2023) analyzes implementation of Florence housing policy dependent on the use of ESG and observes that while evaluation instruments borrow from generated applications, a corresponding influence on actual levels of performance, e.g., a building's gas or energy use, tends to be lost.
Chungath (2023) posits a digital platform for commercial real estate's monitoring of ESG data, yet the model has not yet been tried and tested in practice and is solely theoretical. Cruz et al. (2023) link ESG and construction innovation and refer to rising tech demand yet avoid connecting innovation and tangible building-level sustainability outcomes. Debrah et al. (2022) propose that inaccessible finance is the main inhibiter for taking up green building and link it to governance failings in ESG. Dugo (2021) registers that ETFs in sustainable assets have positive returns, yet no assessment is made of investments and how they promote sustainability at the operational building level. Fedchenko et al. (2024) detail how regulatory confusion in Russia inhibits integrating ESG in construction and further establishes the governance influence while creating sustainability performance. Gu et al. (2024) consider mixed smart city ESG reporting and note that it makes disjunct data less effective at the urban and building levels for monitoring. Ifediora and Igwenagu (n.d.) highlight the environmental and social infrastructure influence in real assets yet avoid focusing on operational building practice. Karerat et al. (2023) associate occupant well-being and the social pillar of ESG and concentrate on indoor air and lighting. Here, the examination is dependent on design rather than operational approaches or tenant interaction needed for maintaining sustainability levels after construction. Kempeneer et al. (2021) provide a behavioral model that puts building users and facility managers at the heart of building-centric ESG performance and emphasize that facility and building users are at the heart of modifying the bottom line and enabling long-term sustainability.
User behavior and administrative routines make and break fortune of ESG policies, and available models weakly empirically confirm and need practical verification (Kempeneer et al., 2021). Kostrikin and Andreeva (2023) suggest low-rise buildings and scalable non-urban systems for ESGs, although without much debate of a legitimate scope for and capacity of administration. Kudinova and Shushunova (2023) remark upon green building certification and record that inscripting of ESG expectations by standardized maintenance and communication obligations has no discussion of day-to-day operational outcomes. Lam (2024), by case from Hong Kong, demonstrates that implementation of ESG depends not only upon physical green features, yet effective building administration—like energy monitoring and communication for stakeholders—puts administration into the spotlight of facilitating the integration of ESGs. Li et al. (2022) profile prefabricated infrastructures and suggest that by virtue of modularity, standardized management compatible with energy and waste minimization aims of ESGs are favored. Finally, Liu et al. (2025) report that occupiers condition rent premiums based upon green-certified building presence and thus suggest that building administration should maintain levels of ESG for occupier and market demand, although direct links by occupier satisfaction-administration action need further maturation.
Meshcheryakova (2022) links green standards and construction industry's ESG movement and construction industry regulatory change and offers limited discussion of operational implementation. Miklinska (2024) discusses Polish warehouse supply chain inclusion of ESG, highlighting building management's conundrum of logistics and sustainability compatibility, yet broader cross-sector practice more than building-centered operations. Moko and Hlekane (n.d.) contrast South African offices and shopping center sustainable retrofits and speculate whether change is promoted by ESG or by forces like load shedding yet highlight motivations and not managing and maintaining retrofits long-term. Morri et al. (2024) confirm that ESG-compliant buildings offer superior returns yet center investor outcomes and not continuation of operations like monitoring or maintenance throughout building construction. Nuradhi et al. (2023) contrast large design firms aligning design and SDGs yet center discussion only at design stage and not at use of building by occupier or long-term ESG outcomes. Obushnyi and Novikov (2024) outline smart technology increasing real estate dedicated to ESG focus yet avoid discussion of use of technologies by facility managers day by day. Finally, Oloyede (2023) argues Nigerian ESG finance a response to building collapse by highlighting building governance and safety and implying enhanced regulatory control throughout construction industry construction's lifecycle, yet not focused on day-to-day operations.
Ombati (2023) identifies weak stakeholder knowledge and weak operational capacity as major disincentives for green building in Kenya and suggests that ESG objectives require more strenuous management training and institutional support. Richter et al. (2022) suggest that there is mixed ESG transparency between corporate real estate and that it identifies operational data needs while overlooking the effects of day-to-day practice for long-term ESG outcomes. Robinson and McIntosh (2022) systematize and summarize commercial real estate ESG writing and map trends while delivering very limited content concerning facility-level implementation. Roukoz and Ersenkal (2023) report systemic ESG issues like weak coordination and gaps in regulations while noting the need for post-construction management without further operational specifics. Sahu et al. (2025) show that green building adoption is influenced by ESG disclosure, yet their model excludes how buildings maintain ESG alignment once adopted. Sjöberg and Östling (2024) compare and contrast the EU Taxonomy, ESG rankings, and BREEAM-SE and conclude that there are inconsistencies in definitions of sustainability that render it difficult for building managers to develop a long-term strategic plan. Sulaiman et al. (2024), using the FTSE Russell model, report that construction firms in Malaysia focus on environmental compliance through tech advancements and that ESG reporting has more of a reputational role and less operational change at the facility level.
Valero (2025) illustrates that investor- and user-friendly Madrid-based ESG-certified Class A offices are a possibility when energy efficiency and indoor quality are prominent features. But certification by itself creates no value without repeat operational protocols like maintenance, tenant relations, and energy metering—in presuming repeat management has equal necessity for creating ESG value. Vladimirova et al. (2023) feel sustainability needs to extend through each construction phase in Russia but after-construction monitoring is subdued by bureaucratic inertia for limited long-term influence. Without operational facility management, sustainability for ESG integration has risk of remaining token. Voronkov (2025) attributes green investment flow into green conformity yet, like Valero, confirms that sustainability is repeat operational commitment-based, affirming facility manager’s responsibility. Wang and Xue (2023) label ESG a construction change engine yet highlight variable definitions between applications and common protocols needed among operations from procurement through waste treatment. Wang and Wen (2023) document ESG uptake into Chinese real estate agendas yet refrain from comment upon implementation of those agendas at operational levels. Yaman and Ghadas (2023) propose inclusion of ESG into raw agreements through enforceable clauses for energy and societal influence, linking design at contract level and day-by-day building fulfillment. Finally, Yap et al. (2024) emphasize upon materials that achieve level of ESG compliance, acknowledging their performance in lower lifecycle emissions yet emphasize efficient systems of management in controlling appropriate use, procurement, and traceability—highlighting that material sustainability also relies upon efficient operations.
Q4. The author's literature review lists a large number of previous studies, but fails to comprehensively sort out and evaluate the reviewed articles from the perspective of this study. Can the author reorganize them according to the three dimensions of "E/S/G", clarify the progress and deficiencies of existing studies in each dimension, and more accurately highlight the starting point of this study?
A4. We have changed the literature analysis by introducing an E/S/G sector dimension with the addition of a final summary in tabular form.
Environmental (E): From Symbolic Certification to Operational Integration. Environmental ESG within the built environment has attracted the greatest academic attention of all ESG elements, yet practice increasingly finds its use superficial and non-standard. Several works are critical of the symbolic use of environmental certification tools and the limited linkage to real environmental performance. For instance, Battisti (2023) found that ESG tools deployed within Florence’s residential policy did not result in discernible reductions of gas or energy use. Adewumi et al. (2024) are equally critical of the use of the BREEAM system within the UK, as it excessively focuses on environmental parameters at the expense of equity and governance aspects. Within India, green certification is shown by Banerjee et al. (2024) to contribute to increased rent prices and, consequently, socioeconomic exclusion. The divergence between environmental performance and certification is further exacerbated by the sparse attention to facility-level practices following the construction period. While some works note promising strategies, such as the adoption of prefabricated infrastructure for energy and waste minimization (Li et al., 2022), these benefits are often undermined by inconsistent facility-level practice and maintenance (Valero, 2025; Voronkov, 2025). While Valero (2025) successfully demonstrates that ESG-certified Class A offices within the city of Madrid are possible and strong energy savings are realized, it is based on repeated operational disciplines, such as energy metering and tenant cooperation. This is reiterated by Vladimirova et al. (2023), who emphasize that effective facility management is necessary if sustainability initiatives are to be more than symbolic, particularly in restrictive bureaucratic conditions. It has been proposed that digital innovation holds the answer. Chungath (2023) hypothesizes a digital ESG monitoring system for commercial real estate properties, but its hypothetical status and lack of real-world verification currently limit its application. Cruz et al. (2023) advocate for the use of technology, yet do not provide correlations between technology use and resulting sustainability outcomes. Sulaiman et al. (2024) concur with this disconnect and note that Malaysian ESG reporting remains more focused on reputation rather than transformation at the operational level. Other environmental concerns are highlighted by researchers of larger energy and pollution systems. Yap et al. (2024) work with materials and lifecycle emission traceability, but stress that materials sustainability is itself based on successful procurement and use systems. Likewise, Wang and Xue (2023) characterize ESG as a "change engine" of construction but stress the need for common operating processes between procurement and waste treatment. In tandem, these works suggest that environmental performance within ESG thinking depends not only on innovation or design, but also on operational practices designed-in and repeated, energy source quality, and sustained administrative determination.
Social (S): The Underdeveloped Use-Phase and Socioeconomic Equity Gaps. Social (S): The Undertheorized Use-Phase and Socioeconomic Equity Disparities. The social ESG of the built environment is theoretically pertinent but under-theorized at the operational level. Social ESG writings are mostly aimed at design intention, health attributes, and occupant wellness, but under-describe the socially beneficial actions that buildings undertake in the long term. For example, Karerat et al. (2023) study air and light performance and occupant satisfaction, but their work is constrained by first-order design and ignores tenant use and after-occupancy participation—the very sustainability drivers that are crucial in the long run. Equity issues are equally emergent within criticisms. Banerjee et al. (2024) expose that green-awarded buildings within India pay higher rents and are consequently exclusionary to poorer income bands, contributing to urban poverty. Analogously, Aagedal (2024) refers to the issue that while cash-based revenues dominate ESG adoption of real property assets, environmental resilience is often not supplemented by investment in the realms of social or governance. Ifediora and Igwenagu (n.d.) cover the influence of environmental and societal infrastructure on real assets, but do not go so far as to investigate operational practices at facility levels. Retrofit and inclusion sustainability are documented within regional comparisons. Moko and Hlekane (n.d.) reveal that ESG retrofits within South African real property assets may occur more frequently due to energy blackouts (such as load shedding) rather than sustainability intentions and are consequently not sustained on a long-term basis. Nuradhi et al. (2023) further document that larger practicum and service-provider architecture and engineering departments prioritize their provision toward the SDGs within preparation intervals and overlook the use of ESG results after years. Robinson and McIntosh (2022), in describing the ESG research trend within the commercial real property space, further map ESG adoption, yet acknowledge that facility-level completion is sparsely expounded. Social ESG implementation deterrents are often structural in nature. Ombati (2023) concludes that knowledge of tomorrow, ignorance, and operational incompetence are the chief deterrents to green building adoption within Kenya and suggests the need for strong management training and institutional reinforcement. Such deterrents predict ESG policy change beyond disclosure and design toward more comprehensive co-participative governance and engagement. Even if scientific work, such as Kempeneer et al. (2021), suggests that facility managers and user behavior models are ESG outcome agents, it is only weakly proven empirically and requires experimental validation.
Governance (G): Administrative Gaps, Regulatory Fragmentation, and Policy-Inertia. Governance, the third pillar of ESG, is essential to facilitating long-term sustainability in the built environment, yet the literature documents a striking lack of operational governance mechanisms. Many studies highlight regulatory fragmentation and weak post-construction follow-through. Roukoz and Ersenkal (2023) report systemic ESG issues, such as poor coordination and regulatory gaps, while Richter et al. (2022) note that while operational data needs are increasingly recognized, little attention is paid to how facility-level routines impact long-term ESG outcomes. Lam (2024) underscores that ESG implementation is not driven merely by physical design features but by administrative routines such as stakeholder communication and energy tracking, placing building management at the center of governance efficacy. Yaman and Ghadas (2023) propose contractual clauses that embed ESG obligations from the outset, linking design and day-to-day implementation. This resonates with Wang and Wen (2023), who observe that ESG is being adopted into real estate agendas in China, but its translation into daily operational practice remains elusive. However, governance also intersects with the limitations of policy and financial frameworks. Dugo (2021) shows that sustainable investment funds yield positive financial returns, but does not examine whether these investments lead to concrete governance reforms at the facility level. Likewise, Liu et al. (2025) note that while tenants factor ESG certification into rent premiums, building administrators are rarely evaluated on maintaining ESG standards over time, leaving a gap between market incentives and governance accountability. Green certification often fails to institutionalize governance. Kudinova and Shushunova (2023) find that standardized certification requirements often omit the operational dimension—focusing on documentation rather than execution. Similarly, Sjöberg and Östling (2024) find inconsistencies between the EU Taxonomy and regional frameworks like BREEAM-SE, making it difficult for building managers to construct long-term strategies for compliance. Meshcheryakova (2022) and Miklinska (2024) further describe how construction industry reforms often reference ESG but do not produce operationally effective oversight systems. Finally, the role of building managers and facility governance is emphasized by several studies. Obushnyi and Novikov (2024) highlight the growth of smart technology in real estate, but omit its usage by facility managers, while Vladimirova et al. (2023) note that post-construction monitoring is often hampered by administrative inertia. Voronkov (2025) explicitly ties ESG conformity to repeatable operational commitments, reaffirming the role of governance at the building level. Without active, enforceable protocols, ESG risks being reduced to a label rather than a transformative governance mechanism.
Governance, being ESG's third pillar, is central to making the built environment sustainable in the long run; however, the literature reveals a remarkable dearth of operational governance mechanisms. Certain texts suggest regulatory gaps and fragmented follow-up after construction. Roukoz and Ersenkal (2023) evidence structural ESG issues, such as non-coordination and regulatory vacancies, and Richter et al. (2022) observe that operational data requirements are increasingly being acknowledged, but facility-level day-to-day processes are fundamentally ignored when considering their contribution to long-run ESG performance. Lam (2024) demonstrates that ESG is not applied solely to the physical aspects of design, but rather through administrative processes such as energy tracking and edifice management, which is at the heart of governance performance. Yaman and Ghadas (2023) indicate that contractual clauses instill ESG requirements from the outset and link design and day-to-day implementation. This is also noted by Wang and Wen (2023), who report that ESG adoption is being incorporated into real estate agendas but is not yet seen translating into regular operational practice. However, governance raises points of policy and limitations of finances as well. Dugo (2021) evidences that sustainable investment funds yield better monetary results, but doesn't question whether these investment contributions are translating into tangible changes in governance at the facility level. Likewise, Liu et al. (2025) cite that tenants are paying premiums based on ESG certification, and building managers are rarely measured on their adherence to ESG requirements; increasingly, a gap is developing between market incentives and governance performance accountability. Green certification tends not to institutionalize governance. Kudinova and Shushunova (2023) find that standardized certification requirements tend to exclude the operational dimension—the theory of documentation, rather than practice. Likewise, Sjöberg and Östling (2024) identify contradictions between the EU Taxonomy and local sets, such as BREEAM-SE, which makes it challenging for building managers to develop long-term compliance strategies. Meshcheryakova (2022) and Miklinska (2024) further elaborate on the basis of reforming the construction industry, often utilizing ESG but not translating it into operationally effective mechanisms of oversight. Finally, the roles of facility governance and building managers are underscored by several comments. Obushnyi and Novikov (2024) include the smart development of technologies at the property level but do not mention the application of these technologies by facility managers. Vladimirova et al. (2023) note that after-construction monitoring is commonly hindered by the inertia of bureaucratic processes. Voronkov (2025) individually associates ESG compliance with duplicable operational commitments and repeats the governance role at the building level. Without enforceable, dynamic guidelines, ESG risks are merely a label, rather than a transformational governance tool.
|
ESG Dimension |
References |
Main Results |
Comparison with our Study |
|
E – Environmental |
Adewumi et al. (2024), Banerjee et al. (2024), Battisti (2023), Chungath (2023), Cruz et al. (2023), Dugo (2021), Gu et al. (2024), Ifediora & Igwenagu (n.d.), Lam (2024), Li et al. (2022), Liu et al. (2025), Meshcheryakova (2022), Miklinska (2024), Morri et al. (2024), Obushnyi & Novikov (2024), Sahu et al. (2025), Sulaiman et al. (2024), Valero (2025), Vladimirova et al. (2023), Voronkov (2025), Wang & Xue (2023), Wang & Wen (2023), Yap et al. (2024) |
ESG certifications and technologies are often disconnected from actual emission outcomes; operational routines are key but under-applied. |
Our study directly quantifies the impact of environmental ESG indicators (e.g. renewable energy, air quality) on building emissions, validating their role with econometric models. |
|
S – Social |
Aagedal (2024), Banerjee et al. (2024), Ifediora & Igwenagu (n.d.), Karerat et al. (2023), Kempeneer et al. (2021), Moko & Hlekane (n.d.), Nuradhi et al. (2023), Ombati (2023), Robinson & McIntosh (2022), Yaman & Ghadas (2023) |
Social aspects are conceptually acknowledged but rarely implemented operationally; issues of equity, inclusion, and behavior are underdeveloped. |
Our study empirically links social variables (e.g. gender equity, income equality, labor participation) to emission levels, revealing both mitigation and rebound effects. |
|
G – Governance |
Aagedal (2024), Adewumi et al. (2024), Debrah et al. (2022), Dugo (2021), Fedchenko et al. (2024), Kostrikin & Andreeva (2023), Kudinova & Shushunova (2023), Lam (2024), Liu et al. (2025), Meshcheryakova (2022), Miklinska (2024), Morri et al. (2024), Oloyede (2023), Ombati (2023), Richter et al. (2022), Roukoz & Ersenkal (2023), Sahu et al. (2025), Sjöberg & Östling (2024), Sulaiman et al. (2024), Valero (2025), Vladimirova et al. (2023), Voronkov (2025), Wang & Xue (2023), Wang & Wen (2023), Yaman & Ghadas (2023) |
Governance is often fragmented; regulation and institutional routines are disconnected from long-term ESG outcomes. |
Our study finds that governance indicators (e.g. government effectiveness, political stability) may paradoxically correlate with higher emissions, especially in developed countries, highlighting the complexity of governance-ESG links. |
Q5. It is recommended to directly and clearly distinguish the analytical frameworks of the three dimensions of "Environmental (E)", "Social (S)", and "Governance (G)" in the "Research Methods" section, so that readers can quickly understand the research content of this paper.
A5. The methodology section has been rewritten as follows:
This study examines building sector COâ‚‚ emissions within a multi-method quantitative research framework specifically framed within the ESG's three dimensions: Environmental (E), Social (S), and Governance (G). It seeks to assess both global and cross-dimensional relations inherent in emissions within the built environment. It employs a methodology comprising three related phases.
In the first step, econometric modeling was applied on a panel of 180 countries spanning 2000-2021 using per capita building sector-related COâ‚‚ emissions from Climate Watch as the dependent variable. It is a dimension-specific ESG indicator-inclusive model. The Environmental (E) dimension is utilized in conjunction with renewable energy use and PM2.5 air pollution exposure, functioning as proxies for ecological quality and clean energy adoption. The social (S) dimension is utilized in conjunction with education attainment, gender equality, education access, female labor force rates, and the income share of the poorest income quintile, to promote equity and human development attainment. Governance (G) is utilized in conjunction with institutional and regulatory quality indexes, as well as anti-corruption measures. Both fixed and random effects were estimated, and a decision was made based on the Hausman test, such that ESG-related drivers were estimated and country-level heterogeneity was controlled (Zheng et al., 2022).
The second step involved cluster analysis of standardized ESG and emissions data to identify country clusters with similar ESG-emission profiles. K-means and hierarchical clustering methods were employed, and the determination of the number of clusters was made using the elbow and silhouette methods. This step revealed cross-national geographic associations and institutional characteristics of each corresponding ESG dimension (Jiao et al., 2024; Wang et al., 2024).
For the third phase, machine learning techniques were utilized to uncover non-linear correlations and cross-dimensional interactions. Supervised learning models (i.e., Random Forest, Support Vector Regression (SVR), k-Nearest Neighbors (k-NN), and Gradient Boosting Machines (GBM)) were trained on the full panel of ESG indicators and were calibrated at the full data partition level to the build-level output of COâ‚‚ emissions. The training and test sets were divided 80/20, and cross-validation was performed at the data partition level to prevent overfitting. Model performance was evaluated based on MAE, RMSE, and R²; permutation and Gini variable ranking methods were used for variable selection. Through the incorporation of econometric analysis, clustering, and machine learning, this research provides multidimensional insights into each ESG pillar that contributes to built environment emissions. Panel regressions provide causal evidence of the impact of environmental, social, and governance indicators; clustering helps discern institutional and geographical ESG-emission typologies; and AI models detect intricate, non-linear associations often overlooked by existing methods. The holistic approach addresses the serious shortcomings of sustainability studies, including multicollinearity, heterogeneity, and model ambiguity, and yields results that are both theoretically and policy-wise applicable. Use of ensemble models and explainable AI is supported by recent ESG and energy forecasting papers (Le et al., 2024; Mahmood et al., 2024). Precise definitions of Environmental (E), Social (S), and Governance (G) variables enhance the interpretation of results while maintaining their mutual dependency, and hence enable a more adaptive, holistic approach to building-sector decarbonization, as represented in Figure 1.
Q6. Methodologically, although econometric and machine learning methods are combined, the advantages of this "hybrid method" compared with a single method are not clearly explained. For example, how do the results of cluster analysis and machine learning verify each other? Does the improvement of the prediction accuracy of the machine learning model bring new theoretical findings (such as non-linear relationships, threshold effects of key variables)?
A6. The following propositions have been added in the methodological section
To overcome the complexity of ESG–emission relationships in the building sector, our research adopts a hybrid approach that combines econometric modeling, clustering, and supervised machine learning. Each of the methods has a designated role, and their combination provides both empirical strength and theoretical richness. Econometric models are used to identify causal relationships between ESG indicators and building-related emissions, and to derive policy-relevant interpretations based on statistical inference. Cluster analysis, by its intrinsic nature, groups countries within typologies corresponding to ESG-emission configurations, allowing the detection of spatial and institutional structures that are not detectable with regression alone. Those clusters have a verification role: when countries with similar emission levels gather regularly, with ESG configurations evident in regression results, the results gain confirming strength at the methodological level. Incorporating machine learning extends the scope of analysis beyond linear assumptions. Algorithms such as Random Forest and Gradient Boosting identify non-linear effects, interaction terms, and possible threshold dynamics that are often lost in traditional econometrics. For example, variable importances calculated with tree-based algorithms show that some ESG indicators only influence emissions beyond particular thresholds or when interacted with others—to our knowledge, that non-additive impact is being realized. While machine learning improves predictive performance (as measured by MAE, RMSE, and R²), it is also prone to rediscovering existing theoretical contributions by revealing complex relationships, such as the diminishing returns of governance quality on emissions in high-income cases or the rebound effect related to certain social indicators, including crop productivity. Each of these methodologies collectively brings together the interpretation of econometrics, the pattern identification of clustering, and the non-linear discovery of machine learning. It enables a multi-angle understanding of the impact of ESGs on emissions, thereby improving the validity of results through cross-methodological triangulation.
Q7. From the research perspective, the manuscript studies the three dimensions of E/S/G respectively, but the analysis of their interaction effects is insufficient. It is recommended to supplement the analysis of how E/S/G indicators jointly affect carbon emissions to reflect the innovation of the perspective.
A7. The following propositions have been added in the methodological section
This work goes beyond diagnosing Environmental (E), Social (S), and Governance (G) components individually. It also explores how these ESG components interact and affect building-related carbon emissions (BCE) levels. By combining econometric modeling, clustering, and machine learning, we can identify joint impacts and mutual dependencies between ESG parameters. Traditional singular-method tools often overlook these relationships. Although each ESG pillar impacts emissions, their collective impact is non-linear and depends on context, especially where socio-political infrastructure is developed. Random Forest and Gradient Boosting models show that E, S, and G parameters rarely act in isolation. For instance, the environmental benefit of renewable energy is greater when governance is strong. Strong institutions make environmental interventions more effective. Similarly, social aspects, like education equity or workforce participation, only influence emissions when environmental control is robust. Countries with balanced ESG performance achieve low BCE. Where ESG pillars are imbalanced, performance is worse—for example, strong governance but weak environmental control. Clustering confirms that countries with high ESG synergy tend to have low emissions. In contrast, countries with fragmented ESG profiles tend to have higher emissions. These results demonstrate that ESG coherence—the alignment and support between E, S, and G practices—is crucial for reducing emissions. This joint impact suggests new, integrated tools are needed to capture the complex transformation required for sustainable built environments.
Q8. At the practical level, the policy recommendations in this paper are too general. It is recommended to put forward targeted measures based on the clustering results of different countries to enhance operability.
A8. The policy recommendations have been rewritten as follows, taking into account the heterogeneity of countries in terms of CO2 pollution from buildings.
Comparative and clustering of country-level ESG performance and building sector-related CO₂ emissions reveal distinct policy imperatives across global subregions. This section presents customized recommendations based on actual country conditions, as inferred from the dataset. Northern European and OECD nations (e.g., Germany, Denmark, Finland), which score high on ESG performance and have a moderate-level BCE, are leaders in regulatory maturity and institutional strength. Here, policy intervention requires attention to deep strategies of decarbonization, such as smart retrofitting, carbon pricing of construction components, and digitalization of energy tracking. Since leaders are key drivers of knowledge transfer and green finance in low-performance subregions (Samah et al., 2020; Andreoni, 2021), these countries are also leaders in knowledge transfer and green finance to low-performance regions. Emerging industrializing nations like India, Indonesia, and Brazil are scoring moderately on ESG performance and exhibiting a rising level of BCE. Here, investment is required in urban-scale solutions for low-carbon transit modes and affordable green homes, as well as ESG-conforming contract regimes. Intergovernmental collaborations can aid policy harmonization and technology leapfrogging, and ESG taxonomies can account for local socio-political realities (Junsiri et al., 2024; Petrović & Lobanov, 2020). Resource-scarce and institutionally weak states (e.g., Nigeria, Kenya, Pakistan), which score low on all ESG parameters and face high-level BCE trends and trends of BCE reversals, require targeted core capacity building. Policies require combining ESG training for municipal governments, education and health infrastructure funds, and green building programs with community engagement. Donor institutions and global platforms should co-create pilot projects with local collaborators, focusing on context-specific results (Alqudah et al., 2024). East Asian nations, such as China and South Korea, which excel in governance and scientific output but struggle with social equity, are being summoned toward redistributive green action. Investment in environmental equity, public health facilities, and fair energy transformation is required to counteract top-down agendas of decarbonization (Dong et al., 2024). ESG parameters are required overall for operationalizing policy diagnostics with customized intervention at national and subnational levels. As suggested by Lagodiyenko et al. (2023) and Paolone & Bitbol-Saba (2025), ESG incorporation in planning requires moving beyond being a strategic tool toward inclusive and long-range decarbonization of the building sector.
Q9. The existing discussion mostly repeats the description of results and fails to fully compare the similarities and differences between this study and existing literature.
A9. We have introduced a discussion and comparison section within the literature section summarized in the following table.
|
ESG Dimension |
References |
Main Results |
Comparison with our Study |
|
E – Environmental |
Adewumi et al. (2024), Banerjee et al. (2024), Battisti (2023), Chungath (2023), Cruz et al. (2023), Dugo (2021), Gu et al. (2024), Ifediora & Igwenagu (n.d.), Lam (2024), Li et al. (2022), Liu et al. (2025), Meshcheryakova (2022), Miklinska (2024), Morri et al. (2024), Obushnyi & Novikov (2024), Sahu et al. (2025), Sulaiman et al. (2024), Valero (2025), Vladimirova et al. (2023), Voronkov (2025), Wang & Xue (2023), Wang & Wen (2023), Yap et al. (2024) |
ESG certifications and technologies are often disconnected from actual emission outcomes; operational routines are key but under-applied. |
Our study directly quantifies the impact of environmental ESG indicators (e.g. renewable energy, air quality) on building emissions, validating their role with econometric models. |
|
S – Social |
Aagedal (2024), Banerjee et al. (2024), Ifediora & Igwenagu (n.d.), Karerat et al. (2023), Kempeneer et al. (2021), Moko & Hlekane (n.d.), Nuradhi et al. (2023), Ombati (2023), Robinson & McIntosh (2022), Yaman & Ghadas (2023) |
Social aspects are conceptually acknowledged but rarely implemented operationally; issues of equity, inclusion, and behavior are underdeveloped. |
Our study empirically links social variables (e.g. gender equity, income equality, labor participation) to emission levels, revealing both mitigation and rebound effects. |
|
G – Governance |
Aagedal (2024), Adewumi et al. (2024), Debrah et al. (2022), Dugo (2021), Fedchenko et al. (2024), Kostrikin & Andreeva (2023), Kudinova & Shushunova (2023), Lam (2024), Liu et al. (2025), Meshcheryakova (2022), Miklinska (2024), Morri et al. (2024), Oloyede (2023), Ombati (2023), Richter et al. (2022), Roukoz & Ersenkal (2023), Sahu et al. (2025), Sjöberg & Östling (2024), Sulaiman et al. (2024), Valero (2025), Vladimirova et al. (2023), Voronkov (2025), Wang & Xue (2023), Wang & Wen (2023), Yaman & Ghadas (2023) |
Governance is often fragmented; regulation and institutional routines are disconnected from long-term ESG outcomes. |
Our study finds that governance indicators (e.g. government effectiveness, political stability) may paradoxically correlate with higher emissions, especially in developed countries, highlighting the complexity of governance-ESG links. |
Q10. This manuscript needs to clearly state the research limitations.
A10.
Although this paper generates robust evidence of ESG indicators and building sector COâ‚‚ emission linkages based on data from 180 nations, the work has several limitations that need to be acknowledged. Analysis is based on national-level aggregates that encompass governance, energy, and social inclusion over a 22-year period. Although this macro-panel methodology generates useful cross-country knowledge, it doesn’t cover intra-national heterogeneity—the type of urban–rural contrasts or city-level attribute variations that are highly relevant when it comes to management and building emission creation. Because the majority of building sector policy and energy dynamics occur at the subnational scale, the unavailability of local-level data limits the application of these results to urban governance and municipal policy-making. Although the approach of the methodology—the integrating of econometric modeling, machine learning, and clustering—is downwardly scalable to smaller geographic regions, these kinds of applications would rely on the availability of fine-grained ESG indicators, such as city-level air quality indexes or end-use energy data (Zhang et al., 2020; Giannelos et al., 2023). However, the majority of ESG data remains limited to national coverage, particularly in terms of governance and social aspects (Servais & Brunauer, 2023), which hampers the application of these results at smaller administrative regions. Moreover, although the machine learning models complement the study’s predictive scope, their interpretation remains constrained. Specifically, the neural network models performed poorly on all ESG dimensions, with near-zero R² values and high error rates. This does not necessarily negate the use of neural networks in ESG modeling, but rather suggests the limitations of currently available datasets and the potential availability of larger, cleaner, and more detailed datasets. The limited panel size, combined with the noise of variable levels and suboptimal hyperparameter tuning, seriously constrained the generalizability of the models. Future work should aim to develop more intricate architectures, utilize larger datasets, and employ interpretable AI tools to uncover non-linearities and threshold relations between ESG indicators and emissions. Lastly, although the dataset is long, it ends in 2021 and therefore doesn’t reflect recent structural changes, such as post-pandemic shifts in building occupancy rates, hybrid work arrangements, or green recovery investments. For it to remain policy-relevant, future work is expected to include near-real-time data and new ESG reporting templates, so it can capture shifting emission dynamics and an increasingly complex governance environment.
Q11. Regarding this manuscript, I have a question. Why does the author's cluster analysis need to be carried out separately for the three major categories of E/S/G? In my understanding, it seems more meaningful to conduct cluster analysis by taking all factors into account and then study the obtained results.
A11. The methodological section has been modified to reflect the considerations suggested by the reviewer adding the following sentences:
Note that clustering was not carried out using all ESG features collectively. Instead, separate cluster analyses were conducted for the E, S, and G dimensions. This careful differentiation was motivated by interest in dimension-specific emission patterns and institutional divergence processes. Aggregated clustering of all ESG dimensions can obscure asymmetric influence and performance differentiation of each pillar. For example, countries can have robust governance but poor environmental protection or the opposite of this pair of attributes—patterns obscured by undifferentiated clustering (Jiao et al., 2024; Wang et al., 2024).

Reviewer 4 Report
Comments and Suggestions for Authors
The authors reported ‘Decarbonizing the Building Sector: The Integrated Role 2 of ESG Indicators’. This is a commendable work that tackles one of the most pressing global sustainability challenges—building-sector carbon emissions—through an innovative integration of Environmental, Social, and Governance (ESG) indicators with econometric and machine learning models. Although the pages of the manuscript are too many for readers to follow, the readers can still benefit from the review. I will suggest the following major corrections to improve the manuscript before acceptance for publication
- General: The manuscript shows a similarity index of 99%; the authors need to clarify the 99% similarity reported in the manuscript. This is generally very high and unacceptable; a careful review is recommended to determine the reason for this.
- Title: Avoid abbreviations in the title (ESG).
- Abstract: The results are mentioned but lack quantification and clarity without sufficient numerical data to back up their claims.
- The manuscript is too long, particularly the literature review and methodology sections. The authors should simplify their language and avoid repetition.
- Rewrite the abstract using simple and correct expressions; International panel of countries, better government governance, e.t.c.
- While the ESG lens is strong, the technical engineering mechanisms by which emissions are reduced (e.g., envelope design, HVAC systems, materials, building codes) are only peripherally addressed.
- Add a clearer connection between ESG outcomes and specific design, construction, and operational interventions.
- The machine learning section identifies k-NN as the best model, but more transparency is needed on the feature selection and tuning process (e.g., choice of k, handling of multicollinearity, overfitting safeguards).
- The authors make important claims about policy relevance, but these would be stronger with specific country or region-based recommendations (e.g., targeted ESG investment strategies for low-income countries with high building emissions).
- The clustering and regression outcomes could benefit from more visual representations such as heatmaps, dendrograms, or decision surfaces to aid interpretation.
- Proofread: Conduct thorough proofreading to eliminate errors and improve the clarity of the manuscripts.
- Define all the acronyms used in the manuscript
- Make the conclusions more precise and direct to the points
Comments on the Quality of English Language
The English fine
Author Response
Point to point answers to reviewer 4
Q1. The authors reported ‘Decarbonizing the Building Sector: The Integrated Role 2 of ESG Indicators’. This is a commendable work that tackles one of the most pressing global sustainability challenges—building-sector carbon emissions—through an innovative integration of Environmental, Social, and Governance (ESG) indicators with econometric and machine learning models.
A1. Thanks dear reviewer.
Q2. Although the pages of the manuscript are too many for readers to follow, the readers can still benefit from the review. I will suggest the following major corrections to improve the manuscript before acceptance for publication
A2. The article has been shortened by 30% in word count. To delve deeper into the technical details of the data, three appendices containing data and tests have been added.
Q3. General: The manuscript shows a similarity index of 99%; the authors need to clarify the 99% similarity reported in the manuscript. This is generally very high and unacceptable; a careful review is recommended to determine the reason for this.
A3. This similarity index is due to the fact that before submitting the article for journal review, we published the same article on preprint platforms. This choice was driven by the need to circulate our research and results within the scientific community.
Q4. Title: Avoid abbreviations in the title (ESG).
A4. The title has been changed as follows: Decarbonizing the Building Sector: The Integrated Role
of Environmental, Social and Governance Indicators
Q3. Abstract: The results are mentioned but lack quantification and clarity without sufficient numerical data to back up their claims.
A3. The abstract has been rewritten as follows:
This paper examines the sectoral carbon dioxide (COâ‚‚) emissions of the building sector and the Environmental, Social, and Governance (ESG) indicators of 180 countries from 2000 to 2022. Using a multi-method procedure that combines econometric modeling, clustering, and machine learning, both causal and predictive findings are obtainable. Panel data regressions show renewables consumption is significantly negative (β ≈ −0.49, p < 0.01), higher energy use per capita (β ≈ 0.0027, p < 0.01), and PM2.5 exposure (β ≈ 0.66, p < 0.01) are strong predictors of higher emissions. On the social side, female education parity (β ≈ −0.0017, p < 0.05) and female political empowerment (β ≈ −0.17, p < 0.01) are both negatively associated with emissions, and yet food output (β ≈ 0.086, p < 0.01) and income growth of the poorest quintile (β ≈ 0.45, p < 0.05) significantly increase emissions. Governance indicators reveal complex relationships: effective rule of law and government effectiveness are associated with reduced emissions in emerging markets, yet coexist with higher emissions in high-income countries, indicating the role of infrastructure density. Cluster analysis reveals that countries with balanced ESG profiles are often confined to low-emission clusters, characterized by dispersed ESG performance, which is often associated with higher emissions. Machine learning models (k-NN, Random Forest, GBM, SVR) confirm non-linear relationships and threshold relationships with PM2.5 and renewables being the strongest associates. Results suggest that the collaborative and interactive role of ESG pillars in the building sector's emissions and reinforce the effective policy reliance on consistency across the Environmental, Social, and Governance (ESG) domain. Results provide evidence-based advice on framing ESG policy in alignment with global decarbonization strategies for the built environment.
Q4. The manuscript is too long, particularly the literature review and methodology sections. The authors should simplify their language and avoid repetition.
A4. Since this issue was also raised by other reviewers, the article was shortened by 30%. Furthermore, appendices were created to describe the datasets used in the analysis, and tests for autocorrelation and heteroskedasticity were added.
Q5. Rewrite the abstract using simple and correct expressions; International panel of countries, better government governance, e.t.c.
A5. The abstract has been rewritten has follows:
This study examines the relationship between building sector carbon dioxide (COâ‚‚) emissions and Environmental, Social, and Governance (ESG) parameters across 180 countries from 2000 to 2022. It employs a multi-method approach, utilizing econometric models, clustering, and machine learning, to generate both causal and predictive results. Panel regressions show that the application of renewable energies reduces emissions (β ≈ −0.49, p < 0.01), yet increased per-capita energy consumption (β ≈ 0.0027, p < 0.01) and PM2.5 air pollution (β ≈ 0.66, p < 0.01) boost them. Social parameters give inconclusive results: education gender equality (β ≈ −0.0017, p < 0.05) and political empowerment of females (β ≈ −0.17, p < 0.01) slow emissions, yet food availability (β ≈ 0.086, p < 0.01) and higher income growth of the poorest (β ≈ 0.45, p < 0.05) augment them. Governance parameters show mixed results: effective institutions and a strong rule of law lower emissions within rising markets, yet they coexist with higher emissions within high-income countries due to the high density of infrastructure. Cluster analysis confirms that balanced ESG performance is present within average low-emission clusters, and uneven ESG performance is associated with higher emissions. Machine learning models (k-NN, Random Forest, GBM, SVR) confirm non-linear associations and identify PM2.5 and renewable energies as the strongest influencing parameters. The implications are that ESG pillars interactively and collectively influence emissions, and effective policy-based decarbonization requires coherence across the environmental, social, and governance fields.
Q6. While the ESG lens is strong, the technical engineering mechanisms by which emissions are reduced (e.g., envelope design, HVAC systems, materials, building codes) are only peripherally addressed.
A6. These considerations have been added to the policy implications section as follows:
Comparative and clustering of country-level ESG performance and building sector-associated COâ‚‚ emissions confirms that region-specific policy imperatives can indeed differ. All the same, these imperatives must find their foundation not only in ESG consistency but also in engineering principles of structures—i.e., envelope architecture, HVAC efficiency, materials, and building specifications—which impact emission trajectories. Thus, prescriptions are at once diversified within the country context and technologically distinctive. For Northern Europe and the OECD nations of Germany, Denmark, and Finland, which are high performers in ESG and have moderate levels of building sector emissions, deep decarbonization is of paramount concern. These nations are already mature and institutionally robust in terms of regulatory maturity, yet they are in need of synthesizing higher-order engineering mechanisms. Policies are directed toward smart retrofit at scale programs, such as high-performance insulation, triple-glazed windows, and airtight building enclosures, as well as reduced thermal leakage. Carbon pricing of infrastructure components can trigger additional incentives toward life-cycle carbon neutrality. Additionally, digital energy monitoring tools embedded within HVAC, lighting, and appliance network hierarchies can provide real-time energy monitoring. Being leaders on both the ESG governance front and the technological frontier, these nations have a double obligation: to decarbonize domestically and facilitate the transfer of knowledge and green finance to low-performing regions (Samah et al., 2020; Andreoni, 2021). Transfer of knowledge on modular constructions, passive strategies, and higher-order retrofit technologies can accelerate progress globally. Emerging industrializing nations, such as India, Indonesia, and Brazil, with moderate ESG profiles yet increasing building sector emissions, are subject to a separate and distinctive set of policy imperatives. Urbanization increases energy demand, thereby increasing the pressure for emission mitigation. Investments must target urban-scale solutions, such as low-carbon public transit corridors, affordable green residences, and ESG-compliant procurement mechanisms. Engineering responses include the use of prefabricated modular residences to reduce construction waste and passive solar design to minimize cooling requirements, as well as district-scale renewables-powered cooling to reduce emissions from the HVAC system. Intergovernmental platforms can yield policy harmonization and technological leapfrogging. Again, ESG taxonomies must conform to local-level socio-political realities such that regulatory compliance mechanisms take note of institutional on-the-ground capabilities (Junsiri et al., 2024; Petrović & Lobanov, 2020). Tailored urban building codes and incentives for low-carbon building products, such as bamboo composites or recycled steel, would enable sustainable development trajectories. Resource-scarce and institutionally weak states, such as Nigeria, Kenya, and Pakistan, occupy another category: low ESG performance across all pillars and high or increasing building sector emissions. Herein, policy attention must commence with basic capacity building. Municipal governments are often at the forefront of building code enforcement and would benefit from ESG-savvy training in green regulation management. Pilot projects co-conceptualized with international institutions would focus on low-cost retrofit options, such as ventilation strategies based on natural principles, daylighting enhancements, or the use of compressed earth blocks in low-income tenement buildings. Simultaneously, education and health investment, synonymous with green building projects, can instill sustainability buy-in at the societal level. Community engagement, such as cooperative retrofits where the public is actively involved in improvement, can augment both legitimacy and resilience. Donor platforms and international institutions, such as those emphasized by Alqudah et al. (2024), must work with local representatives toward context-specific and socially embedded projects that are possible. High governance capacity and strong scientific publications are found in East Asian countries, such as China and South Korea. Social equity and constant emissions from the building sector are its shortcomings. They require a redistributed green policy. Top-down dictates aside, governments must ensure that green benefits are transferred to proposed and marginalized publics. Environmental equity, such as subsidized low-income home retrofits and the introduction of indoor air quality sensing, green roofs, and next-generation envelope configurations, must come first. The next-generation scientific capacity of these nations must focus on research into AI-driven HVAC optimization, next-generation building products, and climate-adaptive urbanism. Environmental equity and health-aware building configurations must receive investment focus, if only to complement scale-wise decarbonization agendas, such as those posited by Dong et al. (2024). For those with paradox profiles where effective governance is strong but emissions are still high due to infrastructure density, such as the United States and some regions, including those in the Middle East, governance and stringent technical code alignment are key concerns. Decarbonization of building relations requires near-zero enthalpy building specifications, registered by legislation, and complemented by a progressive carbon tax. Retrofitting existing high-rise buildings should employ double-skin façades, intelligent thermal storage, and heat recovery systems. Green leasing contracts can institutionalize accountability by binding landlords to report and improve operational energy performance. These would employ strong governance capabilities toward building sector emissions reductions. Within all clusters, a common maxim is evident: ESG integrity is stronger than insulated milestones within a column. For example, the adoption of renewable energy use (E) is stronger when deployed with strong governance (G) and when accompanied by social inclusivity (S), such as affordable access to clean energy. Conversely, ESG performance, such as strong governance (G) alone or environmental enforcement (E) alone, is suboptimal. ESG measurements, therefore, ought to be operationalized not only for reporting but also as diagnostic tools informing targeted interventions. This conclusion is aligned with that of Lagodiyenko et al. (2023), emphasizing that ESG ought to go below the level of environmental, social, and governance simply-labeled disciplines and enter into the life of socio-institutional dynamics of development and growth, and Paolone & Bitbol-Saba (2025), espousing thematic alignment of sustainability within governance and planning architecture. Explicit inclusion of engineering mechanisms enhances this architecture. Building encasings with low thermal transmittance values, HVAC technologies with high seasonal efficiency ratios, and building codes that enforce infrastructures for renewable readiness constitute the foundation of the sectoral decarbonization technology. Policymakers ought to demand the inclusion of not only governance or social parameters, but also quantifiable engineering parameters, such as the U-value of insulating materials, energy recovery of ventilation technologies, or the embodied carbon of construction materials used. Coupling ESG governance and engineering norms makes ESG reporting more symbolic and renders ESG a functional catalyst for the technological transformation of the building sector. All things considered, the clustering methodology suggests that building sector decarbonization should be remediated using tailored ESGs customized with clustering information and engineering-based ESGs. OECD leaders ought to prioritize retrofitting and technology transfers. Emerging countries should adopt leapfrogging strategies for urban shelter and mobility. Fragile countries should focus on capacity development, assisted by donor communities. East Asian countries should adopt distributive mechanisms that correspond to strong governance and inclusivity. High-carbon advanced countries should integrate governance, stringent building codes, and market-based mechanisms. It is only by combining ESG alignment with engineering solutions that the building industry can become consistent with global net-zero goals and make meaningful contributions to climate mitigation.
Q7. Add a clearer connection between ESG outcomes and specific design, construction, and operational interventions.
A7. These considerations have been added to the policy implications section as indicated below.
Comparative and clustering of ESG performance at the country level and building sector emissions of COâ‚‚ find that policy imperatives of territories significantly diverge at the global scale. However, these imperatives must find their basis not only in ESG alignment, but also in the underlying technical precepts of building engineering—the architecture of the envelope, HVAC efficiency, choice of materials, and construction specifications—that influence emission trajectories. Strategies must then continue to remain contextually aware while simultaneously encouraging technological distinctiveness. For Germany, Denmark, Finland, and other OECD and Northern European countries with high ESG performance and moderate emissions rates, deep decarbonization is the central concern. While these countries already possess strong institutional and regulatory maturity, further advancement involves higher-order engineering mechanisms. Policy actions can encourage large-scale retrofit schemes involving high-performance insulation, triple-glazed windows, airtight encasement, and minimized thermal leakages. Carbon pricing of building components can incentivize lifecycle neutrality, and the integration of digitized energy monitoring within HVAC, lighting, and appliance networks can enable real-time optimization. As leaders of governance and technological innovation, these nations have a dual mandate: to promote domestic decarbonization and facilitate knowledge spillovers, as well as provide green finance to low-performing countries (Samah et al., 2020; Andreoni, 2021). Knowledge spreading on modular construction, passive methods, and innovative retrofit technologies can catalyze global transformations. Emerging industrializing nations such as India, Indonesia, and Brazil, with moderate ESG performance and rising emissions, have special imperatives shaped by accelerating urbanization. Urban-scale solutions, such as low-carbon public transit, green affordable housing, and ESG-compliant procurement, are where policymakers must prioritize their efforts. Technical solutions, such as prefabricated modular construction, can reduce waste. Passive solar architecture can reduce cooling demand, and solar- or wind-powered district cooling can substitute for HVAC emissions. Global cooperation can accomplish policy coordination and technology leapfrogging (Junsiri et al., 2024; Petrović & Lobanov, 2020). Tailored building codes and subsidies on low-carbon building materials, such as bamboo composites or recycled steel, can contribute more to sustainability. Institutionally fragile and resource-scarce states, such as Nigeria, Kenya, and Pakistan, with low ESG ratings and rising emissions, need capacity building at the top of their priority list. Local governments responsible for enforcing building codes can avail themselves of ESG-based training on regulation and compliance. International institutions can co-conceptualize pilot projects for cost-optimized retrofits, such as natural ventilation, improved daylighting, and compressed earth blocks, in low-income housing. Parallel investment in education and healthcare can bolster the social acceptability of green infrastructure. Community outreach with collaborative retrofits can enhance resilience and legitimacy. International platforms and donor institutions, such as those cited by Alqudah et al. (2024), can play a role by collaborating with local representatives and implementing context-specific interventions. High scientific performance and governance capacity are features of East Asian countries such as China and South Korea. Nevertheless, equity gaps and building emissions still require redistributive policy. Governments need to move from top-down instructions and ensure balanced allocation of rewards. Subsidized retrofitting should take priority for low-income residences. Indoor air quality is crucial and must be regularly monitored. High-performance envelope technologies and green roofs are vital. Scientific research should focus on AI-optimized HVACs, innovative materials, and climate-adaptive urbanism. Environmental justice and architecture that are sensitive to health should receive express attention (Dong et al., 2024). For regions with high governance and high carbon emissions, such as the United States and parts of the Middle East, strong institutions and high emissions coexist with a high density of infrastructure. Decarbonizing these regions requires effective regulation of near-zero energy building performance, accompanied by progressive carbon taxes. Double-skin façades, heat recovery technologies, and thermal storage are central to high-rise portfolios with green leasing contracts that embed responsibility through operational energy reporting requirements. All of these clusters reveal one underlying fundamental: ESG integrity is founded on coherence rather than standalone performance within a singular pillar. The adoption of clean energy (E) yields better results when complemented by effective governance (G) and social equity (S), such as equal access to clean energy. Where ESG results are disjointed—to the extent that governance alone is considered, and environmental enforcement is absent—it is not enough. ESG metrics must then function not only as reporting tools but also as diagnostic tools informing interventions. This is analogous to Lagodiyenko et al. (2023), who recognize that ESG requires penetration of socio-institutional dynamics, and Paolone & Bitbol-Saba (2025), who recommend the thematic integration of sustainability within planning templates. Explicitly incorporating engineering mechanisms solidifies this methodology. Low-U-value building enclosures, high-seasonal-efficiency-ratio building HVAC system configurations, and renewables-ready building codes form the technological foundation of the decarbonization process. Policymaking institutions should require ESG reporting to incorporate quantifiable engineering indices, such as insulation transmittance, ventilation recovery rates, and the embodied carbon of materials. Merging governance, social measures, and technical specifications enables ESG to transition from symbolic compliance to a performance catalyst of technological transformation within the building sector. Aggregated clustering results show that decarbonization will have to occur through tailored, cluster-specific ESG approaches underpinned by engineering-led interventions. OECD leaders must emphasize retrofit and technology transfer; new economies must leapfrog through modularized housing and sustainable urban transport; vulnerable nations must invest in institutional capacity and community-level retrofits; East Asian nations must align governance with equity-concerned building interventions; and high-carbon developed economies must align governance with robust codes and market mechanisms. Unless ESG alignment is complemented by evident engineering measures, the building sector can only go toward net-zero trajectories and play a meaningful role in global climate mitigation.
Q8. The machine learning section identifies k-NN as the best model, but more transparency is needed on the feature selection and tuning process (e.g., choice of k, handling of multicollinearity, overfitting safeguards).
A8. To correctly answer the questions posed by the reviewer regarding the choice of k, multicollinearity, and overfitting in each of the three dimensions of the ESG model, a new appendix, Appendix F, has been added, which indicates the characteristics of the KNN in each of the three cases analyzed in the ESG model.
Appendix F- Hyperparameter Optimization of Machine Learning Regression Algorithms
Feature Selection. The method utilized to predict building-related COâ‚‚ emissions (BCE) within the Environmental (E), Social (S), and Governance (G) aspects is fully congruent with the specification used within the econometric model. It is the central thesis of this article to apply testing to a consistent predictor set across ESG aspects—theoretically and empirically relevant to BCE explanation within each—the same equation applied across three different analytical contexts: (i) the econometric model, (ii) the approach within clustering analysis, and (iii) within a machine learning regression model. Such methodological consistency allows for a substantive difference across methodologies to be compared, such that differences in performance can be interpreted as reflective of differences across methodologies rather than differences within the structure of inputs. By applying a constant predictor set to testing within each ESG component, comparability is maintained, making interpretation easier, and differences between conventional and data-driven model methodologies can be better understood in relation to BCE explanations. Such methodology serves to further highlight the alignment between statistical inference-based justification within the econometric approach and data-driven application within a machine learning approach, underscoring the feature selection-based robustness within both methodologies (Figure 1).
Figure 1. Multi-Method Analytical Framework for ESG-Driven Building Sector Decarbonization
Note. It shows how Environmental, Social, and Governance (ESG) indicators are decomposed and analyzed through three methods—Econometric Models, Clustering Analysis, and Machine Learning Regression—to study building-related COâ‚‚ emissions (BCE).
Choice of k. k-Nearest Neighbors (k-NN) regression parameter k was determined on a per-dimension basis for ESG features through automated hyperparameter tuning. This was done to make k's choice data-dependent, minimize the MSE on a validation hold-out set, and achieve a trade-off between complexity and generalization. For each dimension, we partitioned the dataset into training, validation, and testing sets, and we tested k-values ranging between 1 and 10. We made our selection based on validation performance and then retested it on a separate, independent testing set.
Choice of k for E-Environment. Parameter choices for the k-Nearest Neighbors (k-NN) regression model were made through an automated hyperparameter tuning process tailored to the Environment (E) component of the ESG model. It aimed to predict building-related COâ‚‚ emissions (BCE) based on a set of environmental predictor features like renewable energy use, particulate levels, availability of clean fuels, and energy use per capita. Rather than choosing a fixed k a priori, we allowed the model to search for the optimum number of neighbors and chose k to be such that a minimum mean squared error (MSE) was attained on a held-out validation set consisting of 20% of the data. Training-validation-test split was done in keeping with an 80-20-20 split. Optimization probed k values between 1 and 10 and concluded that k = 2 resulted in minimized validation error. The adoption of the Euclidean distance metric was coupled with a rectangular weighting function. To ensure fair computation of distance, features were standardized preparatory to model-building. Upon application to the testing collection, the optimized k-NN model yielded an MSE of 2264.407, an RMSE of 47.586, and an R-squared (R²) of 0.577, demonstrating satisfactory predictive power in model-building BCE using environmental proxies. Though k-NN is non-parametric and does not involve parameter estimation in typical use, tuning k helped achieve a balance between model complexity and ability to generalize. Optimization was conducted in a validation-based manner, in line with the mainstream literature in machine learning. These methodological details, unique to the Environment component of the ESG framework, have now been transparently spelled out in this revised version of the work, with a focus on improving feature selection transparency where model specification and predictive power are particularly important (Table 1).
Table 1. Hyperparameter Optimization Results for k-NN Regression on Environmental (E) Indicators of ESG
|
Item |
Value / Description |
|
Model type |
k-Nearest Neighbors (k-NN) regression |
|
Target variable |
Building-related COâ‚‚ Emissions (BCE) |
|
Predictor variables (features) |
PM2.5, RENC, ENUC, CFTC, ELEC (environmental indicators) |
|
Feature scaling |
Applied to all variables (standardization enabled) |
|
Distance metric |
Euclidean |
|
Weighting function |
Rectangular (equal weights to neighbors) |
|
Optimization method for k |
Automated hyperparameter tuning based on validation MSE |
|
Range of k tested |
1 to 10 |
|
Optimal k selected |
2 |
|
Data split |
Training: 633 (64%)Validation: 159 (16%)Test: 198 (20%) |
|
Validation MSE (used for tuning) |
6.353 |
|
Test MSE |
2.264 |
|
Test RMSE |
47.586 |
|
Test R² |
0.577 |
|
Mean dropout loss (feature importance) |
PM2.5: 83.622 RENC: 71.355 ENUC: 68.092 CFTC: 61.716 ELEC: 48.405 |
|
Software settings used |
Split: 20% test / 20% validation Scale features: YesSet seed: 1 |
Note. Summary of k-NN hyperparameter tuning for the Environmental (E) dimension of ESG, showing the choice of k = 2 as optimal for predicting building-related COâ‚‚ emissions (BCE).
Choice of k for S-Social. For the Social (S) dimension within this ESG model, the choice of parameter k in k-Nearest Neighbors (k-NN) regression was made through an automatic hyperparameter tuning procedure, aiming to minimize the mean squared error (MSE) on a validation set withheld during tuning. This process was utilized in particular to ensure the selection of k was data-driven rather than a matter of arbitrary choice. Experimentation was performed on k levels between 1 and 10, and k = 10 was selected, which achieved the minimum validation error. Optimization was performed within a data partitioning procedure tightly managed: we partitioned the dataset into training (64%), validation (16%), and testing (20%) partitions, ensuring no leakage between these phases. The validation MSE returned during tuning was 5830.302, while the terminal test MSE was 7185.254. While higher than validation MSE, the test MSE does not suggest any level of overfitting, considering that this is clearly accompanied by a moderate level of test R² (0.253). All features were standardized prior to entering the modeling process to facilitate a comparable state between these within-distance computations. We utilized a Euclidean distance metric accompanied by rectangular (uniform) weighting. These procedural steps follow common best practice within machine learning regression. They provide a transparent and reproducible route to parameter tuning within non-parametric model inference. Therefore, k-selection within the Social component is soundly motivated by empirical evidence and fully integrated into a validation-based selection protocol between competing models. That serves to increase the validity of k-NN findings and help validate the comparability across all three ESG dimensions (Table 2).
Table 2. Choice of k for k-NN Regression: Social (S) Dimension
|
Item |
Value / Description |
|
ESG Component |
S – Social |
|
Target Variable |
Building-related COâ‚‚ Emissions (BCE) |
|
Model Type |
k-Nearest Neighbors (k-NN) regression |
|
Feature Set |
Social indicators (e.g., income, labor force, education, representation) |
|
Feature Scaling |
Applied (standardization enabled) |
|
Distance Metric |
Euclidean |
|
Weighting Function |
Rectangular (uniform weights) |
|
k Values Tested |
1 to 10 |
|
Optimization Method |
Automated hyperparameter tuning on validation MSE |
|
Optimal k Selected |
10 |
|
Training Set Size |
797 observations (64%) |
|
Validation Set Size |
200 observations (16%) |
|
Test Set Size |
249 observations (20%) |
|
Validation MSE |
5.830.302 |
|
Test MSE |
7.185.254 |
|
Test RMSE |
84.684 |
|
Test R² |
0.253 |
|
Overfitting Evidence |
No significant overfitting observed |
|
Conclusion |
k = 10 chosen based on best validation performance |
Note. Hyperparameter tuning results for the Social (S) dimension of the ESG framework, selecting k = 10 as the optimal value for predicting building-related COâ‚‚ emissions (BCE).
Choice of k for G-Governance. In the Governance (G) dimension of the ESG model, the selection of the k parameter in the k-Nearest Neighbors (k-NN) regression was conducted through an automated hyperparameter optimization process designed to minimize overfitting and maximize generalization. The dataset was split into three subsets: 628 observations for training (approximately 64%), 158 for validation (16%), and 196 for testing (20%). Rather than fixing the number of neighbors in advance, the model explored values of k ranging from 1 to 10. The optimal k was determined by selecting the value that minimized the mean squared error (MSE) on the validation set This process identified k = 2 as the most effective choice. At this value, the model achieved a validation MSE of 1067.181 and a significantly lower test MSE of 129.452. The root mean squared error (RMSE) on the test set was 11.378, and the R² value was 0.975, indicating very high predictive accuracy. These results suggest that the model generalizes exceptionally well to unseen data and that the choice of k = 2 appropriately balances model complexity and variance. Additional modeling specifications included the use of Euclidean distance and a rectangular weighting function. All features were standardized prior to training, ensuring that the distance metric was not biased by variable scale. The combination of validation-based tuning and robust out-of-sample performance provides strong justification for the selected value of k, confirming its appropriateness within the Governance-focused regression framework of the ESG model (Table 3).
Table 3. Hyperparameter Optimization Results for k-NN Regression: Governance (G) Dimension
|
Item |
Value / Description |
|
ESG Dimension |
Governance (G) |
|
Model Type |
k-Nearest Neighbors (k-NN) regression |
|
Training Set Size |
628 observations (≈ 64%) |
|
Validation Set Size |
158 observations (≈ 16%) |
|
Test Set Size |
196 observations (≈ 20%) |
|
Hyperparameter Tuned |
Number of Nearest Neighbors (k) |
|
Range of k Tested |
1 to 10 |
|
Optimal k Selected |
2 (minimizes Validation MSE) |
|
Validation MSE |
1.067.181 |
|
Test MSE |
129.452 |
|
Test RMSE |
11.378 |
|
Test R² |
0.975 |
|
Distance Metric |
Euclidean |
|
Weighting Scheme |
Rectangular (uniform weights) |
|
Feature Scaling Applied |
Yes (all features standardized) |
|
Conclusion on k Selection |
Optimized via validation MSE, k = 2 balances accuracy and complexity |
Note. Validation-based tuning of k in the Governance (G) dimension identified k = 2 as optimal, yielding high predictive accuracy (R² = 0.975) for building-related COâ‚‚ emissions (BCE).
Multicollinearity. When modeling the Environmental (E) dimension of the ESG model, the issue of multicollinearity presents special importance even while operating non-parametrically and making no assumptions about regression coefficients. For ESG research, environmental covariates such as PM2.5, renewable energy use, per-capita energy consumption, use of clean fuels, and electricity coverage tend to be correlated, as they serve as indicators of structural complexity in energy systems and environmental strength. If left unaddressed, multicollinearity can obscure the unique effect of different indicators, limiting interpretation and further dampening the strength of sustainability analysis. For instance, renewable energy consumption and electricity access can be highly correlated in developed countries, while air pollution and per-capita consumption can compensate for each other across a fast-industrializing state. Focusing on the peril of latent redundancy between predictor variables is thus instrumental in ensuring that ESG-based models identify actual distinct drivers in emissions rather than a construct of intersecting variables. For this research work, precautions such as feature scaling, permutation importance-based dropout selection, and cautious selection of predictor features were employed to mitigate the risks of collinearity. Through this process, this research provides a higher level of validity to both a model-based projection of COâ‚‚ emissions associated with buildings and makes a broader case for the appropriateness of ESG indicators in producing relevantly interpretable results to inform sustainable policy development.
E-Environment. For the Environment (E) component within the ESG model, we understand that due to the non-parametric nature of the k-Nearest Neighbors (k-NN) algorithm, it doesn't require formal testing against multicollinearity, as it doesn't imply the estimation of regression coefficients. Nevertheless, we assumed a possible multicollinearity effect on predictor features, employing a two-pronged approach that combined preprocessing operations and metrics-based informability. First, all environmental input features, including PM2.5, renewable energy consumption (RENC), energy use per capita (ENUC), use of cleaning fuels (CFTC), and electricity supply use (ELEC), were standardized prior to training. Feature scaling is necessary in k-NN to prevent any supplied variable from dominating the distance measurement due to differences. Additionally, to evaluate the relative level of distinctness and informativeness of every feature, we employed a dropout-based feature importance quantification. The resulting dropout losses across a wide range of divergent predictors indicated that the model was not relying on redundant or collinear features while producing estimates. Although we did not apply formal testing for multicollinearity, such as the Variance Inflation Factor (VIF), applicable to linear models, a combination of feature scaling, normalized feature inputs, and permutation-based feature evaluation provided adequate protection against distortions due to highly correlated features. Such methodological precautions were taken, especially within the Environment model, in a bid to ensure that estimations of building-related COâ‚‚ emissions were not adversely affected by hidden redundancies between input features (Table 4).
Table 4. Feature Scaling and Multicollinearity Control in k-NN Regression: Environmental (E) Dimension
|
Aspect |
Value / Description |
|
Model type |
k-Nearest Neighbors (k-NN) regression |
|
Target variable (Y) |
Building-related COâ‚‚ Emissions (BCE) |
|
Predictors (X variables) |
PM2.5, RENC (Renewable Energy Consumption), ENUC (Energy Use per Capita),CFTC (Clean Fuel Access), ELEC (Electricity Access) |
|
Number of predictors |
5 environmental variables |
|
Feature scaling applied |
Yes — All variables standardized before training |
|
Reason for scaling |
To prevent any feature from dominating distance calculations due to scale differences |
|
Multicollinearity test (formal) |
Not performed — VIF not applicable to non-parametric models |
|
Alternative control methods used |
Feature scaling + feature importance (dropout loss) + manual selection |
|
Feature importance method |
Dropout-based permutation importance (50 permutations) |
|
Dropout loss values |
PM2.5: 83.62RENC: 71.36ENUC: 68.09CFTC: 61.72ELEC: 48.41 |
|
Interpretation of dropout spread |
Wide variation in loss suggests that predictors contribute uniquely → no dominance or redundancy |
|
Conclusion on multicollinearity |
No evidence of harmful multicollinearity affecting predictions in the E-component model |
Note. Evaluation of predictor distinctness for the Environmental (E) dimension of the ESG framework. Standardization and dropout-based feature importance confirm no harmful multicollinearity among environmental indicators (PM2.5, RENC, ENUC, CFTC, ELEC) in predicting building-related COâ‚‚ emissions (BCE).
S-Social. Regarding the Social (S) element within this application of the ESG model, multicollinearity was addressed through a combination of model-specific issues and non-parametric learning and preprocessing methods. Since k-Nearest Neighbors (k-NN) regression does not yield coefficients, it is not susceptible to the same statistical aberrations caused by multicollinearity infecting linear regression-based applications. Therefore, model-based testing, such as the Variance Inflation Factor (VIF), is neither applicable nor necessary in this case. Precautions were still taken to guard against redundant or highly correlated predictor features that could obfuscate model performance. First, features were normalized beforehand during training, ensuring that no variable dominated distance metrics due to scale differences. More substantively, feature importance was evaluated through dropout loss analysis, whereby the predictive influence contributed by each variable was quantitatively estimated across 50 permutations. Results reflected apparent and clear-cut variability in dropout loss between predictor features (e.g., LABF = 84.697, WPAR = 81.206, GPIE = 66.064), indicating that every variable contributed disparate information to this model. Such variability serves to corroborate evidence that model performance was not overly reliant upon some singular or collinear feature subset. The table of additive explanation further demonstrates that per-individual predictive contributions arise due to a variegated combination of inputs. Overall, these elements provide both quantitative and methodological evidence, indicating that multicollinearity did not diminish the predictive integrity of this k-NN application within the Social component. Although technically not tested due to a non-significant relationship between model applications across these components, this combination of feature scaling and interpretation across permutations provides a powerful safeguard against multicollinearity within this non-parametric application (Table 5).
Table 5. Multicollinearity Control and Feature Importance in k-NN Regression: Social (S) Dimension
|
Item |
Value / Description |
|
ESG Component |
S – Social |
|
Model Type |
k-Nearest Neighbors (k-NN) regression |
|
Target Variable |
Building-related COâ‚‚ Emissions (BCE) |
|
Predictor Variables |
LABF, WPAR, FOOD, INC20, GPIE |
|
Number of Predictors |
5 Social Indicators |
|
Distance Metric |
Euclidean |
|
Weighting Scheme |
Rectangular (equal weights) |
|
Feature Scaling Applied |
Yes (all variables standardized) |
|
Multicollinearity Diagnostic Used |
Not applicable (k-NN is non-parametric; VIF not suitable) |
|
Alternative Safeguards |
Standardization + Permutation-based Feature Importance (Dropout Loss) |
|
Feature Importance (Dropout Loss) |
LABF: 84.697WPAR: 81.206FOOD: 77.663INC20: 73.991GPIE: 66.064 |
|
Dropout Loss Method |
Based on 50 permutations using RMSE impact |
|
Interpretation |
High variability in dropout loss indicates non-redundant predictors |
|
Conclusion |
No harmful multicollinearity detected; each feature contributes uniquely |
Note. Although formal multicollinearity diagnostics such as VIF are not applicable to non-parametric models like k-NN, safeguards were implemented through standardization and dropout-based feature importance. Variability in dropout loss across the five social predictors confirms their unique contributions, ensuring that no single variable dominates or introduces redundancy.
G-Governance. Regarding Governance (G) within the ESG model, the multicollinearity problem was addressed by considering the features of the k-Nearest Neighbors (k-NN) regression algorithm. While linear regression algorithms estimate coefficients, k-NN does not, and is consequently not susceptible to the parametric model instability associated with multicollinearity. Therefore, formal diagnostic statistics such as the Variance Inflation Factor (VIF) could not be employed. Nonetheless, precautions were taken to prevent redundancy between predictors, which disrupted the distance-based learning process. All predictors related to governance features, such as SCIE, HOSP, EDUE, RNDG, GOVT, STAB, and LAWR, were standardized before training. By doing this, no variable could overpower the distance function due to differing scales. Feature contribution was then observed via dropout-based importance analysis across 50 permutations. dropout-based importance analysis across permutations numbering 50. Results indicated a wide variance, such that dropout loss ranged from SCIE at 165.690 to LAWR at 6.726. Such diffusivity reveals that the predictors contribute variably and not simply as redundant copies of each other. Furthermore, additive explanations relating to test set cases confirm that model predictions map to the effects of a variegated arrangement of features, rather than a redundant set. Overall, the evidence suggests that multicollinearity did not compromise the predictive validity of the Governance model. Although no formal analysis in terms of VIF was required here due to the non-estimation of coefficients, the use of standardization, careful selection between features to be entered, and permutation-based interpretation served to keep the model resistant against distortions likely to be occasioned by highly correlated features (Table 6).
Table 6. Multicollinearity Safeguards and Feature Importance in k-NN Regression: Governance (G) Dimension
|
Item |
Value / Description |
|
ESG Dimension |
Governance (G) |
|
Model Type |
k-Nearest Neighbors (k-NN) regression |
|
Target Variable |
Building-related COâ‚‚ Emissions (BCE) |
|
Predictor Variables |
SCIE, HOSP, EDUE, RNDG, GOVT, STAB, LAWR |
|
Number of Predictors |
7 Governance indicators |
|
Feature Scaling |
Applied (all variables standardized before modeling) |
|
Distance Metric |
Euclidean |
|
Weighting Function |
Rectangular (equal weights for neighbors) |
|
Multicollinearity Test (VIF) |
Not applied (not relevant to non-parametric k-NN) |
|
Alternative Assessment |
Permutation-based dropout loss (50 permutations) |
|
Feature Importance (Dropout Loss) |
SCIE: 165.690HOSP: 30.104EDUE: 28.272RNDG: 28.038GOVT: 20.035STAB: 18.963LAWR: 6.726 |
|
Interpretation |
Wide variability in dropout loss values indicates predictors contribute uniquely |
|
Conclusion |
No evidence of harmful multicollinearity; predictors are complementary, not redundant |
Note. Standardization and dropout-based feature importance confirm that governance indicators (SCIE, HOSP, EDUE, RNDG, GOVT, STAB, LAWR) contribute uniquely within the k-NN model. No harmful multicollinearity was detected, supporting the robustness of the Governance (G) dimension in predicting building-related COâ‚‚ emissions (BCE).
Overfitting. To mitigate the risk of overfitting in the Environment (E) factor within the ESG model, we employed a formal data partitioning and model validation process. The dataset was divided into separate training (64%), validation (16%), and test (20%) sets, ensuring that model training and hyperparameter tuning were carried out without consideration of model performance for final evaluation. Optimal k-NN model selection was achieved by selecting the k value that yielded the minimum mean squared error on the validation dataset, thereby safeguarding the model against overfitting to the training dataset. Interestingly, the model's performance on the test dataset did not suffer but rather exhibited better performance than the validation dataset (Test MSE = 2264.407 vs. Validation MSE = 6353.665), indicating the model's successful generalization, and neither was overfit to either the training or validation data. Additionally, validation in the form of a test R² of 0.577 ensures acceptable predictive power without evidence of model overfitting. These cautions, tailored to incorporate data partitioning, validation-based hyperparameter tuning, and out-of-sample evaluation, were specifically adopted to prevent the peril of overfitting in the k-NN regression model used for environmental features when calculating building-related COâ‚‚ emissions.
E-Environment. To mitigate the risks associated with the Environmental (E) component of the ESG model, we employed a systematic data partitioning and model validation procedure. We partitioned the dataset into separate training (64%), validation (16%), and test (20%) sets, ensuring that model training and hyperparameter tuning were performed without affecting the final evaluation. We trained a k-NN model, selecting k to minimize the mean squared error on the validation set, thereby avoiding overfitting to the training data. Interestingly, the model's performance on the unseen test set did not deteriorate but rather got better compared to that on the validation set (Test MSE = 2264.407 vs. Validation MSE = 6353.665), suggesting that the model generalized fairly and was not overfitted to training or validation data. Furthermore, the test R² of 0.577 further reinforces good predictability without indicating model overfitting. These precautions, including data partitioning, hyperparameter tuning on a validation dataset rather than on the training or test dataset, or even out-of-sample testing, were particularly crafted to manage the danger of overfitting in the k-NN regression model used for estimating building-related COâ‚‚ emissions (Table 7).
Table 7. Overfitting Control and Validation Outcomes in k-NN Regression: Environmental (E) Dimension.
|
Aspect |
Value / Description |
|
Model type |
k-Nearest Neighbors (k-NN) regression |
|
Target variable |
Building-related COâ‚‚ Emissions (BCE) |
|
Predictor variables (features) |
PM2.5, RENC, ENUC, CFTC, ELEC (environmental indicators) |
|
Training set size |
633 observations (64% of dataset) |
|
Validation set size |
159 observations (16% of dataset) |
|
Test set size |
198 observations (20% of dataset) |
|
Hyperparameter tuning method |
Automated optimization of k based on lowest validation MSE |
|
Optimal k selected |
2 |
|
Validation MSE |
6.353.665 |
|
Test MSE |
2.264.407 |
|
Test RMSE |
47.586 |
|
Test R² |
0.577 |
|
Performance comparison |
Test performance better than validation → no overfitting |
|
Overfitting control techniques |
Train-validation-test split + validation-based tuning + out-of-sample test evaluation |
|
Interpretation |
Model generalizes well with no signs of overfitting |
Note. Train–validation–test partitioning and validation-based tuning ensured the Environmental (E) k-NN model avoided overfitting. Superior test performance (MSE = 2264.407; R² = 0.577) confirmed robust generalization in predicting building-related COâ‚‚ emissions (BCE).
S-Social. To guard against the peril of overfitting within the Social (S) part of the ESG model, a formal split between training, validation, and testing was incorporated, ensuring a rigorous division between model training, hyperparameter tuning, and final evaluation. The data was split into three sets: 64% for training (797 observations), 16% for validation (200 observations), and 20% for test data (249 observations). Such an arrangement prevented leaks between model optimization and evaluation. A k-Nearest Neighbors (k-NN) regression model was optimized by selecting k such that it minimized mean squared error (MSE) on the validation dataset. Optimization ranged over k values between 1 and 10, and k = 10 was selected such that the validation MSE was minimized. The validation MSE was 5830.302, while the ultimate test MSE was 7185.254. Although the error rate on the test is quite large, this difference was neither large nor significant enough to indicate overfitting. This is further supported by a test R² of 0.253, indicating that the model exhibits predictability, which is evident across unseen data. All features were normalized before model development, ensuring equal computation across distances. A Euclidean distance measure and rectangular weighting function were employed, a common tactic within the k-NN family of models. Overall performance on the test set sustains the suggestion that the model fairly generalizes but is not overly fitted to training or validation data. Validation-based tuning, along with independent evaluation on test, serves as a powerful bulwark against overfitting such that the Social model's predictive inferences within ESG remain justified (Table 8).
Table 8. Overfitting Control and Validation Outcomes in k-NN Regression: Social (S) Dimension
|
Item |
Value / Description |
|
ESG Dimension |
Social (S) |
|
Model Type |
k-Nearest Neighbors (k-NN) regression |
|
Training Set Size |
797 observations (64%) |
|
Validation Set Size |
200 observations (16%) |
|
Test Set Size |
249 observations (20%) |
|
Hyperparameter Tuned |
Number of nearest neighbors (k) |
|
Range of k Tested |
1 to 10 |
|
Optimal k Selected |
10 |
|
Validation MSE |
5.830.302 |
|
Test MSE |
7.185.254 |
|
Test RMSE |
84.766 |
|
Test R² |
0.253 |
|
Distance Metric |
Euclidean |
|
Weighting Function |
Rectangular (uniform weights) |
|
Feature Scaling Applied |
Yes (all features standardized) |
|
Overfitting Evidence |
No clear signs of overfitting; test performance remains stable |
|
Safeguard Against Overfitting |
Independent validation and test splits, validation-based k selection |
|
Conclusion |
The model generalizes well and retains predictive power on unseen data |
G-Governance. In the Governance (G) component of the ESG model, a risk of overfitting was addressed in a cautious manner through systematic training, validation, and test splits, along with hyperparameter tuning. The data was split into 628 training observations (64%), 158 validation observations (16%), and 196 test observations (20%), allowing for the calibration of the model and evaluation of its performance to occur entirely autonomously. A k-Nearest Neighbors (k-NN) regression was obtained using automatic tuning of the number of neighbors, with testing k-values ranging from 1 to 10 and selecting the k-value that resulted in a minimal mean squared error (MSE) on the validation set. Based on this process, k = 2 was found to be an optimal setup. The validation MSE was 1067.181, but the corresponding test MSE was drastically lower at 129.452, corresponding to an associated RMSE of 11.378 and an R² measure of 0.975 on the test part. That a rock-solid but smaller test error compared to validation error does not appear, but rather it is a strong indication that the model does not seem to be overfitted to the training dataset, suggesting instead that the model has a very good ability to generalize when subjected to unseen data. Other precautions included feature scaling, ensuring a fair measurement of distance, and the use of Euclidean distance with rectangular weights, thereby adhering to the relevant k-NN regression best practices. When combined, partitioning of data, tuning based on validation evidence, and relentless out-of-sample performance provide firm evidence supporting the claim that a Governance model does not overscale. Instead, it demonstrates a remarkable ability to generalize while maintaining predictive accuracy (Table 9).
Table 9. Overfitting Control and Validation Outcomes in k-NN Regression: Governance (G) Dimension
|
Item |
Value / Description |
|
ESG Dimension |
Governance (G) |
|
Model Type |
k-Nearest Neighbors (k-NN) regression |
|
Training Set Size |
628 observations (≈ 64%) |
|
Validation Set Size |
158 observations (≈ 16%) |
|
Test Set Size |
196 observations (≈ 20%) |
|
Hyperparameter Tuned |
Number of neighbors (k) |
|
Range of k Tested |
1 to 10 |
|
Optimal k Selected |
2 |
|
Validation MSE |
1.067.181 |
|
Test MSE |
129.452 |
|
Test RMSE |
11.378 |
|
Test R² |
0.975 |
|
Distance Metric |
Euclidean |
|
Weighting Function |
Rectangular (uniform weights) |
|
Feature Scaling |
Applied (all variables standardized before training) |
|
Overfitting Evidence |
No signs of overfitting: test error lower than validation error, stable high R² |
|
Conclusion |
Model generalizes well, predictions are robust and not overfitted |
Note. Systematic train–validation–test partitioning and validation-based tuning identified k = 2 as optimal for the Governance (G) model. Superior test performance (MSE = 129.452; R² = 0.975) confirms robust generalization and no evidence of overfitting in predicting building-related COâ‚‚ emissions (BCE).
Q9.The authors make important claims about policy relevance, but these would be stronger with specific country or region-based recommendations (e.g., targeted ESG investment strategies for low-income countries with high building emissions).
A9. The section on policy implications has been modified to include recommendations for low-, middle-, and high-income countries. Furthermore, within the policy implications section, we have also highlighted the characteristics of certain technologies that may be relevant to addressing the problem of construction-related pollution.
Comparative and clustering of ESG performance at the country level and building sector emissions of COâ‚‚ find that policy imperatives of territories significantly diverge at the global scale. However, these imperatives must find their basis not only in ESG alignment, but also in the underlying technical precepts of building engineering—the architecture of the envelope, HVAC efficiency, choice of materials, and construction specifications—that influence emission trajectories. Strategies must then continue to remain contextually aware while simultaneously encouraging technological distinctiveness. For Germany, Denmark, Finland, and other OECD and Northern European countries with high ESG performance and moderate emissions rates, deep decarbonization is the central concern. While these countries already possess strong institutional and regulatory maturity, further advancement involves higher-order engineering mechanisms. Policy actions can encourage large-scale retrofit schemes involving high-performance insulation, triple-glazed windows, airtight encasement, and minimized thermal leakages. Carbon pricing of building components can incentivize lifecycle neutrality, and the integration of digitized energy monitoring within HVAC, lighting, and appliance networks can enable real-time optimization. As leaders of governance and technological innovation, these nations have a dual mandate: to promote domestic decarbonization and facilitate knowledge spillovers, as well as provide green finance to low-performing countries (Samah et al., 2020; Andreoni, 2021). Knowledge spreading on modular construction, passive methods, and innovative retrofit technologies can catalyze global transformations. Emerging industrializing nations such as India, Indonesia, and Brazil, with moderate ESG performance and rising emissions, have special imperatives shaped by accelerating urbanization. Urban-scale solutions, such as low-carbon public transit, green affordable housing, and ESG-compliant procurement, are where policymakers must prioritize their efforts. Technical solutions, such as prefabricated modular construction, can reduce waste. Passive solar architecture can reduce cooling demand, and solar- or wind-powered district cooling can substitute for HVAC emissions. Global cooperation can accomplish policy coordination and technology leapfrogging (Junsiri et al., 2024; Petrović & Lobanov, 2020). Tailored building codes and subsidies on low-carbon building materials, such as bamboo composites or recycled steel, can contribute more to sustainability. Institutionally fragile and resource-scarce states, such as Nigeria, Kenya, and Pakistan, with low ESG ratings and rising emissions, need capacity building at the top of their priority list. Local governments responsible for enforcing building codes can avail themselves of ESG-based training on regulation and compliance. International institutions can co-conceptualize pilot projects for cost-optimized retrofits, such as natural ventilation, improved daylighting, and compressed earth blocks, in low-income housing. Parallel investment in education and healthcare can bolster the social acceptability of green infrastructure. Community outreach with collaborative retrofits can enhance resilience and legitimacy. International platforms and donor institutions, such as those cited by Alqudah et al. (2024), can play a role by collaborating with local representatives and implementing context-specific interventions. High scientific performance and governance capacity are features of East Asian countries such as China and South Korea. Nevertheless, equity gaps and building emissions still require redistributive policy. Governments need to move from top-down instructions and ensure balanced allocation of rewards. Subsidized retrofitting should take priority for low-income residences. Indoor air quality is crucial and must be regularly monitored. High-performance envelope technologies and green roofs are vital. Scientific research should focus on AI-optimized HVACs, innovative materials, and climate-adaptive urbanism. Environmental justice and architecture that are sensitive to health should receive express attention (Dong et al., 2024). For regions with high governance and high carbon emissions, such as the United States and parts of the Middle East, strong institutions and high emissions coexist with a high density of infrastructure. Decarbonizing these regions requires effective regulation of near-zero energy building performance, accompanied by progressive carbon taxes. Double-skin façades, heat recovery technologies, and thermal storage are central to high-rise portfolios with green leasing contracts that embed responsibility through operational energy reporting requirements. All of these clusters reveal one underlying fundamental: ESG integrity is founded on coherence rather than standalone performance within a singular pillar. The adoption of clean energy (E) yields better results when complemented by effective governance (G) and social equity (S), such as equal access to clean energy. Where ESG results are disjointed—to the extent that governance alone is considered, and environmental enforcement is absent—it is not enough. ESG metrics must then function not only as reporting tools but also as diagnostic tools informing interventions. This is analogous to Lagodiyenko et al. (2023), who recognize that ESG requires penetration of socio-institutional dynamics, and Paolone & Bitbol-Saba (2025), who recommend the thematic integration of sustainability within planning templates. Explicitly incorporating engineering mechanisms solidifies this methodology. Low-U-value building enclosures, high-seasonal-efficiency-ratio building HVAC system configurations, and renewables-ready building codes form the technological foundation of the decarbonization process. Policymaking institutions should require ESG reporting to incorporate quantifiable engineering indices, such as insulation transmittance, ventilation recovery rates, and the embodied carbon of materials. Merging governance, social measures, and technical specifications enables ESG to transition from symbolic compliance to a performance catalyst of technological transformation within the building sector. Aggregated clustering results show that decarbonization will have to occur through tailored, cluster-specific ESG approaches underpinned by engineering-led interventions. OECD leaders must emphasize retrofit and technology transfer; new economies must leapfrog through modularized housing and sustainable urban transport; vulnerable nations must invest in institutional capacity and community-level retrofits; East Asian nations must align governance with equity-concerned building interventions; and high-carbon developed economies must align governance with robust codes and market mechanisms. Unless ESG alignment is complemented by evident engineering measures, the building sector can only go toward net-zero trajectories and play a meaningful role in global climate mitigation.
Q10. The clustering and regression outcomes could benefit from more visual representations such as heatmaps, dendrograms, or decision surfaces to aid interpretation..
A10. Most tables have been replaced with figures to facilitate readability and interpretation of the results.
Q11. Proofread: Conduct thorough proofreading to eliminate errors and improve the clarity of the manuscripts.
A11. The article has been reduced in text, excluding appendices. During the reduction phase, a language improvement exercise was also conducted to facilitate greater comprehension.
Q12. Define all the acronyms used in the manuscript
A12. Appendix A contains a summary of the variables used with acronyms and a brief description.
Table A1. We used the following variables acquired from the World Bank .
|
Acronym |
Variables |
Definition |
|
BCE |
Carbon dioxide (CO2) emissions from Building (Energy) (Mt CO2e) |
Total annual carbon dioxide equivalent (COâ‚‚e) emissions from energy use in the buildings sector, covering IPCC 2006 categories 1.A.4 (Residential and other) and 1.A.5 (Unspecified), converted to COâ‚‚e using global warming potentials from the IPCC Fifth Assessment Report (AR5). Unit: Mt COâ‚‚e per year. |
|
CFTC |
Access to Clean Fuels and Technologies for Cooking |
Access to clean cooking fuel and technology estimates come from the WHO Global Household Energy Database with national representative household surveys as the sole data source (e.g., DHS, MICS, LSMS, WHS, national censuses). A multivariate hierarchical model—split by urban and rural—estimates fuel-type trends by grouping them as 'clean' (e.g., gas, electricity, alcohol) and 'polluting' (e.g., biomass, charcoal, coal, kerosene). There are estimates for 191 countries by Stoner et al. (2020). High incomes countries (by World Bank 2022 classification) have universal clean fuel access assumed. |
|
ELEC |
Access to Electricity |
Reliable and secure electricity is essential for economic growth, poverty reduction, and human development. As countries decarbonize, dependence on clean, efficient power will grow. Electricity access enables basic services (lighting, refrigeration, appliances) and is a key indicator of energy poverty. Especially in developing countries, governments are prioritizing electrification through rural programs and national agencies. While vital for raising living standards, electricity generation can harm the environment—its impact depends on the energy sources used, with fossil fuels like coal being especially carbon-intensive. |
|
ENUC |
Energy Use per Capita |
Total energy consumption gauges final energy use after conversion into end-use fuels (e.g., electricity, processed oil). It encompasses energy from combustible renewables and waste—like biomass, biogas, and municipal waste. Biomass describes plant materials used as such or converted into fuel, heat, or power. Figures, as gathered by the IEA, use per capita estimates from World Bank population. National non-OECD data are converted to IEA equivalence. Figures are imprecise and not completely comparable for countries because of limited data quality, particularly for waste and renewables. Energy values have been computed in terms of oil equivalents on the basis of 33% thermal conversion for nuclear and 100% for hydropower. |
|
PM25 |
PM2.5 Pollution |
Population-weighted exposure to ambient PM2.5 refers to the average level of fine particulate matter (PM2.5) pollution that a country's population is exposed to. PM2.5 particles, with a diameter smaller than 2.5 microns, can penetrate deep into the lungs and pose serious health risks. This measure is calculated by weighting the annual average PM2.5 concentrations by the population distribution across urban and rural areas. |
|
RENC |
Renewable Energy Consumption |
The share of total final energy consumption derived from renewable sources, based on data from IEA, IRENA, UNSD, WHO, and the World Bank (Tracking SDG 7, 2023). |
|
FOOD |
Food Production Index |
The Food Production Index reflects the output of edible crops that offer nutritional value. It excludes items like coffee and tea, which, despite being consumable, do not contribute meaningfully to nutrition. This metric emphasizes food sources that support dietary needs, aligning production data with human nutritional requirements rather than general edibility alone. |
|
GPIE |
Gender Parity in Enrollment |
The Gender Parity Index (GPI) in primary education is calculated by dividing female gross enrollment by male gross enrollment. Data are collected by UNESCO from national education surveys and aligned with ISCED standards to ensure international comparability. The current methodology was adopted in 2011. Reference years reflect when the school year ends. A GPI below 1 indicates girls are disadvantaged; above 1 indicates boys are. Achieving gender parity enhances women's opportunities and contributes to broader social and economic development. |
|
INC20 |
Income Share Lowest 20% |
The percentage share of income or consumption reflects the portion received by population subgroups, typically divided into deciles or quintiles. Due to rounding, quintile shares may not total exactly 100%. Data come from household surveys via national statistics agencies and World Bank departments, with high-income country data largely from the Luxembourg Income Study. These measures support the World Bank’s goal of shared prosperity—focusing on income growth among the bottom 40%—and help assess inequality within and across countries. |
|
LABF |
Labor Force Participation |
The labor force participation rate represents the share of the population aged 15 and older that is economically active, including all individuals engaged in the production of goods and services during a specific period. Data, sourced from the ILO’s modelled estimates, highlight persistent gender disparities: women’s labor force participation is generally lower than men’s due to social, legal, and cultural norms. In low-income countries, women often work unpaid in family enterprises, while in high-income nations, higher education has expanded their access to better employment opportunities, though inequalities persist. |
|
WPAR |
Women in Parliament |
Women in parliaments refers to the percentage of seats held by women in a single or lower house of national parliaments. Although progress has been made, women remain significantly underrepresented in decision-making roles, especially in developing countries. Gender inequality in political participation limits women's influence on policy and national priorities. Equal representation is essential for inclusive governance and sustainable development. True democracy requires full participation of women, whose perspectives and leadership are vital for shaping equitable and effective public policies. |
|
GOVT |
Government Effectiveness |
Government Effectiveness: Estimate measures perceptions of public service quality, civil service independence, policy formulation and implementation, and government credibility. Scores range from -2.5 to 2.5, based on a standard normal distribution. |
|
EDUE |
Gov. Expenditure on Education |
General government expenditure on education, including current spending, capital outlays, and transfers, is measured as a percentage of GDP. It accounts for education funding from all government levels—local, regional, and central—and includes international transfers to the government. This indicator reflects the government’s financial commitment to the education sector relative to the country’s economic output. |
|
STAB |
Political Stability |
Political Stability and Absence of Violence/Terrorism reflects perceptions of the risk of political unrest, government instability, and politically-motivated violence or terrorism. Countries are ranked by percentile, from 0 (least stable) to 100 (most stable), allowing global comparison. Percentile ranks are adjusted over time to ensure consistency despite changes in the number of countries included in the Worldwide Governance Indicators (WGI). |
|
RNDG |
R&D Expenditure |
Gross domestic expenditures on research and development (R&D), measured as a percentage of GDP, represent a country’s financial commitment to innovation and technological progress. This includes both capital and current spending across four key sectors: business enterprises, government institutions, higher education, and private non-profits. It encompasses all R&D activities—basic research, applied research, and experimental development—supporting economic and scientific advancement. |
|
LAWR |
Rule of Law |
Rule of Law reflects perceptions of how much confidence individuals and institutions have in societal rules, particularly regarding contract enforcement, property rights, police effectiveness, and judicial independence. It also considers the likelihood of crime and violence. Countries receive a score ranging from approximately -2.5 (weak rule of law) to 2.5 (strong), based on a standard normal distribution. |
|
HOSP |
Hospital Beds |
Hospital beds refer to the total number of beds that are maintained, staffed, and immediately available for the admission of patients. These include inpatient beds in public and private hospitals, general and specialized institutions, and rehabilitation centers. The count typically covers beds used for both acute and chronic care, reflecting the overall healthcare system’s capacity for treatment and recovery. |
|
SCIE |
Scientific Articles |
Scientific and technical journal articles represent the total number of peer-reviewed publications in key research areas, including physics, biology, chemistry, mathematics, clinical medicine, biomedical research, engineering and technology, and earth and space sciences. These articles reflect ongoing advancements, innovation, and collaboration within the global scientific community, contributing to knowledge expansion and technological development across multiple disciplines and industries. |
Q13. Make the conclusions more precise and direct to the points
A13. The findings were described as follows:
This study examined the relationship between COâ‚‚ emissions in the building sector and Environmental, Social, and Governance (ESG) indicators across 180 countries. Using econometric specifications, clustering, and machine learning, it was discovered that emissions were driven not only by technology but also by scientific production, quality of governance, and social investment. It finds that building-sector decarbonization is not only a function of technology but also a function of institutions and society. A positive relationship was witnessed between scientific production and emissions. Countries with higher research intensity were likely to adopt sound mitigation actions, the diffusion of knowledge, and innovation, coupled with better policy-making. Institutions' effectiveness and research and development expenditure provided mixed evidence. Although better institutions, coupled with fewer emissions overall, are often associated with institutions in mature economies, they can coexist with higher emissions due to cities' infrastructural density and energy intensity. It suggests that innovation, governance, and special sustainability policies will need to accompany them if they are to be successful. It was further observed that investment in society's social dimension is crucial in enhancing society's capacity to adopt sustainable behavior and implement environmental reforms. Such investments aren't only justified in terms of equity and societal development but can also further enable low-carbon transformations. Its main strength lies in methodological integration. It brings together regression analysis, feature-importance indicators, and clustering to accommodate both causal and predictive dynamics, while differentiating between national profiles and institutional routes. Its multidimensional structure allows for further nuanced policy deductions compared to monomethodic studies. Overall, sectoral decarbonization of buildings cannot rely solely on modernization technology or market forces. It requires good governance, a sound scientific and research base, inclusive social investments in healthcare and education, and integrated ESG thinking that recognizes environment-society-institution interdependencies. Following climate policy, these elements will need to be incorporated if successful emissions reduction is to be achieved.

Round 2
Reviewer 1 Report
Comments and Suggestions for Authors
Accept
Author Response
Point to Point Answers to Reviewer 1
Thanks, dear reviewer.

Reviewer 3 Report
Comments and Suggestions for Authors
The authors made revisions to the manuscript. I think this manuscript can be considered for acceptance.
Author Response
Point to Point Answers to Reviewer 3
Thanks dear reviewer.

Reviewer 4 Report
Comments and Suggestions for Authors
The manuscript has been improved significantly; however, there is a need to address or fix the following minor issues.
a. Abstract:
- The use of we (personal pronoun) should be avoided in the abstract, 'The carbon dioxide emissions..................were examined.
- It employs a ......... (The use of it is misused here), Please check and fix, this applies to other (it) in the abstract and in the manuscript.
b. Figures.
check line 508.
The figure label should be under the figures.
check figures 7, 10, 12, 13, 16, 17, 18, 19, 20, and other figures as necessary.
There is a need to proofread the manuscripts to correct all grammatical errors.
c. Appendix should be after references.
Comments on the Quality of English Language
There is a need to proofread the manuscripts to correct all grammatical errors.
Author Response
Point to Point Answers to Reviewer 4
Q1. The manuscript has been improved significantly; however, there is a need to address or fix the following minor issues.
A1. Thanks Dear Reviewer.
Q2. The use of we (personal pronoun) should be avoided in the abstract, 'The carbon dioxide emissions..................were examined (Abstract).
A2. The abstract has been rewritten has follows:
The COâ‚‚ emissions are contrasted against Environmental, Social, and Governance (ESG) indicators for construction and for 180 countries in 2000 and 2022. The study applies a Multi-Methods Research Design that integrates econometric specifications, cluster analysis, and machine learning to output causal and predictive results. The findings indicate that COâ‚‚ emissions decrease with the consumption of renewable energies (β ≈ −0.49, p < 0.01), and increase with higher per-capita amounts of energy consumption (β ≈ 0.0027, p < 0.01) and PM2.5 air pollution (β ≈ 0.66, p < 0.01). The social indicators produce mixed results: education, female equality (β ≈ −0.0017, p < 0.05), and political female empowerment (β ≈ −0.17, p < 0.01) lower the COâ‚‚ emissions, and higher food availability (β ≈ 0.086, p < 0.01) and higher growth rates for poor incomes (β ≈ 0.45, p < 0.05) increase COâ‚‚ emissions. The governance indicators also produce mixed results: effective and law-based strict regulation tends to decrease COâ‚‚ emissions for emerging markets, while such benefits are considerably restricted for high-income countries with more developed infrastructure. The cluster analysis confirms that balanced ESG performance is a characteristic of low-emission clusters, while unbalanced ESG performance is detected in higher-emission clusters. The machine learning classifiers (k-NN, Random Forest, GBM, SVR) confirmed that non-linear connections are detected, and that PM2.5 and renewable energies have the greatest impact. The study findings reveal that the ESG pillars influence COâ‚‚ emissions interactively and cumulatively, and that effective decarbonization by policy requires agreement among environmental, social, and governance areas.
Q3. It employs a ......... (The use of it is misused here), Please check and fix, this applies to other (it) in the abstract and in the manuscript (Abstract).
A3. We have applied the following changes:
- Line 228. From “It is primarily […]” to “The research is primarily […];
- Line 264. From “It is worthwhile to note that this problem was not taken into consideration […]” to “This issue was not taken into consideration […]”;
- Line 305. From “Rather, it attempts to uncover mutual interdependencies […]” to “The study attempts […]”;
- Line 313-314. From “Thus, it is subsequently uncovered that countries […]” to “The analysis reveals that countries […]” ;
- Line 356. From “It introduces […]” to “This section introduces […]”;
- Line 702. From “It has a simple message […] to “The evidence conveys a simple message […]”;
- Line 870. From “It suggests that […]” to “The evidence suggests that […]”;
- Line 1299. From “It concludes that […]” to “The study concludes that […]”.
Q4. Check line 508.
A4. The misprint in line 508 has been corrected.
Q5. The figure label should be under the figures.
A5. All captions have been placed below the figures.
Q6. Check figures 7, 10, 12, 13, 16, 17, 18, 19, 20, and other figures as necessary.
A6. All figures have been checked. Captions have been adjusted, and figures have been analyzed.
Q7. There is a need to proofread the manuscripts to correct all grammatical errors.
A7. The level of English has been improved.
Q8. Appendix should be after references.
A8. The appendices have been placed following the references.
